# Description and evaluation of the Diat-HadOCC model v1.0: the ocean biogeochemical component of HadGEM2-ES

Ian Totterdell[1]

[1]Met Office, Fitzroy Road, Exeter, EX1 3PB, UK

*Correspondence to:* Dr Ian Totterdell (ian.totterdell@metoffice.gov.uk)

**Abstract.** The Diat-HadOCC model (version 1.0) is presented. A simple marine ecosystem model with coupled equations representing the marine carbon cycle, it formed the ocean biogeochemistry sub-model in the Met Office's HadGEM2-ES Earth System Model. The equations are presented and described in full, along with the underlying assumptions, and particular attention is given to how they were implemented for the CMIP5 simulations. Results from the CMIP5 Historical simulation (particularly those for the simulated 1990s) are shown and compared to data: dissolved nutrients and dissolved inorganic carbon, as well as biological components, productivity and fluxes. Where possible, the amplitude and phase of the predicted seasonal cycle is evaluated. Since the model was developed to explore and predict the effects of climate change on the marine ecosystem and marine carbon cycle, the response of the model to the RCP8.5 future scenario is also shown. While the model simulates the historical and current global annual mean air-sea $CO_2$ flux well, and is consistent with other modelling studies about how that flux will change under future scenarios, several of the ecosystem metrics are less well simulated. The total chlorophyll is higher than observations, while the primary productivity is just below the estimated range. In the CMIP5 simulations certain parameter choices meant that the diatoms and the misc-Phytoplankton state variables behave more similarly than they should, and the surface dissolved silicate concentration drifts to excessively-high levels. The main structural problem with the model is shown to be the iron sub-model.

## 1   Introduction

The recent publication of the 5th Assessment Report of Working Group 1 of the Inter-Governmental Panel on Climate Change (IPCC, 2013) includes analysis of four possible future scenarios of how the global climate might change over the next few decades in response to anthropogenic emissions of carbon dioxide ($CO_2$) and other anthropogenic influences (e.g. changes to land use). These future scenarios are informed by the results of the 5th Climate Model Intercomparison Project, CMIP5 (Taylor et al., 2012), for which 47 different climate models ran one or more of the scenarios. Models are of course an absolute necessity for predicting future climate, since no observations can exist.

The number of general circulation models (GCMs) available to study climate has increased rapidly in recent years, and the range of processes and feedbacks that they can represent has also become more comprehensive. Initially there were just physical models, describing the circulation of the atmosphere and the ocean and how those circulations redistributed and stored heat, as well as the response of the system to rising atmospheric $CO_2$. The first coupled climate model to include representations of the land and marine carbon cycles, including terrestrial vegetation and soils and marine ecosystems and capable of representing their basic feedbacks on the climate, was HadCM3LC (Cox et al., 2000). In that model, the terrestrial vegetation was described by the TRIFFID model (Cox, 2001), while the chemistry of carbon dioxide in sea-water and the marine ecosystem were described by the Hadley Centre Ocean Carbon Cycle (HadOCC) model (Palmer and Totterdell, 2001). The latter is a simple Nutrient-Phytoplankton-Zooplankton-Detritus (NPZD) model, using nitrogen as the limiting element.

A brief overview of Met Office model nomenclature is useful here. The Met Office modelling system used (over a time period of several decades) for climate studies and for numerical weather prediction is known as the Unified Model, and the coupled climate models exist as various versions of it. The HadCM3LC model mentioned above featured a lower-resolution ("L") ocean sub-model than the HadCM3C model, which itself was the member of the HadCM3 family of coupled climate models (Gordon et al., 2000; version 4.5 of the Unified Model) that featured an interactive carbon cycle ("C") in the atmosphere, on land and in the ocean. The HadGEM2 family of climate models (The HadGEM2 Development Team, 2011), a development of HadCM3 with enhanced resolution and improved parameterisations that was used for CMIP5 simulations, was version 6.6 of the Unified Model. In particular HadGEM2-ES (Collins et al., 2011), featuring active Earth System components including version 1.0 of the Diat-HadOCC sub-model, was version 6.6.3.

The aim of this paper is to describe and validate version 1.0 of the Diat-HadOCC model, as used in HadGEM2-ES to run simulations for the CMIP5 experiment. Although the simulations were run several years ago this decription of the model is important as a record and can inform other modellers of potential parameterisations that succeeded (or not) here. The equations are presented and described in detail, and reasons are given for certain choices made in the representation of processes and in the values of parameters. Where potential other uses of the model (e.g. in ocean-only simulations forced by re-analysis fluxes) differs from its use here, this is mentioned. The publicly-available model output submitted to CMIP5 is used to evaluate the model, and its successes and weaknesses discussed.

## 2   Description of the Diat-HadOCC model, version 1.0

As shown in Table 1 and Figure 1 the Diat-HadOCC model has thirteen biogeochemical state variables, representing three dissolved nutrients (nitrate, silicate and iron), two phytoplankton (diatoms and misc-Phyto; plus diatom silicate), one zooplankton, three detritus compartments (detrital nitrogen, carbon and silicon), dissolved oxygen, dissolved inorganic carbon and alkalinity. "misc-Phyto(plankton)" refers to the "Miscellaneous Phytoplankton" term used in the CMIP5 database, i.e. any phytoplankton that is not specified to be a particular functional type. All the state variables are advected by the ocean currents and mixed by physical processes such as the isopycnal diffusion, diapycnal diffusion and convective mixing. The biogeochemical processes that affect the biogeochemical state variables are shown below in basic form, with greater detail on the processes given in

subsequent paragraphs. In the following equations all flows are body (point) processes except those in [ square brackets ] which are biogeochemical flows across layer interfaces.

$$\frac{dDIN}{dt} = ph_{resp} + dm_{resp} + ph_{mort} \cdot f_{nmp} + dm_{mort} \cdot f_{nmp} + grz_{DIN} + zp_{lin} + zp_{mort} \cdot f_{zmrt}$$
$$+ dtn_{remin} + dtn_{bedrmn} - ph_{PP} - dm_{PP} \tag{1}$$

$$\frac{dSi}{dt} = dtsi_{remin} + dtsi_{bedrmn} - dm_{PP} \cdot R_{si2n}^{Dm} \tag{2}$$

$$\frac{dDIC}{dt} = ph_{resp} \cdot R_{c2n}^{Ph} + dm_{resp} \cdot R_{c2n}^{Dm} + ph_{mort} \cdot f_{nmp} \cdot R_{c2n}^{Ph} + dm_{mort} \cdot f_{nmp} \cdot R_{c2n}^{Dm} + grz_{DIC}$$
$$+ zp_{lin} \cdot R_{c2n}^{Zp} + zp_{mort} \cdot f_{zmrt} \cdot R_{c2n}^{Zp} + dtc_{remin} + dtc_{bedrmn} + crbnt$$
$$- ph_{PP} \cdot R_{c2n}^{Ph} - dm_{PP} \cdot R_{c2n}^{Dm} + [CO2_{asf}] \tag{3}$$

$$\frac{dTAlk}{dt} = 2 \cdot crbnt - \frac{dDIN}{dt} \tag{4}$$

$$\frac{dOxy}{dt} = [Oxy_{asf}] - \left( \frac{dDIC}{dt} - crbnt - [CO2_{asf}] \right) \cdot R_{o2c}^{eco} + resetO_2 \tag{5}$$

The terms in Equation 1 show that the concentration of dissolved inorganic nitrogen is increased by, in order: a release of nitrogen associated with respiration by misc-phytoplankton (to keep the cell's molecular C:N ratio constant: Equation 37); a corresponding release associated with diatom respiration (Equation 36); fractions of the nitrogen released by the natural mortalities of misc-phytoplankton and of diatoms (the rest of the nitrogen in each case passes to sinking detritus $DtN$, see Equations 40 and 38); a release of nitrogen due to grazing by zooplankton on misc-phytoplankton, diatoms and detritus (Equation 34); losses from zooplankton (mainly associated with respiration; Equation 41); a fraction of the loss due to zooplankton mortality (natural and due to unmodelled grazing by higher trophic levels; Equation 42); and nitrogen returned to the dissolved state by the remineralization of sinking detritus in the water-column (Equation 46) and at the sea-floor (Equation 51). Conversely, the final two terms show that the concentration is decreased by uptake by misc-phytoplankton and diatoms to fuel photosynthesis and primary production (respectively Equations 79 and 80). The processes of nitrogen deposition from the atmosphere, inflow from rivers and estuaries, release from sediments, nitrogen fixation and denitrification are not included in the Diat-HadOCC model.

Equation 2 shows that the concentration of dissolved silicate is increased by the dissolution of detrital silicate in the water-column (Equation 48) and at the sea-floor (Equation 51), while it is decreased by uptake by diatoms to produce opaline shells in association with growth (Equation 80; the Si:N ratio $R_{si2n}^{Dm}$ is a function of the dissolved iron concentration following Equation 9). As with DIN, there are no inputs/losses of Si from/to the atmosphere, rivers, estuaries or sediments.

Each of the processes increasing or decreasing the dissolved inorganic nitrogen concentration has a counterpart that increases or decreases the dissolved inorganic carbon concentration; Equation 3 shows those processes and also the two processes that affect $DIC$, namely the formation and dissolution of solid calcium carbonate ($crbnt$, Equation 63) and the air-sea flux of $CO_2$ (Equation 82). Apart from the air-sea flux of $CO_2$ there are no other inputs/losses of inorganic carbon to the ocean.

In this model, biologically-mediated changes to the total alkalinity are associated with either the formation and dissolution of solid calcium carbonate or the uptake and release of dissolved inorganic nitrogen; Equation 4 shows how these processes are related to the alkalinity. Because the carbonate ion $CO_3^{2+}$ has two charges the change in the alkalinity due to $crbnt$ is double the change in $DIC$, and of opposite sign. Although uptake by phytoplankton of dissolved nitrate does not directly change the alkalinity it is usually associated with a balancing release of $OH^-$ ions which does change it (Goldman and Brewer , 1980). In the model all the $DIN$ taken up is assumed to be nitrate, but in the real ocean some of the nutrient will be dissolved ammonia, $NH_4^+$, which is associated with a release of $H^+$ ions that change the alkalinity in the opposite sense to the $OH^-$ ions; the model's omission of ammonium ions is not a great problem as any that is taken up for growth will likely have been produced locally shortly before, given that ammonium has a short residence time in the upper water-column.

Dissolved oxygen is included in the model as a diagnostic tracer: its concentration is changed by biological processes (as well as physical and chemical ones) but does not affect any other model state variable. It has particular value as a diagnostic of the respiration of organic matter at depth in the water-column, but also allows for the simulation of oxygen-minimum zones, and their evolution under climate change. It is assumed for the model that all respiration of organic matter is aerobic, so the same O:C ratio $R_{o2c}^{eco}$ can be used for all ecosystem processes, including both uptake and release of $O_2$; the second term in Equation 5 (i.e. within the large brackets) connects such oxygen fluxes to those of organic carbon. The first term in that equation relates to the air-sea flux of oxygen. The third term, $resetO_2$, is included to prevent the dissolved oxygen concentration going negative: at the end of each time-step, if the combination of physical fluxes and biological processes have taken the concentration in any grid-cell below zero, the concentration is re-set to zero and the amount that has been added to the model recorded. The column inventory of such re-set additions is calculated and subtracted from the surface layer; because that layer is in close contact with the atmosphere this adjustment should never reduce the surface concentration to zero (and in the CMIP5 simulations never came close to doing so anywhere). This approach was adopted in the model to prevent negative concentrations of dissolved $O_2$ while conserving the global $O_2$ inventory.

## 2.1 Diatoms and misc-Phytoplankton

$$\frac{dPh}{dt} = ph_{PP} - ph_{resp} - ph_{mort} - ph_{grz} \tag{6}$$

$$\frac{dDm}{dt} = dm_{PP} - dm_{resp} - dm_{mort} - dm_{grz} - [dm_{sink}] \tag{7}$$

$$\frac{dDmSi}{dt} = dm_{PP} \cdot R_{si2n}^{Dm} - dmsi_{mort} - dmsi_{grz} - [dmsi_{sink}] \tag{8}$$

In the model misc-Phytoplankton and Diatoms are both quantified by their nitrogen content, and have units of mMol N m$^{-3}$. Their carbon contents are related to their nitrogen contents by fixed elemental ratios, respectively $R_{c2n}^{Ph}$ and $R_{c2n}^{Dm}$. Equation 6 shows that, in terms of biological processes, the misc-Phytoplankton concentration is increased by growth and decreased by respiration, mortality and grazing by zooplankton. Equation 7 shows that the Diatom concentration is increased and decreased by analogous biological processes, but is additionally subject to sinking at a constant velocity $V_{Dm}$ because of gravity. Equation 8 describes the (analogous) biological processes that increase or decrease the concentration of opal shells

attached to living diatoms (Diatom Silicate), which is also subject to sinking (at velocity $V_{Dm}$); since the ratio of silicon in the diatom shell to nitrogen in the organic tissue of the diatom cell can vary Diatom Silicate has to be represented as a distinct model state variable.

The growth of diatoms and misc-Phytoplankton (respectively $dm_{PP}$ and $ph_{PP}$) is a function of the availability of macro- and micro-nutrients, the temperature and the availability of light. The growth limitation by dissolved nitrate (and, in the case of Diatoms, also by dissolved silicate) in the model has a hyperbolic form, while that by dissolved iron is represented in a different way. The effect of dissolved iron ($FeT$) in the Diat-HadOCC model is to vary certain parameter values: the assimilation numbers (maximum growth rates) for diatoms and misc-Phytoplankton (respectively $P_m^{Dm}$ and $P_m^{Ph}$), the silicon:nitrogen ratio for diatoms $R_{si2n}^{Dm}$, the zooplankton base preference for feeding on diatoms $bprf_{Dm}$ and the zooplankton mortality $\Pi_{mort}^{Zp}$. (Note that, because the base feeding preferences are subsequently normalised so that their sum is 1, changing the preference for diatoms will mean the preferences for misc-Phytoplankton and for detritus also change.) The dependence of zooplankton parameters on the dissolved iron concentration is not intended to suggest a direct causal relation (or that the parameters relating to any single species of zooplankton are iron-dependent) but rather reflect a change in the types and species of zooplankton that dominate the ecosystem when their phytoplankton prey-species respond to greater iron-stress by becoming more silicified; larger phytoplankton cells with thicker and more protective shells will be less palatable to predators and predated by larger meso- and macro-zooplankton species, multi-cellular and with different life-cycles and lower specific mortality. Since there is only one zooplankton compartment in the Diat-HadOCC model its parameters must change to accurately represent such a shift. The parameterisation used here is based on the results of earlier, but unpublished, 1-D modelling work by the late Dr M.J.R. Fasham (pers. comm.), an extension of the work described in Fasham et al. (2006). Each of the iron-dependent parameters has an iron-replete value (the standard) and an iron-deplete value, and the realised value at a given time and location will be:

$$\Pi = \Pi_{replete} + (\Pi_{deplete} - \Pi_{replete})/\left(1 + \frac{FeT}{k_{FeT}}\right) \tag{9}$$

where $k_{FeT}$ is a scale factor for iron uptake. In the CMIP5 simulations run using HadGEM2-ES (with the Diat-HadOCC model as the ocean biogeochemical component) only the value of $P_m^{Dm}$ varied (i.e. the iron-replete and -deplete values of the other parameters were set equal).

The Diat-HadOCC model, as coded, includes an option for the growth-rate to vary exponentially with temperature according to Equation 1 of Eppley (1972) (normalised so that default rates occur at 20°C). However for the CMIP5 simulations run using HadGEM2-ES the temperature variation of phytoplankton growth-rate was switched off and the default values were used (i.e. in the equation below $f_{Temp}$ was always equal to 1).

$$P^{Ph} = \left(P_{m,r}^{Ph} + \frac{(P_{m,d}^{Ph} - P_{m,r}^{Ph})}{(1 + \frac{FeT}{k_{FeT}})}\right) \cdot MIN\left(1.0, f_{Temp} \cdot \frac{DIN}{k_{DIN}^{Ph} + DIN}\right) \tag{10}$$

$$P^{Dm} = \left(P_{m,r}^{Dm} + \frac{(P_{m,d}^{Dm} - P_{m,r}^{Dm})}{(1 + \frac{FeT}{k_{FeT}})}\right) \cdot MIN\left(1.0, f_{Temp} \cdot \frac{DIN}{k_{DIN}^{Dm} + DIN} \cdot \frac{Si}{k_{Si}^{Dm} + Si}\right) \tag{11}$$

In the above equations the combined effect of the temperature and the macro-nutrient concentrations is limited to a maximum factor of 1.0 to guard against excessively-fast growth should the water temperature should become very high (when the temperature factor is switched on).

The light-dependency of the growth rates of misc-Phytoplankton and Diatoms is calculated using an implementation of the scheme presented in Anderson (1993); it is described in detail in Annex A. In addition, although prescribed constant carbon:chlorophyll ratios (with the value 40.0 mg C / mg Chl for each phytoplankton type) were used in the CMIP5 simulations the option exists in the Diat-HadOCC model to calculate a variable ratio (similar to that used in conjunction with the HadOCC model in Ford et al. (2012)), and this is described in Annex B.

## 2.2 Zooplankton and grazing

$$\frac{dZp}{dt} = grz_{Zp} - zp_{lin} - zp_{mort} \tag{12}$$

Zooplankton biomass (quantified by its nitrogen content) is increased by the grazing (of misc-phytoplankton, diatoms and detrital particles; see Equation 30) and decreased by losses such as respiration (Equation 41) and by density-dependent predation by the un-modelled higher trophic levels (Equation 42).

The grazing function used in the Diat-HadOCC model differs from that used in the HadOCC model in that it uses a 'switching' grazer similar to that used in Fasham et al. (1990; hereafter FDM90). It is noted that some authors (e.g. Gentleman et al., 2003) recommend against using such a formulation because it can lead to reduced intake when food resources are increasing. The single zooplankton consumes diatoms, misc-Phytoplankton and (organic) detrital particles. As in FDM90 the realised preference $dprf_X$ for each food type depends on that type's abundance and on the base preferences $bprf_X$:

$$dprf_{denom} = bprf_{Dm} \cdot R_{b2n}^{Dm} \cdot Dm + bprf_{Ph} \cdot R_{b2n}^{Ph} \cdot Ph + bprf_{Dt} \cdot (R_{b2n}^{DtN} \cdot DtN + R_{b2c}^{DtC} \cdot DtC) \tag{13}$$

$$dprf_{Dm} = \frac{bprf_{Dm} \cdot R_{b2n}^{Dm} \cdot Dm}{dprf_{denom}} \tag{14}$$

$$dprf_{Ph} = \frac{bprf_{Ph} \cdot R_{b2n}^{Ph} \cdot Ph}{dprf_{denom}} \tag{15}$$

$$dprf_{Dt} = \frac{bprf_{Dt} \cdot (R_{b2n}^{DtN} \cdot DtN + R_{b2c}^{DtC} \cdot DtC)}{dprf_{denom}} \tag{16}$$

where, if $M_N$ and $M_C$ are the respective atomic weights of nitrogen and carbon (14.01 and 12.01 g Mol$^{-1}$) and $R_{c2n}^{Rdfld}$ is the Redfield C:N ratio (106 Mol C : 16 Mol N), then the $R_{b2Y}^X$ terms convert from nitrogen or carbon units to biomass units that

allow the various potential food items to be compared:

$$E = (M_N + M_C \cdot R_{c2n}^{Rdfld})^{-1}$$

$$R_{b2n}^{Ph} = E \cdot (M_N + M_C \cdot R_{c2n}^{Ph}) \tag{17}$$

$$R_{b2n}^{Dm} = E \cdot (M_N + M_C \cdot R_{c2n}^{Dm}) \tag{18}$$

$$R_{b2n}^{Zp} = E \cdot (M_N + M_C \cdot R_{c2n}^{Zp}) \tag{19}$$

$$R_{b2n}^{DtN} = E \cdot M_N \tag{20}$$

$$R_{b2c}^{DtC} = E \cdot M_C \tag{21}$$

Note that the base preference values supplied (or calculated as a function of iron-limitation) $bprf_X$ are normalised so that they sum up to 1. The available food is:

$$food = dprf_{Dm} \cdot R_{b2n}^{Dm} \cdot Dm \; + \; dprf_{Ph} \cdot R_{b2n}^{Ph} \cdot Ph \; + \; dprf_{Dt} \cdot (R_{b2n}^{DtN} \cdot DtN + R_{b2c}^{DtC} \cdot DtC) \tag{22}$$

and the grazing rates on the various model state variables are:

$$dm_{grz} = \frac{dprf_{Dm} \cdot Dm \cdot g_{max} \cdot R_{b2n}^{Zp} \cdot Zp}{g_{sat} + food} \tag{23}$$

$$dmsi_{grz} = \frac{dprf_{Dm} \cdot DmSi \cdot g_{max} \cdot R_{b2n}^{Zp} \cdot Zp}{g_{sat} + food} \tag{24}$$

$$ph_{grz} = \frac{dprf_{Ph} \cdot Ph \cdot g_{max} \cdot R_{b2n}^{Zp} \cdot Zp}{g_{sat} + food} \tag{25}$$

$$dtn_{grz} = \frac{dprf_{Dt} \cdot DtN \cdot g_{max} \cdot R_{b2n}^{Zp} \cdot Zp}{g_{sat} + food} \tag{26}$$

$$dtc_{grz} = \frac{dprf_{Dt} \cdot DtC \cdot g_{max} \cdot R_{b2n}^{Zp} \cdot Zp}{g_{sat} + food} \tag{27}$$

A fraction $(1 - f_{ingst})$ of the grazed material is not ingested: of this, a fraction $f_{messy}$ returns immediately to solution as $DIN$ and $DIC$ while the rest becomes detritus. All of the grazed diatom silicate $DmSi$ immediately becomes detrital silicate $DtSi$. Of the organic material that is ingested, a source-dependent fraction $(\beta^X)$ of the nitrogen and of the carbon is assimilatable while the remainder is egested from the zooplankton gut as detrital nitrogen $DtN$ or carbon $DtC$. The amount of assimilatable material that is actually assimilated by the zooplankton $grz_{Zp}$ is governed by its C:N ratio compared to that of

the zooplankton: as much as possible is assimilated, with the remainder passed out immediately as $DIN$ or $DIC$.

$$assim_N = f_{ingst} \cdot (\beta^{Dm} \cdot dm_{grz} + \beta^{Ph} \cdot ph_{grz} + \beta^{Dt} \cdot dtn_{grz}) \tag{28}$$

$$assim_C = f_{ingst} \cdot (\beta^{Dm} \cdot R^{Dm}_{c2n} \cdot dm_{grz} + \beta^{Ph} \cdot R^{Ph}_{c2n} \cdot ph_{grz} + \beta^{Dt} \cdot dtn_{grz}) \tag{29}$$

$$grz_{Zp} = MIN\left(assim_N, \frac{assim_C}{R^{Zp}_{c2n}}\right) \tag{30}$$

$$grz_{DtN} = (1 - f_{ingst}) \cdot (1 - f_{messy}) \cdot (dm_{grz} + ph_{grz} + dtn_{grz})$$
$$+ f_{ingst} \cdot ((1 - \beta^{Dm}) \cdot dm_{grz} + (1 - \beta^{Ph}) \cdot ph_{grz} + (1 - \beta^{Dt}) \cdot dtn_{grz}) \tag{31}$$

$$grz_{DtC} = (1 - f_{ingst}) \cdot (1 - f_{messy}) \cdot (R^{Dm}_{c2n} \cdot dm_{grz} + R^{Ph}_{c2n} \cdot ph_{grz} + dtc_{grz})$$
$$+ f_{ingst} \cdot ((1 - \beta^{Dm}) \cdot R^{Dm}_{c2n} \cdot dm_{grz} + (1 - \beta^{Ph}) \cdot R^{Ph}_{c2n} \cdot ph_{grz} + (1 - \beta^{Dt}) \cdot dtc_{grz}) \tag{32}$$

$$grz_{DtSi} = dmsi_{grz} \tag{33}$$

$$grz_{DIN} = (1 - f_{ingst}) \cdot f_{messy} \cdot (dm_{grz} + ph_{grz} + dtn_{grz}) + MAX\left(0, assim_N - \frac{assim_C}{R^{Zp}_{c2n}}\right) \tag{34}$$

$$grz_{DIC} = (1 - f_{ingst}) \cdot f_{messy} \cdot (R^{Dm}_{c2n} \cdot dm_{grz} + R^{Ph}_{c2n} \cdot ph_{grz} + dtc_{grz})$$
$$+ MAX(0, assim_C - assim_N \cdot R^{Zp}_{c2n}) \tag{35}$$

## 2.3 Other processes

The other loss terms for diatoms, misc-Phytoplankton and zooplankton are:

$$dm_{resp} = \Pi^{Dm}_{resp} \cdot Dm \tag{36}$$

$$ph_{resp} = \Pi^{Ph}_{resp} \cdot Ph \tag{37}$$

$$dm_{mort} = \Pi^{Dm}_{mort} \cdot Dm^2 \tag{38}$$

$$dmsi_{mort} = \Pi^{Dm}_{mort} \cdot Dm \cdot DmSi \tag{39}$$

$$ph_{mort} = \Pi^{Ph}_{mort} \cdot Ph^2 \qquad (Ph > ph_{min})$$
$$= 0 \qquad (Ph < ph_{min}) \tag{40}$$

$$zp_{lin} = \Pi^{Zp}_{lin} \cdot Zp \tag{41}$$

$$zp_{mort} = \Pi^{Zp}_{mort} \cdot Zp^2 \tag{42}$$

In the above equations $ph_{min}$ is a set (low) concentration of $Ph$ below which the natural mortality of misc-Phytoplankton is set to zero; the inclusion of this term was a pragmatic and necessary choice in an early version of the model to prevent the misc-Phytoplankton dying out in certain parts of the seasonal cycle at high latitudes (it was not found to be necessary to include a similar term for diatoms). It can be rationalised as representing the ability of phytoplankton to enter a "cyst" state under certain stressful conditions. Although respiration involves a release of carbon (as $CO_2$) the fixed C:N ratios used in the models for misc-Phytoplankton, Diatoms and Zooplankton require a balancing release of nitrogen from those model compartments. The "natural mortality" of both phytoplankton variables refers to cell-death, particularly including that caused by viral infections,

which will be density-dependent. The $zp_{mort}$ refers primarily to zooplankton losses due to predation by un-modelled higher trophic levels, and is the closure term of the modelled ecosystem.

### 2.3.1 Detrital sinking and remineralisation

$$
\begin{aligned}
\frac{d\,DtN}{dt} &= ph_{mort} \cdot (1 - f_{nmp}) + dm_{mort} \cdot (1 - f_{nmp}) + grz_{DtN} + zp_{mort} \cdot (1 - f_{zmrt}) + dm_{bedmrt} \\
&\quad - dtn_{grz} - dtn_{remin} - [dtn_{sink}] \tag{43} \\
\frac{d\,DtSi}{dt} &= dmsi_{mort} + grz_{DtSi} + dmsi_{bedmrt} - dtsi_{remin} - [dtsi_{sink}] \tag{44} \\
\frac{d\,DtC}{dt} &= ph_{mort} \cdot (1 - f_{nmp}) \cdot R_{c2n}^{Ph} + dm_{mort} \cdot (1 - f_{nmp}) \cdot R_{c2n}^{Dm} + grz_{DtC} + zp_{mort} \cdot (1 - f_{zmrt}) \cdot R_{c2n}^{Zp} \\
&\quad + dm_{bedmrt} \cdot R_{c2n}^{Dm} - dtc_{grz} - dtc_{remin} - [dtc_{sink}] \tag{45}
\end{aligned}
$$

All detrital material sinks at a constant speed $V_{Dt}$ at all depths. Diatoms (and its associated silicate) sinks at a constant speed $V_{Dm}$ at all depths. Detrital remineralisation (of $DtN$ and $DtC$) is depth-dependent, the specific rate varying as the reciprocal of depth but with a maximum value. This functional form gives a depth variation of detritus consistent with the Martin et al. (1987) power-law curve. Dissolution of opal does not vary with depth.

$$
dtn_{remin} = DtN \cdot MIN\left(\Pi_{rmnmx}^{DtN}, \frac{\Pi_{rmndd}^{DtN}}{z}\right) \tag{46}
$$

$$
dtc_{remin} = DtC \cdot MIN\left(\Pi_{rmnmx}^{DtC}, \frac{\Pi_{rmndd}^{DtC}}{z}\right) \tag{47}
$$

$$
dtsi_{remin} = DtSi \cdot \Pi_{rmn}^{DtSi} \tag{48}
$$

$$
dt(n,c,si)_{sink} = V_{Dt} \cdot \frac{d\,Dt(N,C,Si)}{dz} \tag{49}
$$

$$
d(m,msi)_{sink} = V_{Dm} \cdot \frac{d\,D(m,mSi)}{dz} \tag{50}
$$

Since there are no sediments in the Diat-HadOCC model, all detritus that sinks to the sea-floor is instantly remineralised to N, C or Si and spread through the lowest three layers (above the sea-floor). Spreading over the bottom three levels is a numerical artifice to prevent excessive build-up of high concentrations (below regions of high primary productivity and sinking detritus) in bathymetric canyons that are too narrow to support advection and so rely on weak vertical mixing to redistribute N, C or Si being introduced by the instant sea-floor remineralisation (such high concentrations would themselves be artifacts of the model). It is reasoned that where the ocean is (thousands of metres) deep the time required for dissolved inorganic nutrients and carbon to return to the euphotic zone will be dominated by the slow deep circulation and mixing, and shortening the path by at most a couple of levels will not significantly affect this time; while on the shallow shelves the instant transport upwards through two levels will actually partially mitigate the absence from the model of tidal mixing, which is very important in such environments in the real ocean. Diatoms (and associated silicate) that sink to the sea-floor instantly die and become $DtN$, $DtC$

and $DtSi$, as appropriate, in the lowest layer. Therefore, if $btmflx_Y$ is the value of $[Y_{sink}]$ at the sea-floor:

$$
\begin{aligned}
dt(n,c,si)_{bedrmn} &= \frac{btmflx_{Dt(N,C,Si)}}{\Delta_{b3l}} & (btm\ 3\ lyrs) \\
&= 0 & (above\ btm\ 3\ lyrs)
\end{aligned}
\tag{51}
$$

$$
\begin{aligned}
(dm,dmsi)_{bedmrt} &= \frac{btmflx_{(dm,dmsi)}}{\Delta_{b1l}} & (bottom\ lyr) \\
&= 0 & (other\ lyrs)
\end{aligned}
\tag{52}
$$

where $btmflx_X$ is the sinking flux of $X$ to the sea-floor and $\Delta_{bMl}$ is the combined thickness of the bottom $M$ layers (of course, which layers those are will vary according to the location).

### 2.3.2 The iron cycle

$$
\begin{aligned}
\frac{dFeT}{dt} &= (ph_{resp} \cdot R_{c2n}^{Ph} + dm_{resp} \cdot R_{c2n}^{Dm} + ph_{mort} \cdot R_{c2n}^{Ph} + dm_{mort} \cdot R_{c2n}^{Dm} + grz_{DIC} + grz_{DtC} - dtc_{grz} \\
&\quad + zp_{lin} \cdot R_{c2n}^{Zp} + zp_{mort} \cdot R_{c2n}^{Zp} - ph_{PP} \cdot R_{c2n}^{Ph} - dm_{PP} \cdot R_{c2n}^{Dm}) \cdot R_{fe2c}^{eco} + [fe_{dust}] - fe_{adsorp}
\end{aligned}
\tag{53}
$$

Iron is added to the ocean by dust deposition from the atmosphere (prescribed or passed from the atmospheric sub-model in coupled mode; penultimate term in Equation 53), with a constant proportion (by weight) of the dust being iron which immediately becomes part of the total dissolved iron pool $FeT$. Iron is taken up by diatoms and misc-Phytoplankton during growth in a fixed ratio to the carbon taken up ($R_{fe2c}^{eco}$), and moves through the ecosystem in the same ratio, except that any flow of carbon to $DtC$ is associated with a flow of iron back to solution, as there is no iron in organic detritus in the model. Since the iron sub-model was developed there have been many experimental and observational studies of the marine iron cycle (e.g. Boyd et al., 2017) which have shown that this assumption (which was a pragmatic decision to maintain adequate levels of dissolved iron in the euphotic zone) is a bad one; the performance of the iron model is discussed further in the Conclusions.

While all iron that flows through the ecosystem is returned to solution, there is a final loss term for dissolved iron, namely (implicit) adsorption onto pelagic sinking mineral particles (*not* the model's detrital particles) and thence to the (implicit) sediments (last term in Equation 53). Only the fraction of $FeT$ that is not complexed to organic ligands can be adsorbed. The un-complexed (free) iron concentration $FeF$ and the complexed concentration $FeL$ are found by assuming a constant uniform total ligand concentration $LgT$ and a partition function $K_{FeL}$, and the adsorption flux $fe_{adsorp}$ derived from that:

$$
FeT = FeL + FeF
\tag{54}
$$

$$
LgT = FeL + LgF
\tag{55}
$$

$$
K_{FeL} = \frac{FeL}{FeF \cdot LgF}
\tag{56}
$$

$$
B = K_{FeL} \cdot (LgT - FeT) - 1
\tag{57}
$$

$$
FeF = FeT - LgT + \frac{1}{2 \cdot K_{FeL}} \cdot \left( B + \sqrt{B^2 - 4 \cdot K_{FeL} \cdot LgT} \right)
\tag{58}
$$

$$
fe_{adsorp} = \Pi_{ads}^{FeF} \cdot FeF
\tag{59}
$$

In the above equations, $LgF$ is the portion of the ligand concentration that is not bound to iron.

### 2.3.3 The calcium carbonate sub-model

Solid calcium carbonate is implicitly produced in a constant ratio ($R_{c2n}^{Ph}$) to organic carbon produced by misc-Phytoplankton. Note that this ratio is not the rain ratio, which compares the ratio of inorganic to organic carbon in the particulate sinking flux below the euphotic zone, since it compares the production of the respective carbon types, and while all the model inorganic carbon is exported from the surface layer only a fraction of the organic carbon is, and so this ratio represents the average of the product of the rain ratio and the organic export ratio. The total production (of solid calcium carbonate) is summed over the surface layers (those where production is non-zero) and instantly re-dissolved equally through the water column below the (prescribed) lysocline. If the sea-floor is shallower than the lysocline, then the dissolution takes place in the bottom layer (there being no sediments). The depth of the lysocline is always co-incident with a layer interface, and is constant both geographically and in time. In the following equations, $ccfrmtn$ and $ccdsltn$ are respectively the rate of formation and dissolution of solid calcium carbonate in a given layer, $xprt_{cc}$ is the export of calcium carbonate from the surface layers, and $crbnt$ is the net flux of carbon from solid calcium carbonate to DIC:

$$ccfrmtn = R_{cc2pp}^{Ph} \cdot R_{c2n}^{Ph} \cdot ph_{PP} \tag{60}$$

$$xprt_{cc} = \sum_n (ccfrmtn_n \cdot \Delta_n) \tag{61}$$

$$ccdsltn = \frac{xprt_{cc}}{\Delta_{dsl}} \qquad (valid\ lyrs)$$
$$= 0 \qquad (other\ lyrs) \tag{62}$$

$$crbnt = ccdsltn - ccfrmtn \tag{63}$$

where $\Delta_n$ is the thickness of layer $n$ and $\Delta_{dsl}$ is the total thickness of the valid layers (where dissolution can occur) in that water column, which is equal to the distance between the lysocline and the sea-floor if the lysocline is shallower than the sea-floor and the thickness of the deepest layer otherwise.

### 2.3.4 Air-Sea fluxes

Finally, the calculations of the air-to-sea fluxes of $O_2$ and $CO_2$ (respectively $[Oxy_{asf}]$ and $[CO2_{asf}]$) follow the methodology of OCMIP. The flux is the product of the gas-specific gas transfer (piston) velocity $Vp$ and the difference between the gas concentrations in the atmosphere (just above the sea-surface), $X_{sat}$, and in the (surface) ocean, $X_{surf}$:

$$X_{asf} = Vp_X \cdot (X_{sat} - X_{surf}) \tag{64}$$

The details can be found in Annex C.

## 3 Description of experiments

The Diat-HadOCC model formed the ocean biogeochemical component of the HadGEM2-ES Earth System model (Collins et al., 2011), which is part of the HadGEM2 family of coupled climate models (The HadGEM2 Development Team,

2011). Full details of the model set-up for the experiments described here can be found in those references, but a brief description is given here.

The atmospheric physical model has a horizontal resolution of 1.25° latitude by 1.875° longitude, and a vertical resoltion of 38 layers (to a height of 39 km). A timestep of 30 minutes is used. Eight species of aerosol are included in the atmosphere, as well as a representation of mineral dust (described in more detail below). The UK Chemistry and Aerosols (UKCA) model (O'Connor et al., 2014) describes the atmospheric chemistry. MOSES II (Essery et al., 2003) is used for the land surface scheme, with additional processes and components as described in papers about the derived JULES scheme by Best et al. (2011) and Clark et al. (2011). The hydrology includes a river-routing sub-model based on the TRIP scheme (Oki and Sud, 1998), which supplies freshwater (but not nutrients, carbon or alkalinity) to the ocean. The TRIFFID dynamic vegetation model (Cox, 2001; Clark et al. 2011) and a four-pool implementation of the RothC soil carbon model (Coleman and Jenkinson 1996,1999) are used to represent the terrestrial carbon cycle. TRIFFID calculates the growth and phenology of five plant functional types (broad-leaf trees, needle-leaf trees, C3 grasses, C4 grasses and shrubs) so that the (terrestrial) Gross Primary Production (GPP), and the Net Primary Production (NPP) can be determined, and thereby also the terrestrial sources and sinks of atmospheric carbon.

The ocean physical model is based on that described in Johns et al. (2006), with developments as detailed in the paper by The HadGEM2 Development Team (2011). It has a longitudinal resolution of 1°, while the latitudinal resolution is also 1° poleward of 30° (N or S) but increasing from that latitude to $\frac{1}{3}^{\circ}$ at the equator. In the vertical there are 40 levels with thicknesses increasing monotonically from 10 m in the top 100 m to 345 m at the bottom, and with a full depth of 5500 m. A timestep of 1 hour is used. The computer code is based on that of Bryan (1969) and Cox (1984). The active ocean tracers (temperature and salinity) use a pseudo fourth-order advection scheme (Pacanowski and Griffies, 1998), while the passive tracers (including all the ocean biogeochemical tracers) use the UTOPIA scheme (Leonard et al., 1993) with a flux-limiter. The Gent and McWilliams (1990) adiabatic mixing scheme is used in the skew flux form due to Griffies (1998), and with coefficient that varies spatially and temporally following Visbeck et al. (1997). An implicit linear free-surface scheme (Dukowicz and Smith, 1994) is included for freshwater fluxes. A simple upper mixed-layer scheme (Kraus and Turner, 1967) is used for vertical mixing due to surface fluxes of heat and freshwater for both active and passive tracers. The sea-ice model is based on the Los Alamos National Laboratory sea-ice model, CICE (Hunke and Lipscomb, 2004), including five thickness categories, elastic-viscous-plastic ice dynamics (Hunke and Dukowicz, 1997) and ice ridging. The presence of sea-ice of any thickness reduces to zero the light entering the water-column (so preventing photosynthesis by marine phytoplankton) and blocks completely the transfer of gases between the atmosphere and ocean.

Coupling between the atmosphere and ocean models happens every 24 model hours. After 48 atmospheric timesteps (of 30 minutes each) have been run the fluxes of heat, freshwater, wind-stress and wind mixing energy, along with any necessary biogeochemical quantities, are determined (usually as a time-mean over the 24 hours) and passed via the coupler to the ocean. Because the atmosphere and ocean models use different grids this involves re-gridding, with special care needing to be taken at the coasts where an atmospheric grid-box may correspond to both an ocean and a land grid-box. The ocean is then run for 24 timesteps (of 1 hour each) and the relevant fluxes calculated and passed to the atmosphere.

The biogeochemical quantities passed from the atmosphere to the ocean are the deposition flux of mineral dust and the concentration of $CO_2$ in the lowest atmospheric level, while the flux of $CO_2$ and the flux of Dimethyl Sulphide (DMS) are passed from ocean to atmosphere. Note however that in the concentration-driven simulations for which the results are presented here the atmospheric $CO_2$ concentration "seen" by the ocean is not passed from the atmosphere but prescribed in the ocean model (in such a way that it agrees with the atmospheric concentration prescribed in the atmosphere, once the different units are taken into account), and while the flux of $CO_2$ between the ocean and the atmosphere is calculated in the ocean model it is purely diagnostic and is not passed to the atmosphere.

The DMS sub-model is a simple empirical model based on Simo and Dachs (2002), in which the surface ocean DMS concentration is a function of the surface chlorophyll concentration (in the Diat-HadOCC model only chlorophyll associated with the non-diatom phytoplankton is considered) and the mixed layer depth. If the mixed layer depth is very deep (greater than 182.5m) the scheme of Aranami and Tsunogai (2004) is used. The implementation is described in more detail in Halloran et al. (2010). The same piston velocity function is used as for $CO_2$ (except, of course, that the appropriate Schmidt numbers are used).

The dust deposition flux is calculated in the atmosphere as part of the dust sub-model, which is based on that described in Woodward (2001) but with developments as detailed in Woodward (2011). Six size-classes of mineral dust particles are used (up to 30 $\mu$m radius), and deposition can be by four mechanisms: wet deposition from convective precipitation and from large-scale precipitation and dry deposition (i.e. settling under the force of gravity) from the lowest level and from levels above. For each size-class, the flux of dust being deposited is summed over the four mechanisms and separately passed to the ocean. Although not used in the simulations presented here, this separate passing allows for different size dust particles to have different soluble iron contents (supply of iron is the sole reason the dust deposition flux is passed to the ocean).

## 3.1 Simulations

The HadGEM2-ES model was used to run a wide range of simulations for CMIP5, the 5th Climate Model Intercomparison Project (Taylor et al., 2012); Jones et al. (2011) gives a detailed overview of the HadGEM2-ES simulations. The results presented here relate to a sub-set of three simulations, all with prescribed atmospheric $CO_2$ concentration. The first is the pre-industrial control ("piControl" in the CMIP5 terminology), the historical simulation ("historical"; from December 1859 to December 2005) and the RCP8.5 future simulation ("rcp85"). The historical simulation branched from the piControl, and rcp85 was a continuation of the historical to simulated year 2100.

The model was spun-up before the piControl commenced. The ocean has particular issues with spin-up, because ideally several cycles of the ocean overturning circulation are needed to bring the tracers into equilibrium with the circulation and the driving climatological fluxes from the atmosphere, and each cycle lasts 500-1,000 model years. It was therefore deemed impractical to spin the full coupled model for the required time, and in any case the atmosphere and land-surface models would reach equilibrium much faster.

The World Ocean Atlas (hereafter WOA) provides comprehensive gridded fields for the active tracers, temperature and salinity, and the processes affecting these quantities at the surface are relatively well understood and parameterised, so it was possible to initialise the ocean with fields close to equilibrium. The biogeochemical tracer fields however were not so easy

to initialise. WOA gridded fields are available for the nutrients nitrate and silicate and for oxygen, but they are based on many fewer data than those for temperature and salinity. Gridded fields are available for dissolved inorganic carbon (DIC) and total alkalinity (TAlk) from GLODAP (Sabine et al., 2005; Key et al., 2004) but these are based on even fewer data and relate to the present day with a substantial storage of anthropogenic carbon rather than the pre-industrial distribution (a correction for anthropogenic storage is available, but the method used for its production introduces many more uncertainties). At the time that the model spin-ups were started the 2009 edition of the WOA database was the most recent, so those fields were used. In addition, while the Diat-HadOCC model was developed to represent the main ocean biogeochemical processes which (along with the physical circulation) determine the horizontal and vertical distributions of these tracers the incomplete knowledge of these processes, particularly quantitatively, and the model's necessary simplicity mean that the simulated fields may be significantly different from those measured in the real ocean (even with an accurate circulation). Therefore the ocean biogeochemical tracers, even if initialised from the best-available gridded fields, required a significant period of spin-up before the drifts became acceptably small. The main criterion for "acceptably small" was a net pre-industrial air-sea flux of $CO_2$ that was below 0.2 Pg C / year (averaged over a decade, so inter-annual variability was smoothed out).

The tracers were therefore initialised as follows:

– Temperature and salinity: WOA 2009: Locarnini et al. (2010), Antonov et al. (2010)

– Nitrate, silicate (i.e. silicic acid), oxygen: WOA 2009: Garcia et al. (2010b), Garcia et al. (2010a)

– Iron: an initial field was produced from measurements reported in Parekh et al. (2004), on which the iron model used in Diat-HadOCC was based.

– misc-Phytoplankton, diatoms, zooplankton, and also C-, N-, and Si-detritus: a nominal small value ($10^{-6}$ mMol / m$^3$) was used, because these quantities (being mainly confined to the surface levels) would very quickly come into a pseudo-equilibrium with the climatological fluxes and the initial nutrient distributions, and then be able to track the decadal and centennial changes to those distributions.

– DIC and TAlk: these were initialised from (re-gridded) fields from an earlier pre-industrial simulation by the HadCM3C model, where the net air-sea $CO_2$ flux had been within the criterion; it was expected that the large-scale ocean circulation would not differ greatly between the models.

The early stages of the spin-up were done incrementally: while parameterisations of the land-surface and the dust sub-models were being tested forty-year simulations were run for each trial sequentially, and around 200 years of spin-up were obtained this way. It was reasoned that the different versions of the land and dust models would not produce significantly different equilibria for the ocean tracers, and the ocean biogeochemical model, which was unchanged, would be a more-dominant influence. After this period, another 100 years of simulation was completed with the finalised model, and during this average fields (one for each month of the year) were calculated for the climatological fluxes between the atmosphere and ocean. These average annual cycle fields were then used to force a coarse-resolution ocean-only model (a low-resolution version of the ocean component

of HadCM3 - see Gordon et al., 2000 - with Diat-HadOCC embedded) which could be run extremely efficiently. This ran for 2,000 simulated years, after which the biogeochemical fields (but NOT temperature or salinity) were re-gridded back to the HadGEM2-ES ocean resolution and put back in that model (at the point immediately following the 100-year coupled spin-up. HadGEM2-ES was subsequently run in coupled mode for a further 50 years, during which it was found that the main criterion of the net air-sea $CO_2$ flux being below 0.2 Pg C / year was comfortably satisfied, and the drifts in the other biogeochemical fields were reduced compared to before the ocean-only phase. However, there were still significant drifts in the silicate and dissolved iron fields: in the pre-industrial control simulation the silicate concentration in the top 100m increased by around 4.8 and 3.3 mMol-Si / $m^3$ during the first and second centuries respectively, while that in the lowest 2000m decreased by around 4.0 and 2.2 mMol-Si / $m^3$, and the dissolved iron increased at all depths, in the top 100m by 0.12 mMol-Fe / $m^3$ / century and below 1000m by 0.055 mMol-Fe/$m^3$/century.

The pre-industrial control (piControl) simulation was started from the end of the coupled spin-up, with its date set to 1st December 1859. (Note that HadGEM2-ES, like previous Met Office climate models, uses a 360-day year of 12 months each of 30 days, and begins its simulations on the 1st December, the start of meteorological winter, rather than 1st January.) It ran to the year 2100 and beyond. The atmospheric $CO_2$ concentration was prescribed at a constant value, and the concentration (strictly, the partial pressure) seen by the ocean was also held at the same constant value. The historical simulation began from the same date, using the same initial fields. It ran to the end (31st December) of 2005. The atmospheric $CO_2$ concentrations were prescribed according to the CMIP5 dataset (http://cmip-pcmdi.llnl.gov/cmip5/forcing.html). The future simulation, rcp85, began at 1st December 2005 and was initialised using the fields from the historical simulation that were valid for that time. Again, the atmospheric $CO_2$ was prescribed, but this time according to a future scenario (also to be found in the CMIP5 dataset). This was one of 4 RCPs (Representative Concentration Pathways; see Moss et al., 2010) calculated using an Integrated Assessment Model using projections of future anthropogenic emissions and other changes. RCP8.5 is the scenario with the highest atmospheric $CO_2$ concentrations, and the radiative forcing at year 2100 due to additional $CO_2$ is 8.5 W / $m^2$. Changes in the Earth System due to climate change will in general show most clearly in this scenario, and so, although HadGEM2-ES ran all four RCP simulations (Jones et al. 2011; which also gives more details of other climatically-active gases, etc. in these experiments) it is the results from RCP8.5 that are considered in the following section.

## 4   Results from the Diat-HadOCC model

The primary purpose of the Diat-HadOCC model is to represent the marine carbon cycle, along with the factors and feedbacks influencing and controlling it, in the past, in the present and in the future; and therefore initially the results described here relate to those quantities most directly connected with that cycle. However, it is also important to know that where the model results closely agree with observations they do so for the right reasons, rather than by coincidence, so certain other quantities are also presented.

## 4.1 Results for the present day (2010s)

### 4.1.1 Total Chlorophyll

Figure 2 shows the annual mean surface total chlorophyll predicted by the model for the (simulated) decade 2010-2019 in the upper panel and that derived from satellite retrievals in the lower panel. The satellite-derived data are from the GlobColour surface chlorophyll product (Fanton d'Andon et al., 2010; Maritorena et al., 2010) for the years 1998-2007, with further processing as described in Ford et al. (2012) to produce a monthly climatology, which has then been averaged to give the annual mean. Two things are immediately apparent: the geographical distributions are similar but the actual values in the model are noticeably more extreme: higher where the data are high (Southern Ocean, sub-polar gyres in the North Pacific and North Atlantic, eastern Equatorial Pacific) and lower where the data are low (mainly the sub-tropical gyres). In fact in the centres of the sub-tropical gyres the model chlorophyll is very slightly negative. Comparing the area-means of the respective annual mean fields, the model has an average of 0.812 mg Chl $m^{-3}$ while the average of the data is 0.213 mg Chl $m^{-3}$. However the seasonal cycle is also important, and Figure 3 shows (top panel) the seasonal cycle of the zonally-meaned model chlorophyll; (middle panel) the same but scaled by the factor 0.213/0.812 (so that the global annual mean is the same as that of the data); and (bottom panel) the seasonal cycle of the zonally-meaned data. It can be seen by comparing the middle and bottom panels that the excess Chlorophyll is accentuated by a greater-than-average factor when the observed chlorophyll is high. It is possible to find the best-fitting sine-curve through the monthly mean values at any points (assuming they form a repeating cycle); points are only shown where the variance of the residual cycle (after the best-fitting curve has been subtracted off) is less than half that of the original cycle (meaning that a sine-curve is a good first-order description of the seasonal cycle). Figure 4 shows the amplitude (left panels) and phase (right panels) of the seasonal cycle so derived of the model chlorophyll (upper panels, amplitude adjusted by factor 0.213/0.812 so that patterns can be better compared) and the satellite-derived data (lower panels). In the model, the seasonal cycle is larger (even when adjusted) in much of the Southern Ocean and in the Equatorial Pacific, and slightly lower in the sub-polar North Atlantic. In terms of the phase, and high latitudes there is good agreement between model and the data, though the model misses the late-summer peak that dominates the sub-polar North Pacific. In the tropics and sub-tropics there is less agreement; in particular across all basins the model shows peak chlorophyll around September/October in the Southern Hemisphere between 10° and 35° but the data show the peak occuring two to three months earlier. Figure 5 compares the model total chlorophyll to the GlocColour product in a Taylor diagram. The mean concentration and the mid-point, amplitude and phase of the sine-curve that best fits the seasonal cycle are shown (only points that satisfy the variance condition are considered for the seasonal cycle). The left panel shows the comparisons for the actual model results, while the right panel shows those for the model results scaled by the 0.213/0.812 factor. It can be seen that there is an excellent fit in terms of the standard deviation for the seasonal cycle phase, but that correlation is poor, and for the concentrations and cycle amplitude the (un-scaled) model greatly over-estimates the values (and again the correlation is poor). The scaled comparison is of course better for the standard deviation, but shows that the seasonal cycle in the model is generally of lower amplitude compared to the mean than is found in the data. The satellite-derived chlorophyll products usually accurately show very high concentrations in shelf-seas and close to the coasts, but models often struggle to show those features:

the coarse physical resolution means that in many locations there is no coastal shelf (ocean water-columns adjacent to land can be as deep as the rest of the basin) and most models (including HadGEM2-ES) do not represent the tidal mixing processes that in the real ocean are vital to supply nutrients that have been regenerated in the shallow sediments to the surface where they can drive the high growth rates. Therefore the Taylor diagram also shows a comparison (open symbols) that excludes two grid-boxes around the coasts, so that it is only between open-ocean points. The result is that the model over-estimates the concentrations even more, though the correlation is slightly improved.

### 4.1.2   Diatoms and Misc-Phytoplankton

Figure 6 shows the total surface biomass of phytoplankton (in mMol N m$^{-3}$), and also, separately, that of the two phytoplankton types, diatoms and misc-Phyto: the mean for the model years 2010-2019. The geographical patterns are naturally very similar to that of the model's total surface chlorophyll, since the CMIP5 simulations used a fixed carbon:chlorophyll ratio for each of the phytoplankton (and the same value, 40.0 mg C / mg Chl, for each type). The geographical patterns for each type are also very similar to each other, with the diatoms having a slightly greater value than the misc-Phyto (global averages 1.486 and 1.223 mMol C m$^{-3}$ respectively, so diatoms make up 55% of the total surface biomass). The diatoms are slightly more dominant than the global average in the North Atlantic Ocean and in the Southern Ocean, both areas where surface silicic acid (needed by diatoms for shell formation) is plentiful. An issue with these results is that the distributions of the two phytoplankton types are more similar than they should be. This is due to two factors: the parameter values used (for growth rate, etc.) are similar, and the concentrations of dissolved silicate and dissolved iron, which should produce contrasting responses in the two types, are less limiting in the model than they are in the real ocean and so fail to distinguish them. In terms of the parameter values, the growth rate of diatoms was 1.85 $d^{-1}$ iron-replete and 1.11 $d^{-1}$ iron-deplete while that of misc-Phytoplankton was 1.50 $d^{-1}$, and diatoms had a sinking rate of 1.0 $md^{-1}$ while misc-Phytoplankton did not sink, but the majority of other parameters were identical and there was no difference between the iron-replete and iron-deplete values where those could vary (except the diatom growth rate, as described above). These parameter choices were made after a limited sensitivity analysis that was constrained by the time and computing resources available, and it was reasoned that only if that analysis showed a significant reason for choosing different values for corresponding diatom and misc-Phytoplankton parameters should they not be identical. The surface silicate concentration was, during the historical and future RCP simulations, much too high because the dissolution (remineralisation) rate was too high so diatom growth was not restricted by silicate-limitation in areas and in parts of the seasonal cycle when it should have been. In particular the diatoms do relatively well in the oligotrophic gyres compared to misc-Phytoplankton because they have a nitrate half-saturation constant that is not very different (in absolute terms) from that of the misc-Phytoplankton and the erroneously-high silicate concentration does not limit their growth; in the real ocean they would be strongly silicate-limited in these areas and their large cell-size would mean they were at a competitive disadvantage compared to other phytoplankton. Similarly the surface iron concentration was higher than observed in many parts of the ocean and so did not limit the production at times and places when it should have. These factors mean that the ability of the model to represent two different phytoplankton has not been explored as well as was intended. Figure 7 shows the amplitude and the phase of the seasonal cycle of the total surface biomass, and also a Hovmöller diagram of the zonal-mean

seasonal cycle. As in the case of the total chlorophyll, the amplitude and phase have been obtained by fitting a sine-curve to the monthly mean values at each point. The amplitude of the cycle is very similar to the mean biomass, except in the equatorial latitudes (and especially in the Equatorial Pacific) where the amplitude is significantly less; this implies that in those latitudes there is significant biomass all year round, whereas in the high latitudes where the cycle amplitude and the mean are similar the biomass drops to near-zero for at least some of the year. The phase of the seasonal cycle of surface biomass (time of year of maximum) of the seasonal cycle of surface biomass is very similar to that of total surface chlorophyll, as is to be expected. The Hovmöller diagram clearly shows the pole-ward progression of the high latitude blooms.

### 4.1.3 Primary Production

The vertically-integrated global total primary production during the years 2010-2019 in the model is 35.175 Pg C / yr; of this 19.791 Pg C / yr (56.3%) is due to the diatoms and 15.384 PgC / yr is due to the misc-Phyto. The total is slightly below the generally-quoted range of global primary production, 40-60 Pg C / yr (e.g. Carr et al. 2006). However that total includes the high-production areas along the coasts and in shelf-seas, which the coarse physical resolution and the structure of the model do not allow to be realistically represented: there are no sediments, no tidal mixing, no riverine supply of nutrients or run-off from land and the circulation over the shelf (where that exists) is not accurate. Figure 8 shows the total primary production (in $gC\,m^{-3}\,d^{-1}$). The geographical pattern (of the decadal mean, upper left panel) is very similar to that of the total phytoplankton biomass in Figure 6, as expected. The Hovmöller diagram of the seasonal cycle of the zonal mean (upper right) is also very similar to that in Figure 7. The geographical patterns of the amplitude (lower left panel) and the phase (lower right) of the seasonal cycle (determined, as before, by the best-fitting sine-curve, with only points satisfying the variance condition shown) have many similarities with the corresponding plots for total biomass, though relative to its mean the amplitude of the production cycle in the Sub-Polar North Atlantic is greater than that of the biomass. The pole-ward progression of the peak of the production can clealy be seen in the plot of the phase.

### 4.1.4 Export flux

The export flux of particulate organic matter at 100m in the model during the 2010s decade is $5.58 \pm 0.11$ Pg C / yr, of which 4.95 Pg C / yr is due to sinking detritus and 0.63 Pg C / yr due to sinking living Diatoms. This gives an export ratio of 0.16. The flux figure is at the lower end of the range found in other studies: Bopp et al. (2013) find a range of 4.9 to 8.1 Pg C / yr from a range of CMIP5 models (including HadGEM2-ES), while Siegel et al. (2014) find a slightly wider range of 4.0 to 9.1 Pg C / yr using satellite-driven biogeochemical models (and this latter study considers export out of the euphotic zone, rather than through an 100m depth-horizon). Figure 2 of the review of Boyd et al. (2019) indicates a range of 4 to 9 Pg C / yr for the sinking flux process as represented in this model, but indicates between 1 and 7 Pg C / yr total additional export due to other "particle injection processes", of which only the weakest two are represented explicitly in HadGEM2-ES (and not included in the export flux reported here). The calcium carbonate export is 0.26 Pg C / yr, giving an effective rain-ratio of 0.053. This is equal to the formation of calcium carbonate, since in the model none is re-dissolved above the lysocline.

### 4.1.5 DIC

Figure 9 compares the model's surface DIC (means over the years 2010-2019, in the upper panel, and 1990-1999, in the middle panel) with that from the GLODAPv2 gridded field (lower panel). The data from the second release of the GLODAP project (downloaded from https:/www.nodc.noaa.gov/ocads/oceans/GLODAPv2/) have been re-gridded to the HadGEM2-ES ocean grid, and converted from Mol C $kg^{-1}$ to mMol C $m^{-3}$ using a mean surface water density of 1025 kg $m^{-3}$. The global mean surface values are 2068 mMol C $m^{-3}$ for the model in the years 2010-2019 (and 2054 mMol C $m^{-3}$ averaged over the years 1990-1999), while the data (referenced to the year 2000) have a global average of 2066 mMol C $m^{-3}$. Both these quantities, of course, include anthropogenic $CO_2$ present in the surface waters. The geographical pattern can be seen to be very similar, with the only area showing significant disagreement being the Atlantic Ocean basin, and in particular the northern-hemisphere sub-tropical and sub-polar gyres therein, where the surface concentration in the model is significantly higher. There has been a substantial increase in the model's surface concentration in that basin between the 1990s and the 2010s, and the agreement between model and data is noticeably better for the earlier date (which is closer to the data's reference date).

Figure 10 compares meridional sections of the model's DIC concentration to the gridded GLODAPv2 field in the Atlantic Ocean (upper panels; along 330°) and in the Pacific Ocean (lower panels; along 190°). In the Atlantic section the model underestimates the concentration in the Southern Ocean below about 150m depth (the surface values there are comparable, so the gradient in the upper 200m is too weak in the model) and in the Antarctic Intermediate Water (AAIW) and in the bottom water (below 4000m). These last two errors will be related to the underestimation of the deep Southern Ocean concentration (since that is a source for the AAIW and the bottom water) but the physical model also under-produces AAIW and does not transport what it does produce far enough north. Outside of those regions however the model's representation is good. In the Pacific section the model underestimates the concentration throughout the section below 1000m, and up to depths as shallow as 200m in the Southern Ocean, under the Equator and around 45°N (all sites where there is significant upwards vertical transport). In particular, the model substantially underestimates the meridional gradient between 1000m and 3000m depth: the increase from south to north is up to 150 mMol C $m^{-3}$ in the gridded data, but only around 50 mMol C $m^{-3}$ in the model. This reduced gradient is also seen in Total Alkalinity and (to a reduced extent) in dissolved Nitrate, so the physical deep circulation is likely to be at least a partial cause.

Figure 11 shows the amplitude and the phase (time of year of the maximum) of the seasonal cycle of surface DIC. This is determined by a number of factors: vertical mixing, vertical transport, air-sea $CO_2$ flux and biological uptake and release. All of these factors vary seasonally and their relative contributions are different from place to place, and so the phase of the cycle (and how well a sine-curve represents it) varies more with location than many other cycles. In the sub-polar North Atlantic, for example, relatively high DIC water is mixed (by convective and by wind-induced mixing) from depth to the surface during the winter, and the low surface temperature keeps the ocean $pCO_2$ lower than the atmosphere, so there is ingassing of $CO_2$. As the season passes to spring the increased solar irradiance warms the surface water, vertical mixing is suppressed, and there is net uptake of DIC by the phytoplankton for growth. Those factors tend to cause a reduction in surface DIC concentration and so reduce the $pCO_2$, but at the same time the increased temperature will increase it (for a given DIC concentration); which is the

dominant effect, and so whether the air-sea $CO_2$ flux moves towards greater ingassing or greater outgassing, depends on the local conditions. The phase varies by up to 6 months across the North Atlantic at a latitude of 50°, while at a similar latitude across the Pacific the phase is almost constant.

### 4.1.6 Total Alkalinity

Figure 12 compares the model's surface Total Alkalinity (means over the years 2010-2019, in the upper panel, and 1990-1999 in the middle panel) with that from GLODAPv2 gridded field (lower panel; Lauvset et al., 2016, and Key et al., 2015). As with the corresponding DIC plot (Figure 9) the data from the GLODAPv2 project have been re-gridded to the model grid and converted using a mean water density of 1025 kg m$^{-3}$ to the model units, in this case mEq m$^{-3}$. The model's global surface mean values are 2343 mEq m$^{-3}$ in the 1990s and 2340 mEq m$^{-3}$ in the 2010s, while the global surface average of the gridded data is 2352 mEq m$^{-3}$; the approximately 10 mEq m$^{-3}$ deficit in the model compared to the data is consistent with the 12 mMol C m$^{-3}$ deficit in 1990s surface DIC compared to the DIC surface data (referenced to the year 2000). The model's Total Alkalinity is high in the sub-tropical gyres, especially in the Atlantic Ocean, and this pattern is also seen in the GLODAPv2 gridded field. The correlation between the 2010s model surface field and the (re-gridded) data is 0.78 and the ratio of the standard deviations is 1.29, as shown on Figure 18; these figures are consistent with Figure 12, where the highest concentrations in the Atlantic are higher than the corresponding highs in the data. Compared to DIC, the correlation is lower, and the ratio is higher.

The biological processes that affect the model's Total Alkalinity are shown in Equation 4 to be solid calcium carbonate formation and dissolution and processes linked to the uptake of dissolved nitrate (inorganic nitrogen). At the ocean surface these processes are in opposition (net uptake of DIN and formation of solid carbonate) but, given the low value (0.0195 mMol $CaCO_3$ (mMol C)$^{-1}$, corresponding to a rain-ratio of about 0.053) chosen for the molar ratio of carbonate formation to organic production for misc-Phytoplankton and the proportion of primary production due to that phytoplankton type, the effect of the DIN-uptake (organic production) dominates. In mid-depths of the model, for example between 500m and 1500m, there is no carbonate formation or dissolution and no organic growth but there is significant remineralisation of sinking detritus which releases nitrate into the water and, since the model links that with an uptake of hydroxyl ions, reduces the Total Alkalinity in that depth range. Conversely, in depths below the model lysocline (fixed at 2113m) there is no organic growth or carbonate formation and what little remineralisation does occur is greatly outweighed by carbonate dissolution, which increases the local alkalinity in the bottom waters. Therefore the general biological effect on Total Alkalinity should be an increase in deep water and at the surface but a decrease in mid-water. Figure 13 compares meridional sections of the model's Total Alkalinity to the gridded GLODAPv2 field in the Atlantic Ocean (upper panels; along 330°) and in the Pacific Ocean (lower panels; along 190°). In the Atlantic it is confirmed that the model overestimates the concentration in the top 1000m between 40°S and 40°N, expecially north of the equator, and underestimates the concentration in the Antarctic Bottom Water (AABW). In the Pacific there is an underestimate in the upper water-column under the equator in the model, and again an underestimate in the AABW, but also in the waters above that, and especially in the deep North Pacific where the model has a much lower inventory of Total Alkalinity than is observed. The underestimates at depth in both basins is due to the relatively low value given to

the (effective) rain-ratio, and occurs despite the crude representation of the sinking particulate carbonate flux placing all the carbonate dissolution (and so also all the return of alkalinity to the water column) in the layers below 2000m depth, whereas in the real world a significant proportion occurs in the upper levels.

Figure 14 shows the amplitude (upper panel) and phase (time of year of maximum concentration; lower panel) of the best-fitting sine-curve through the surface seasonal cycle at each point. As in other plots of this type, values are only shown if the variance of the residual (after the sine-curve has been subtracted) is less than half that of the original seasonal cycle; for model Total Alkalinity this test is passed at most points. The corresponding GLODAPv2 gridded field only provides an annual mean, not a seasonal cycle, so no comparison to data is possible. Comparing this figure to the corresponding one for the DIC seasonal cycle (Figure 11) shows that, while the amplitudes are similar in the tropics, in the Sub-Polar North Pacific and North Atlantic, and in the Southern Ocean, that of Total Alkalinity is noticeably smaller than that of DIC. This relates to the counter-acting effects of organic and inorganic production on alkalinity in the surface ocean, as discussed above, which contrast with their re-inforcing effects on DIC. In terms of the phase, the peak of Total Alkalinity occurs two or three months later than that of DIC in the high-latitude regions.

### 4.1.7  pCO$_2$

Figure 15 compares the model surface ocean pCO$_2$ field, meaned over the period 1990 to 2009 (upper panel), with the Takahashi gridded annual mean surface pCO$_2$ field referenced to the year 2000 (lower panel). The fields have global means that show a consistent rise from the preindustrial value, to 364.2 ppmv in the model and 357.9 ppmv in the gridded data product; in the year 2000 the atmospheric partial pressure was specified to be 368.8 ppmv. However, there are significant differences in the geographical distribution. The data show a narrow ridge of high pCO$_2$ in the Eastern Equatorial Pacific, but the corresponding high-pCO$_2$ water in the model is more widespread, does not reach the same extremes as the data, and actually shows a local minimum where the data-product values are highest. This is due to the much higher chlorophyll (and therefore also higher primary production) in that area dragging down the surface DIC. In the Atlantic basin there is a significantly greater area with very high pCO$_2$ in the model than in the gridded field, especially in the northern and southern sub-tropical gyres. Finally in the Southern Ocean there is a zonal band of high pCO$_2$ water in the model just south of 45°S while the gridded fields only shows some elevated values close to the Antarctic continent; the 45°S band is driven by upwelling of carbon-rich water in the model, which overcomes the pCO$_2$-lowering effect of the over-estimated primary production there.

Figure 16 compares the amplitude (left-hand panels) and the phase (right-hand panels) of the seasonal cycle in the model (mean of years 1990 to 2009; upper panels) and the data-product (referenced to year 2000; lower panels). As in other plots of this type, the amplitude and phase are only shown at points where the variance of the residual is less than half that of the original seasonal cycle. It can be seen that the model produces a substantially greater seasonal cycle than is observed in the data, though some of the patterns are similar: the data-product shows a relatively large amplitude of the cycle in the northern sub-tropical and sub-polar Pacific, where the model does as well, and in the areas closest to the Antarctic continent. However the strong seasonal cycle seen in the model in the North Atlantic is largely absent from the data, as is the band covering the southern sub-tropical gyres in all three ocean basins. There is good agreement between the model and the data-product for the

phase of the seasonal cycle at points in the tropics and sub-tropics, but there are substantial differences at higher latitudes: in the Southern Ocean the model phase peaks in May to July, but in the data-product it mainly peaks in August to November, while in the North Atlantic the model phase peaks in August and September but the data-product peaks in January and February. In the latter case the model underestimates the primary production and so also $CO_2$ uptake in spring and summer; therefore when the surface waters warm the $pCO_2$ rises above its winter value (when there was more DIC but a lower temperature) and the annual maximum occurs in summer rather than in winter, as observed.

Figure 17 shows the fraction of the seasonal cycle of $pCO_2$ that is not driven by the temperature (and salinity) seasonal cycles. It has been calculated using a mean seasonal cycles of sea-surface temperature, sea-surface salinity, surface DIC and surface Total Alkalinity from the decade of the pre-industrial control run of HadGEM2-ES corresponding to 2010-2019. The seasonal cycle of $pCO_2$ was calculated first using all four seasonal cycles, and then using the cycles of DIC and Total Alkalinity but annual mean values of SST and SSS. The first run includes the effects of the seasonal variations of temperature (and salinity) as well as the biological uptake and respiration cycles, some effect of the seasonal uptake of $CO_2$ from the atmosphere and the seasonal variation of mixing DIC and Total Alkalinity from the sub-surface ocean; the second run does not include the seasonal variations of SST and SSS, but does include the other cycles. The best-fitting sine-curve was found in each case, and the ratio of the amplitudes (second run divided by first run) calculated. Where the effect of SST (and also SSS) dominates, the value of the ratio will be less than 0.5, while ratios greater than 0.5 indicate that the effects of biological uptake and respiration (and the mixing) dominate. Where the ratio is greater than 1.0, the two effects are of comparable size but opposed. From the Figure it can be seen that the SST cycle is dominant in the tropics and sub-tropics, and also in the North Atlantic, while biological seasonality plays an important role in the sub-polar North Pacific and in the Southern Ocean. The dominance of the SST in the North Atlantic is due to the model having too-low primary production and carbon drawdown there.

The Taylor diagram in Figure 18 shows (blue symbols) the correlation and ratio of standard deviations of the $pCO_2$ in the model and the Takahashi data-product (alongside similar for surface DIC and Total Alkalinity, discussed in earlier sections). The annual means, calculated using all open-ocean points and denoted by the blue square, have a correlation of 0.53 and a ratio of standard deviations of 1.12. The remaining blue symbols relate to the mean seasonal cycle, and have been calculated only at open-ocean points where a sine-curve was a valid fit (in terms of reducing the variance of the residual, as discussed) in both the model and the data (of course, the best-fitting curves will normally be different in model and data). The correlation and the ratio of standard deviation are respectively 0.51 and 1.31 for the mid-point of the fitted sine-curve (circle), 0.49 and 1.42 for the amplitude (upward-pointing triangle) and 0.51 and 0.89 for the phase. The low correlations are a result of the poor match in the higher latitudes mentioned above.

### 4.1.8 Air-Sea $CO_2$ flux

Figure 19 shows the air-to-sea flux of $CO_2$ (i.e. positive for net flux into the ocean) meaned over the decade 2010 to 2019. The upper panel shows the total flux (i.e. the natural cycle of $CO_2$ and the anthropogenic perturbation combined), while the lower panel shows just the anthropogenic perturbation. This perturbation has been calculated by subtracting the mean of the air-to-sea flux in the piControl run from the total flux at each point. The annual mean $CO_2$ flux in the piControl simulation

averaged just 0.0237 Pg C yr$^{-1}$ over the period 1860 to 2099, with a standard deviation of 0.1036 Pg C yr$^{-1}$ and no significant trend; this average is clearly well within the 0.2 Pg C yr$^{-1}$ criterion for a successful spin-up. The annual mean $CO_2$ flux in the RCP8.5 simulation was 2.529 Pg C yr$^{-1}$ averaged over the years 2010 to 2019, and was 2.117 and 1.960 Pg C yr$^{-1}$ in the 2000s and 1990s respectively. These figures are in good agreement with the figures quoted by the IPCC 5th Assessment Report (IPCC, 2013) of $2.3 \pm 0.7$ and $2.2 \pm 0.7$ Pg C yr$^{-1}$ for the 2000s and 1990s respectively. Given the method for calculating the anthropogenic perturbation to the flux there is no way to distinguish between the two separate components to it: namely the (i) ingassing of anthropogenically-emitted $CO_2$ (mainly fossil fuel combustion) and (ii) changes to the natural cycle caused by climate change (itself mainly due to increasing atmospheric $CO_2$). Whereas the first component would be expected to give a net flux into the ocean the second can be either into or out of the ocean, and careful examination of the lower panel reveals a few areas in the sub-tropical Pacific where the perturbation flux is negative (out of the ocean). But predominantly the perturbation flux is into the ocean, and co-incident with some of the largest fluxes in the total flux (and also the natural cycle flux): the sub-polar North Atlantic and the adjacent sector of the Arctic, the area where the Kuroshio current becomes zonal and the seas surrounding the Antarctic continent. It is notable that although (on a per unit area basis) the northern sub-polar Atlantic dominates the total flux it is only comparable with the Southern Ocean in terms of the anthropogenic perturbation. Figure 20 shows Hovm$\ddot{o}$ller plots of the seasonal cycle of the total flux of $CO_2$, zonally meaned globally and separately for each of the three ocean basins: Atlantic, Indian and Pacific. The Atlantic has the largest per unit area fluxes, and these occur in winter and early spring months when low temperatures reduce the surface ocean pCO$_2$ and deep convective mixing carries ingassed $CO_2$ away from the atmosphere. However, that pattern is reversed in the Pacific north of 45°N and in the most southerly latitudes of all three basins, where the most intense uptake is in the local summer months. This is due to strong biological activity taking DIC out of the water and lowering the pCO$_2$ despite the warmer summer temperatures acting to raise it. The model has only weak primary production in the North Atlantic so that effect is reduced there, whereas the winter subduction is particularly strong, and so winter uptake dominates in that region in this model. Figure 21 shows the seasonal cycle of the anthropogenic perturbation flux in a similar way. Similar patterns are observed, but the North Atlantic is less dominant in winter.

### 4.1.9  Nutrients: nitrate, silicate, iron

Figure 22 compares the model surface nitrate field (mean over the years 2010 to 2019) with the corresponding field from Volume 4 of the World Ocean Atlas 2013 (hereafter WOA13V4; Garcia et al. 2014). Strictly the model nitrate field represents the sum of all dissolved inorganic nitrogen compounds (nitrate, nitrite and ammonium) but in many circumstances the first of those is dominant. Nitrogen is the "currency" of the model ecosystem and the main limiting nutrient. To first order the geographical distributions compare fairly well, with high concentrations in the Southern Ocean, the Eastern Equatorial Pacific, and the northern sub-polar regions of the Pacific and Atlantic Oceans. The gridded data from WOA13V4 is slightly higher than the model in the Eastern Equatorial Pacific and in the sub-polar North Atlantic; in the former region this is due to higher production in the model than is observed in the real ocean taking up more nitrate for phytoplankton growth, while in the latter the lower-than-observed production is due to low nitrate concentrations at the start of the growing season, in turn due to a tendency of the model to lose nutrient from that region through the deep circulation. It can also be seen that in the model the

nitrate concentration has slipped to be slightly negative in some sub-tropical regions, particularly the centres of the gyres; in such circumstances the ecosystem model (but not the advection or mixing processes of the physical model) treats the value as zero. As shown in Figure 29 (solid green square), the correlation of the decadal mean of the model and the gridded data is 0.96, while the ratio of the standard deviations is 1.01; note that to make these comparisons the gridded data was re-gridded to the model grid. Figure 23 compares full-depth meridional sections of the nitrate concentration in the Pacific and the Atlantic Oceans (at 190° and 330° respectively) from the model and WOA13V4; the upper $500m$ are shown with an expanded vertical scale. In the Atlantic sections, the model fails to simulate the northwards intrusion of nitrate-rich water at around 1000m depth, and its subsequent upwelling under the tropics; this is due to weak formation of Antarctic Intermediate Water, a know issue of the physical model. Also, in the model the high northern latitudes the nitrate concentration is much lower than the data at all depths, and the deficit is clearly carried with the North Atlantic Deep Water at depth to tropical and even high southern latitudes. This inability to retain high nutrent levels in the sub-polar Atlantic has been seen in previous versions of the model (both physical and biogeochemical), and may be partially due to the absence of riverine inputs of nutrients into the Arctic Ocean and the high northern latitudes. In the Pacific section the comparison is better, though the model lacks the very high nitrate concentrations revealed in the WOA13V4 data at around 1000m depth north of 30°N. Figure 29 (open green square) shows that at around 1050m depth globally the correlation of the model and data is 0.89 and the ratio of the standard deviations is 1.30 (i.e. the model varies more). Figure 24 compares the amplitude and phase of the seasonal cycle in the model and WOA13V4 nitrate fields, determined by the best-fitting sine-curve as described in previous sections; as in earlier figures of this type, the value of the amplitude and phase is only shown where the variance of the residual seasonal cycle is less than half that of the original cycle, determined separately for model and data fields. The seasonal cycle will be determined by a number of factors, including vertical advection and mixing and the uptake and remineralisation of nitrate by the ecosystem, all of which can vary through the year. The most obvious feature is that, while the seasonal cycle at most locations in the model (at both high and low latitudes) is well-represented by a sine-curve, there are far fewer locations in the gridded data where this is so, and they are mainly at high latitudes (and particularly in the Northern Hemisphere). Where comparison can be made the model amplitude field is similar in pattern and scale to the mean concentration as presented in Figure 22, but the WOA13V4 field shows some interesting differences from its concentration field: the scale of the seasonal cycle is much lower in the Southern Ocean (0.5 to 5 mMol N m$^{-3}$ amplitude compared to greater than 20 mMol N m$^{-3}$ mean, while the model has an amplitude of 5 to 15 mMol N m$^{-3}$ with a similar mean). This suggests that the model is not fully limiting the phytoplankton growth in that region: this limitation will not be from low nitrate levels as they are always higher than needed for growth, but could be from other nutrients (probably dissolved iron; see Martin et al. 1992) or from light limitation. In terms of the phase of the cycle, the model shows much greater consistency than WOA13V4: almost all the areas poleward of 30° in the model show the highest concentration at the end of local winter, but the data product shows much more variability in the Southern Hemisphere (both models show variability in the tropics). In Figure 29, the Taylor diagram shows (in green) the correlations and ratios of the standard deviations of the mid-points (circle), amplitudes (upward-pointing triangle) and phases (downward-pointing triangle) of the best-fitting sine-curves to the seasonal cycles of model and data; the data has been re-gridded to the model's grid, and

only points where the sine-curve is an acceptable fit are considered in the analysis. The mid-point, amplitude and phase have correlations of 0.94, 0.49 and 0.63 respectively, while the ratios of the standard deviations are 1.00, 1.54 and 0.90.

Figure 25 compares the model silicate field (i.e. dissolved silicic acid; meaned over the years 2010-2019) with the corresponding gridded field from WOA13V4. Unfortunately the dissolution/remineralisation parameter for sinking detrital silicate $\Pi_{rmn}^{DtSi}$ was given a value that was too high; this meant that too much returned to dissolved silicate in the upper water-column, leaving too little in the lower water-column. Over the period of the simulations therefore the surface concentration of dissolved silicate continually increased (while that in the deep ocean continually decreased) leading to high surface values everywhere. This has the effect that, while it would normally be expected that silicate values will be low enough to limit the growth of diatoms (which require it to form their shells) in some areas all the time and in others at certain times of the seasonal cycle (after a bloom, for instance), in these model simulations silicate is never a limiting nutrient for diatoms, which are therefore only limited by nitrate, iron and light-availability. Atlantic and Pacific Ocean meridional sections (at 330° and 190° respectively) in Figure 26 show how the implementation error has raised the concentration throughout the upper water-column in both oceans. Additionally, the Pacific section shows that the strong build-up of silicate in North Pacific below 1000m (and especially around 2000m depth) that is seen in WOA13V4 is not simulated to the same extent in the model. The Taylor diagram in Figure 29 shows (red symbols) the correlations and ratios of the standard deviations of the surface annual mean concentration (filled square, correlation 0.69, ratio 0.64), annual mean concentration at 1050m (open square, 0.78, 0.62), and the mid-point (circle, 0.64, 0.76), amplitude (upward-pointing triangle, 0.70, 0.36) and phase (downward-pointing triangle, 0.64, 0.89) of the best-fitting sine curve to the seasonal cycle. The data has been re-gridded to the model grid for this calculation and in the case of the best-fitting curve only those points with a good fit are considered. Although the silicate is too high across the ocean (as discussed) its seasonal cycle is strongly influenced by nitrogen- and light-limited diatom growth, so there is value in comparing it to the observations.

Figure 27 presents the surface dissolved iron concentration in the model (upper panel) and the amplitude of the sine-curve that best fits the seasonal cycle (lower panel; as in previous plots of this type values are only shown if the variance of the residual is less than half that of the original cycle). In each case the period considered covers the years 2010 to 2019. Note that different scales are used for the two plots: contour intervals for the lower plot are one tenth of those for the upper. In the upper plot, the effects of high inputs of iron-rich dust can be seen in the northern sub-tropical Atlantic (from the Sahara desert), in the northern Indian Ocean (from the Arabian Peninsula and the Indian Sub-continent) and east of Australia; most of the iron that is supplied to the surface layers of the Southern Ocean is upwelled or mixed from below. Given that the half-saturation concentration for iron limitation in both types of phytoplankton was set at 0.2 $\mu$ Mol Fe m$^{-3}$ it can be seen that in the model there are few areas of the ocean where the decadal mean concentration of dissolved iron limits the growth of either misc-Phyto or diatoms. However, there are significant areas, including the Southern Ocean, the Eastern Equatorial Pacific and the North Pacific, where iron is limiting at certain times of the seasonal cycle, though even this is different from the observed situation where, for instance, iron is limiting in the Southern Ocean at all times of the seasonal cycle. Figure 28 compares the model to observations along a roughly-meridional section in the western Atlantic Ocean. The data were collected for the eGEOTRACES GA02 transect (Schlitzer et al., 2018), and the model section follows the exact same path. The units in the upper panel (model)

are $\mu$Mol Fe m$^{-3}$ while those in the lower panel (data; from http://www.egeotraces.org/sections/GA02_Fe_D_CONC.html) are nMol Fe kg$^{-1}$; since the model's sea-water density is set at 1025 kg/m$^{-3}$ (other than for calculations of density and pressure gradients, of course) these units are roughly comparable (2.5% different, and it can be seen that the differences between the fields are considerably larger than that!). The model shows high concentrations in the surface water (due to strong atmospheric inputs) while the observations show low concentrations there. In contrast, the eGEOTRACES section (which mainly follows the shelf-break) has a number of high-concentration patches well below the surface, which presumably are caused by dissolved iron being released from shelf-edge and basin margin sediments (which the model lacks). Analysis of the long-term behaviour of the dissolved iron field in the piControl simulation shows a drift to higher concentrations at all depths including the surface levels, due to parameter values in the iron sub-model not being optimal and this field not being fully spun-up. There is still much uncertainty in the quantitative understanding of the processes affecting iron in the ocean, especially those relating to organic ligands, and the representation used here can surely be improved.

### 4.1.10 Oxygen and Apparent Oxygen Utilisation

Dissolved oxygen is present in the model (Equation 5) as a diagnostic tracer. It has particular value as a diagnostic of the respiration of organic matter at depth in the water-column, but also allows for the simulation of oxygen-minimum zones, and their evolution under climate change. The surface oxygen concentration is not shown, since it is dominated by the temperature-dependent physical solubility process, but Figure 30 compares the Atlantic and Pacific Ocean meridional sections (at 330° and 190° respectively). In both sections the overall patterns are very similar, with similar concentrations persisting in the model's plume of North Atlantic Deep Water as are seen in the data (the gridded field from WOA13V4). The major difference is that the model's oxygen miniumum concentrations are not as low as in the data: in the Atlantic around 130 mMol O$_2$ m$^{-3}$ compared to around 70 mMol O$_2$ m$^{-3}$, and in the Pacific around 70 mMol O$_2$ m$^{-3}$ below the tropics compared to as low as 20 mMol O$_2$ m$^{-3}$ below the sub-tropics in the data. This discrepancy could be due to the model having too little remineralisation in the relevant depth-ranges, or having too much mixing (of higher-oxygen water into the minimum zone). To assess the extent, geographically and vertically, of the low-oxygen regions, Figure 31 compares the depth-range of the water-column with dissolved oxygen concentrations of less than 50 mMol O$_2$ m$^{-3}$ (upper panels) and 100 mMol O$_2$ m$^{-3}$ (lower panels) in the model (left panels) and WOA13V4 (right panels). The model almost-exclusively produces such zones in the Equatorial Pacific (and particularly the eastern part of that), whereas WOA13V4 additionally shows oxygen-depleted water in the North Pacific and in the northern Indian Ocean. In the Equatorial Pacific, however, the thicknesses of the low-oxygen zones are comparable in model and data. In the Taylor diagram of Figure 29 the blue symbols refer to oxygen variables: the filled square refers to the annual-mean surface concentration (correlation 0.99, standard deviation ratio 1.01), the open square to the mean concentration at 1050m (0.86, 0.96), the filled star to the thickness of the below-50 mMol O$_2$ m$^{-3}$ zone (0.30, 0.61) and the open star to the thickness of the below-100 mMol O$_2$ m$^{-3}$ zone (0.57, 0.87).

The Apparent Oxygen Utilisation (AOU; units mMol O$_2$ m$^{-3}$) is the difference between the oxygen solubility and the actual oxygen concentration in a water sample, and is a measure of the accumulated biological activity in that sample since it was last at the surface (and in contact with the atmosphere). Values tend to be low (and negative) at the surface where oxygen-producing

photosynthesis dominates but significantly higher in deeper, poorly-ventilated water in which there has been much respiration. Figure 32 compares the geographical distribution of AOU at around 1050m depth in the model (upper panel) and in the gridded data (WOA13V4). The model matched the data over much of the ocean, but significantly underestimates the value in the Indian Ocean and in the North Pacific; in the latter the model shows the highest values in the mid-latitudes (and particularly on the eastern side of the basin) while the observed AOU increases northwards from the Equator and peaks in the Northeast Pacific. The failure of the model to simulate extreme values at 1000m depth under the North Pacific has already been seen in the DIC, Total Alkalinity and Nitrate sections. Figure 33 compares meridional sections of the model's AOU to the gridded GLODAPv2 field in the Atlantic Ocean (upper panels; along 330°) and in the Pacific Ocean (lower panels; along 190°). The simulation of the Atlantic section is mostly excellent, except for a slight underestimate at about 500m depth around 20°N, but the model misses the high values under the North Pacific as noted above; in these plots it can be seen that the error extends from 1000m to 3000m depth. In the Taylor diagram of Figure 29 the purple symbols refer to AOU: the filled square refers to the surface value (correlation 0.57, standard deviation ratio 0.45) and the open square to the value at 1050m depth (0.84, 0.89) the mis-match in the latter being mainly due to the failure to simulate the high North Pacific and Indian Ocean values.

## 4.2 Response to climate change

This section presents key results of the response of the model to climate change in the RCP8.5 scenario simulation, in particular between the decade 2010-2019 ("the 2010s") and the decade 2090-2099 ("the 2090s"), and also through the historical simulation from which the future run is initialised.

Figure 34 shows the global zonal mean surface nitrate concentration through the historical and RCP8.5 scenario period (years 1860 to 2099), allowing trends to be identified. The corresponding period of the piControl simulation (not shown) has no trend or drift, so the changes with time seen in this plot are all due to climate change. It can be seen that at almost all latitudes the concentration decreases through the 21st century, and that the rate of decrease becomes more marked towards the end of the simulation. This trend can be understood in terms of the vertical supply of nitrate being reduced as the surface ocean is warmed and becomes more stratified. Although phytoplankton growth (and nitrate uptake) is also reduced because of the reduced nutrient availability the net effect is a decrease in the surface nitrate concentration, and this drives many of the changes seen in the model and presented in this section.

Figure 35 presents Hovmöller plots of the total chlorophyll anomaly (a measure of the abundance of both types of phytoplankton) from 1860 to 2099 for the Atlantic basin (upper panel) and the Pacific basin (lower panel). The anomaly has been calculated by subtracting the chlorophyll in the piControl simulation (the mean from 1860 to 2099) from the annual mean chlorophyll in the historical+RCP8.5 simulation. The piControl chlorophyll showed no significant trend or drift. In addition to inter-annual and inter-decadal variability in both basins it can be seen that trends become apparent in the climate change scenario, mainly after the year 2000. In both basins the chlorophyll close to the Antarctic continent increases substantially, as does that in the Atlantic Basin around 45°S. In contrast there is a clear reduction in chlorophyll at the Equator, present in both basins but particularly marked in the Pacific. Between 30 and 60°N there is a smaller reduction in chlorophyll in each basin, while in the Pacific just north of that band there is a marked increase. These trends can be understood as increased stratifica-

tion both reducing the vertical nutrient supply and reducing the depth of the mixed layer during the growing season (and so improving the available light for phytoplankton in the surface layer): in the tropics the former dominates so production (and chlorophyll) is reduced, but at high latitudes the latter is more important and leads to higher production. In addition, around Antarctica warming seas mean that ice-cover is reduced, allowing more primary production. Similar results have been reported previously in future scenario simulations (e.g. Bopp et al., 2001).

Figure 36 shows how the seasonal cycle of total chlorophyll changes from the 2010s to the 2090s in the Atlantic basin (upper panel) and the Pacific (lower panel). In both basins the reduction in chlorophyll at Equatorial latitudes is seen to be present throughout the year, though it is most intense in the Atlantic between July and November and in the Pacific during March and April. In the Southern Ocean sectors of each basin the change is an increase between October and February in the most southerly latitudes, and no change in other months; however slightly further north, around 45°S, there is an increase during those austral summer months in the Atlantic but a decrease in the Pacific. In the northern hemisphere, poleward of 40°N, the Atlantic sees a reduction between April and September but the Pacific sees a strong increase in the Spring (March to May) followed by an equally-strong reduction in the summer (June to August). This "dipole" change in the North Pacific is a signature of the seasonal cycle shifting forward by several months, in response to changing physical conditions.

Figure 37 shows the difference, between the 2090s and the 2010s, in the mean total primary production (upper panel) and in the mean seasonal cycle of that quantity (lower panel). The mean field displays strong reductions in the Equatorial Pacific and Atlantic Oceans, because of reduced nitrate availability, and also in the sub-polar North Atlantic and the eastern sub-polar North Pacific. In contrast the Southern Ocean close to the Antarctic continent shows strong increases in production, for the reasons outlined above: shallower surface mixed layers allowing the phytoplankton to remain for longer in well-lit depths near the surface, and reduced seasonal ice-cover allowing more time for growth. The seasonal cycle shows a pattern of changes that is very similar to the change in the mean, except in the Eastern Equatorial Pacific where the amplitude of the cycle is little changed but the mean has been substantially reduced; note that in the 2010s the seasonal cycle was also relatively small, while the mean was high in that area. Figure 38 shows the change through time of the total production, separated into the Atlantic and Pacific basins (upper and lower panels respectively). There are similarities as well as some differences between basins: both show a poleward spread (and consequently a slight overall increase) of production in the Southern Ocean, and both show a reduction in equatorial production through the 21st Century; however while the Northern Hemisphere Sub-Polar production shows a marked decrease in the Atlantic basin throughout the last 100 years no such change is seen in the Pacific (and there are hints of a slight increase. The global annual mean total primary production in the 2090s is 30.494 Pg C yr$^{-1}$ (compared to 35.175 Pg C yr$^{-1}$ in the 2010s, so a 13.3% reduction), which is apportioned 17.227 Pg C yr$^{-1}$ (c.f. 19.791; -13.0%) to the diatoms and 13.267 Pg C yr$^{-1}$ (c.f. 15.384; -13.7%) to the misc-Phyto; therefore there is only a very small shift towards increased dominance by the diatoms. Bopp et al. (2005) saw a decrease in the prevalence of diatoms under a warming scenario, and the opposite result obtained in this study is due to the lack of silicate limitation which means that the diatoms are not prevented from utilising their higher growth rate; in fact because the upwards drift in surface silicate concentrations is ongoing throughout the period of the future scenario the silicate is less limiting in the future, rather than more limiting as would be expected with increased stratification.

Figure 39 shows how the surface ocean $pCO_2$ varies through the historical and RCP8.5 scenario. The top panel shows the change with time of the global zonal mean $pCO_2$ anomaly (i.e. the difference between the scenario and the piControl). As expected, the surface $pCO_2$ increases smoothly with time, increasing its rate in keeping with the prescribed atmospheric concentration. Most of the rise therefore occurs during the 21st century. It is notable that all latitudes increase at a substantially similar rate. The middle panel shows the geographical distribution of the anomaly averaged over the period 2090-2099. Here the colour-scale has been set to show up what differences there are: the rise is greatest in the arctic and in the sub-tropical gyres, and in the northern sub-polar Atlantic. The bottom panel shows that the distribution of the anomaly of the seasonal cycle amplitude is very similar to that of the mean concentration, except around the Antarctic continent. The phase of the seasonal cycle in the 2090s (not shown) has changed little from that in the 2010s.

Finally, the air-to-sea flux of $CO_2$ is considered. Figure 40 shows the global total flux through the historical+RCP8.5 simulation from 1860 to 2099 (the piControl over that period showed no trend). It is clear that the flux increases with time; this is to be expected, since the atmospheric $pCO_2$ was increasing monotonically through the simulation. By the 2090s the net flux is 4.8 Pg C yr$^{-1}$.

Figure 41 shows the evolution of the zonal mean flux globally (top panel) and in the Atlantic and Pacific basins separately (middle and bottom panels respectively). It can be seen that, while the global total flux continued to increase throughout the period, there were certain latitudes in some basins where the flux peaked and then began to decline - despite the atmospheric $CO_2$ concentration continuing to increase. This effect is particularly noticeable in the Atlantic between 50 and 60 °N, with the peak uptake occurring between 1980 and 2030 before an accelerating decrease. Such a "peak and decline" feature is seen in many CMIP5 model simulations as well as in other future simulations, and the causes are examined in Halloran et al. (2015). In the Southern Ocean, meanwhile, the uptake shows a monotonic and significant increase, particularly in the second half of the 21st century.

Figure 42 shows the seasonal cycle of the zonally-meaned total flux during the 2090s globally and in each ocean basin separately. It can be compared to Figure 20, which shows the same cycles during the 2010s. It is clear that there has been a substantial shift towards net uptake, particularly where there was substantial uptake already in the 2010s; but there are some areas which were sources at the earlier time that became sinks for atmospheric $CO_2$ at the later time. There are also regions (e.g. the Atlantic around 45°N) which were weak sources in the summer months during the 2010s but which have become strong sources by the 2090s; and this is despite those latitudes being stronger sinks in the winter and spring months at the later time. Overall, therefore, the cycling of $CO_2$ between the ocean and atmosphere seems to have generally intensified. This result is consistent with the conclusions of Hauck and Völker (2015) who argued that, due to a reduction in the Revelle (or buffer) factor of the surface waters the seasonal cycle due to biological growth will become relatively more important.

## 5   Conclusions

The Diat-HadOCC model is a development of the earlier HadOCC model, including separate diatom and misc-Phytoplankton components and representations of the dissolved silicate and iron cycles in the ocean and through the marine ecosystem. The

model forms the ocean biogeochemistry component of the Met Office's coupled Earth System model HadGEM2-ES, and has been used to run a wide-ranging suite of simulations for the CMIP5 experiment. This paper has described the model in detail, presenting the equations and explaining choices made in the parameterisations. The Diat-HadOCC model's performance has been evaluated by comparing a selection of results from the CMIP5 simulations to publicly-available data products such as the World Ocean Atlas 2013 and GLODAPv2. The model results shown (and many more) are freely available from the Earth System Grid website (https://pcmdi.llnl.gov/projects/esgf-llnl/).

The model has been shown to be capable of reproducing to a reasonable extent many of the important features of the marine carbon cycle, including annual mean surface concentrations of dissolved inorganic carbon and total alkalinity and the annual air-sea flux of $CO_2$. However there are also significant differences from the real-world observations in these quantities, both in the surface layer (where the effect on the air-sea $CO_2$ flux is direct) and in the deep and mid-waters (where model errors will take decades to centuries to affect that flux). Some of these differences may be due to errors in the physical ocean model's deep circulation, but some will be due to errors in the ecosystem performance. The climate change response of the marine carbon cycle in the model is also shown to be in accordance with similar modelling studies.

In terms of the ecosystem, the model does less well. The model's total chlorophyll (the sum of diatom chlorophyll and misc-Phytoplankton chlorophyll) is too high in many areas (by a factor of 2 or more), including in the Eastern Equatorial Pacific and the Southern Ocean, while being lower than observed in the oligotrophic gyres. In contrast, the model's primary production (global mean 35.2 Pg C / yr) is slightly below the range estimated from observations, even when the highly-productive coastal regions are ignored (the physical structure of the model means those regions will not be adequately represented). Therefore, the model produces too much chlorophyll that does not do enough. The split between diatoms and misc-Phytoplankton is roughly even, with the former having 55% of the biomass and being responsible for 56% of the primary production. The geographical distributions of the two phytoplankton types are also very similar, and this similarity is roughly maintained even under the RCP8.5 climate change scenario. The reason for the two types being so similar is due to many of their parameters having the same values (an exception is the maximum growth-rate, which is higher for diatoms), and due to the dissolved silicate and dissolved iron fields not being limiting to diatom growth as much as they should be. The dissolved nitrate field is represented fairly well, though its surface concentration is low in the North Atlantic due to circulation issues (and a lack of riverine inputs). The dissolved silicate field, by contrast, suffers from a poor-choice of the detrital silicate dissolution parameter which leads to a drift to excessively-high surface values through the run and so is rarely limiting. Surface concentrations of dissolved iron, which should be limiting in most areas of the ocean for at least some of the year, are also too high because the iron in the particulate biology is remineralised too shallow in the water-column. The iron sub-model is not a success, and is discussed below, whereas the silicate problem, not being due to any inherent flaw in the model structure or equations, can be corrected by choosing a more suitable (i.e. lower) value for the relevant parameter.

The iron sub-model was developed for the Diat-HadOCC model (and so for use in HadGEM2-ES) at a time (circa 2007) when much less was known about the cycling of iron through the ocean ecosystem. This was particularly the case for a quantitative understanding of the system, which is required to produce a predictive numerical model: it is not enough to know that a certain process happens, in order to include it successfully in an Earth System Model the rate at which it happens and how

it depends (or not) on temperature and other factors (including concentrations of state variables) has to be known. It is certainly the case that if the Diat-HadOCC model was being developed now the iron sub-model would be very different from the one used as part of HadGEM2-ES in the CMIP5 experiments. In particular the forced remineralisation of all iron at the point at which material enters the detrital compartment(s) would not be repeated: there is incontrovertible evidence (Boyd et al., 2017) that iron is found in sinking detrital particles, and even in the model the problem that that choice was pragmatically made to address - too little iron in the surface waters - has ceased to exist since subsequent changes to the land surface scheme in HadGEM2-ES led to increased dust deposition to the ocean and so a greater surface iron supply. The result in the simulations was that the surface iron concentration was too high and was rarely limiting to phytoplankton growth.

One innovation used in the Diat-HadOCC model relates to how various phytoplankton and zooplankton processes respond to iron stress. Originally suggested by the late Professor Mike Fasham based on unpublished work, it provides separate iron-replete and iron-deplete parameter values, with the realised value at any time and location being determined by the dissolved iron concentration. The intention was to provide an effective short-cut where a quantitative mechanistic understanding of how iron affects certain biological processes is lacking or where an accurate representation would require extra state variables (e.g. for internal pools of stored nutrients, etc.). The model allows five processes to be modified this way: the growth-rate of diatoms, the growth-rate of misc-Phytoplankton, the Si:N ratio for uptake by diatoms, the preference for zooplankton feeding on diatoms and the natural mortality of zooplankton. The last two are not meant to suggest that dissolved iron directly affects any individual zooplankton, or indeed any particular zooplankton species, in that way, but rather recognises that the single Zooplankton compartment used in the Diat-HadOCC model has necessarily to represent an assemblage of different zooplankton species and iron-stress will lead to diatoms being more heavily silicified and so affect the relative palatability of diatoms as prey and the make-up of that assemblage. In a different model that has two zooplankton state variables it might be possible to produce such a shift in the assemblage more explicitly, and so it might not be necessary to use that last short-cut. The success of this innovation was not fully tested in the CMIP5 experiments, as the iron-replete and iron-deplete parameter values were set to be equal in all except the case of diatom growth-rate. The decision to do that was taken after a limited sensitivity analysis showed no great benefits in making the values significantly different, and it was reasoned that, as just part of a much larger ESM running predictive simulations over several hundred years, it was better to "play it safe" and err on the side of caution where there was no strong reason to do otherwise.

The problems of the too-high surface dissolved silicate and dissolved iron concentrations, while scoring poorly on some ocean ecosystem metrics, do not invalidate the air-sea flux of $CO_2$, or the ocean carbon storage, in the simulated results submitted to the CMIP5 experiment. The effect of those too-high concentrations is to make the diatom phytoplankton state variable to be not limited by silicate and iron, and so behave more similarly to the misc-Phytoplankton state variable than it should; therefore the total primary production and carbon drawdown is like that that would be seen if there was a single phytoplankton state variable, limited only by dissolved inorganic nitrogen (and light). While such a single-phytoplankton ecosystem model would lack some of the climate responses that it was hoped the Diat-HadOCC model would explore it would still be a valid model, so the representation of the wider carbon cycle (including the ocean carbon cycle) is not impaired. It is a disappointment that the Diat-HadOCC model as implemented for the CMIP5 experiments was not able to fully explore the

intended range of potential feedbacks (e.g. changes in iron-limitation due to changes in dust deposition, the effect of changes in the relative abundance of the two phytoplankton types, etc.). However, this failing was largely due to cautious choices of certain parameter values, which led to the phytoplankton types being very similar in behaviour, and poor choices of others, which led to the drift in surface dissolved silicate concentrations; with different parameter choices the model structure and equations could explore those potential feedbacks. The main structural problems concern the iron sub-model, in particular the forced remineralisation of iron rather than letting it become part of sinking detritus, and in the light of significant research undertaken since the model was developed this sub-model would benefit from being significantly changed.

The Diat-HadOCC model took part in the comparative study (Kwiatkowski et al., 2014) to choose the ocean biogeochemical sub-model for the first UK Earth System Model (UKESM1), but was not chosen. In the light of that decision there are no plans to develop the Diat-HadOCC model further. However, this paper achieves the important task of giving a detailed description of the Diat-HadOCC model that was used as part of HadGEM2-ES to run simulations for the CMIP5 experiment, which informed the 5th Assessment Report of the IPCC, and as such is a valuable record. Certain parameterisations uniquely used by the model have been highlighted. The successes, and weaknesses, of the model have been presented and discussed, making it clear where the latter are due to the model structure and where they are the result of parameter choices.

**Annex A: The photosynthesis sub-model**

The variation with light availability of the primary production of each phytoplankton type is calculated using the production scheme of Anderson (1993; hereafter TRA93). This models the preferential absorption of longer-wavelength light by seawater, so that the spectrum of light available for growth is shifted towards blue deep in the euphotic zone. Note that consequently the light calculated and used for photosynthesis in these functions at a given depth will not be the same as that available to the physics (for heating): the physics could easily be made to use the biological light field but does not do so as standard (and did not in the CMIP5 simulations). The functions also integrate production over a day, based on the noon surface irradiance and the number of daylight hours (from Equation 5 of Platt et al., 1990). This is consistent with the once-daily frequency of atmosphere-ocean coupling used in HadGEM2-ES (and previously in HadCM3C), because daily-average light is passed through the coupler and noon irradiance can easily be calculated given the daily-average and the number of daylight hours (and assuming, as Platt et al. did, that the light varies sinusoidally within the daylight hours only). Note that although the light will stay the same for each time-step between couplings the other factors determining production (e.g. phytoplankton abundance and nutrient concentration) will not, so the production is re-calculated every time-step and the appropriate proportion of daily production added to the phytoplankton state variable (e.g. 1/24 for a 1-hour time-step). When the HadOCC model (which uses the same productivity model) has been forced by 6-hourly re-analysis fluxes, for example, a daily-average irradiance field has been calculated and passed in for use in this scheme. When used in coupled models with shorter coupling periods, either a running 24-hour average of irradiance could be calculated and the scheme used as designed (and as described in the following paragraphs), or the daily integral part of the scheme could be removed and instantaneous production calculated using the remainder of the scheme.

TRA93 built on earlier work by Morel (1988,1991) which measured the absorption of light due to water and chlorophyll in 61 wavelength-bands, each 5 $nm$ wide, across the visible spectrum between 400 and 700 $nm$. Considering six typical chlorophyll depth-profiles TRA93 showed that the changing spectrum of light with depth (due to red light being more readily absorbed than blue) could be taken into account by splitting the water-column into three depth-ranges, allowing the absorption in each depth range to be modelled by a different function of the chlorophyll concentration. It was found that the best-fitting solution put the boundaries between the ranges at $5m$ and $23m$ depth, and the parameters for the three functions published in TRA93 related to those splits. However, since the physical ocean model in HadGEM2-ES (and also in previous Met Office GCMs, including HadCM3) has layer interfaces at $10m$ and $20m$ the scheme was re-parameterised for depth-range boundaries at those depths, and the model described here uses those new values. Note however that in other implementations of the Diat-HadOCC model (e.g. Kwiatkowski et al., 2014) the original TRA93 parameter values are used; where a light-scheme boundary (at $5m$ or $23m$) falls within a model layer that model layer is split in two at the appropriate depth for the purposes of calculating the primary production, and the results from the two sub-layers is then combined to update the phytoplankton biomass, etc.

Using the notation of TRA93, the spectrally-averaged vertical attenuation coefficient for layer $n$ within depth-range $L$, $k_n$ (units: $m^{-1}$), is given by that paper's Equation 16:

$$k_n = b_{0,L} + b_{1,L} \cdot c_n + b_{2,L} \cdot c_n^2 + b_{3,L} \cdot c_n^3 + b_{4,L} \cdot c_n^4 + b_{5,L} \cdot c_n^5 \tag{65}$$

where $c_n$ is the square-root of $G_n$, the total pigment concentration in layer $n$ (units: $mg\ m^{-3}$), and the re-parameterised coefficient values $b_{i,L}$ are given in Table 2. TRA93 assumed the chlorophyll biomass is always 80% of the total pigment biomass $G$ (the remainder being pheophytin) and the HadOCC and Diat-HadOCC models make the same assumption.

A derived parameter $a^{\#}$, required to calculate light absorption by phytoplankton, is then calculated by finding its surface value $a_{s,G}^{\#}$ (TRA93 Equation 20) and integrating down the water-column, $\frac{da^{\#}}{dz}$ being parameterised in terms of $c$ and the depth $z$ (TRA93 Equations 21-23). The paper's equations allow for the pigment concentration to have a depth-profile that varies continuously with depth, but as implemented in Met Office GCMs the concentration is taken as being constant within a model layer and changing suddenly at the depth-interfaces. TRA93 showed that this requires an offset to $a^{\#}$ when crossing between model layers: this offset is equal to the difference between $a_{s,G}^{\#}$ calculated using the $G$ for each layer.

The calculation (in layer $n$) of the model variable $astar_n$, which corresponds to $a^{\#}$ in TRA93, is performed layer-by-layer, stepping down from the surface; the value is calculated at the mid-point of each layer:

$$astar_1 = astar0_0 + 0.5 \cdot dastar_1 \qquad\qquad (n=1) \tag{66}$$

$$astar_n = astar_{n-1} + (dastar_{n-1} + dastar_n)/2 + astar0_n - astar0_{n-1} \qquad (n>1) \tag{67}$$

where $astar_1$ is the model variable corresponding to TRA93's $a^{\#}_{L=1}$, $astar0_0 = astar0_1$ and corresponds to $a^{\#}_{s,G_1}$, $dastar_1$ corresponds to $\frac{da^{\#}}{dz}(c,\nu)$ integrated over depth from the top to the bottom of layer 1 and where

$$
\begin{aligned}
astar0_n &= 0.36796 + 0.17537c_n - 0.065276c_n^2 + 0.013528c_n^3 - 0.0011108c_n^4 & (68)\\
dastar_n &= (gcof_1 + gcof_2 \cdot c_n + gcof_3 \cdot c_n^2 + gcof_4 \cdot c_n^3) \cdot DLCO0_n + (gcof_5 + gcof_6 \cdot c_n \\
&\quad + gcof_7 \cdot c_n^2) \cdot DLCO1_n + (gcof_8 + gcof_9 \cdot c_n) \cdot DLCO2_n + gcof_{10} \cdot DLCO3_n & (69)\\
c_n &= G_n^{0.5} \\
&= 1.25(\frac{w_C \cdot R_{c2n}^{Ph}}{R_{c2chl}^{Ph}} \cdot Ph + \frac{w_C \cdot R_{c2n}^{Dm}}{R_{c2chl}^{Dm}} \cdot Dm) & (70)\\
\nu_n &= 1 + Z_n
\end{aligned}
$$

$$
DLCO0_n = \nu_n - \nu_{n-1} \tag{71}
$$

$$
DLCO1_n = (\nu_n \cdot \log(\nu_n) - \nu_n) - (\nu_{n-1} \cdot \log(\nu_{n-1}) - \nu_{n-1}) \tag{72}
$$

$$
DLCO2_n = (\nu_n \cdot (\log(\nu_n))^2 - 2\nu_n \cdot \log(\nu_n) + 2\nu_n) - (\nu_{n-1} \cdot (\log(\nu_{n-1}))^2 - 2\nu_{n-1} \cdot \log(\nu_{n-1}) + 2\nu_{n-1}) \tag{73}
$$

$$
\begin{aligned}
DLCO3_n = \;&(\nu_n \cdot (\log(\nu_n))^3 - 3\nu_n \cdot (\log(\nu_n))^2 + 6\nu_n \cdot \log(\nu_n) - 6\nu_n) - (\nu_{n-1} \cdot (\log(\nu_{n-1}))^3 \\
&- 3\nu_{n-1} \cdot (\log(\nu_{n-1}))^2 + 6\nu_{n-1} \cdot \log(\nu_{n-1}) - 6\nu_{n-1}) \tag{74}
\end{aligned}
$$

In the above equations $R_{c2chl}^{Ph}$ is the carbon to chlorophyll ratio (units: mgC mgChl$^{-1}$), which is either calculated according to Equation 81 or fixed, $w_C$ is the molecular weight of carbon, 12.01 mg Mol$^{-1}$, and $Z_n$ is the depth (in metres) of the base of layer n, with $Z_0 = 0.0$m. Note that the $gcof$ coefficients relate to the 'g' coefficients in TRA93's Equations 18 and 21, but are numbered in a different order, as shown in Table 3; in TRA93 they were ordered by the total exponent of $c$ and $\nu$ combined, but the Diat-HadOCC model (like the HadOCC model) orders them by the exponent of $\nu$.

Based on TRA93's Equation 29 (itself derived from work described in Platt et al., 1990) the primary production for each phytoplankton type ($Dm$ or $Ph$) in layer n during a whole day can then be calculated using a fitted 5th-order polynomial. In that equation, a quantity shown as $(\alpha_{max}^B \cdot a_n^{\#} \cdot I_{n,\Phi,1}/P_m^B)$ is calculated; Platt et al.'s polynomial is fitted for values of that quantity between 0.0 and 15.8 and the fitted function oscillates wildly outside that range, but in the model the value of the corresponding quantity can be larger than 15.8. Therefore a rational function with non-oscilliatory behaviour was calculated (Geoff Evans, pers. comm) which matches the 5th-order polynomial at an input of 15.8 in both value and first derivative, and

this is used for higher input values. For phytoplankton type $X$ and layer n (of thickness $\Delta_n$):

$$solbio_n = solbio_{n-1} \cdot exp(-k_n \cdot \Delta_n) \tag{75}$$

$$psmaxs_n^X = P_n^X \cdot R_{c2chl}^X / 24 \tag{76}$$

$$V_a = \alpha_{mx}^X \cdot astar_n / psmaxs_n^X \tag{77}$$

$$V_b = V_a \cdot solbio_{n-1}$$

$$V_c = V_a \cdot solbio_n$$

$$V_d = MIN(15.8, V_b)$$

$$V_e = MIN(15.8, V_c)$$

$$V_f = MAX(15.8, V_b)$$

$$V_g = MAX(15.8, V_c)$$

$$psynth_n^X = \sum_{i=1}^{5} \Omega_i [V_d^i - V_e^i] + \left( \frac{V_f \cdot (\gamma_1 + \gamma_2 \cdot V_f)}{(1.0 + \gamma_3 \cdot V_f)} - \frac{V_g \cdot (\gamma_1 + \gamma_2 \cdot V_g)}{(1.0 + \gamma_3 \cdot V_g)} \right) \tag{78}$$

The values of the coefficients $\Omega$ and $\gamma$ are given in Table 4. In the above equations, $\alpha_{mx}^X$ is the maximum photosynthetic efficiency ($\alpha_{max}^B$ in TRA93) and has the value 2.602 times $\alpha^X$, the initial slope of the photosynthesis-light curve (Equation 26 in TRA93). $P_n^X$ is the maximum growth rate for the phytoplankton type and layer, taking into account the temperature and the nutrient limitations, as calculated in Equations 10 and 11. $solbio_0$ is the solar radiance just below the ocean surface. The total daily production in that layer is then:

$$ph_{PP} = Ph \cdot \frac{dlh \cdot P^{Ph}}{\pi \cdot k \cdot \Delta} \cdot psynth^{Ph} \tag{79}$$

$$dm_{PP} = Dm \cdot \frac{dlh \cdot P^{Dm}}{\pi \cdot k \cdot \Delta} \cdot psynth^{Dm} \tag{80}$$

where $dlh$ is the number of daylight hours at that location and time of year and $k$ is the attenuation coefficient calculated in Equation 65. All terms in these equations (except $dlh$ and the constant $\pi$) vary between layers. Where a number of layers are part of a surface mixed layer at a given time-step the production in those layers is averaged over those layers.

**Annex B: Carbon-to-chlorophyll ratio**

The carbon to chlorophyll ratio for each phytoplankton type, $R_{c2chl}^X$, can either be prescribed or updated using a scheme based on Geider et al. (1996,1997,1998). In the CMIP5 simulations run using HadGEM2-ES the constant values $R_{c2chl,0}^X$ shown in Table 5 were used. However, for completeness the time-varying scheme as implemented in the Diat-HadOCC model is described briefly.

Re-arranging Equations A1-A5 in Geider et al. (1997; hereafter G97) produces (using that paper's notation, including $\theta = (chl/C)$, so corresponding to the reciprocal of the ratio used in this model):

$$\frac{d\theta}{dt} = \frac{k_{chl}}{\theta} \cdot \frac{(P_m^C)^2}{\alpha^{chl} I} \cdot \left(1 - exp\left(\frac{-\alpha^{chl} I\theta}{P_m^C}\right)\right) - \theta \cdot \left(P_m^C \cdot \left(1 - exp\left(\frac{-\alpha^{chl} I\theta}{P_m^C}\right)\right) - (R^{chl} - R^C)\right) \tag{81}$$

where G97's $P_m^C$ corresponds to this model's $P^X$, $\alpha^{chl}$ corresponds to $\alpha_{mx}^X \cdot astar$, $I$ is the irradiance (in the middle of the layer) and $R^{chl}$ and $R^C$ are respectively the specific removal rates of chlorophyll and carbon from the phytoplankton. Finally, $k_{chl}$ is the 'maximum proportion of photosynthesis that can be directed to chl a synthesis', but in a number of conditions is equal to the maximum $(chl/C)$ ratio, and in this model it is represented by $1/R_{c2chl,min}^X$.

The equation above has no analytical solution for $\theta$, and it is intended that the model should be able to operate with long time-steps if required (up to 1 day), so a semi-implicit finite-difference solution was found. $\frac{d\theta}{dt}$ is represented as $(\theta_{t+1} - \theta_t)/\delta t$, and the $\theta$s inside the exponents take the value $\theta_t$ (i.e. the reciprocal of the value of $R_{c2chl}^X$ from the previous time-step) while those outside take the value $\theta_{t+1}$. $R^C$ is set equal to $\Pi_{resp}^X + \Pi_{mort}^X \cdot X$ (where $X$ is $Ph$ or $Dm$ as appropriate), and $R^{chl}$ is set equal to $R^C$ (so the difference is zero). Then a simple re-arrangement results in a quadratic equation in $\theta_{t+1}$ which can be easily solved. The updated value of $R_{c2chl}^X$ is then the reciprocal of the resulting $\theta$ (though it can be necessary on occasions to apply upper and lower bounds to the ratio, respectively $R_{c2chl,max}^X$ and $R_{c2chl,min}^X$). Ratios calculated in layers that are part of the surface mixed layer are averaged. As implemented, the ratio is stored from one time-step to the next and not advected or mixed as a tracer; the change in the ratio due to biological processes is much larger than that due to mixing with the ratio in adjacent grid boxes. It would be possible to use the ratio and the concentration of the appropriate phytoplankton type to create a phytoplankton-chlorophyll state variable which could be advected and mixed as a tracer, but that is not how the scheme is currently used in the Diat-HadOCC model.

## Annex C: Air-sea fluxes

Finally, the calculations of the air-to-sea fluxes of $O_2$ and $CO_2$ (respectively $[Oxy_{asf}]$ and $[CO2_{asf}]$) follow the methodology of OCMIP. The flux is the product of the gas-specific gas transfer (piston) velocity $Vp$ and the difference between the gas concentrations in the atmosphere (just above the sea-surface), $X_{sat}$, and in the (surface) ocean, $X_{surf}$:

$$X_{asf} = Vp_X \cdot (X_{sat} - X_{surf}) \tag{82}$$

The piston velocity (in m/s) is a function of the 10m wind-speed, $U$ (using the Wanninkhof 1992 formulation, normalised for a Schmidt number of 660), the gas-specific Schmidt number $Sch$ and the fraction of the grid-box area that is open water $A_{ow}$:

$$Vp_X = A_{ow} \cdot (f_U \cdot U^2 \times 0.01/3600.0) \cdot (Sch_X/660)^{-1/2} \tag{83}$$

where $f_U$ is a coefficient taking the value 0.31 if wind-speed averaged over a day or less is used (e.g. in a coupled model) or 0.39 if monthly-mean wind-speed is used (Wanninkhof, 1992).

In the case of oxygen $O_{2,surf}$ is the model oxygen concentration, while the surface ocean is assumed to be fully saturated in equilibrium so $O_{2,sat}$ is equal to the solubility $C_{O_2}$ (calculated in units of ml $O_2$/l, and converted to model units

before use). That is calculated using Equation 8 of (Garcia and Gordon, 1992), but removing the spurious "$A_3 \cdot T_s^2$" term found at the end of the first line (as in the o2sato.f subroutine in the OCMIP-2 Biotic-HOWTO documentation, available at http://ocmip5.ipsl.jussieu.fr/OCMIP/phase2/simulations/Biotic/boundcond/o2sato.f). The solubility coefficients used in the OCMIP-2 subroutine, originally from Benson and Krause (1984) and recommended by Garcia and Gordon (1992), are used here. Note that in HadGEM2-ES the sea-level pressure is assumed to be always 1.0 atmospheres, everywhere. Therefore the equation is:

$$
\begin{aligned}
C_{O_2} \;=\; & exp(2.00907 + 3.22014T_s + 4.05010T_s^2 + 4.94457T_s^3 - 0.256847T_s^4 + 3.88767T_s^5 \\
& -S \cdot (6.24523 + 7.37614T_s + 10.3410T_s^2 + 8.17083T_s^3) \times 10^{-3} - 4.88682 \times 10^{-3} \cdot S^2)
\end{aligned}
\tag{84}
$$

where sea-surface temperature $T$ has units of $^\circ$C, salinity $S$ has units of permil and where $T_s = ln[(298.15-T)(273.15+T)^{-1}]$. $C_{O_2}$ can be converted to units of mol $O_2$/m$^3$ by dividing by the molar volume, 22.3916 l/mol. The Schmidt number is calculated according to Keeling et al. (1998):

$$
Sch_{O_2} = 1638.0 - 81.83T_l + 1.483T_l^2 - 0.008004T_l^3
\tag{85}
$$

where $T_l = max(-2.0, min(40.0, T))$, protecting the calculation from crashing if the physical ocean model should produce unreasonably low or high sea-surface temperatures.

In the case of carbon dioxide $CO_{2,sat} = C_{CO_2} \cdot pCO_{2,atm}$ where $C_{CO_2}$ is the $CO_2$ solubility and $pCO_{2,atm}$ is the partial pressure of $CO_2$ in dry air at 1 atmosphere pressure in the atmospheric level immediately above the ocean surface (note again that the sea-level pressure is always assumed to be 1 atmosphere). The solubility is that due to Weiss (1974):

$$
C_{CO_2} = exp(93.4517/T_h - 60.2409 + 23.3585 \cdot ln(T_h) + S \cdot (0.023517 - 0.023656T_h + 0.0047036T_h^2))
\tag{86}
$$

where $T_h = max(2.71, (273.15+T)/100.0)$ (protecting the calculation from any spuriously-low sea-surface temperatures the physical model might produce). The Schmidt number for $CO_2$ is calculated according to Wanninkhof (1992):

$$
Sch_{CO_2} = 2073.1 - 125.62T_l + 3.6276T_l^2 - 0.043219T_l^3
\tag{87}
$$

where $T_l$ is defined as in the calculation for $Sch_{O_2}$.

The calculation of $CO_{2,surf}$ has to take into account the partitioning of $DIC$ into three forms, namely carbonic acid (taken here to include the dissolved gas phase), bicarbonate ion and carbonate ion, only the first of which contributes to the air-to-sea flux:

$$
DIC = [H_2CO_3] + [HCO_3^-] + [CO_3^{2-}]
\tag{88}
$$

The calculation of the partitioning, which follows the method described by Bacastow (1981), requires as inputs the total Alkalinity $A_T$ and the DIC concentration $DIC$, the temperature, the salinity and the total boron concentration. The method involves using a term $\chi_{x,i}$, which is dependent as shown in Equation 103 on an earlier estimate of the hydrogen ion concentration $[H^+]$,

to calculate the carbonate alkalinity $A_C = A_T - f(\chi_{x,i})$. $A_C$ is then used with $DIC$ to set up a quadratic equation in the related term $\chi_{y,i}$. Bacastow (1981) then used the secant method of similar triangles (Acton, 1970) to produce an updated estimate $\chi_{x,i+1}$ and to minimise the difference between successive estimates. This algorithm is explained in more detail below.

Four equilibrium constants describing the dissociation of carbonic acid ($K_1$, from Roy et al. 1993), bicarbonate ion ($K_2$, also from Roy et al. 1993), boric acid ($K_B$, from Dickson 1990) and water ($K_W$, from Millero 1995) are calculated (in moles/kg):

$$
\begin{aligned}
K_1 &= \frac{[H^+][HCO_3^-]}{[H_2CO_3]} & (89) \\
&= (1 - 0.001005S) \cdot exp(-2307.1266/T_k + 2.83655 - 1.5529413ln(T_k) \\
&\quad -(4.0484/T_k + 0.20760841) \cdot S^{1/2} + 0.08468345S - 0.00654208S^{3/2}) & (90) \\
K_2 &= \frac{[H^+][CO_3^{2-}]}{[HCO_3^-]} & (91) \\
&= (1 - 0.001005S) \cdot exp(-3351.6106/T_k - 9.226508 - 0.2005743ln(T_k) \\
&\quad -(23.9722/T_k + 0.106901773) \cdot S^{1/2} + 0.1130822S - 0.00846934S^{3/2}) & (92) \\
K_B &= \frac{[H^+][B(OH)_4^-]}{[B(OH)_3]} & (93) \\
&= exp(-(8966.90 + 2890.53S^{1/2} + 77.942S - 1.728S^{3/2} + 0.0996S^2)/T_k \\
&\quad +(148.0248 + 137.1942S^{1/2} + 1.62142S) - (24.4344 + 25.085S^{1/2} + 0.2474S) \cdot ln(T_k) \\
&\quad +0.053105S^{1/2} \cdot T_k) & (94)
\end{aligned}
$$

$$
\begin{aligned}
K_W &= [H^+][OH^-] & (95) \\
&= exp(-13847.26/T_k + 148.96502 - 23.6521ln(T_k) \\
&\quad +(118.67/T_k - 5.977 + 1.0495ln(T_k)) \cdot S^{1/2} - 0.01615S) & (96)
\end{aligned}
$$

where $T_k = T + 273.15°C$ is the temperature in Kelvin and S the salinity in per mil. Note that, because these constants are in units of Moles/kg-seawater (strictly, (Moles/kg-seawater)$^2$ in the case of $K_W$), the alkalinity and DIC state variables must be converted to those units from the model units of mMoles/m$^3$ before the partitioning is calculated; all state variables in the converted units have the subscript $u$ (e.g. $A_{T,u}$).

The total borate concentration $B_T$ (in Moles/kg) is set to be proportional to the salinity: $B_T = [B(OH)_3] + [B(OH)_4^-] = 4.16e^{-4}S/35.0$. Then, since the Diat-HadOCC model uses the 5-term expression for total alkalinity (Bacastow, 1981), the carbonate alkalinity is calculated as:

$$
\begin{aligned}
A_{C,u} &= [HCO_3^-] + 2[CO_3^{2-}] & (97) \\
&= A_{T,u} - Q_W \cdot \chi_{x,i} + Q_p/\chi_{x,i} - B_T / \left(1 + \frac{Q_B}{\chi_{x,i}}\right) & (98)
\end{aligned}
$$

where

$$Q_p = \sqrt{K_1 \cdot K_2} \tag{99}$$

$$Q_r = \sqrt{\frac{K_1}{K_2}} \tag{100}$$

$$Q_B = \frac{Q_p}{K_B} \tag{101}$$

$$Q_W = \frac{K_W}{Q_p} \tag{102}$$

$$\chi = \frac{Q_p}{[H^+]} \tag{103}$$

Equations 88 and 97 can be re-arranged and combined with Equations 89, 91, 99, 100 and 103 to give a quadratic in $\chi_{y,i}$:

$$(2DIC_u - A_{C,u}) \cdot \chi_{y,i}^2 - Q_r \cdot (A_{c,u} - DIC_u) \cdot \chi_{y,i} - A_{C,u} = 0 \tag{104}$$

which has the solution

$$\chi_{y,i} = 0.5(Q_r \cdot (A_{c,u} - DIC_u) + \sqrt{(Q_r^2 \cdot (A_{c,u} - DIC_u)^2 + 4A_{C,u} \cdot (2DIC_u - A_{C,u}))})/(2DIC_u - A_{C,u}) \tag{105}$$

When $\chi_{y,i}$ and $\chi_{x,i}$ are equal the value of $\chi$ that is consistent with both the $A_{C,u}$ and the $DIC_u$ values (for the current temperature and salinity) has been found, so [H$_2$CO$_3$] can be found from Equations 88, 89 and 91. While the two estimates of $\chi$ are not equal however, the secant method of similar triangles (Acton, 1970) is used to find an updated estimate $\chi_{x,i+1}$ for input into the next iteration of Equation 98 by minimising $\chi_y - \chi_x$. The two similar triangles are right-angled and have sides of length $(\chi_{x,i+1} - \chi_{x,i}, \chi_{y,i} - \chi_{x,i})$ and $(\chi_{x,i+1} - \chi_{x,i-1}, \chi_{y,i-1} - \chi_{x,i-1})$ respectively; equating the ratios of these two triangles' sides and re-arranging gives

$$\chi_{x,i+1} = \frac{\chi_{x,i-1} \cdot \chi_{y,i} - \chi_{x,i} \cdot \chi_{y,i-1}}{(\chi_{y,i} - \chi_{y,i-1}) - (\chi_{x,i} - \chi_{x,i-1})} \tag{106}$$

This calculation can be iterated until the fractional change in successive estimates is less than a certain amount (e.g. $10^{-5}$). However, in the implementation used for HadGEM2-ES the calculation was iterated eight times; it had been found that the convergence criterion was always satisfied in 6 iterations, and given the computer architecture it was more computationally efficient to run that way than to repeatedly test for convergence.

Once the carbonic acid concentration has been determined (and converted back to model units) it can be used as $CO_{2,surf}$ in the air-sea flux calculation. Other diagnostic quantities can also be calculated: $pCO_2$ and $pH$ (the latter from the H$^+$ concentration).

*Code availability.* Due to intellectual property right restrictions, the author cannot provide either the source code or documentation papers for the Unified Model (UM). The Met Office Unified Model is available for use under licence. A number of research organizations and national meteorological services use the UM in collaboration with the Met Office to undertake basic atmospheric process research, produce

forecasts, develop the UM code and build and evaluate Earth system models. For further information on how to apply for a licence, see http://www.metoffice.gov.uk/research/modelling-systems/unified-model.

*Acknowledgements.* The development of the HadGEM2 family represents the work of a large number of people, to whom the author is indebted. This work was supported by the Joint DECC/Defra Met Office Hadley Centre Climate Programme (GA01101). The author also wishes to thank three anonymous reviewers for their perceptive and detailed comments which greatly helped to improve the manuscript.

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

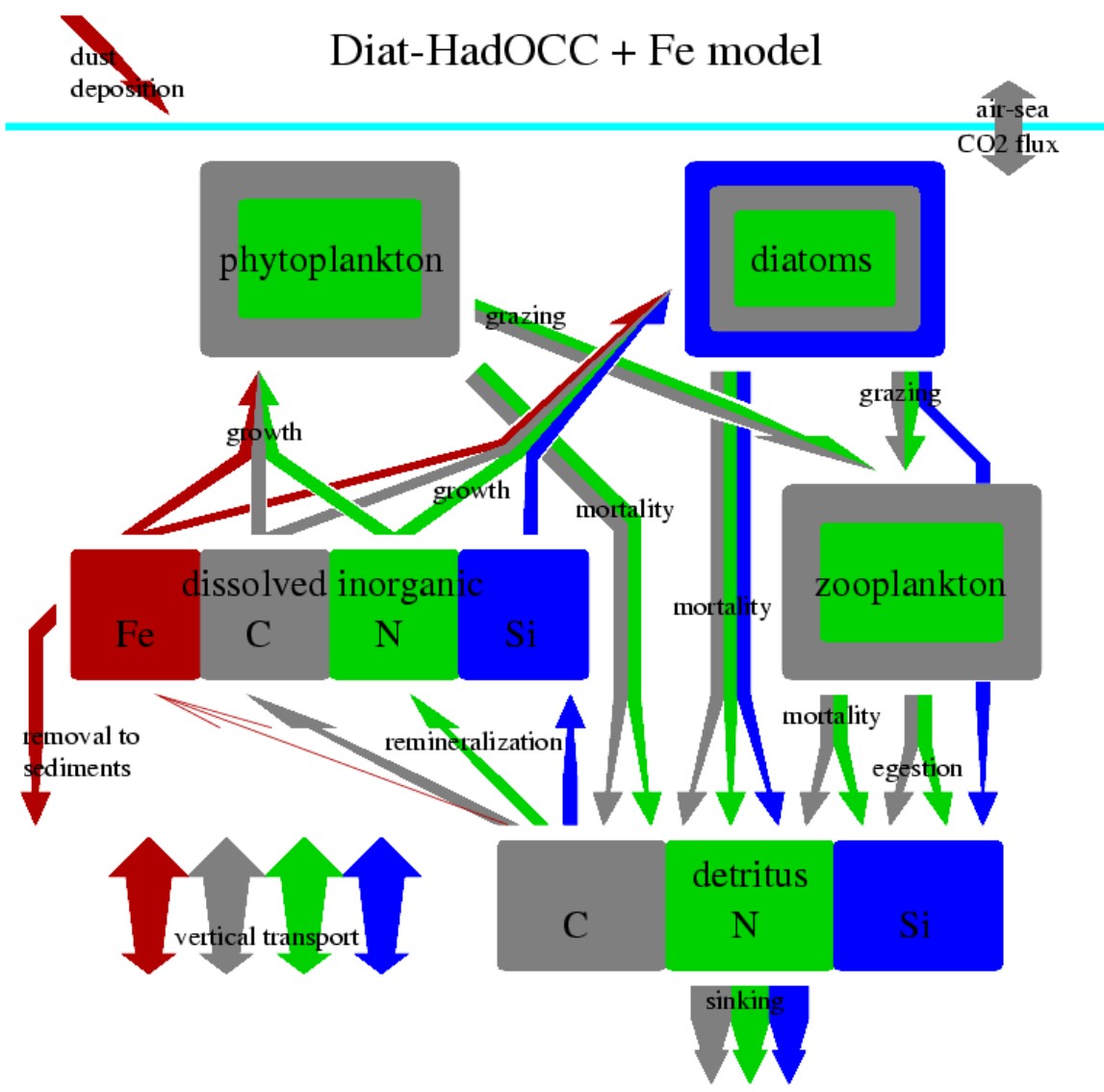

**Figure 1.** Diagram of the Diat-HadOCC model components and flows of nitrogen, carbon, silicon and iron

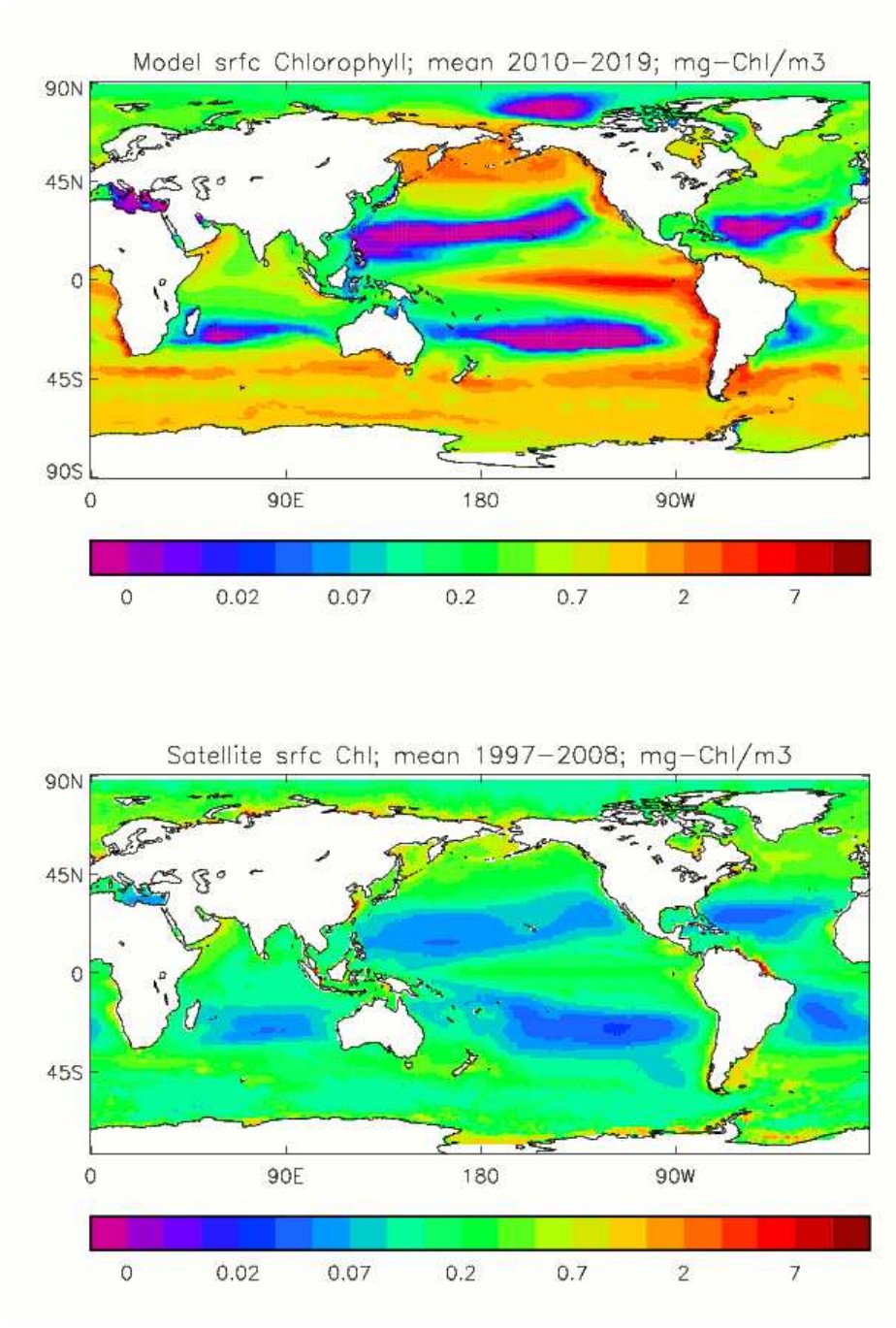

**Figure 2.** Comparison of surface chlorophyll: upper panel, mean over the years 2010-9 inclusive from the model, Historical+ RCP8.5 scenario; lower panel, mean over 1998-2007 from GlobColor, with further processing as described in (Ford et al., 2012). Units are mg Chl m$^{-3}$

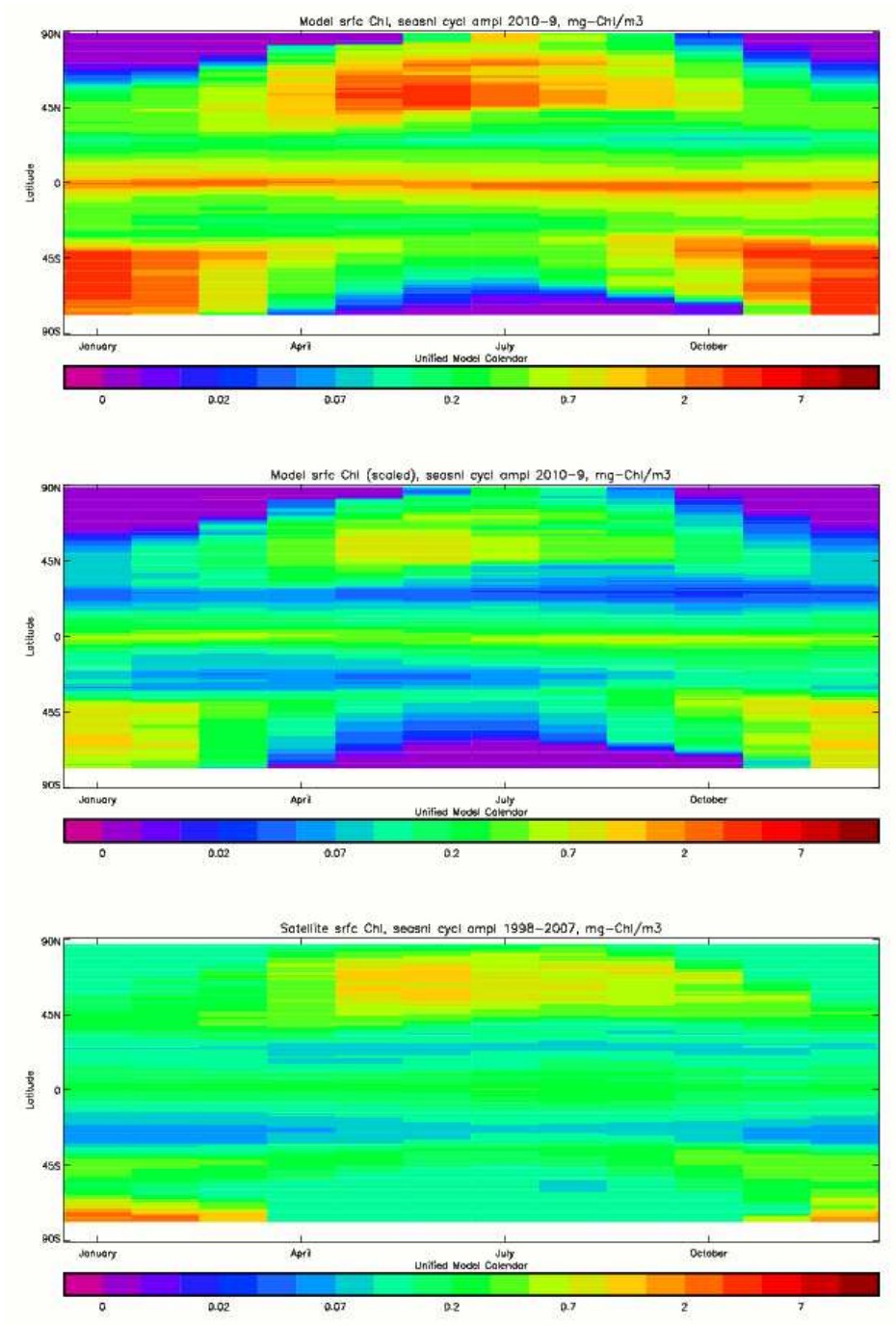

**Figure 3.** Seasonal cycle of global zonal mean surface chlorophyll, in mg Chl m$^{-3}$: top panel, average over the years 2010-9 inclusive from the model, Historical+RCP8.5 scenario; middle, the same but scaled by factor 0.213/0.812 (=0.262) so that the model mean matches the observations; bottom, satellite-derived data from GlobColor, averaged over 1998-2007 inclusive.

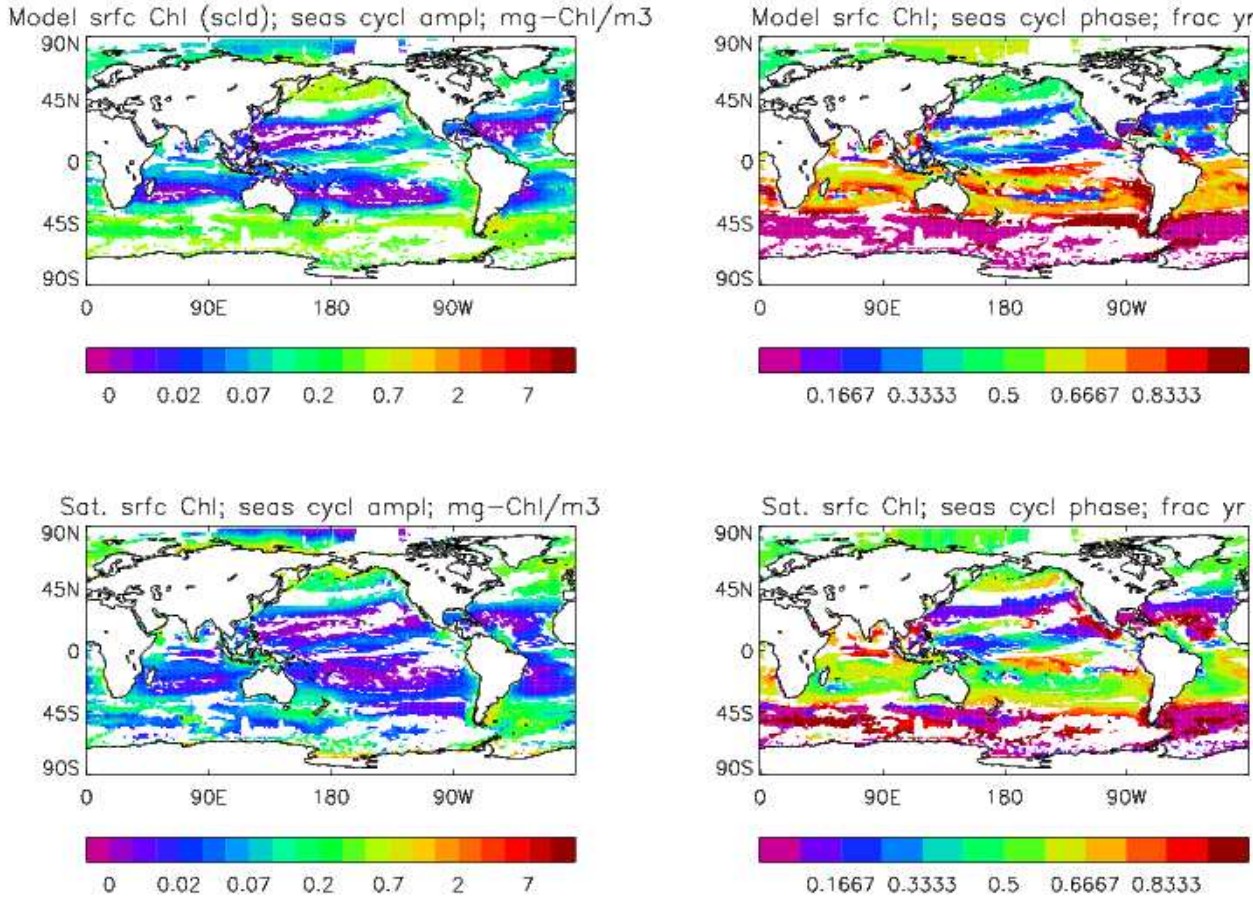

**Figure 4.** The amplitude (left-hand panels; units are mg Chl m$^{-3}$) and phase (right-hand panels; units are 'fraction of year') of the seasonal cycle of surface chlorophyll in the model (upper panels; average over years 2010-9, Historical+RCP8.5 scenario, amplitude scaled by factor of 0.213/0.812) and in the GlobColor data (lower panels; average over years 1998-2007). The amplitude has been determined by finding the best-fitting sine-curve through the monthly-mean values of the average cycle at each point, and the phase refers to the fraction of the year when the fitted curve is at its maximum. Points are left white if the variance of the residual (after the best-fitting sine-curve has been removed) is more than half that of the original seasonal cycle.

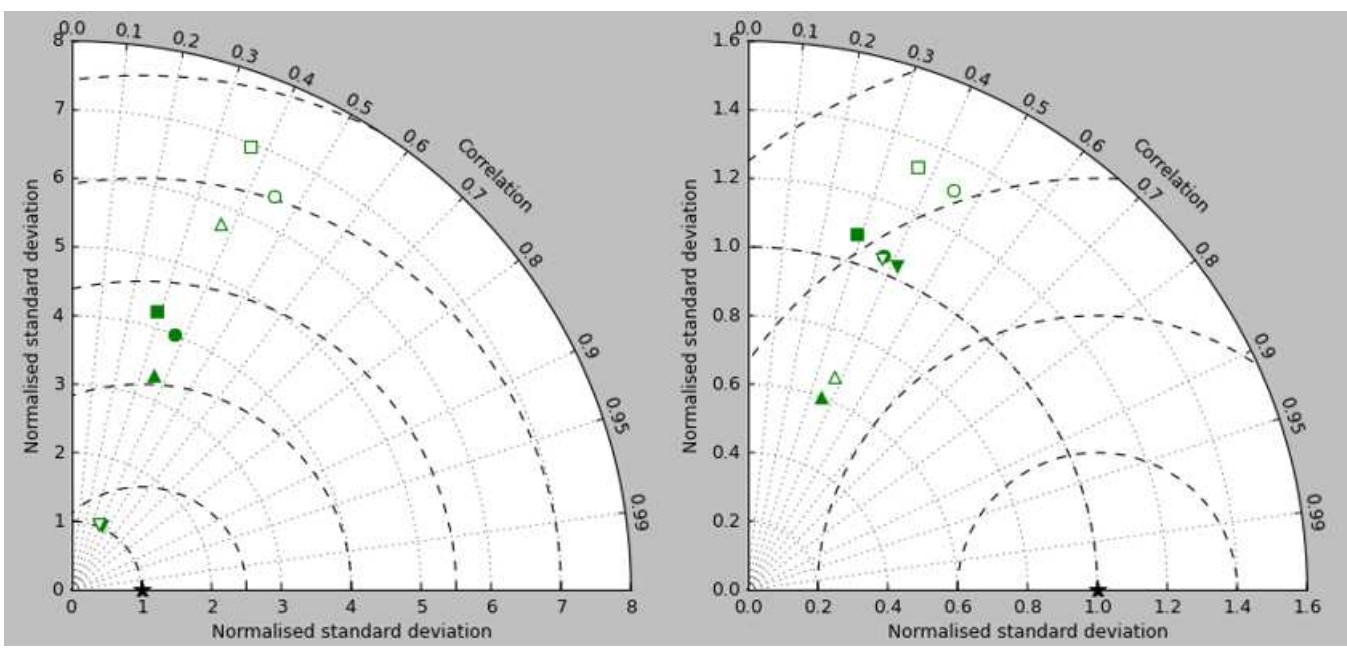

**Figure 5.** Taylor diagrams of model surface chlorophyll compared to the GlobColor product. Solid symbols represent correlations and standard deviations from points in all parts of the ocean (except inland seas), while open symbols have had a mask applied to remove the Arctic Ocean and two grid-boxes around the coast, as explained in the text. Squares represent the annual mean of all points, while circles, up-pointing triangles and down-pointing triangles respectively represent the mid-point, amplitude and phase of the sine-curve that best fits the seasonal cycle (where the variance of the residual is less than half the variance of the cycle). The diagram on the left uses the raw model results, while that on the right uses the model chlorophyll scaled to give a comparable global mean to the observations (again as explained in the text).

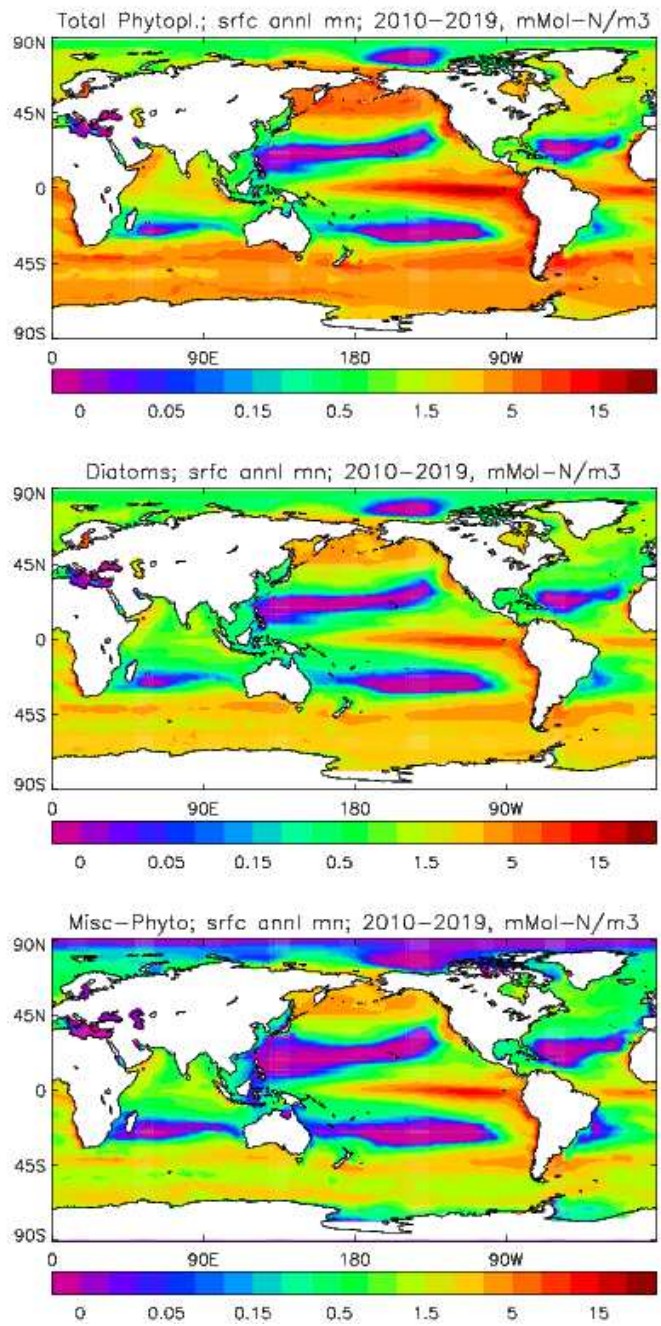

**Figure 6.** Phytoplankton surface biomass (in mMol N m$^{-3}$), averaged over the model years 2010-2019 inclusive: top panel, total phytoplankton; middle panel, Diatoms; bottom panel, misc-Phytoplankton.

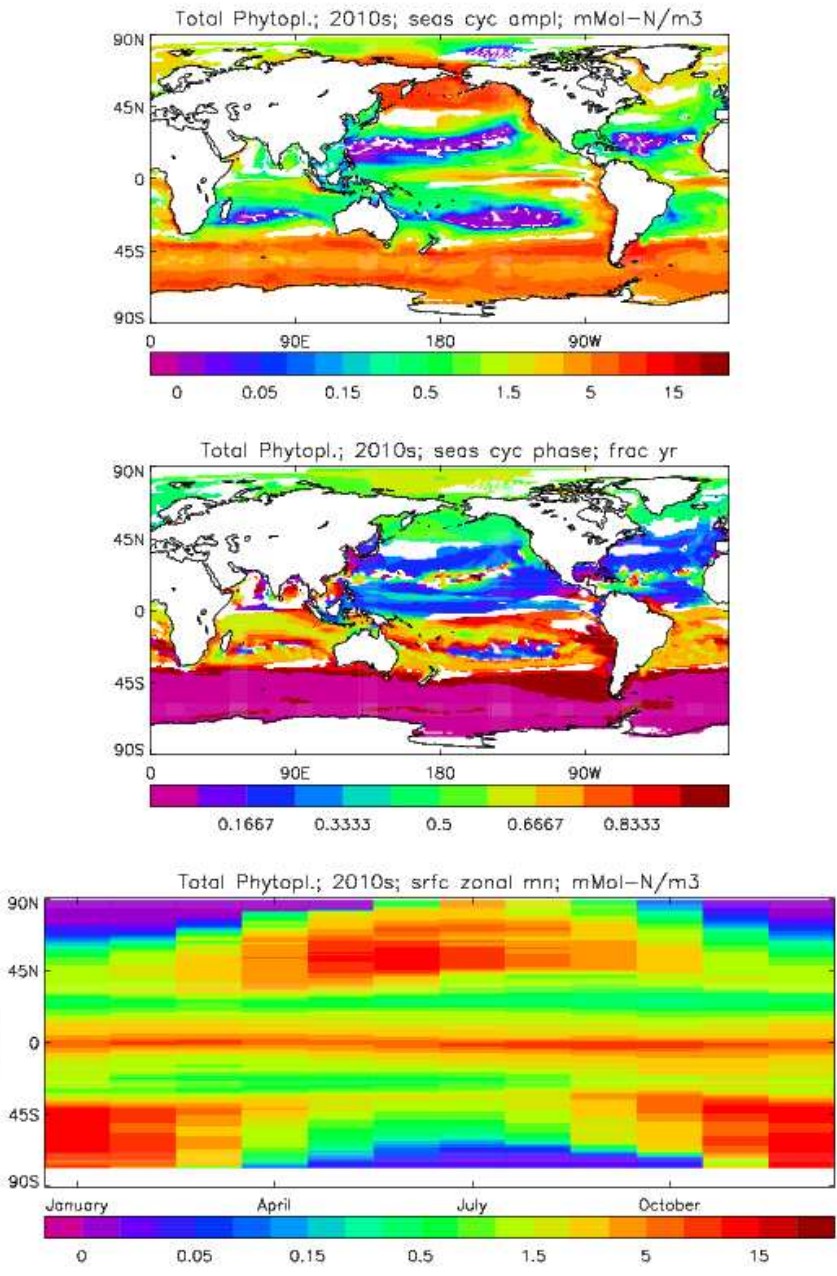

**Figure 7.** Total phytoplankton surface biomass mean seasonal cycle, averaged over model years 2010 to 2019 inclusive. Top panel, amplitude (in mMol N m$^{-3}$) and middle panel, the phase (fraction of year when peak value occurs) of the seasonal cycle, determined by the best-fitting sine-curve (only points where the residual variance is less than half that of the original cycle are shown). Bottom panel, the global zonal mean for each month (mMol N m$^{-3}$).

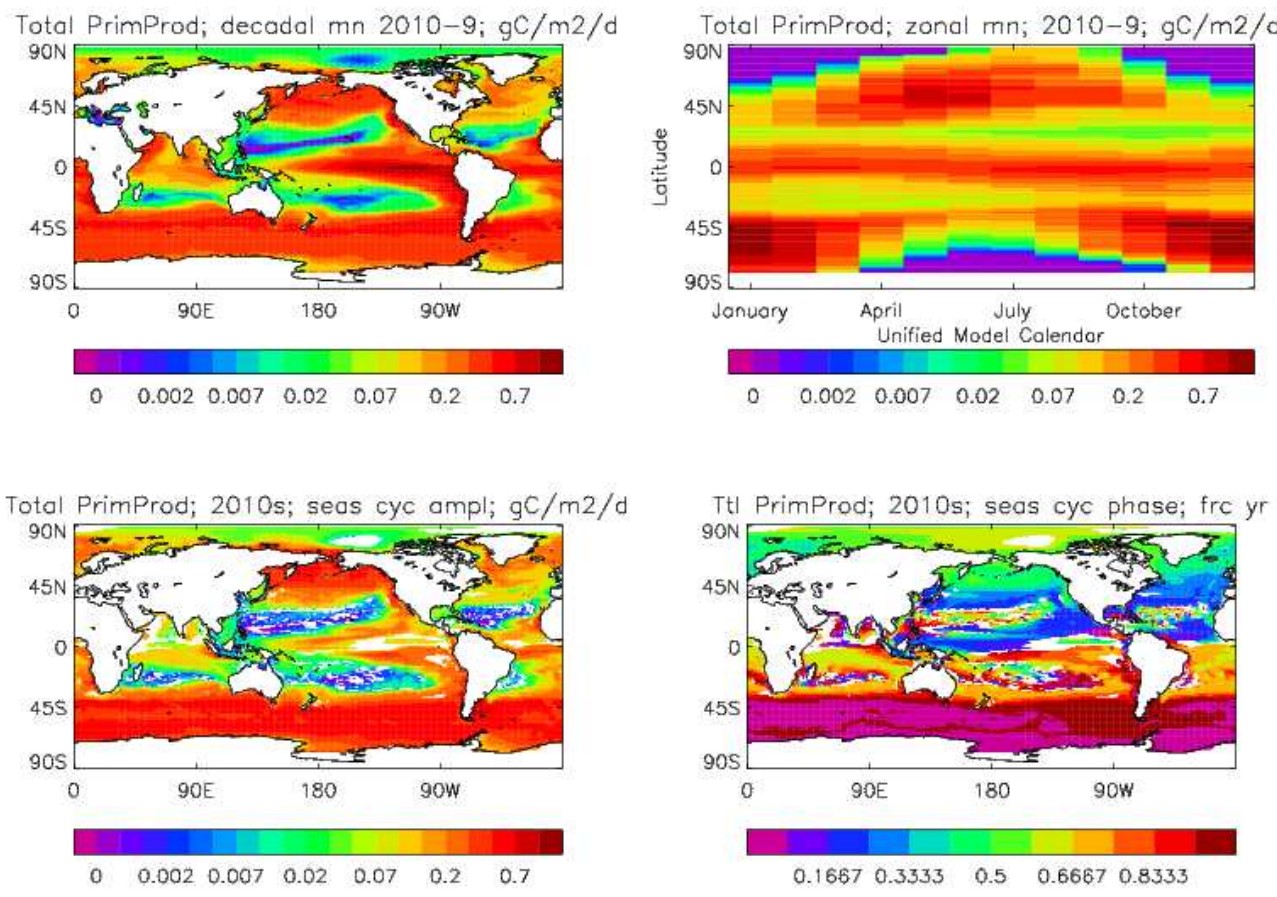

**Figure 8.** Total Primary Production, depth-integrated, averaged over the model years 2010-2019 inclusive: upper left panel, decadal mean (units: g C m$^{-2}$ d$^{-1}$); upper right panel, zonal mean for each month, same units; lower left, amplitude of model seasonal cycle (best fitting sine-curve), same units; lower right, phase of model seasonal cycle (units are fraction of year when peak value occurs). As described in the text, the seasonal cycle has been determined by the best-fitting sine-curve, and points are only shown where the variance of the residual cycle is less than half that of the original cycle.

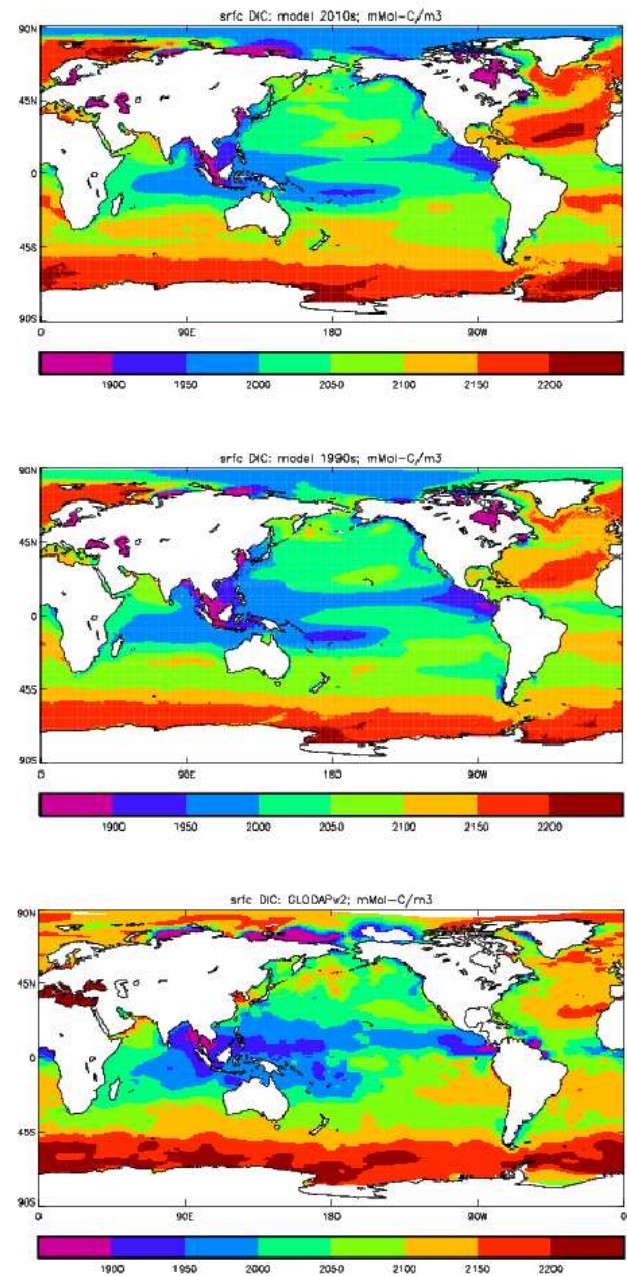

**Figure 9.** Surface concentration of Dissolved Inorganic Carbon (mMol C m$^{-3}$): top panel, model field averaged over model years 2010-2019 inclusive; middle, model field averaged over model years 1990-1999 inclusive; bottom, the gridded field from the GLODAPv2 database

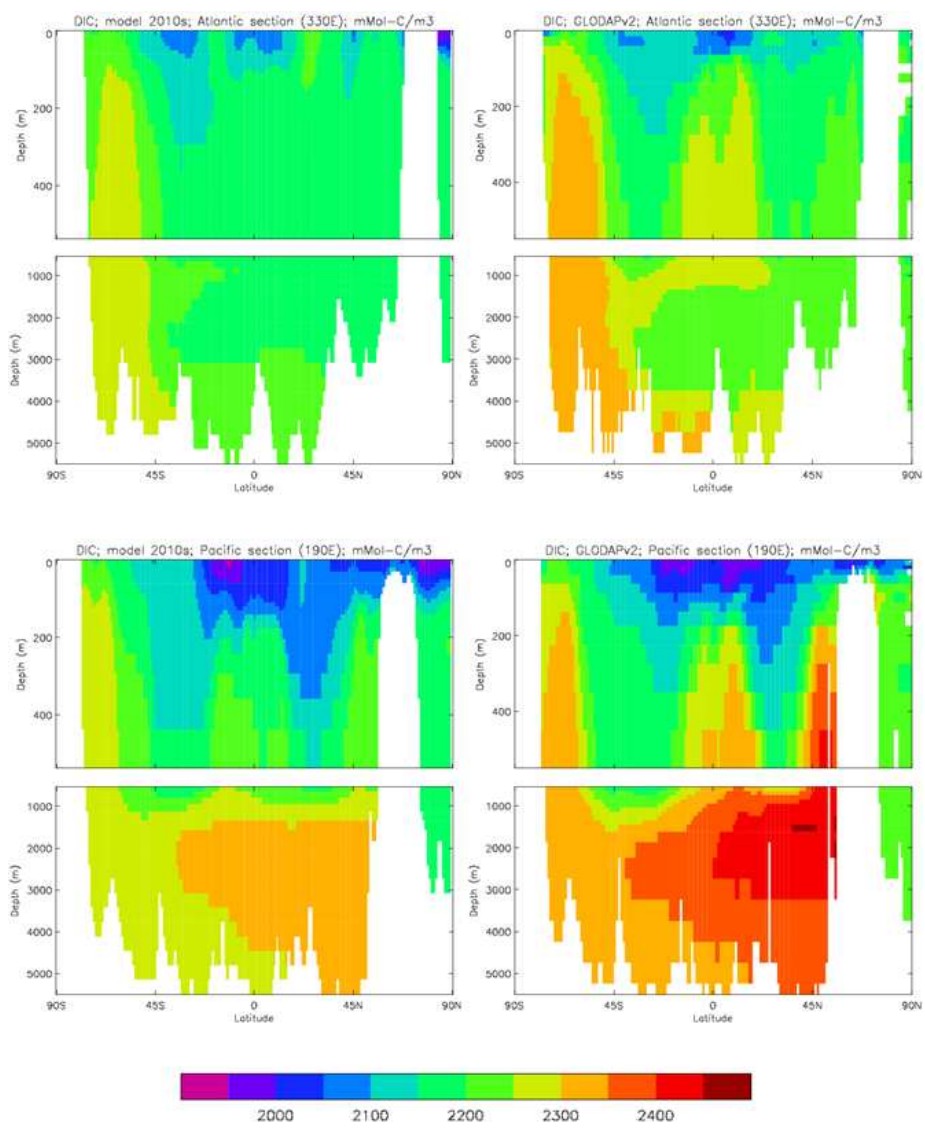

**Figure 10.** Meridional sections of DIC: Upper panels show sections in the Atlantic Ocean along 330°, lower panels Pacific Ocean sections along 190°; left panels show model concentrations averaged over 2010-2019, right panels show concentrations from the GLODAPv2 gridded product.

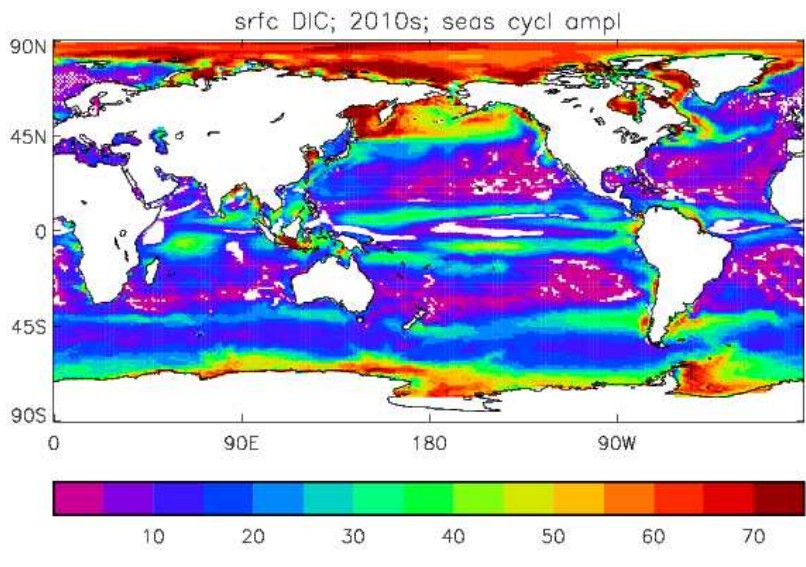

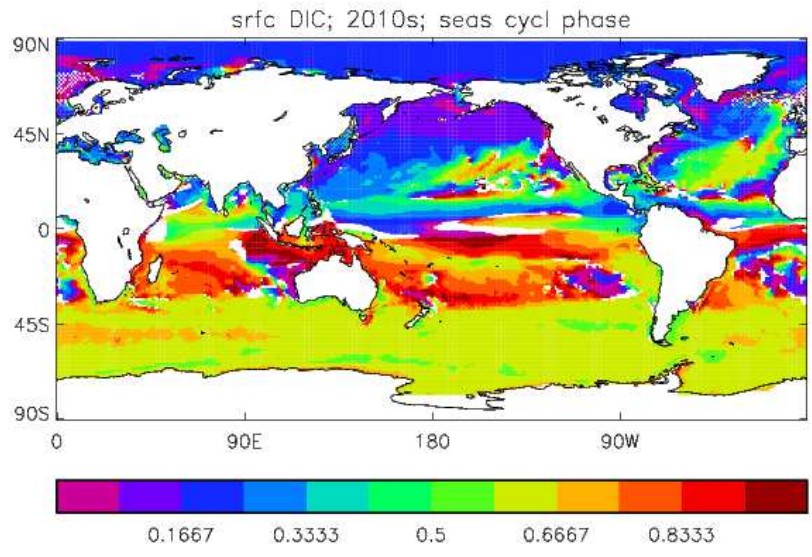

**Figure 11.** Surface DIC, model seasonal cycle, averaged over model years 2010-2019 inclusive: upper panel, amplitude of cycle (mMol C m$^{-3}$); lower panel, phase of cycle (fraction of year). Only points where the residual variance is less than half the original are shown.

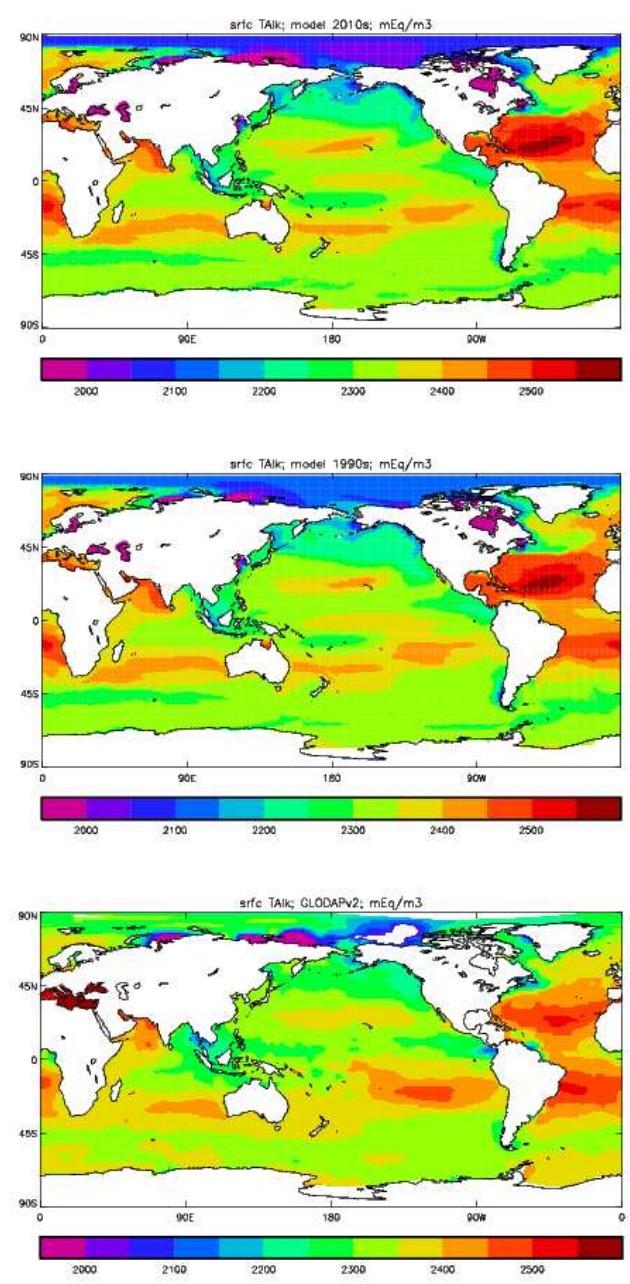

**Figure 12.** Surface concentration of Total Alkalinity (mEq m$^{-3}$): top panel, model field averaged over model years 2010-2019 inclusive; middle, model field averaged over model years 1990-1999 inclusive; bottom, the gridded field from the GLODAPv2 database.

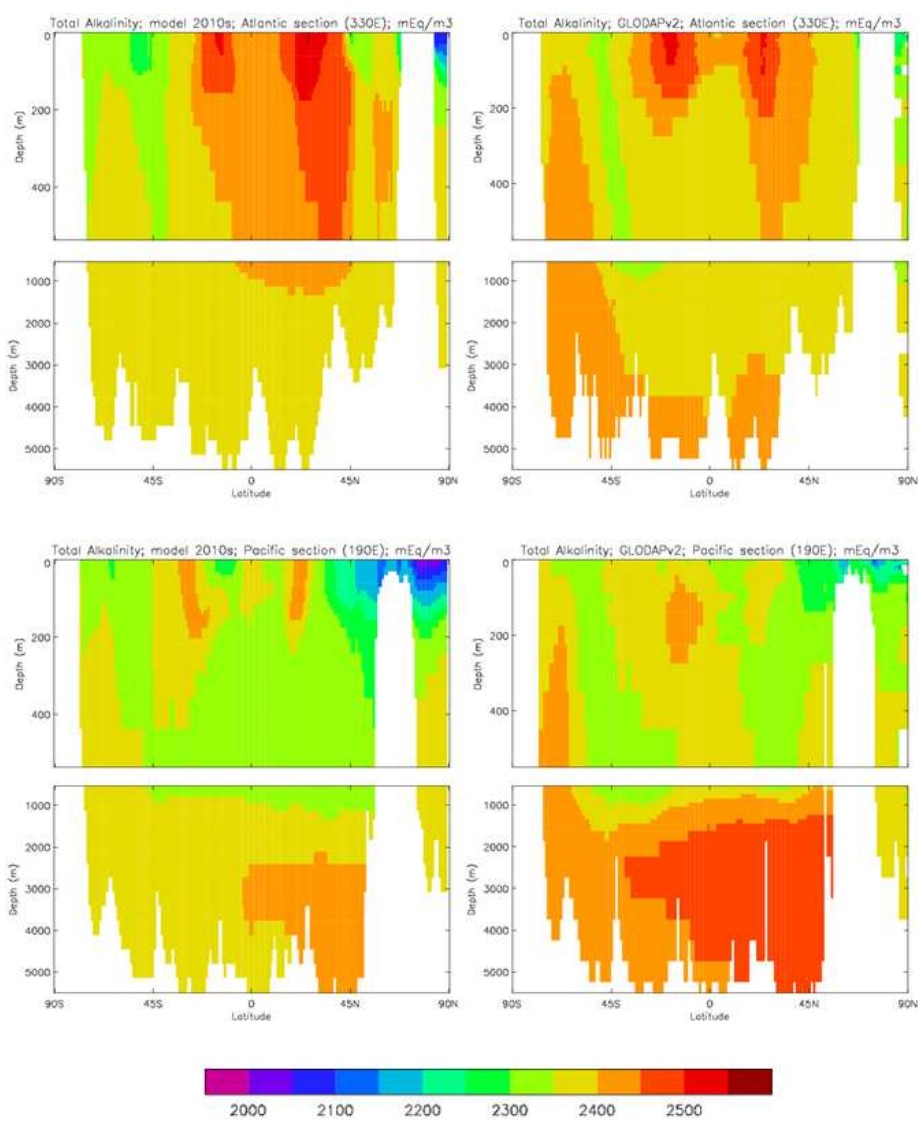

**Figure 13.** Meridional sections of Total Alkalinity: Upper panels show sections in the Atlantic Ocean along 330°, lower panels Pacific Ocean sections along 190°; left panels show model concentrations averaged over 2010-2019, right panels show concentrations from the GLODAPv2 gridded product.

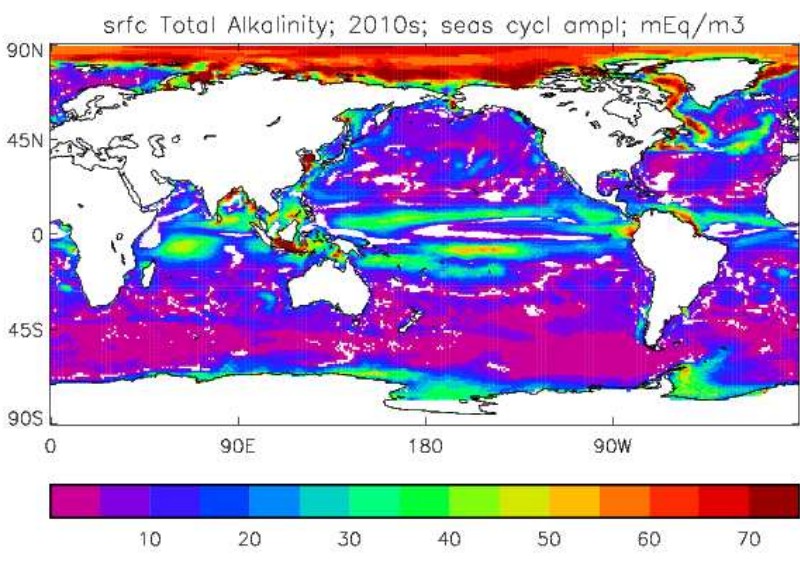

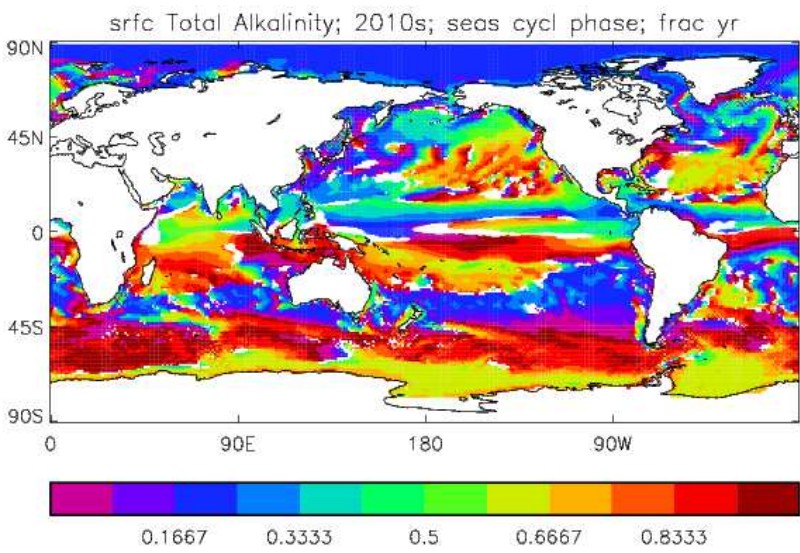

**Figure 14.** Surface Total Alkalinity, model seasonal cycle, averaged over model years 2010-2019 inclusive: upper panel, amplitude of cycle (mEq m$^{-3}$); lower panel, phase of cycle (fraction of year). Only points where the residual variance is less than half the original are shown.

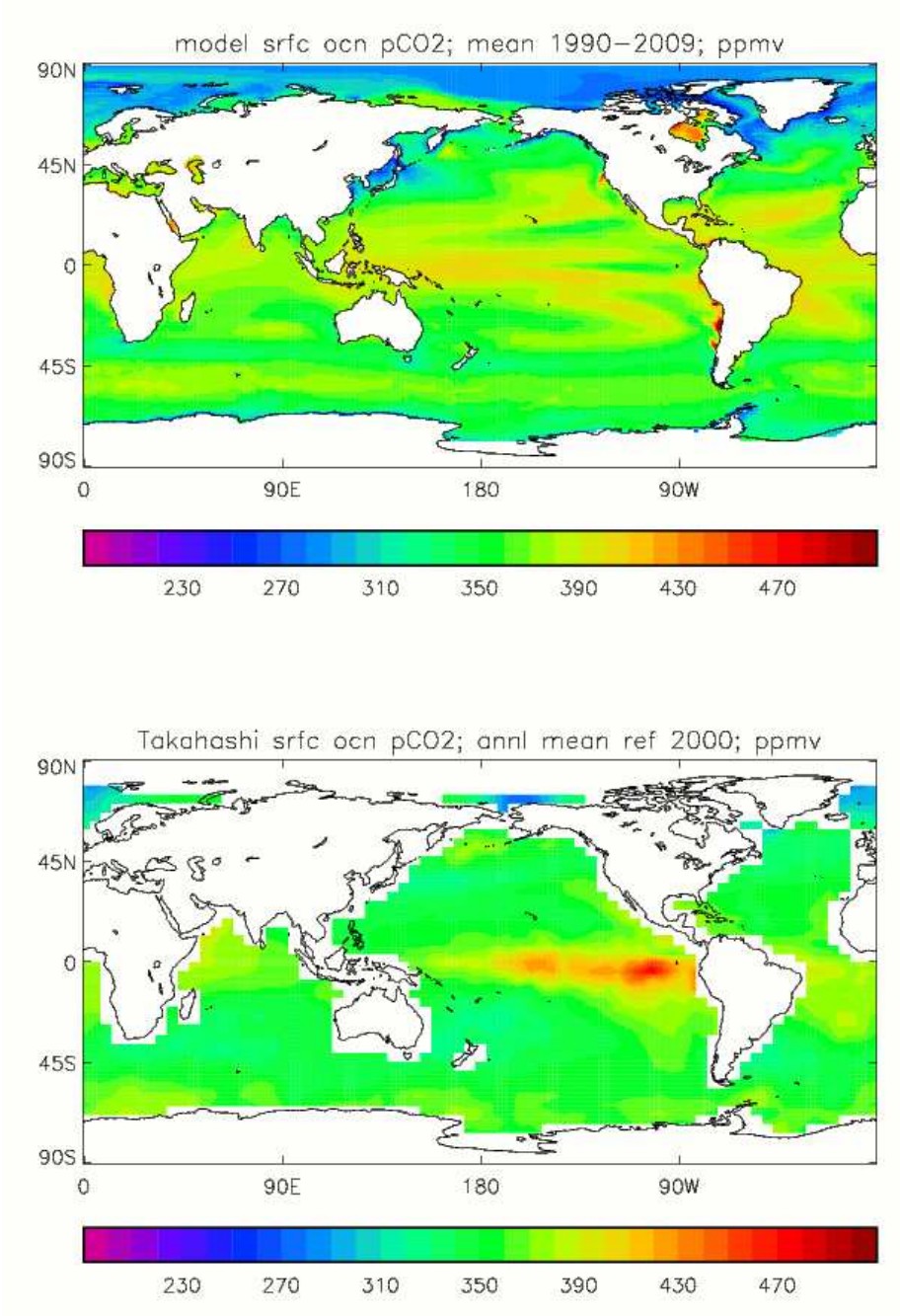

**Figure 15.** Surface ocean pCO2 (in ppmv): upper panel, model field averaged over the model years 1990-2009 inclusive; lower panel, Takahashi gridded field from data, annual mean, referenced to the year 2000

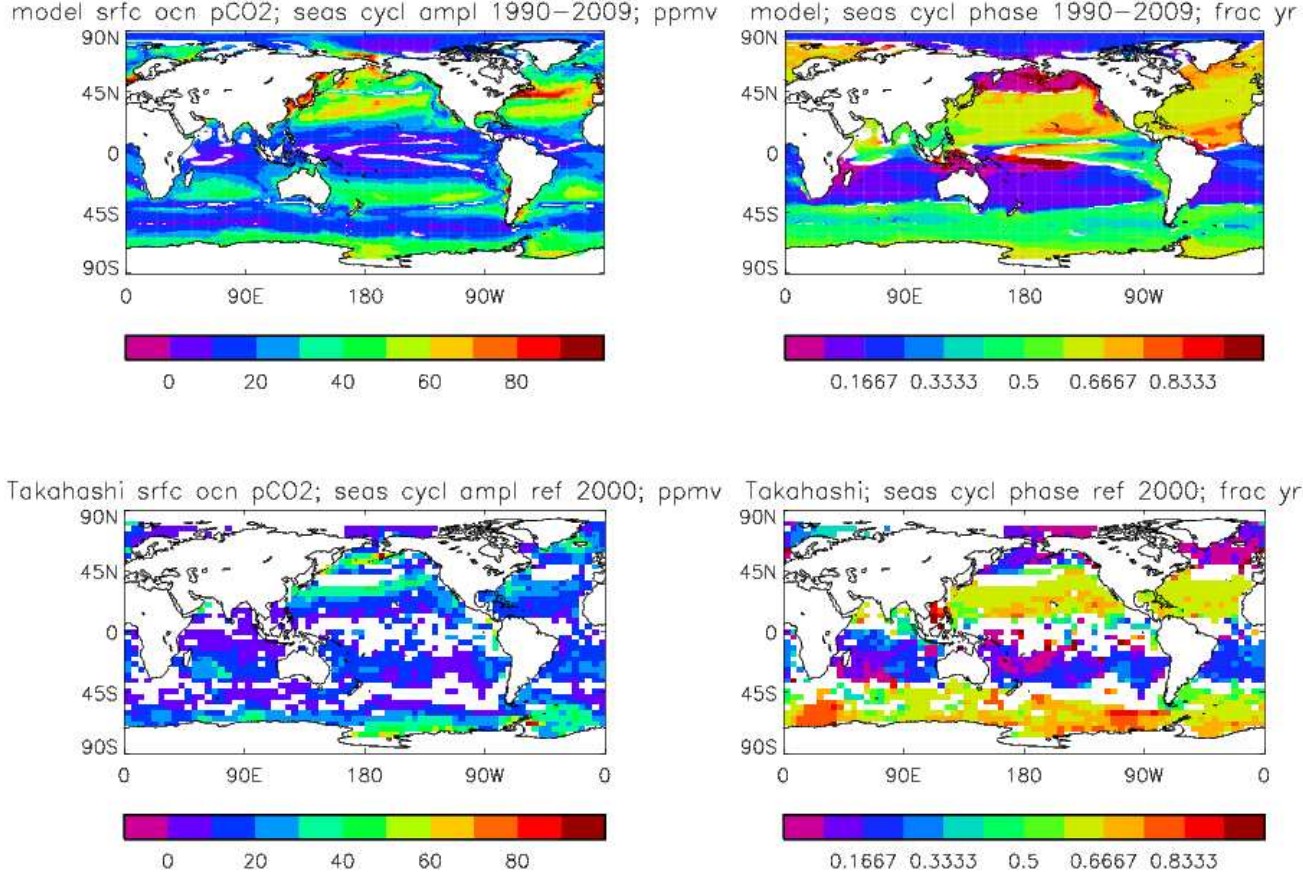

**Figure 16.** Surface ocean pCO2, seasonal cycle: upper panels, model, averaged over model years 1990-2009 inclusive; lower panels, Takahashi gridded data, referenced to the year 2000; left-hand panels, amplitude of the cycle (ppmv); right-hand panels, phase of the cycle (in fraction of year). Only points where the residual variance is less than half the original are shown.

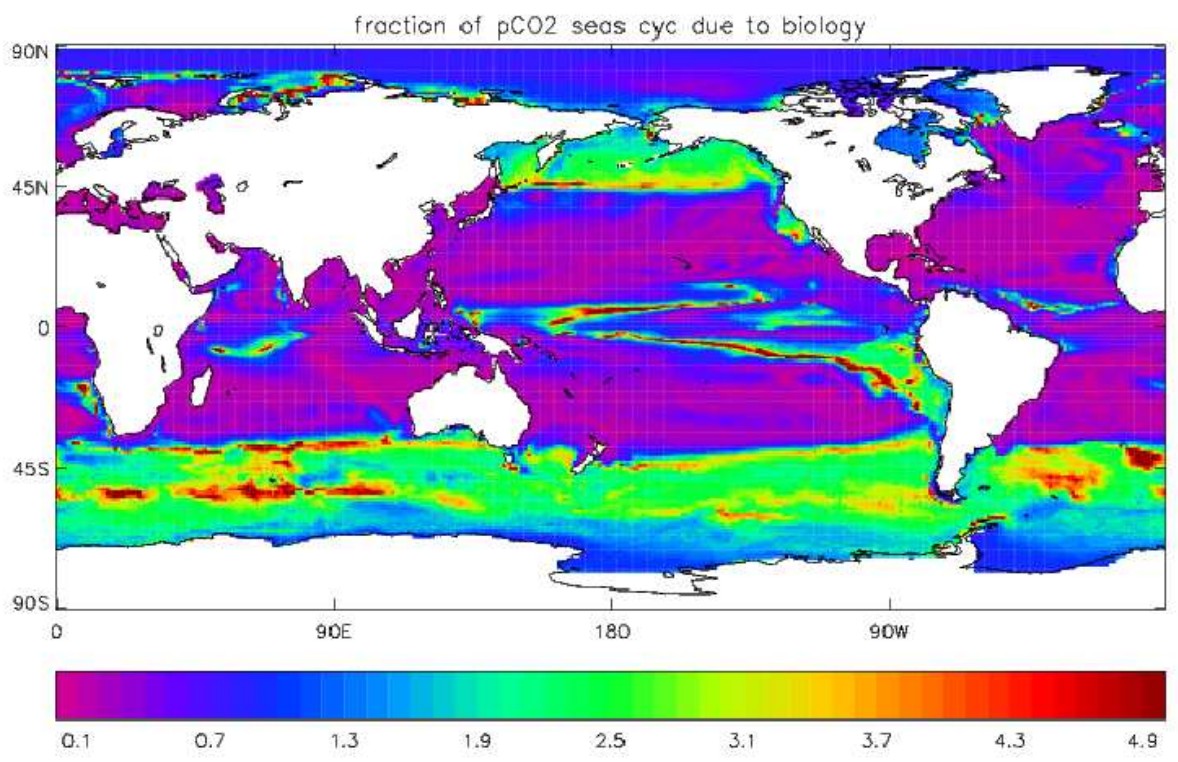

**Figure 17.** Fraction of the seasonal cycle of pCO₂ that is not driven by the temperature (and salinity) seasonal variation. The details of the calculation are given in the text. Where the ratio is less than 0.5, the temperature variation dominates, and where the ratio is greater than 0.5 the biological uptake/respiration (and the air-sea uptake) dominate. Ratios greater than 1.0 indicate that the biologically-driven and temperature-driven cycles are opposed.

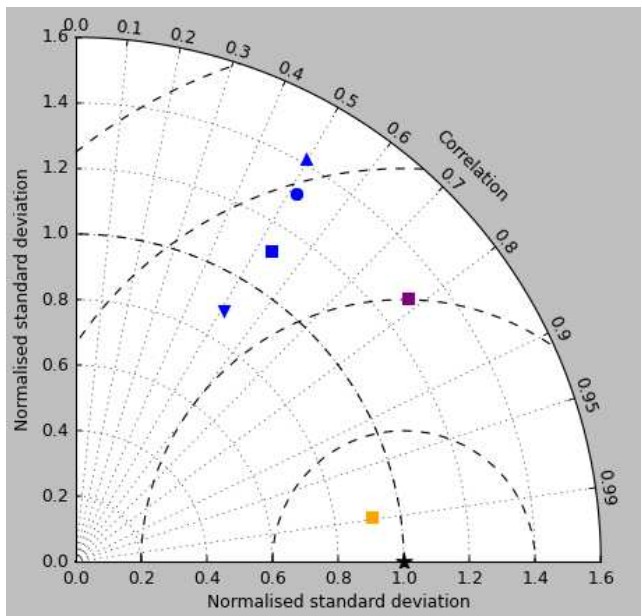

**Figure 18.** Taylor diagram for surface DIC (orange), surface Total Alkalinity (purple) and surface pCO2 (blue). Model DIC and Total Alkalinity from the RCP8.5 simulation (meaned over the years 2010 to 2019 inclusive) are compared to the gridded fields from GLODAPv2, while model pCO2 (meaned over 1990 to 2009 inclusive) is compared to the Takahashi gridded data. Filled squares refer to the raw surface fields, and filled circles, upward-pointing triangles and downward triangles respectively refer to the mid-point, amplitude and phase of the sine-curve that best fits the seasonal cycle (in points where the variance of the residual is less than half that of the original cycle).

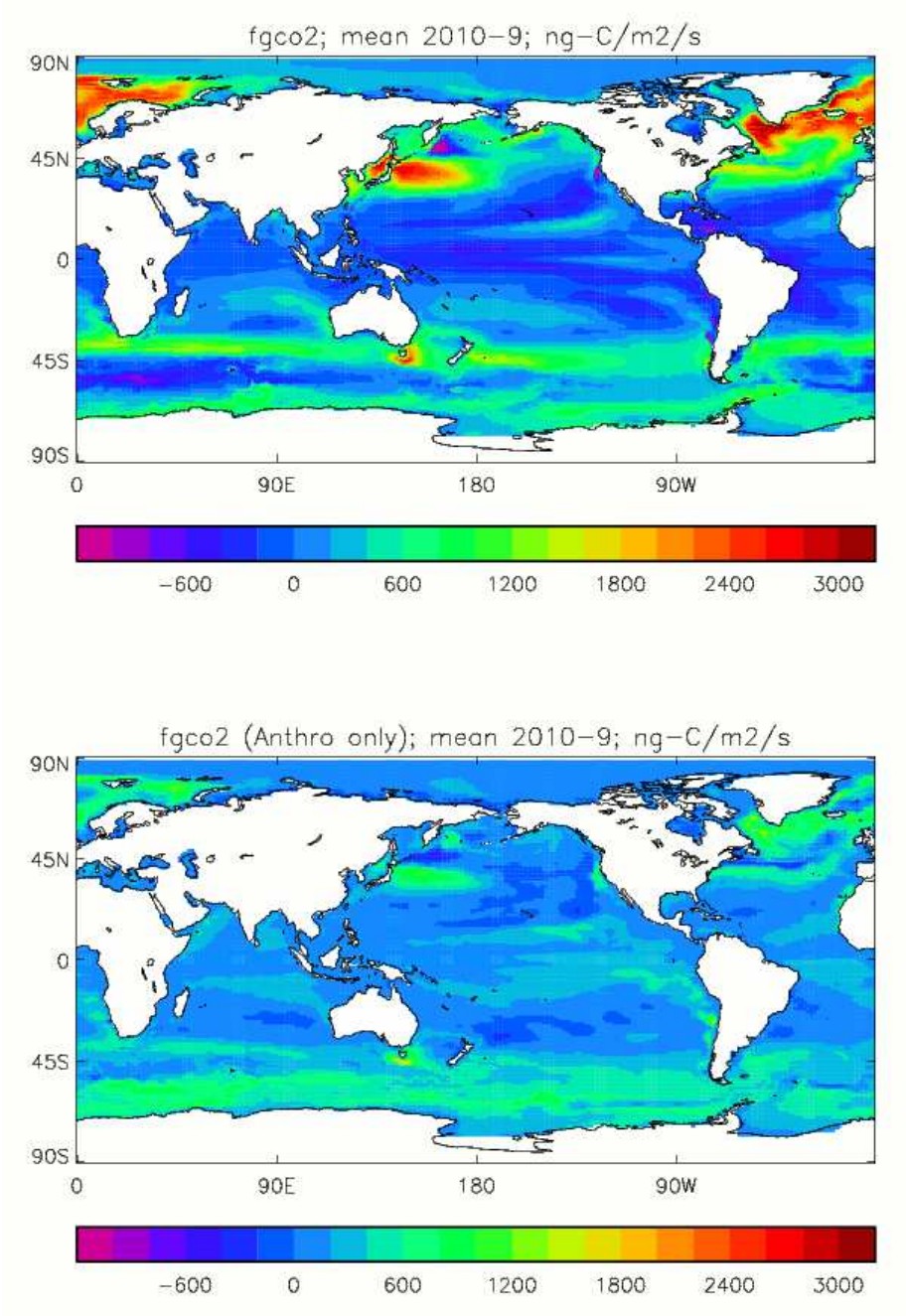

**Figure 19.** Total air-to-sea flux of $CO_2$ (ng C m$^{-2}$ s$^{-1}$; positive values into the ocean), mean over model years 2010-2019 inclusive: upper panel, total flux (natural cycle and anthropogenic perturbation); lower panel, anthropogenic perturbation

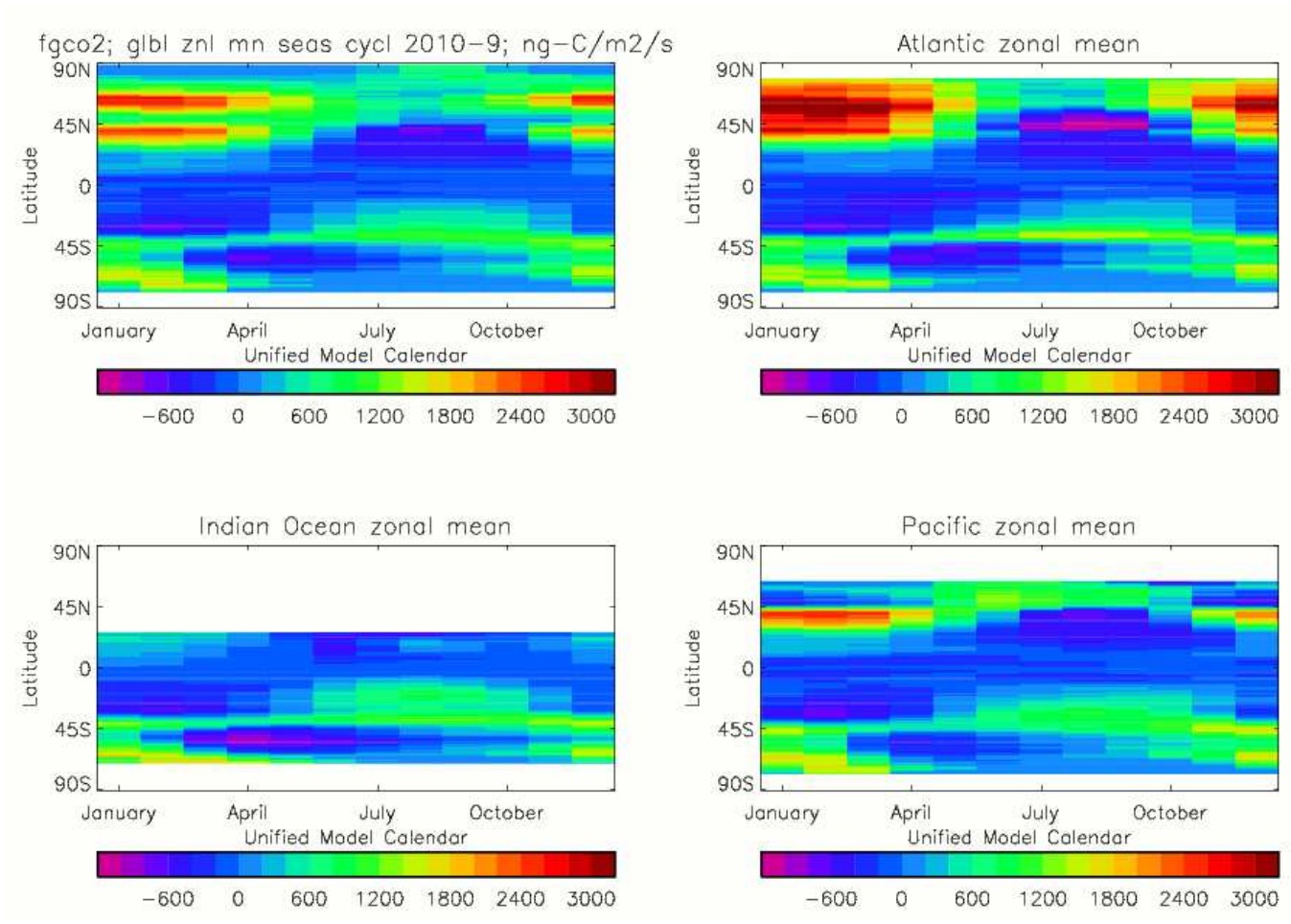

**Figure 20.** Total air-to-sea flux of $CO_2$ (ng C m$^{-2}$ s$^{-1}$), seasonal cycle averaged for each month over the model years 2010-2019 inclusive, zonally-meaned: upper left panel, global zonal mean; upper right, zonal mean of the Atlantic Ocean basin; lower left, zonal mean of the Indian Ocean basin; lower right, zonal mean of the Pacific Ocean basin

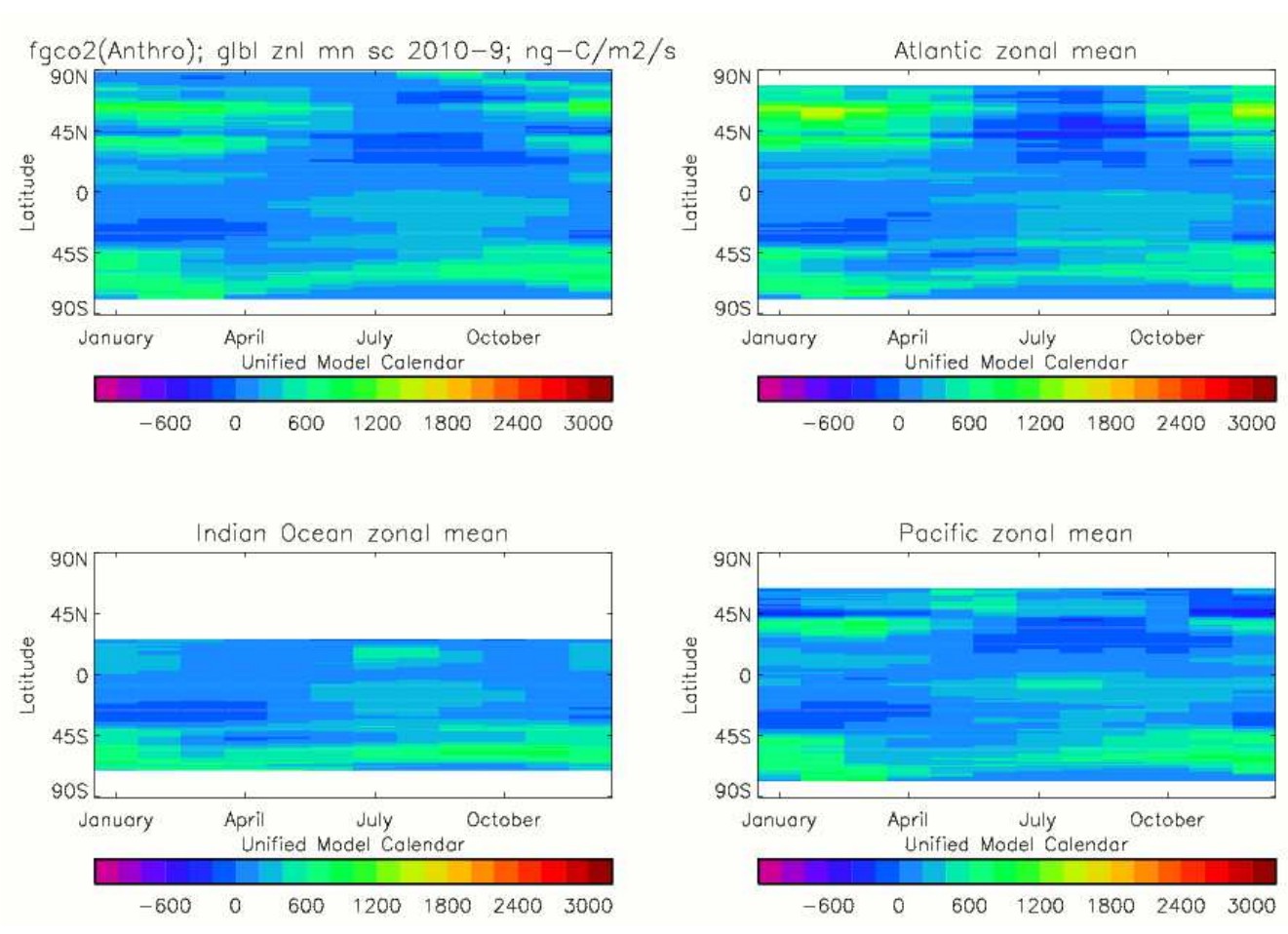

**Figure 21.** As Figure 20, but for the air-to-sea flux of anthropogenic $CO_2$ only (ng C m$^{-2}$ s$^{-1}$)

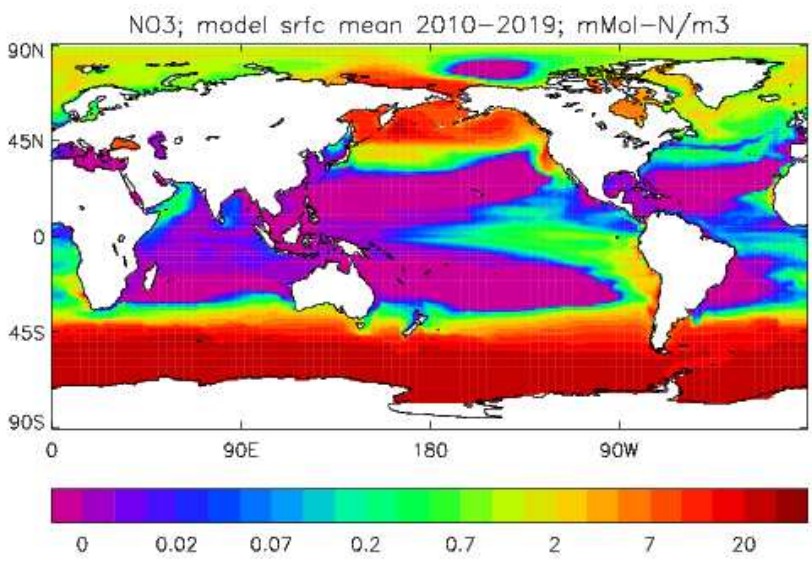

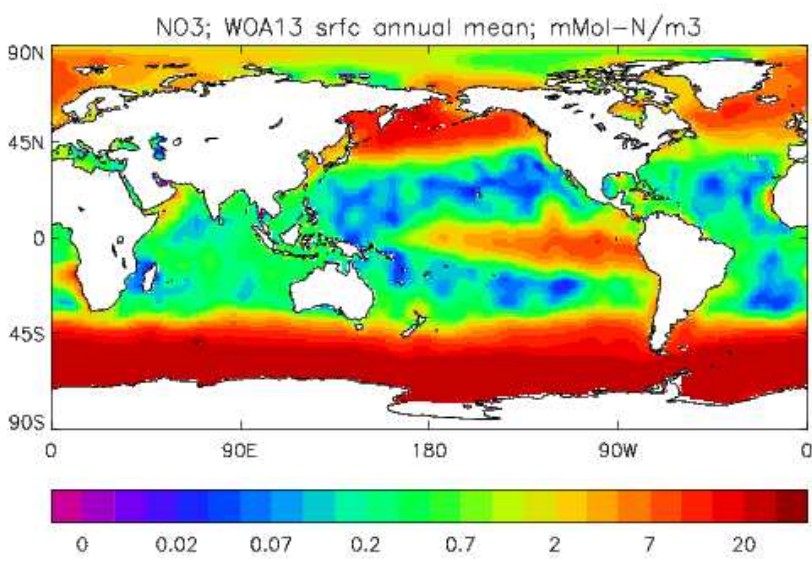

**Figure 22.** Surface dissolved nitrate (mMol N m$^{-3}$): upper panel, model field averaged over model years 2010-2019 inclusive; lower panel, the gridded field from the 2013 World Ocean Atlas

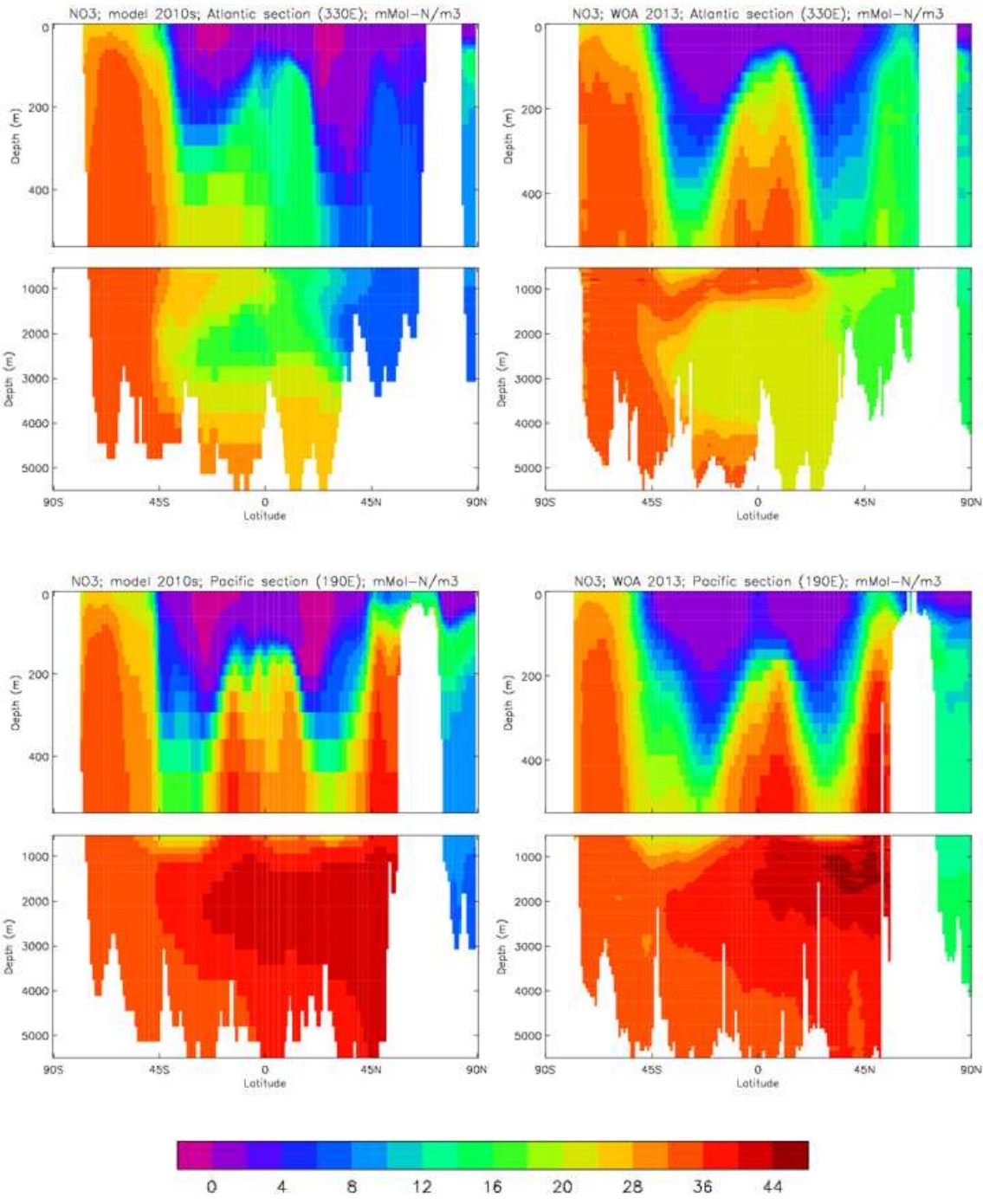

**Figure 23.** Meridional sections of disolved nitrate (mMol N m$^{-3}$): upper panels show sections in the Atlantic Ocean along 330°, lower panels Pacific Ocean sections along 190°; left panels show model concentrations averaged over 2010-2019, right panels show concentrations from the 2013 World Ocean Atlas gridded field.

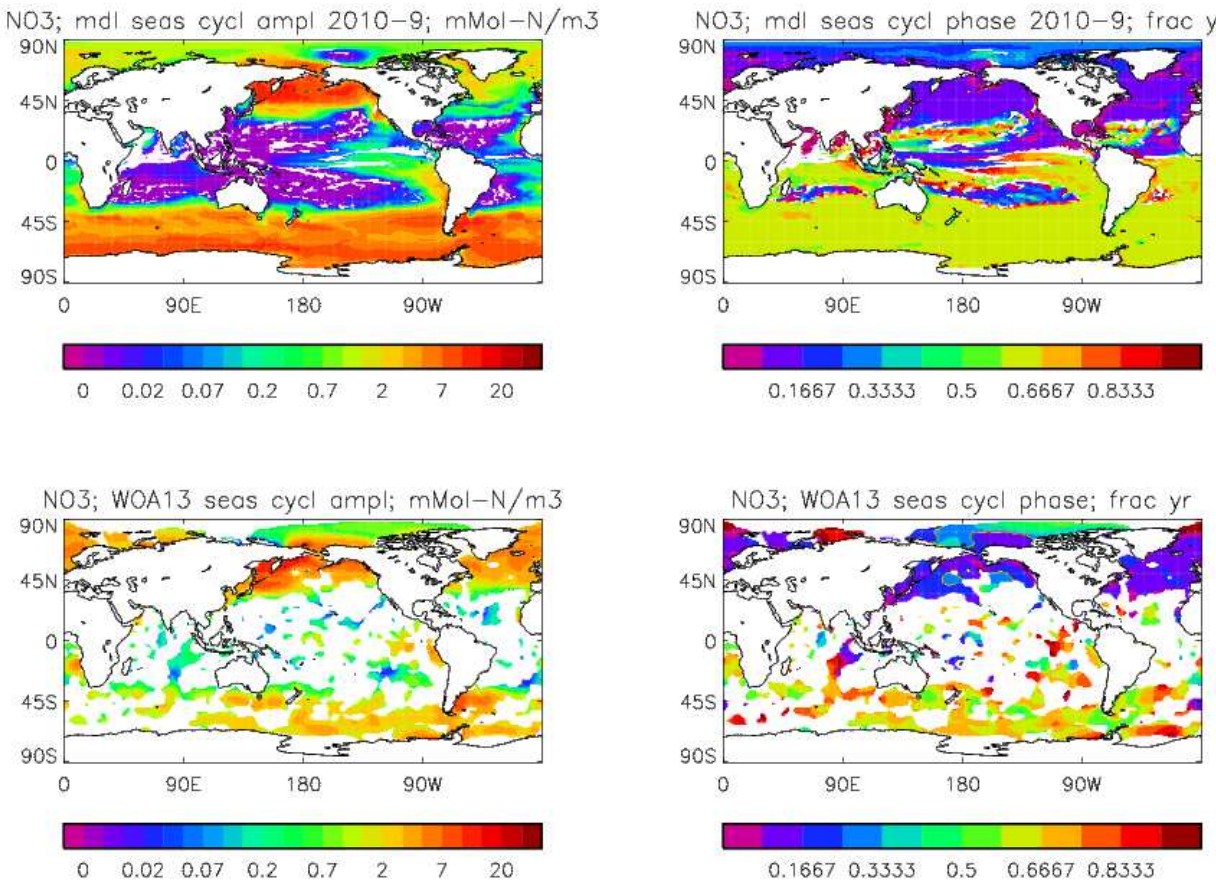

**Figure 24.** Surface dissolved nitrate, seasonal cycle: upper panels, model cycle, averaged over model years 2010-2019 inclusive; lower panels, the cycle from the monthly gridded fields from the 2013 World Ocean Atlas; left-hand panels, the amplitude of the cycle (mMol N m$^{-3}$); right-hand panels, the phase of the cycle (fraction of year). Only points where the residual variance is less than half the original are shown.

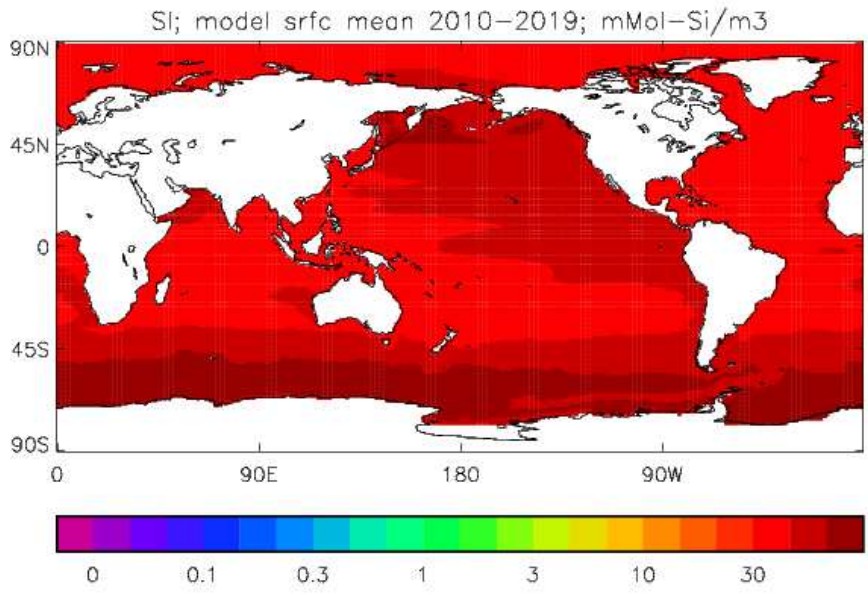

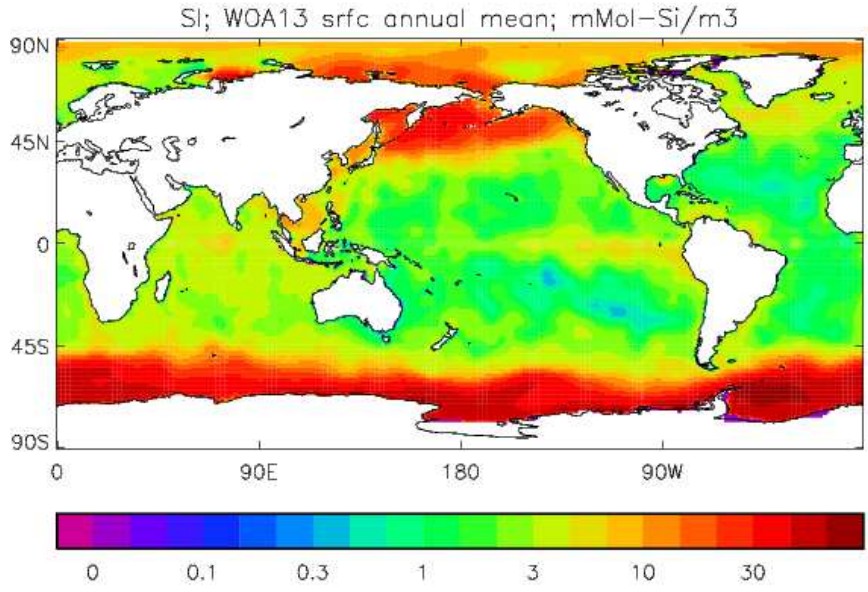

**Figure 25.** Surface dissolved silicate (mMol Si m$^{-3}$): upper panel, model field averaged over model years 2010-2019 inclusive; lower panel, the gridded field from the 2013 World Ocean Atlas

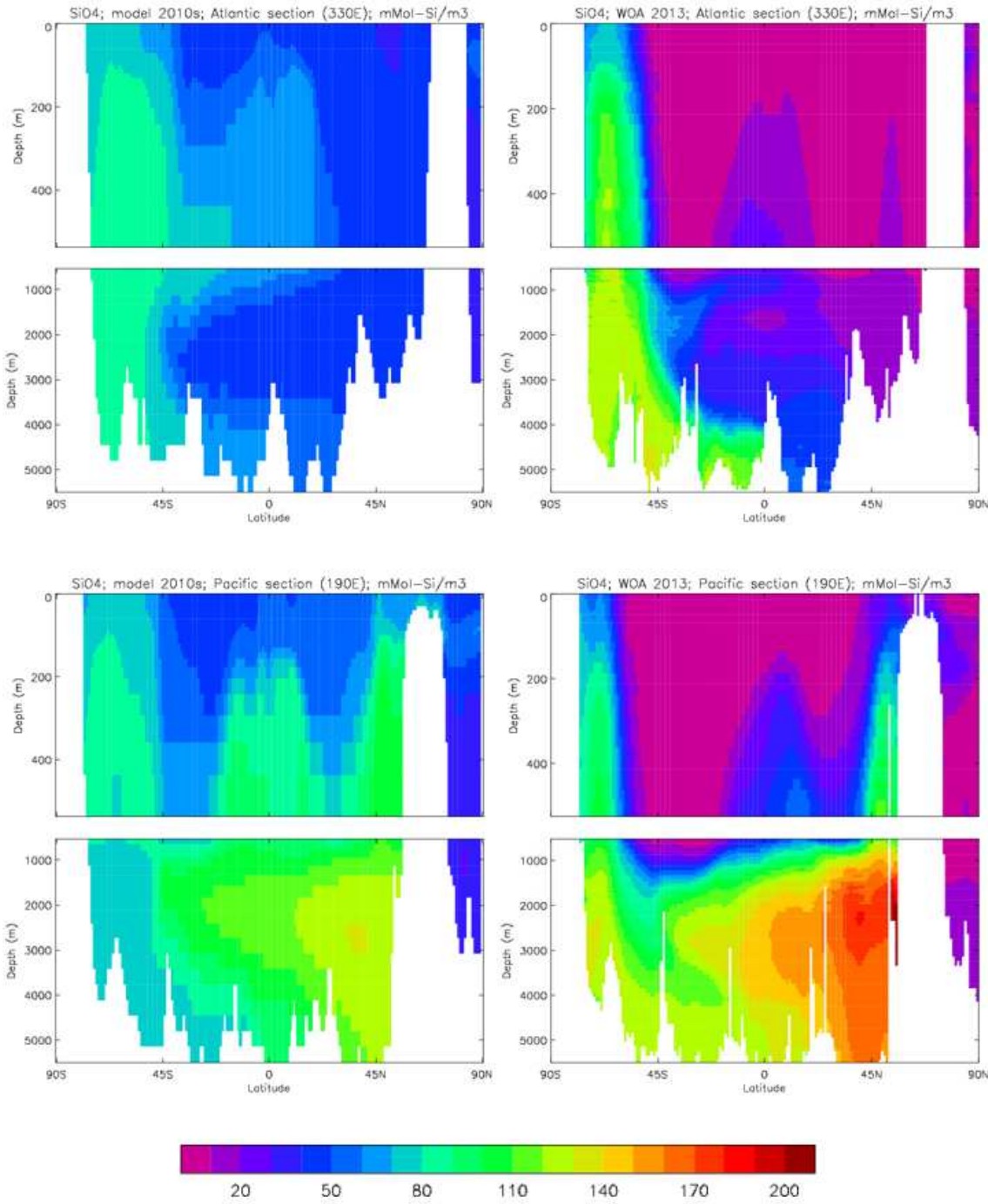

**Figure 26.** Meridional sections of disolved silicate (mMol Si m$^{-3}$): upper panels show sections in the Atlantic Ocean along 330°, lower panels Pacific Ocean sections along 190°; left panels show model concentrations averaged over 2010-2019, right panels show concentrations from the 2013 World Ocean Atlas gridded field.

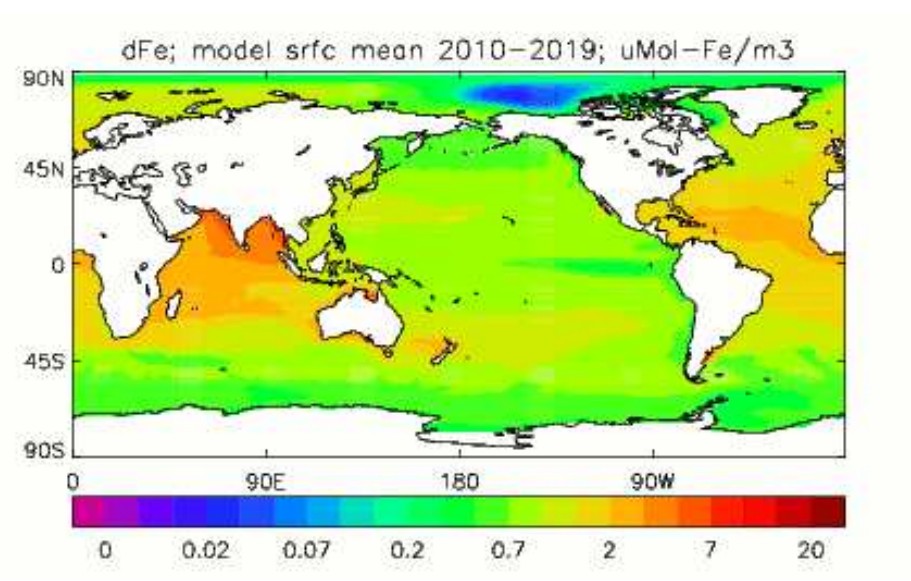

**Figure 27.** Surface dissolved iron, averaged over model years 2010-2019 inclusive (uMol Fe m$^{-3}$).

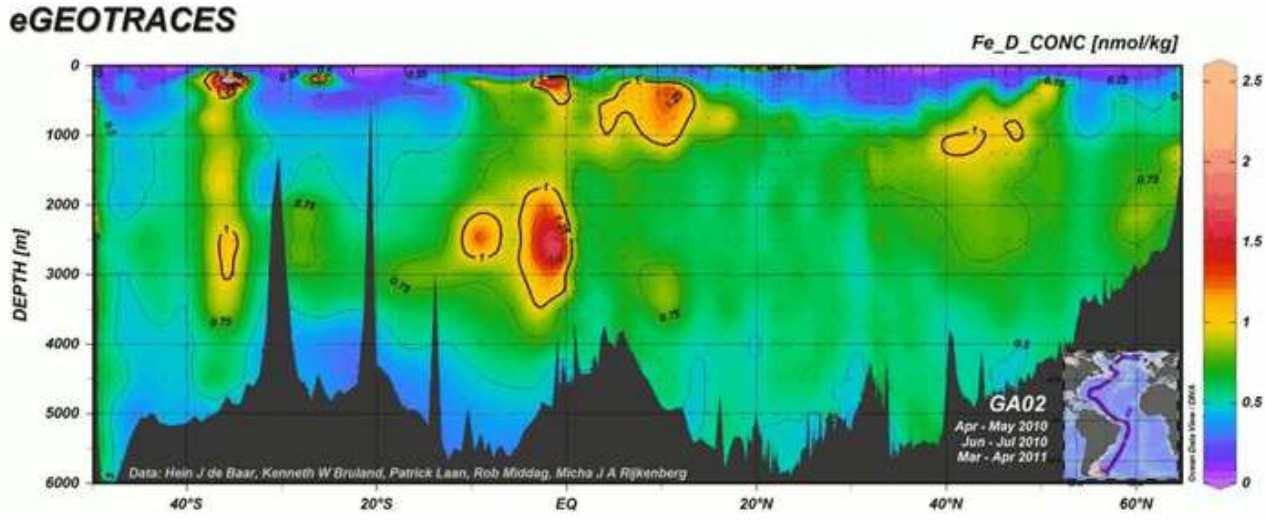

**Figure 28.** Sections of dissolved iron (uMol Fe m$^{-3}$) along the eGEOTRACES GA02 transect (in the Atlantic Ocean): upper panel shows model concentrations averaged over years 2010 to 2019 inclusive, lower panel is reproduced from eGEOTRACES.

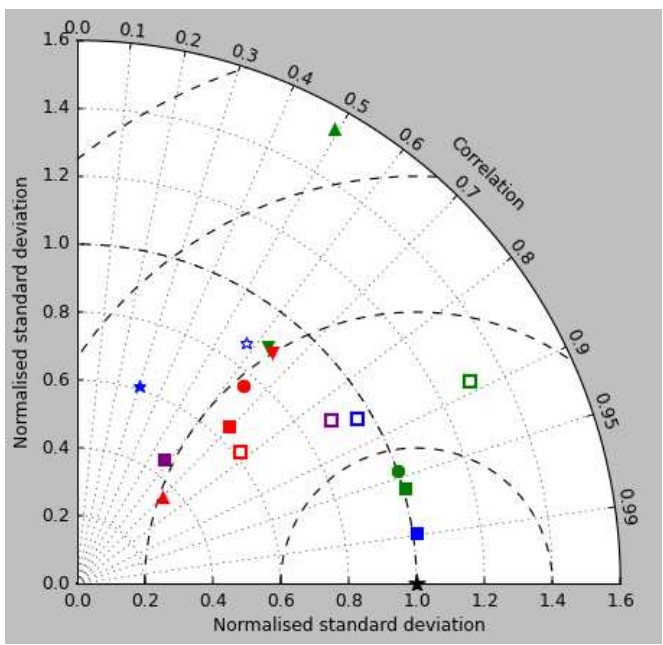

**Figure 29.** Taylor diagram of nitrate (green), silicate (red), oxygen (blue) and Apparent Oxygen Utilisation (purple); model values are averaged over years 2010 to 2019 inclusive, observations are from the 2013 World Ocean Atlas. Filled squares show the surface concentrations, open squares the concentrations at the nearest depth level to 1050 m, and circles, upward-pointing triangles and downward triangles respectively the mid-point, amplitude and phase of the sine-curve that best fits the seasonal cycle (only at points where the residual variance is less than half the original). The filled and open stars show respectively the vertical extent of the water-column where $O_2$ concentration is below 50 and 100 mMol $O_2$ m$^{-3}$.

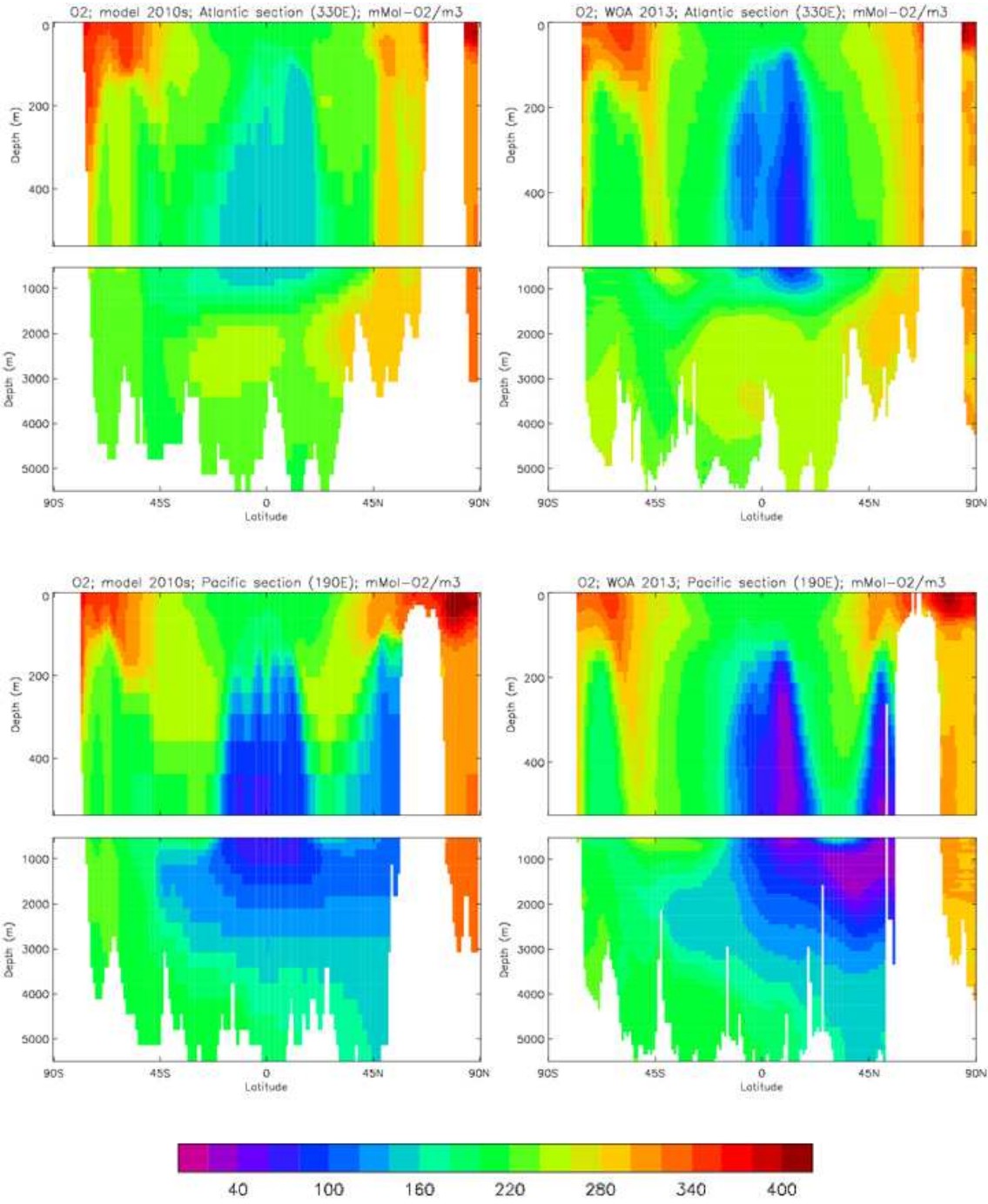

**Figure 30.** Meridional sections of disolved oxygen (mMol $O_2$ m$^{-3}$): upper panels show sections in the Atlantic Ocean along 330°, lower panels Pacific Ocean sections along 190°; left panels show model concentrations averaged over 2010-2019, right panels show concentrations from the 2013 World Ocean Atlas gridded field.

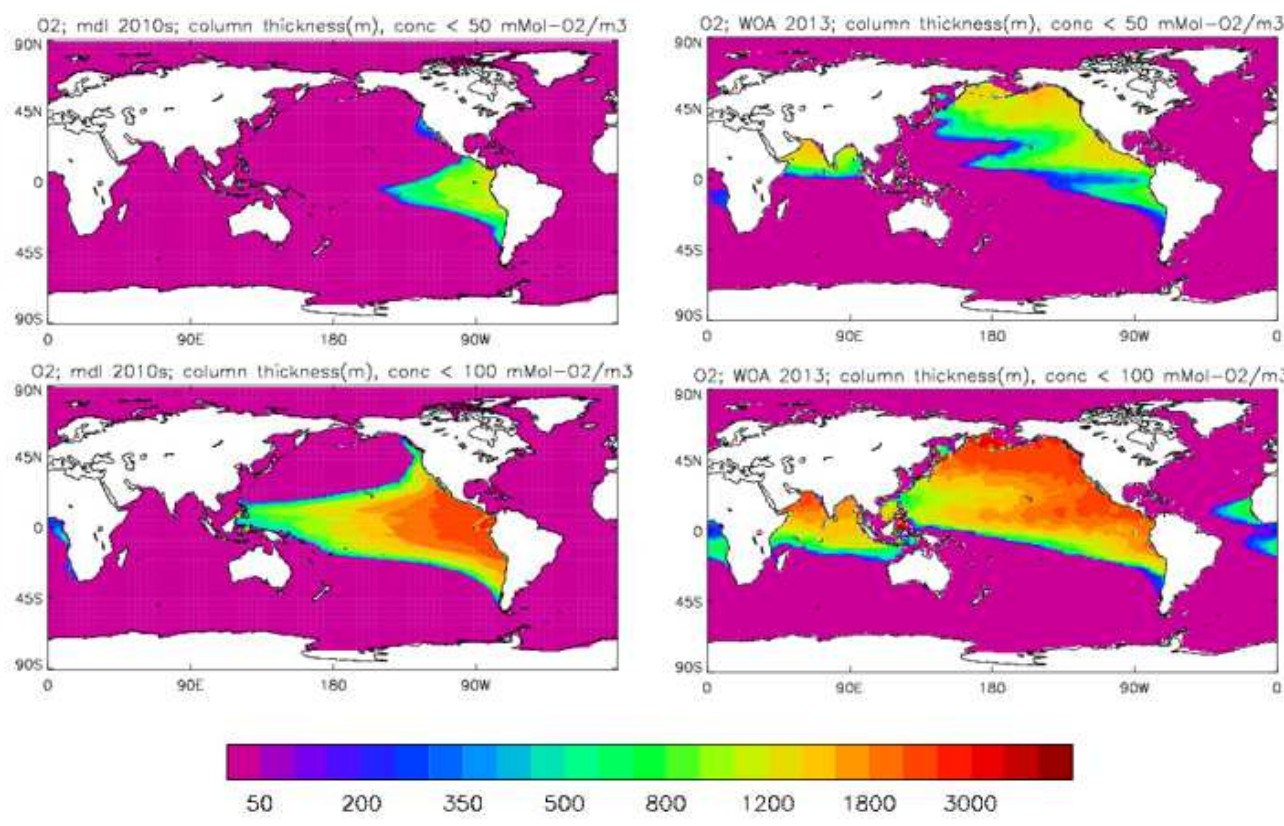

**Figure 31.** Thickness (m) of the oxygen depletion zone in the water-column; left panels from the model (averaged over the years 2010 to 2019 inclusive), right panels from the 2013 World Ocean Atlas. Upper and lower panels show the extent of the water-column in which $O_2$ concentrations are respectively below 50 and 100 mMol $O_2$ m$^{-3}$.

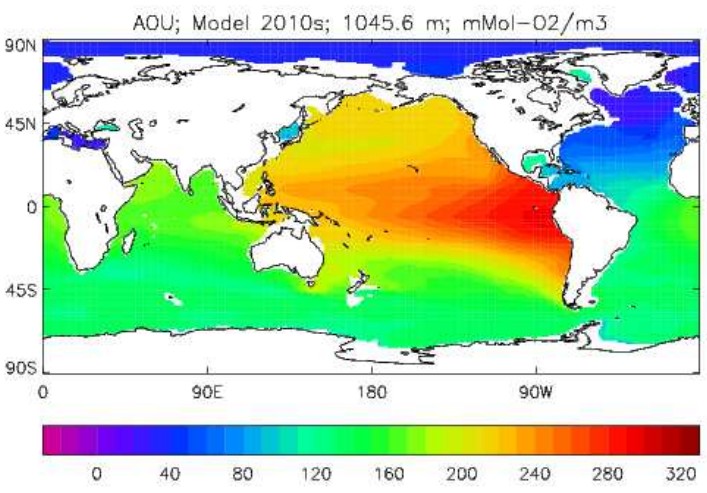

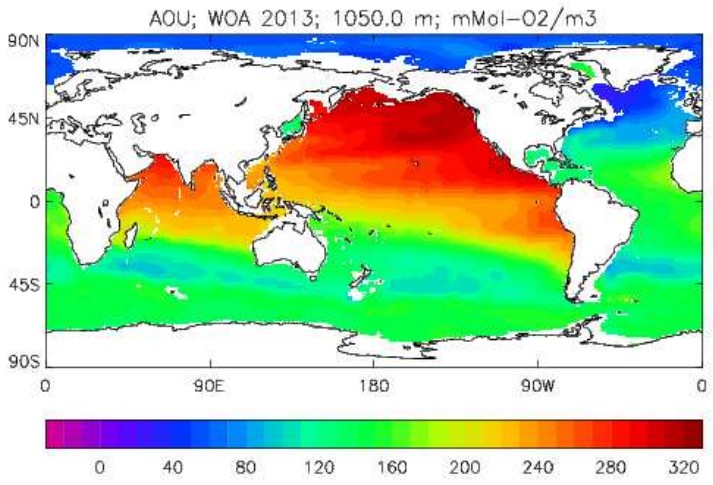

**Figure 32.** Apparent Oxygen Utilisation (mMol $O_2$ m$^{-3}$): upper panel, model field at 1045.6 m, averaged over years 2010 to 2019 inclusive; lower panel, field at 1050 m from the 2013 World Ocean Atlas.

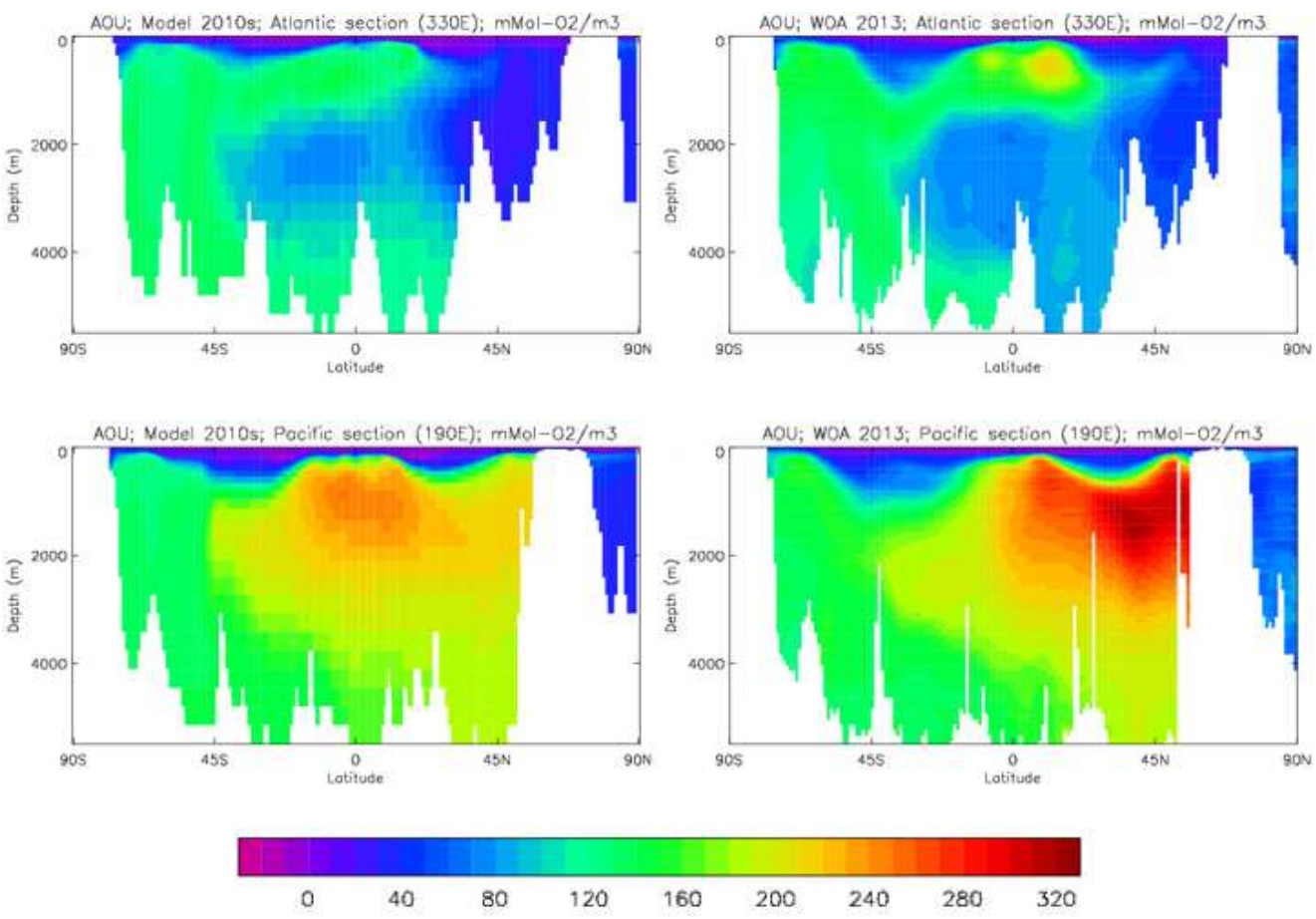

**Figure 33.** Meridional sections of Apparent Oxygen Utilisation (mMol $O_2$ m$^{-3}$): upper panels show sections in the Atlantic Ocean along 330°, lower panels Pacific Ocean sections along 190°; left panels show model concentrations averaged over 2010-2019, right panels show concentrations from the 2013 World Ocean Atlas gridded field.

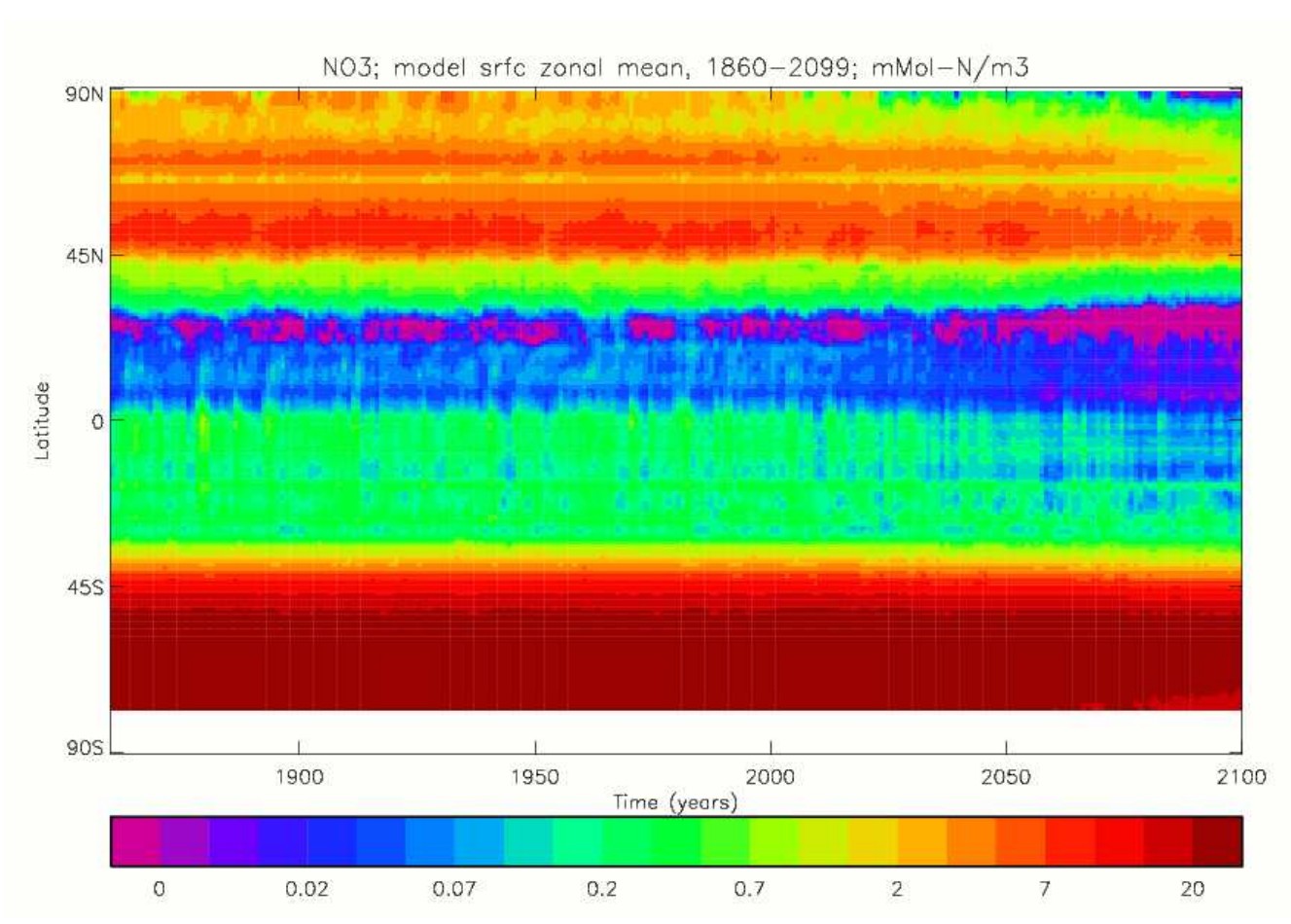

**Figure 34.** Surface dissolved nitrate concentration (mMol N m$^{-3}$), global zonal and annual means for model years 1860 to 2099, from the CMIP5 Historical and RCP8.5 simulations, showing the response to changing climatic forcing

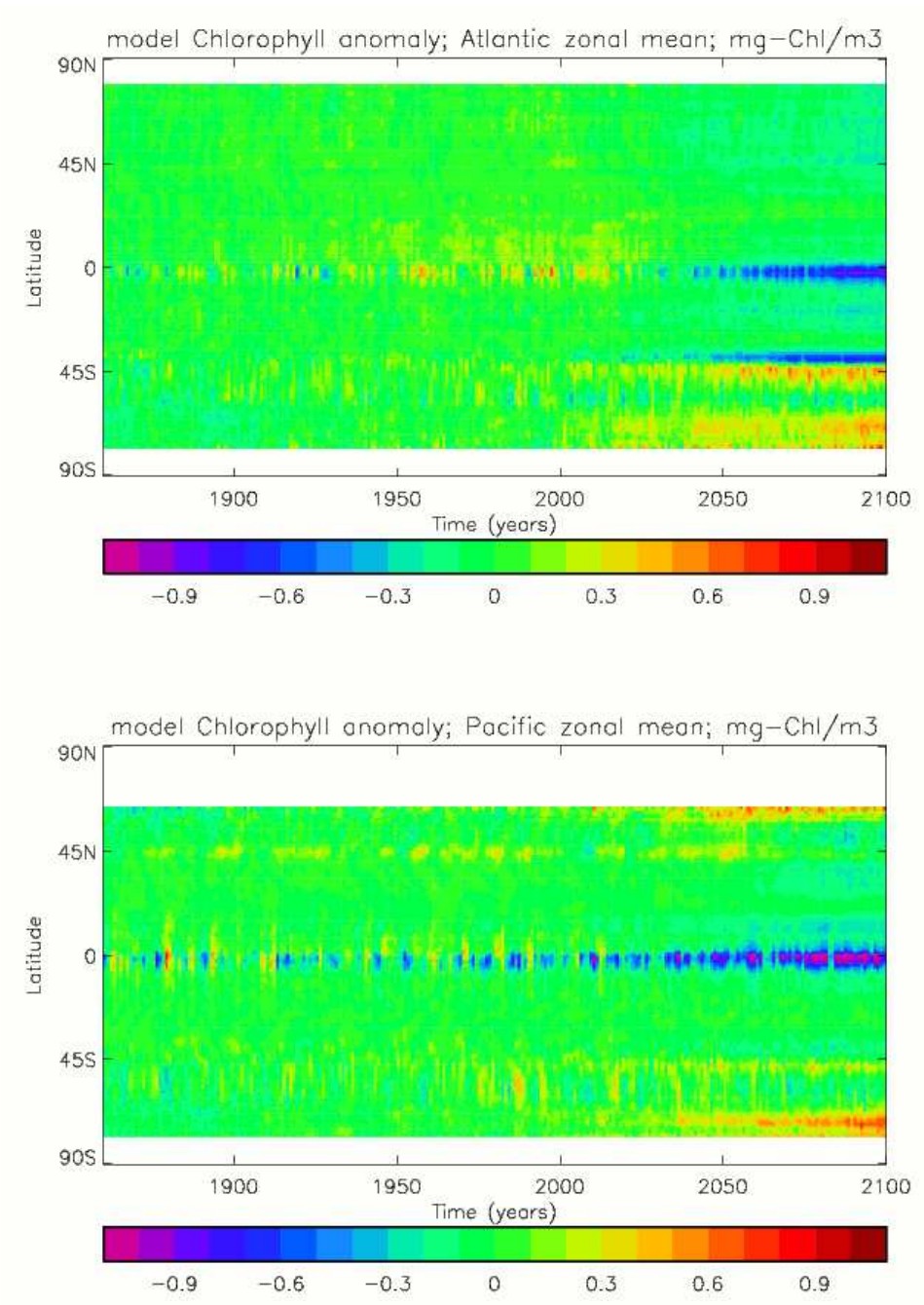

**Figure 35.** Surface total chlorophyll concentration anomaly (mg Chl m$^{-3}$), zonal and annual means for model years 1860 to 2099, from the CMIP5 Historical and RCP8.5 simulations: upper panel, zonal mean of the Atlantic Ocean basin; lower panel, zonal mean of the Pacific Ocean basin. The anomaly has been calculated by subtracting the surface chlorophyll concentration field, meaned over the years 1860 to 2099 inclusive, as produced by the piControl simulation from the annual means of the Historical and RCP8.5 simulations

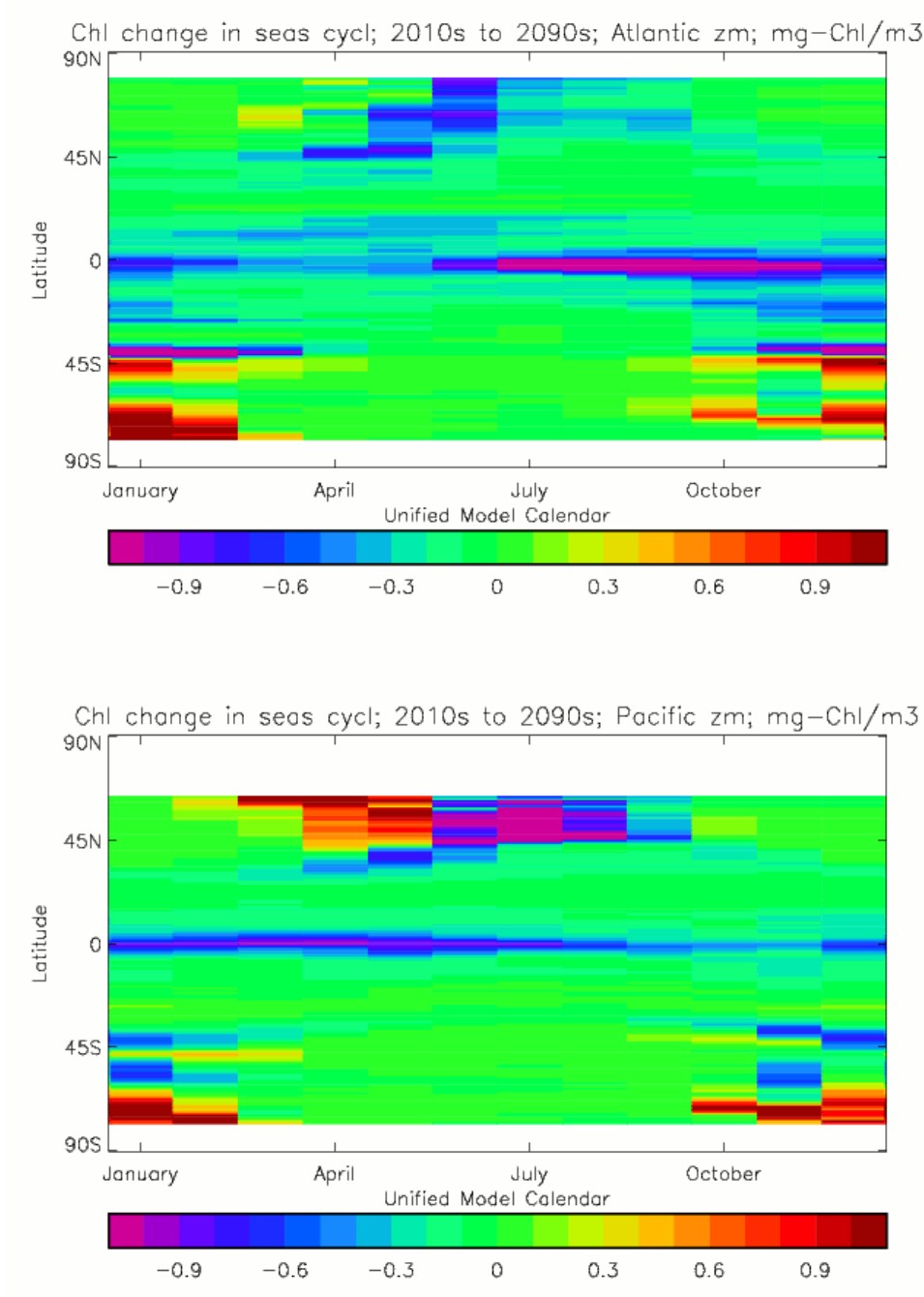

**Figure 36.** Change in the seasonal cycle of surface chlorophyll concentration in the CMIP5 RCP8.5 simulation: change is calculated between the mean seasonal cycles of the model years 2090-2099 and 2010-2019. Zonal means of the (upper panel) Atlantic Ocean basin and (lower panel) Pacific Ocean basin

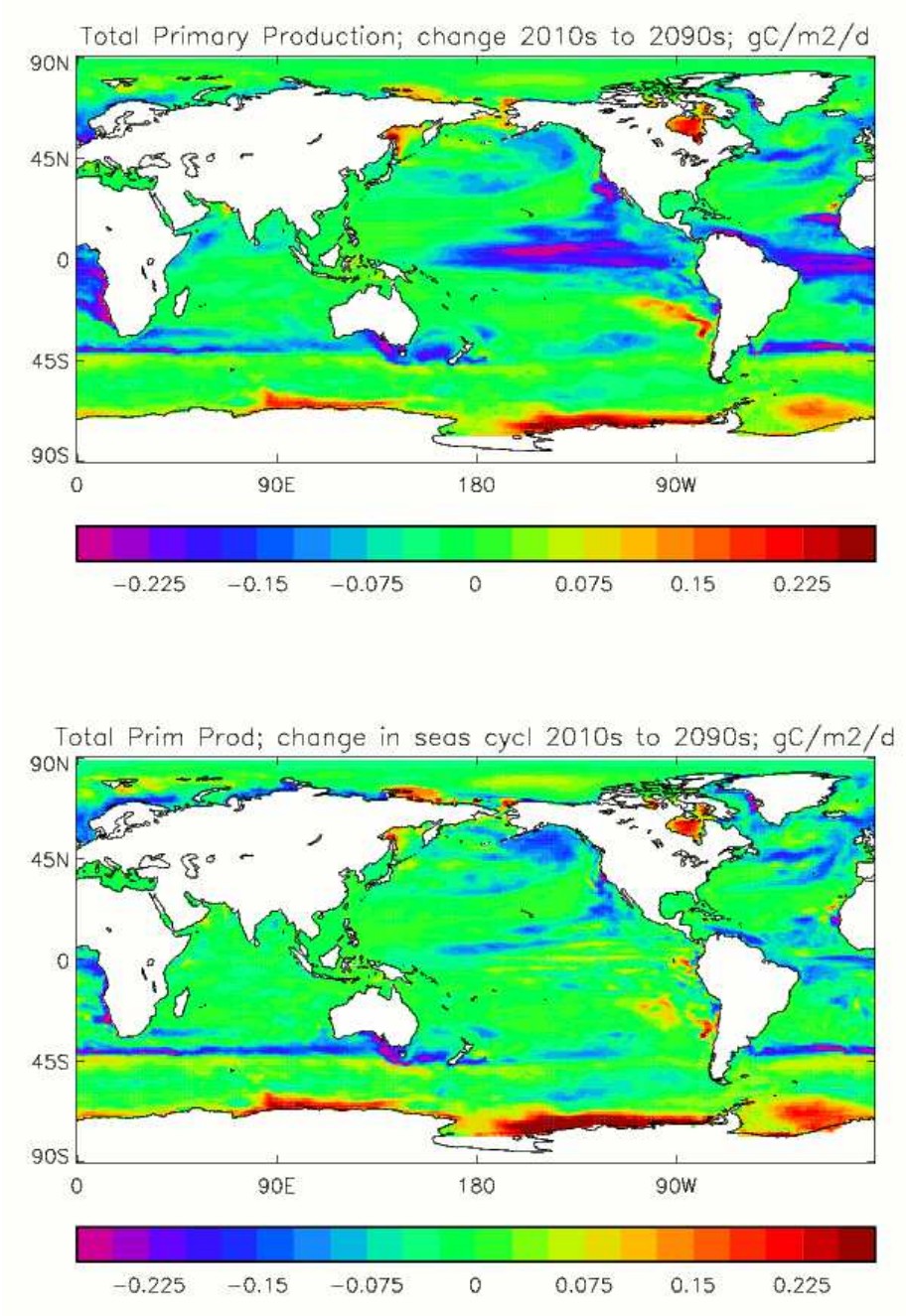

**Figure 37.** Change in the depth-integrated total Primary Production (mg C m$^{-2}$ d$^{-1}$) in the RCP8.5 simulation: difference between the model years 2090-2099 and 2010-2019. Upper panel: difference in decadal means; lower panel: difference in amplitude of mean seasonal cycle

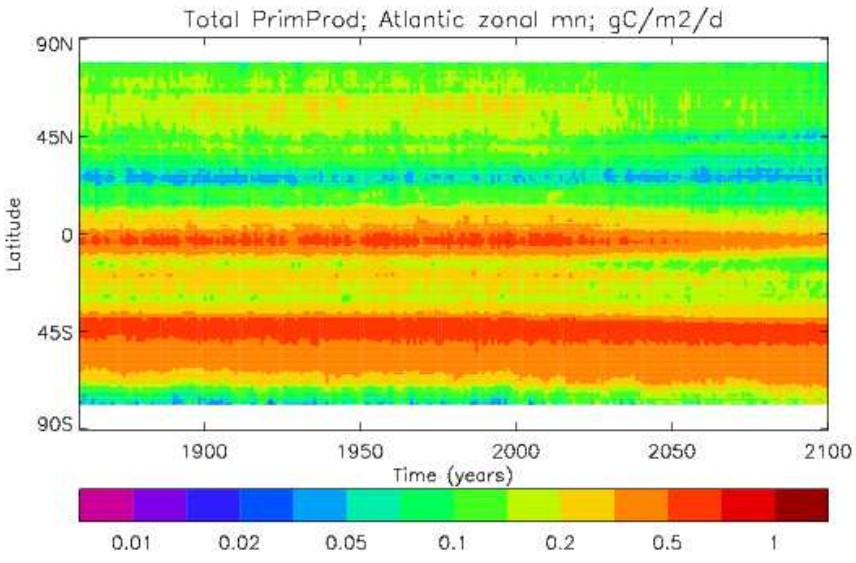

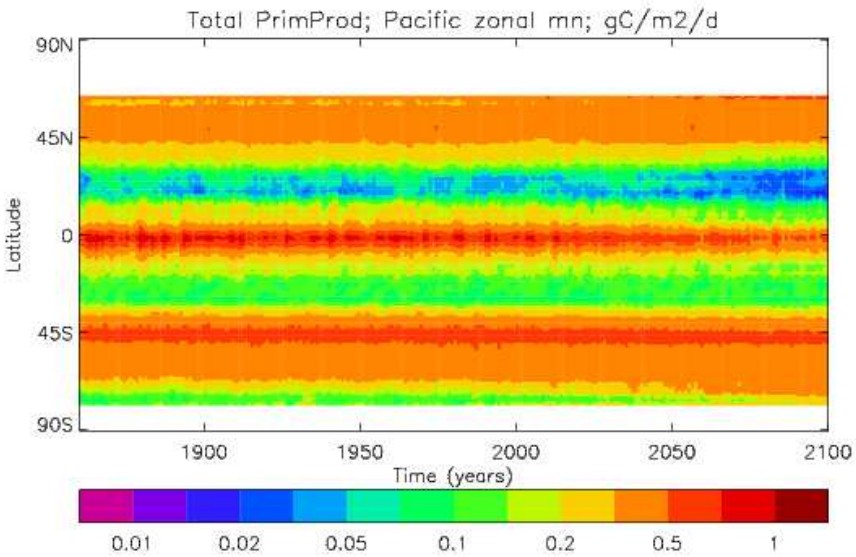

**Figure 38.** Change in annual mean depth-integrated total Primary Production (g C m$^{-2}$ d$^{-1}$) during the model years 1860 to 2099 in the CMIP5 Historical and RCP8.5 simulations. Upper panel, Atlantic Ocean basin zonal mean; lower panel, Pacific Ocean basin zonal mean.

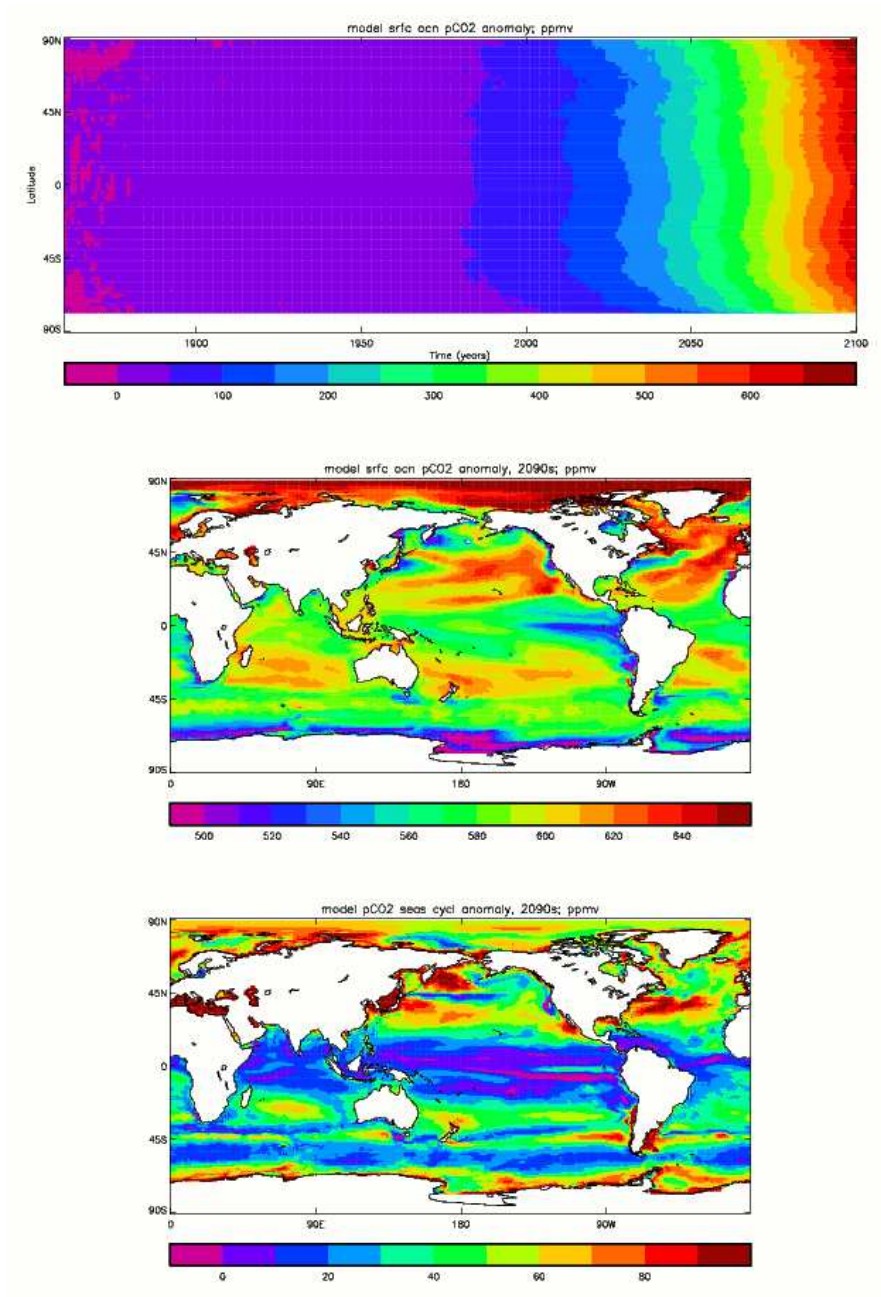

**Figure 39.** Change in surface $pCO_2$ (ppmv) during the model years 1860 to 2099 in the CMIP5 Historical and RCP8.5 simulations. Top panel: the anomaly over the period of the simulations, calculated by subtracting the annual means of the piControl simulation from those of the Historical and RCP8.5 simulations. Middle panel: the decadal mean anomaly during the model years 2090-2099, calculated by subtracting the relevant years of the piControl from those of the RCP8.5 simulation. Bottom panel: the seasonal cycle amplitude anomaly averaged over the model years 2090-2099, calculated as for the middle panel

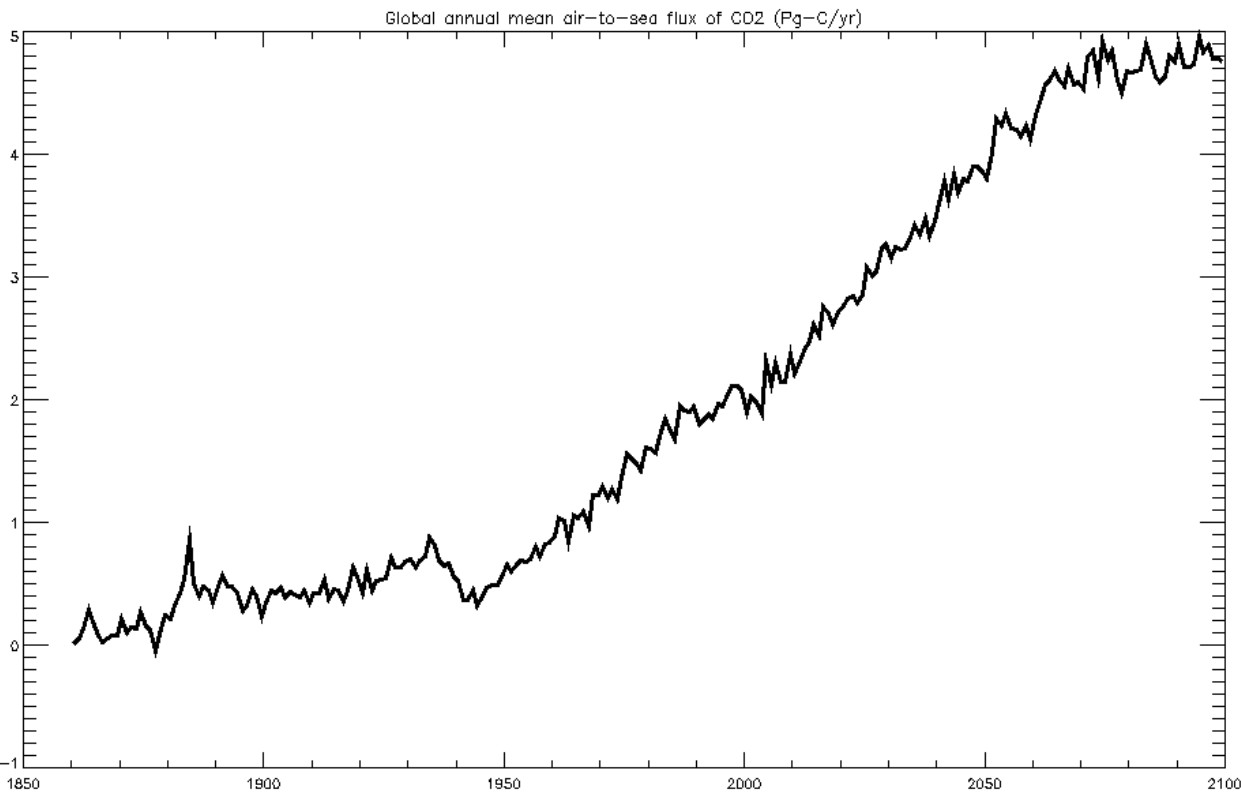

**Figure 40.** Time-evolution of the annual mean global total air-to-sea $CO_2$ flux (Pg C $yr^{-1}$) between model years 1860 and 2099 in the CMIP5 Historical and RCP8.5 simulations

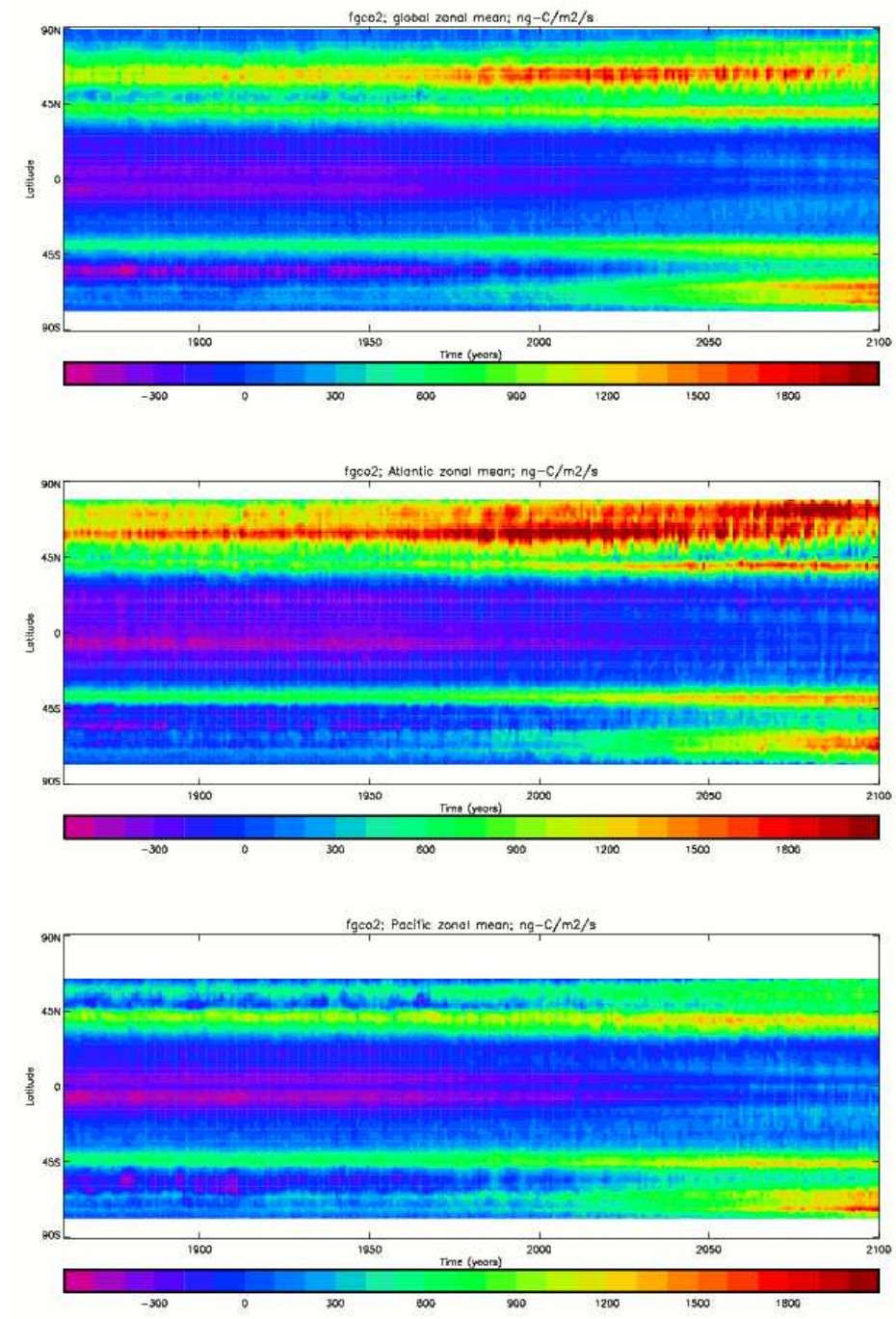

**Figure 41.** Change in the annual mean total air-to-sea CO$_2$ flux (ng C m$^{-2}$ s$^{-1}$) during model years 1860 to 2099 in the Historical and RCP8.5 simulations. Top panel: global zonal mean; middle panel: Atlantic Ocean basin zonal mean; bottom panel: Pacific Ocean basin zonal mean

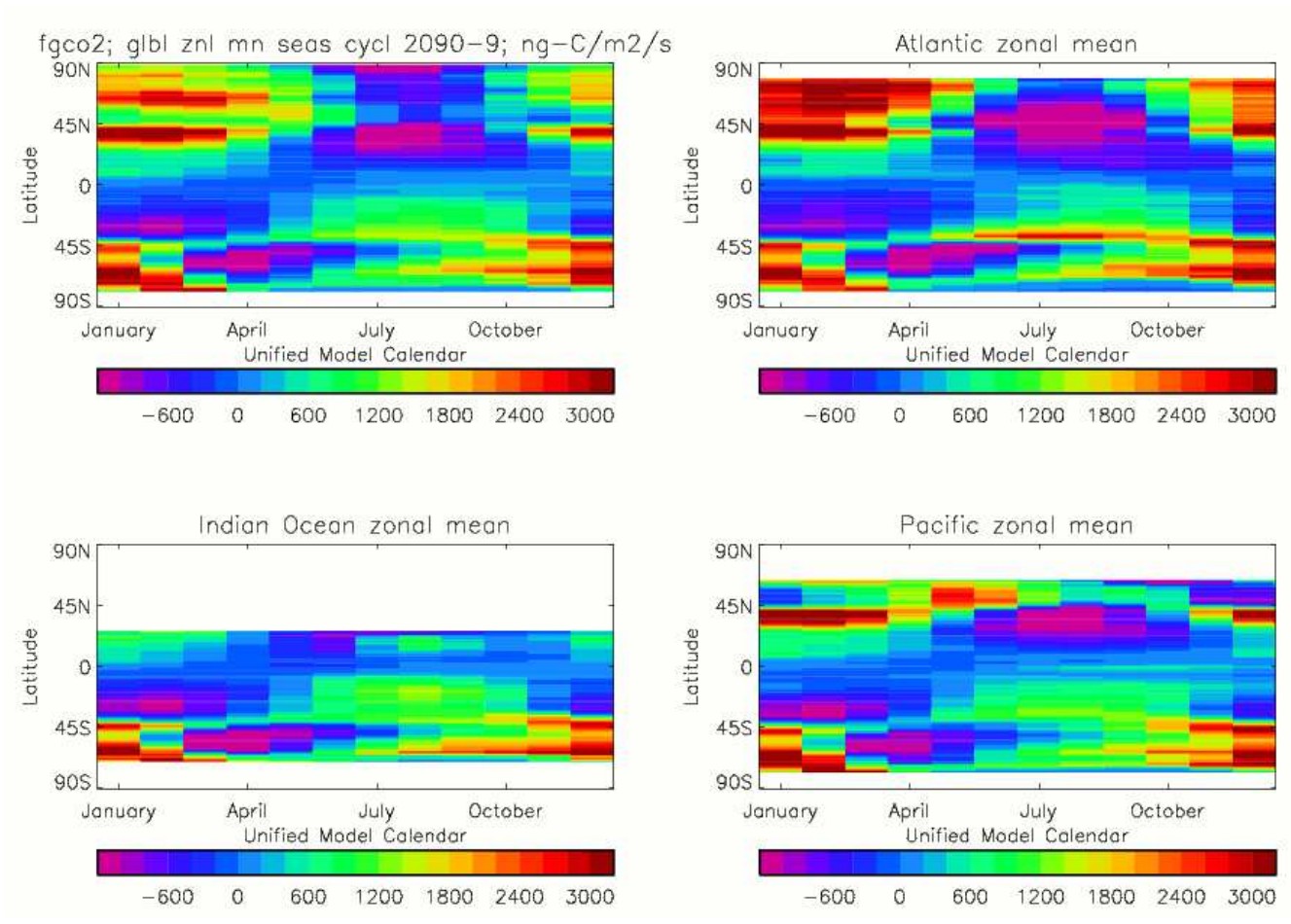

**Figure 42.** The seasonal cycle (monthly means) of the total air-to-sea $CO_2$ flux (ng C m$^{-2}$ s$^{-1}$) averaged over the model years 2090-2099 inclusive. Zonal mean of: upper left panel, global ocean; upper right, Atlantic Ocean basin; lower left, Indian Ocean basin; lower right, Pacific Ocean basin

**Table 1.** Diat-HadOCC model state variables

| Symbol | Description | Units |
| --- | --- | --- |
| $DIN$ | dissolved inorganic nitrogen | mmol-N / m$^3$ |
| $Si$ | silicic acid | mmol-Si / m$^3$ |
| $FeT$ | total dissolved iron | umol-Fe / m$^3$ |
| $Ph$ | miscellaneous (misc-) phytoplankton | mmol-N / m$^3$ |
| $Dm$ | diatom phytoplankton | mmol-N / m$^3$ |
| $DmSi$ | diatom silicate | mmol-Si / m$^3$ |
| $Zp$ | zooplankton | mmol-N / m$^3$ |
| $DtN$ | detrital nitrogen | mmol-N / m$^3$ |
| $DtSi$ | detrital silicate | mmol-Si / m$^3$ |
| $DtC$ | detrital carbon | mmol-C / m$^3$ |
| $DIC$ | dissolved inorganic carbon | mmol-C / m$^3$ |
| $TAlk$ | total alkalinity | meq / m$^3$ |
| $Oxy$ | dissolved oxygen | mmol-O2 / m$^3$ |

**Table 2.** Polynomial coeffs relating $k$ to square root of pigment in depth-range $L$

| $L$ | $b_{0,L}$ | $b_{1,L}$ | $b_{2,L}$ | $b_{3,L}$ | $b_{4,L}$ | $b_{5,L}$ |
|---|---|---|---|---|---|---|
| 1 | 0.095934 | 0.039307 | 0.051891 | -0.020760 | 0.0043139 | -0.00035055 |
| 2 | 0.026590 | 0.016301 | 0.073944 | -0.038958 | 0.0075507 | -0.00054532 |
| 3 | 0.015464 | 0.14886 | -0.15711 | 0.15065 | -0.055830 | 0.0075811 |

**Table 3.** Polynomial coeffs for $\frac{da\#}{dz}$ as a function of pigment and depth

| | |
|---|---|
| $gcof_1 = g_1 = 0.048014$ | $gcof_6 = g_4 = 0.0031095$ |
| $gcof_2 = g_2 = 0.00023779$ | $gcof_7 = g_9 = 0.0012398$ |
| $gcof_3 = g_5 = -0.0090545$ | $gcof_8 = g_6 = 0.0027974$ |
| $gcof_4 = g_7 = 0.00085217$ | $gcof_9 = g_{10} = -0.00061991$ |
| $gcof_5 = g_3 = -0.023074$ | $gcof_{10} = g_8 = -0.0000039804$ |

**Table 4.** Polynomial coeffs and rational function coeffs for psynth calculation

| Coeff | i=1 | 2 | 3 | 4 | 5 |
|---|---|---|---|---|---|
| $\Omega_i$ | 1.9004 | -0.28333 | 0.028050 | -0.0014729 | 0.000030841 |
| $\gamma_i$ | 1.62461 | 0.0045412 | 0.13140 | | |

**Table 5.** Parameter values used in CMIP5 simulations

| Param | Value | Units | Description |
|---|---|---|---|
| $P_{m,r}^{Ph}$ | 1.5 | $d^{-1}$ | Max rate of psynth; misc-Phyto, Fe-replete |
| $P_{m,l}^{Ph}$ | 1.5 | $d^{-1}$ | Max rate of psynth; misc-Phyto, Fe-limited |
| $P_{m,r}^{Dm}$ | 1.85 | $d^{-1}$ | Max rate of photosynthesis; diatom, Fe-replete |
| $P_{m,l}^{Dm}$ | 1.11 | $d^{-1}$ | Max rate of photosynthesis; diatom, Fe-limited |
| $\alpha^{Ph}$ | 0.02 | mg C (mg Chl)$^{-1}$ h$^{-1}$ ($\mu$Einst m$^{-2}$ s$^{-1}$)$^{-1}$ | Initial slope of the psynth-light curve; misc-Phyto |
| $\alpha^{Dm}$ | 0.02 | mg C (mg Chl)$^{-1}$ h$^{-1}$ ($\mu$Einst m$^{-2}$ s$^{-1}$)$^{-1}$ | Initial slope of the psynth-light curve; diatom |
| $k_{DIN}^{Ph}$ | 0.1 | mMol N m$^{-3}$ | Half-saturation const, N uptake; misc-Phyto |
| $k_{DIN}^{Dm}$ | 0.2 | mMol N m$^{-3}$ | Half-saturation const, N uptake; diatom |
| $k_{Si}^{Dm}$ | 1.0 | mMol Si m$^{-3}$ | Half-saturation const, Si uptake; diatom |
| $R_{c2n}^{Ph}$ | 6.625 | mMol C (mMol N)$^{-1}$ | Molar C:N ratio, misc-Phyto |
| $R_{c2n}^{Dm}$ | 6.625 | mMol C (mMol N)$^{-1}$ | Molar C:N ratio, diatom |
| $R_{c2n}^{Zp}$ | 5.625 | mMol C (mMol N)$^{-1}$ | Molar C:N ratio, zoopl |
| $R_{si2n,r}^{Dm}$ | 0.606 | mMol Si (mMol N)$^{-1}$ | Molar Si:N ratio, diatom, Fe-replete |
| $R_{si2n,l}^{Dm}$ | 0.606 | mMol Si (mMol N)$^{-1}$ | Molar Si:N ratio, diatom, Fe-limited |
| $R_{c2chl,0}^{Ph}$ | 40.0 | mg C (mg Chl)$^{-1}$ | default Carbon:Chlorophyll ratio, misc-Phyto |
| $R_{c2chl,min}^{Ph}$ | 20.0 | mg C (mg Chl)$^{-1}$ | minimum Carbon:Chlorophyll ratio, misc-Phyto |
| $R_{c2chl,max}^{Ph}$ | 200.0 | mg C (mg Chl)$^{-1}$ | maximum Carbon:Chlorophyll ratio, misc-Phyto |
| $R_{c2chl,0}^{Dm}$ | 40.0 | mg C (mg Chl)$^{-1}$ | default Carbon:Chlorophyll ratio, diatom |
| $R_{c2chl,min}^{Dm}$ | 20.0 | mg C (mg Chl)$^{-1}$ | minimum Carbon:Chlorophyll ratio, diatom |
| $R_{c2chl,max}^{Dm}$ | 200.0 | mg C (mg Chl)$^{-1}$ | maximum Carbon:Chlorophyll ratio, diatom |
| $g_{max}$ | 0.8 | $d^{-1}$ | Max specific rate of zooplankton grazing |
| $g_{sat}$ | 0.5 | nMol N m$^{-3}$ | Half-saturation const for zoopl grazing |
| $bprf_{Ph}$ | 0.45 | (none) | Zoopl base feeding preference for misc-Phyto |
| $bprf_{Dm,r}$ | 0.45 | (none) | Zoopl base feeding pref: diatom, Fe-replete |
| $bprf_{Dm,l}$ | 0.45 | (none) | Zoopl base feeding pref: diatom, Fe-limited |
| $bprf_{Dt}$ | 0.10 | (none) | Zoopl base feeding preference for detritus |
| $F_{ingst}$ | 0.77 | (none) | Fraction of food that is ingested |
| $F_{messy}$ | 0.1 | (none) | Frac of non-ingstd food to dslvd nutrient/carbon |
| $\beta^{Ph}$ | 0.9 | (none) | Assimilate-able frac of ingested misc-Phyto |
| $\beta^{Dm}$ | 0.9 | (none) | Frac of ingested diatom that can be assimilated |
| $\beta^{Dt}$ | 0.7 | (none) | Frac of ingested detritus that can be assimilated |

**Table 5a.** Parameter values used in CMIP5 simulations (cont)

| Param | Value | Units | Description |
|---|---|---|---|
| $\Pi_{resp}^{Ph}$ | 0.05 | $\text{d}^{-1}$ | misc-Phyto respiration, specific rate |
| $\Pi_{resp}^{Dm}$ | 0.0 | $\text{d}^{-1}$ | Diatom respiration, specific rate |
| $\Pi_{mort}^{Ph}$ | 0.05 | $\text{d}^{-1}\,(\text{mMol N m}^{-3})^{-1}$ | misc-Phyto mortality, density-dep rate |
| $ph_{min}$ | 0.01 | $\text{mMol N m}^{-3}$ | misc-Phyto conc below which mortality is zero |
| $\Pi_{mort}^{Dm}$ | 0.04 | $\text{d}^{-1}\,(\text{mMol N m}^{-3})^{-1}$ | Diatom mortality, density-dep rate |
| $\Pi_{lin}^{Zp}$ | 0.05 | $\text{d}^{-1}$ | Zooplankton losses, specific rate |
| $\Pi_{mort,r}^{Zp}$ | 0.3 | $\text{d}^{-1}\,(\text{mMol N m}^{-3})^{-1}$ | Zoopl. mortality, density-dep, Fe-replete |
| $\Pi_{mort,l}^{Zp}$ | 0.3 | $\text{d}^{-1}\,(\text{mMol N m}^{-3})^{-1}$ | Zoopl. mortality, density-dep, Fe-deplete |
| $F_{nmp}$ | 0.01 | (none) | Fraction of mortality to dissolved nutrient |
| $F_{zmort}$ | 0.67 | (none) | Fraction of zoopl mortality to dissolved nutrient |
| $V_{Dt}$ | 10.0 | $\text{m d}^{-1}$ | Sinking speed, detritus |
| $\Pi_{rmndd}^{DtC}$ | 8.58 | $\text{m d}^{-1}$ | Detrital remineralisation rate factor, carbon |
| $\Pi_{rmnmx}^{DtC}$ | 0.125 | $\text{d}^{-1}$ | Max detrital remineralisation rate, carbon |
| $\Pi_{rmndd}^{DtN}$ | 8.58 | $\text{m d}^{-1}$ | Detrital remineralisation rate factor, nitrogen |
| $\Pi_{rmnmx}^{DtN}$ | 0.125 | $\text{d}^{-1}$ | Max detrital remineralisation rate, nitrogen |
| $\Pi_{rmn}^{DtSi}$ | 0.05 | $\text{d}^{-1}$ | Detrital silicate (opal) remin/dissolution rate |
| $V_{Dm}$ | 1.0 | $\text{m d}^{-1}$ | Diatom sinking speed |
| $R_{fe2c}^{eco}$ | 0.025 | $\mu\text{Mol Fe (mMol C)}^{-1}$ | Molar Fe:C ratio for ecosystem |
| $k_{FeT}$ | 0.2 | $\mu\text{Mol Fe m}^{-3}$ | Scale factor for Fe-limitation |
| $LgT$ | 1.0 | $\mu\text{Mol m}^{-3}$ | Total ligand concentration |
| $K_{FeL}$ | 200.0 | $(\mu\text{Mol m}^{-3})^{-1}$ | Fe-ligand partition function |
| $\Pi_{ads}^{FeF}$ | $5.0\times10^{-5}$ | $\text{d}^{-1}$ | Adsorption rate of iron onto particles |
| $R_{o2c}^{eco}$ | 1.302 | $\text{mMol O}_2\,(\text{mMol C})^{-1}$ | Molar O$_2$:C ratio for ecosystem |
| $R_{cc2pp}^{Ph}$ | 0.0195 | $\text{mMol CaCO}_3\,(\text{mMol C})^{-1}$ | Misc-Phyto molar ratio, carbnt frmtn:organic prodn |
| $Z_{lys}$ | 2113.0 | m | Depth of lysocline |