# Peer review of "Description and evaluation of the Diat-HadOCC model v1.0: the"

_Geoscientific Model Development, 2017_

## Referee Comment (RC1) · Anonymous Referee #1 · 15 Nov 2017

This paper proposes a detailed description of the Diat-HadOCC model v1.0 which is a component of the Earth System model developed and used by the Met Office. This model is a quite simple NPZD-type model which represents ocean biogeochemistry based on two phytoplankton groups (miscellaneous phytoplankton and diatoms), one zooplankton group, detritus (with a variable C/N stoichiometry) and three limiting nutrients (inorganic N, Fe, and Si). The model described here is an upgrade of the HadOCC model which was published almost two decades ago (Totterdell and Palmer, 2001). It is being embedded in the Met Office modeling platform since at least the study by Collins et al. (2011). Thus, this model is not particularly new. The manuscript proposes the first detailed description and validation of this model based on simulations performed

by Collins et al. (2011).

I should admit that I have mixed feelings about this manuscript. It proposes a detailed description and validation of the ocean biogeochemical component of the HadGEM2. The description is rather complete, relatively well written and as a consequence, very useful to understand the model structure. The validation is also interesting and allows to quite correctly highlight the model capabilities, at least in the upper ocean. A validation of the model in the interior of the ocean is lacking as are also lacking some more quantitative diagnostics of the model performance (statistical indices). Thus, the main objective of this paper is fulfilled quite correctly. However I have some serious concerns which I detailed below:

- First, Diat-HadOCC is the ocean biogeochemical component of HadGEM2 which was the MetOffice Earth System model used in CMIP5. For CMIP6, the MetOffice has switched to another ocean dynamical model and another ocean biogeochemical model (MEDUSA). Thus, I am wondering what the status of HadGEM2 in general and of Diat-HadOCC in particular is. Is that model still actively developed and maintained? Will it be developed in the coming years? What is it used for currently? The author should detailed that more clearly in the paper.

- Second, I have some serious concerns about the model parameterizations as well as about the model behavior. I detail these concerns in my specific comments below. They are mainly related to the iron cycle, the grazing parameterization, and the DMS part (which is said to be a component of the model but is not described here). Furthermore, the simulations that are used here to validate the model are bugged. I understand that rerunning the ES model would be extremely expensive but this is quite "disturbing" especially since a main purpose of this study is to prove that the model is suitable for the type of applications it has been designed for. Third, this model is an evolution of HadOCC. As far as I understand, a main difference is the explicit representation of diatoms. However, the results shown here indicate that diatoms and misc. phytopankton behave very similarly. Thus, the interest of an additional plankton functional type is rather tenuous and clearly not really demonstrated. And since the silicon part is bugged, the advantage for the silicon cycle can not be proved here.

- Third, as already mentioned, no validation is proposed for the ocean interior. It remains restricted to the surface. And a more quantitative validation would be nice.

Thus, in its current state, I consider that the paper is not suitable for publication in GMD. Major modifications should be brought to the manuscript to address my main concerns. My advice would be at least to improve the description of the iron cycle, to alter some of the default parameters of the model (especially the feeding preferences), to retune the model and to perform at least one simulation with this updated version, in which the silicon cycle part is debugged. I understand that this requires quite a substantial amount of work but according to me, this is mandatory so that the manuscript becomes suitable for publication.

Specific comments

Pages 2-3: the set of equations is nice but at this point, it may be difficult to read without the complete details that are provided later in the manuscript. Thus, my suggestion is to put this set of equation either at the end of the description or in a table or to split and put them in different parts of the manuscript when appropriate.

Page 3, line 21: I don't understand why the zooplankton mortality term can be made a function of iron limitation. Studies on the impact of iron on zooplankton physiology are rather scarce and as far as I am aware of, I don't think any of them have shown an increased mortality rate (either direct mortality or increased predation by the upper trophic levels).

Page 4, eq. 15-16 and the text below: I don't understand why the temperature effect is removed above 20°C when there is no limitation. In fact, I don't fully understand the reason for the threshold. It should be explained.

Page 5, lines 6-7: The optical scheme includes three layers. This scheme has been

reparameterized to be suitable for the the actual vertical discretization of HadGEM2. Thus, if I understand correctly, this means that this part has to be recoded each time the vertical structure is changed. Not very convenient.

Page 5, line 9: I think there is a typo there. It should be equation 17, shouldn't it ?

Page 8, line 8: Zooplankton grazing is parameterized according to Fasham's active switching scheme. This scheme exhibits a number of drawbacks as detailed in Gentleman et al (2003), Vallina et al. (2014), Morozov and Petrovskii (2013), ... Other schemes present better general properties and may be more appropriate the simulate grazing on multiple preys. I don't suggest to change that but it should be discussed a little bit.

Page 8, Eqs 44-47: Zooplankton here is converted from carbon units to weight units. I think these equations have a problem because that's the prey biomass that should be converted to be consistent with the denominator. Zooplankton should not be converted.

Page 9, Eq 48: The equation should be rewritten. There is a typo there.

Page 10, lines 11-20: I don't understand why the detritus that reach the bottom of the ocean are remineralized in the last 3 vertical layers of the water column. This means that remineralization at the bottom impacts on biogeochemistry over the last three layers. In that specific case, since the deepest vertical layers are about 350m thick, organic matter from the sediments are remineralized over the bottom 1000m of the ocean. Why?

Page 10, bottom paragraph: Iron is not tracked in the detritus. The author assumes that all iron that would be routed to detritus is instantaneously remineralized back to inorganic dissolved iron. Thus, no iron is exported by the sedimentation of organic particles. This is quite a strong assumption that is not supported by observations (see the review by Boyd et al. (2017). Iron parameterization should be changed to remove that assumption. This would not increase the computing cost of the model since the

Fe/C ratios are constant in the model and identical in all organic compartments.

Page 11, Eqs 77-80: The notation should be detailed.

Pages 11-14: I am not convinced that such a high level of details is required here. The equations have been detailed elsewhere in the quoted literature. This section can be considerably shortened.

Page 16, lines 5-6: the author says that CO2 and DMS are exchanged between the ocean and the atmosphere. This means that DMS is explicitly modeled in the ocean biogeochemical model. But, this is not described here. I don't say that DMS should be described but, at least some words should be said about the DMS module.

Page 18, lines 5-6: The silicate and iron fields are said to drift quite significantly. For silicate, since it is bugged in this study, that's not a big issue. For iron, it is more annoying since iron limitation is supposed to control phytoplankton growth (especially diatom growth) in up to 40% of the ocean. More details on the drift would be nice to have an idea of the magnitude and the spatial distribution of this drift.

Page 19, lines 3-7: As acknowledged by the author, chlorophyll concentrations are largely overestimated in the model, especially in the typical HNLC regions. This means that iron limitation is not sufficient (which is confirmed later by the much to large surface iron levels). Thus, either there is problem with iron (see below) or with the parameters chosen to model iron limitation. Anyway, the model behavior is not really satisfactory. And applying a correction factor is not really a very good solution even if it is more convenient to compare with the data. The model should be modified and retuned.

Page 19, section 4.1.2: The model simulates an almost equal contribution between misc. phyto and diatoms, even in the typical HNLC regions and in the oligotrophic subtropical gyres. This is a major model deficiency. This means that either diatoms and misc. phytoplankton are not different enough in the model and that the model cannot simulate contrasting species relative contribution. The authors should try to

explain that deficient model behavior. An aspect of the model that should play a role is the grazing parameterization. Feeding preferences for phytoplankton and diatoms are identical which means that the zooplankton grazing pressure is similar. Furthermore, the active switching parameterization shares some similarities with the kill the winner parameterization. The most successful species is grazed preferentially which tends to even up the relative contributions. Anyway, since this is a major model deficiency, the author should explore with care the mechanisms that generate this deficiency.

Page 20, lines 13-14: Diatoms are simulated to have a better success in oligotrophic areas. That's quite the opposite oif what is commonly observed: Diatoms tend to cope less well with oligotrophic conditions because of their large volume.

Page 20, line 20: The author says that diatoms are more resistant to grazing. This does not seem to be the case in the model because the feeding preferences for diatoms and misc. phyto are identical in Table 5.

Page 20, line 25: The author claims that the pco2 fields look very similar. I think this is quite optimistic!

Page 21, line 1-2: pCO2 levels are overestimated just south of 45°S which is explained in the next sentence by an excessive PP. I don't understand the reasoning. Shouldn't it be the opposite?

Page 21, lines 7-10: On figure 11, one can see that the model simulates a pCO2 maximum in summer in the North Atlantic Ocean. According to the in situ data, this is the opposite. Thus, the seasonal cycle in the subpolar North Atlantic Ocean is inverted. This should be explored.

Page 22: Alkalinity is not shown. As it is an important player of the carbon system, it should be shown.

Page 23, lines 23-32: the silicon cycle is bugged in the model which is acknowledged by the author. Thus, this part should not be discussed. The problem is that the silicon

cycle part of the model cannot be validated. Since a major evolution of that model relative to HadOCC is the representation of diatoms, being unable to validate the silicon cycle is quite annoying. I don't think that validating the seasonal cycle is a good argument because since there is a bug, I have no confidence in that part of the code. My advice is to rerun the model without that bug. Of course, this would be very expensive but one option would be to follow the protocol that has been adopted to spin up the model and run the ocean part only forced by the atmospheric conditions from the Earth System model.

Pages 23-24, the iron cycle part: Values at the surface are much too high over large parts of the ocean. A comparison with the data would be useful, for instance the dataset from Tagliabue et al. (2012). A possible explanation for these too high values is the lack of iron export by sinking particles (which is highly unrealistic).

Page 25, line 22-25: This is the opposite to what has been found in previous studies such as Bopp et al. (2005) and Marinov et al (2010). Why?

---

## Short Comment (SC1) · 17 Nov 2017

I would like to comment on some statements made in the manuscript regarding observations of ocean dissolved inorganic carbon (DIC) and total alkalinity (TAlk). On page 17, lines 1-4 it is stated that gridded DIC and TAlk are available from GLODAP (Sabine et al., 2005; Key et al., 2004) but that these fields are based on much fewer data than the nutrient fields available in World Ocean Atlas (WOA). While this is still true, there is now an updated data product with ocean DIC and TAlk - GLODAPv2 (Lauvset et al., 2016; Olsen et al., 2016) - with is based on many more observations. On page 22, lines 12-13 it is stated that the GLODAP field of DIC ends at 70N. This

is true for the original data product, but the updated version includes the Arctic Ocean. I do understand that these model simulations were started quite some time ago and it then made sense to compare to the original GLODAP data product. I would recommend to now compare, at least in Figure 15, the model output with GLODAPv2. At the very least, a note should be made that an updated version of the observational data now exists, which also includes the Arctic. GLODAPv2 can be downloaded here: https://www.nodc.noaa.gov/ocads/oceans/GLODAPv2/, and links to the publications are also available from that site. Also, on page 22, line 7 a link is given to the GLODAP data. This link is no longer correct since CDIAC no longer exists. The link to the original GLODAP is this: https://www.nodc.noaa.gov/ocads/oceans/glodap/

References: Lauvset, S. K., et al. (2016), A new global interior ocean mapped climatology: the 1x1 GLODAP version 2, Earth Syst. Sci. Data, 8(2), 325-340, doi:10.5194/essd-8-325-2016. Olsen, A., et al. (2016), The Global Ocean Data Analysis Project version 2 (GLODAPv2) – an internally consistent data product for the world ocean, Earth Syst. Sci. Data, 8(2), 297-323, doi:10.5194/essd-8-297-2016.

---

## Referee Comment (RC2) · Anonymous Referee #2 · 11 Dec 2017

**Comments on "*Description and evaluation of the Diat-HadOCC model v1.0: the ocean biogeochemical component of HadGEM2-ES*"**
by Ian Totterdell, submitted to Geosci. Model Dev. (gmd-2017-90).

This paper aims at providing a detailed description of the biogeochemical component of the HadGEM2-ES model used for climate studies. It also describes the results obtained for the CMIP5 simulations. The paper is rather well and clearly written.

However, I have mixed feelings about this work and the actual motivation for submission. The discussion and assessment of the ecosystem and carbon cycle models appear to be given low priority; it seems that the main aim is publicizing results obtained in the framework of the CMIP5 project.

As a consequence the submitted work does not provide a comprehensive assessment of the Diat-HadOCC model. Further, the conclusions are really too optimistic given the present results. Since many information is lacking it is hard to evaluate whether the discrepancies between the model and available climatologies stem from the biogeochemical module or from shortcomings in the ocean circulation model.

In order to be fit for publication in Geosci. Model Dev. the paper should be thoroughly reworked so as to include a real assessment of the model as well as provide missing information on some processes.

**Major comments** The work as presented here suffers from several weaknesses. The most important points to consider are

1. The evaluation of model performance is minimal.

   - A visual comparison is not sufficient for model validation. Statistical tools providing mean and pattern biases both on spatial and temporal scales should be used in that purpose.

   - The paper does not present nor assess the distribution of any of the variable at depth though there are sufficient available data for assessing these distributions.
     Such an assessment is essential since the marine carbon cycle, as well as the ecosystem state, strongly depend on the exchange rate between surface and deep ocean layers.

   - Additionaly, the following fields should also be assessed against data: Fe, $O_2$, and Alk. For iron the data set of Tagliabue et al. (2012) can be downloaded from http://pcwww.liv.ac.uk/~atagliab. $O_2$ and Alk data sets are readily available from the World Ocean Atlas and GLODAP databases. AOU should also be included in the discussion.

2. Iron cycle:

   - The impact of dissolved iron on the ecosystem (page 3, lines 19–21) is not motivated; these processes call for more justification.
     Further, does $Fe$ really influence the feeding preference as well as the mortality of zooplankton. Is there any evidence for that?

- Since there is no iron detritus in the model, all iron is returned back to solution (page 10, lines 25–26). However, most of this flux is expected to happen in the uppermost (euphotic) layers of the ocean since it is concommitant with the C flux from the living to the detritus pool. Is this way of doing coherent with the other cycles?

- The description of the iron cycle (section 2.3.2) could provide more details and adequate references to published works.

3. There is no discussion of the impact of any of the parameterization (sensitivity test).

- It is agreed that performing many experiments with a coupled model is not feasible. However advantage could be taken of the existing ocean-only model (mentioned on page 17, line 31) to fully investigate the Diat-HadOCC component.

- In the description of treatment of detrital material it is written that the depth variation of detritus is consistent with the power-law curve of Martin et al. (1987). As far as I understand the exponent of the power-law is -0.858; however, as stated in Martin et al.(1987) that exponent is representative of an oligotrophic environment. Was the model tested with other sinking velocity or remineralization rate?
  It could be that the poor performance with respect to $pCO_2$ is caused by a too shallow redistribution of material.

4. The analysis of the phase and amplitue of the $pCO_2$ signal over a year would be more interesting if it was accompanied with a similar analysis of the SST-driven part of the signal. As such one could evaluate how much is due to the physics and how much is due to the ecosystem model.

5. Section 4.1.7: the claim that 'The gridded data from WOA05 is *slightly* higher than the model in the Eastern Equatorial Pacific' is questionable; indeed, nitrate concentration in that area is of the order of 7 mMol $N/m^3$ for WOA05 while in the model it is around 0.7 mMol $N/m^3$. It is not clear whether the model-data discrepancy is due to too high primary production or to an inappropriate vertical distribution of detritus.

6. The abstract and conclusion are too optimistic with respect to the actual performance of the model.

**Other comments**

1. Equations (1) to (10): I was a bit lost in all these terms. I would recommend
   1) that these equations be re-ordered, starting with diatoms and phytoplankton, then nutrient, detritus, and finally DIC, Alk , and $O_2$, and,
   2) that each equation be commented (what do represent the various source/sink terms?).

2. Eq. (14) and line 26: in the present form of Eq.(14) $k_{FeT}$ does not represent a half-saturation constant. Indeed, Eq. (14) may be reformulated as

$$\Pi = \Pi_{\text{replete}} + \frac{k_{FeT}}{k_{FeT} + FeT} \left( \Pi_{\text{deplete}} - \Pi_{\text{replete}} \right).$$

For $k_{FeT}$ to represent a half-saturation constant Eq. (14) should read

$$\Pi = \Pi_{\text{replete}} + (\Pi_{\text{deplete}} - \Pi_{\text{replete}}) \left/ \left( 1 + \frac{k_{FeT}}{FeT} \right) \right. .$$

which is equivalent to

$$\Pi = \Pi_{\text{replete}} + \frac{FeT}{k_{FeT} + FeT} (\Pi_{\text{deplete}} - \Pi_{\text{replete}}) .$$

3. The model is apparently built on constant Redfiel ratio; for the sake of clarity could the author give those ratio under the usual form C:N:P:$O_2$, C:Si, C:Fe?

**Minor comments**

- Page 4: There should be a factor relating alkalinity and nitrate in Eq. (13)

- Page 4, line 10: what is meant by '(and the temperature factor is actively used)'?

- Page 4, lines 25–27: the sentence 'When the HadOCC model (which uses the same productivity model) has been forced by 6-hourly re-analysis fluxes, for example, a daily-average irradiance field has been calculated and passed in for use in this scheme.' is not really relevant here.

- Page 9, Eq. (61): what is $ph_{min}$?

- Page 11, Eq. (72): what is $LgF$?

- What is the rationale for providing Fig. 19. Wouldn't it be more sensible to redo the experiment without the bug and present some decent silicate field results?

**References**

Martin et al. (1987). VERTEX: carbon cycling in the northeast Pacific, Deep Sea Research Part A., 34, 267–285, doi:10.1016/0198-0149(87)90086-0.

Tagliabue et al. (2012). A global compilation of dissolved iron measurements: focus on distributions and processes in the Southern Ocean, Biogeosciences, 9, 2333–2349, doi:10.5194/bg-9-2333-2012.

---

## Referee Comment (RC3) · Anonymous Referee #3 · 13 Dec 2017

**1   General remarks**

This manuscript provides a description of the marine biogeochemical and ecosystem model that has been used in the CMIP5 experiments by the Hadley Centre, and provides some evaluation of its performance, compared to climatological chlorophyll, nutrient and carbon system (dissolved inorganic carbon, DIC, $CO_2$ partial pressure, $pCO_2$, and air-sea flux of $CO_2$) data. It is well written, and in its descriptive part rather detailed, which I found extremely useful. One example is a detailed description of the calculation of the vertical attenuation of light, which is treated in a more sophisticated way than in most global biogeochemical models.

[Figure]
The first reviewer has remarked that the model described has been replaced by another one in the current CMIP calculations in the Hadley Centre and questions whether it then makes still sense to publish it. I am not of that opinion, firstly because the model runs performed with this model feature prominently in CMIP5 and in the latest IPCC report; a full description of the model allows to see these in the context of their inherent assumptions. Secondly, the model, while at its core a typical NPPZD model, contains some not-so-typical parameterizations (like the already mentioned treatment of light, or the iron-dependency of grazing preferences) that other modellers may find useful to adopt.

Like the second reviewer, my main criticism of the manuscript concerns the evaluation of model performance, which should be improved significantly before the paper can be published. Specifically, it would be helpful not to limit the evaluation at least of the modeled DIN and DIC distributions to surface values; although from a carbon system perspective it may be the air-sea flux of carbon which is the main metric of model success, the surface values of both DIC and DIN are strongly influenced by upwelling of old water, and hence the representation of both water mass age and the accumulated remineralization at depth. It would also be helpful to quantify the model-data (dis-)agreement somewhat, like e.g. done in the papers by Schneider et al. (2008). Also, since it affects air-sea gas exchange, it would also be helpful if the modeled Alkalinity field would be evaluated as well.

Unlike the second reviewer, however, I do not see too much sense in a qualitative evaluation of the model against observed iron data. Most models do not fare well against that data (see e.g. Tagliabue et al. 2016), firstly because ocean models are still way to simplistic, but also because the data available is still far away from a climatology that represents the average distribution in a similar way as the World Ocean Atlas does for hydrography and macronutrients. It is quite clear that the model will fail against iron data. More interesting is whether there is a specific trend in the model-data mismatch: From the description of the modeled iron cycle I would expect, as explained further

down, surface values to be systematically too high and deep values systematically too low.

Putting more effort into the evaluation of the model results could make this a paper well fitting into Geoscientific Model Development.

**2   Questions and suggestions for improvement**

The manuscript evaluates the annual cycle of several variables by showing the seasonal amplitude and the timing of the maximum value; but these two quantities have been derived by fitting the time-series pointwise to a variation that has a sinusoidal form. But the seasonal cycle of most biologically influenced variables does not nearly follow a sinusoidal pattern. To me this raises the question how robust the method of evaluation is, i.e. would one get similar patters for the timing if one left away the fit, and rather looked at quantities like the ones often defined for analyzing bloom phenology (e.g. in Racauld et al., 2012 or Siegel et al. 2002)?

What would need a bit of explanation is why the zooplankton grazing preference for diatoms, as well as mortality has been made dependent of dissolved iron; this should be explained. My guess for explaining the preference dependency is that it is a way to account for the thicker diatom frustules and hence longer handling time of diatoms by zooplankton under iron limitation. If that is correct, that parameterization is a really clever idea, worthy of an Ernst Maier-Reimer-award for useful tricks that make sense.

The assumption that there is no iron in organic detritus in the model (p.10, l.25-26) effectively means that iron is not transported vertically with the biological pump. Is there a reason why this assumption was made? My suspicion is that this is the reason why the iron fields still show a drift at the end of the simulations (p.18, l.5-6): There are relatively few ways how dust-deposited dissolved iron can be transported into the deep ocean, and hence surface concentrations will have a tendency to increase, and vice

versa for the deep ocean. The author states (p.23, l.15ff) that nitrate in the Southern Ocean shows a strong seasonal cycle in the model. To me this is an indication that iron there may be too high and non-limiting. This should be explored a bit further.

When comparing the modelled with satellite derived annual average chlorophyll fields (section 4.1.1), it is unclear to me how the bias was handled that is introduced in the satellite data by the absence of ocean colour data in polar night.

In the comparison of the $pCO_2$ values, it is stated that the model produces a substantially larger seasonal cycle than is observed in the data (p.21, l.5). Could this perhaps be caused by a too weak buffering of the carbonate system? This is one reason why I think it would make sense also to compare Alk, and not only DIC, to GLODAP data.

I am not entirely convinced of the extra value of comparing the air-sea flux of $CO_2$, in addition to comparing $pCO_2$; the main new ingredient that enters here is the distribution of the piston velocity, which has been parameterized with the same formula (Wanninkhof, 1992) in the model and in the Takahashi climatology.

In the description of the Si field it is acknowledged that these are not realistic and that there has been an error in the implementation of the model in the CMIP5 simulations (p.23, l.24). It would be interesting to know what that error was.

On p.26, l.19, it is stated that the seasonal cycling of $CO_2$ between atmsphere and ocean has intensified in the future model run, compared to the present-day. It would be interesting to know whether this is related to the change in buffer factor with increasing $CO_2$, as argued by Hauck and Völker (2015).

**3 Minor remarks**

page 7, line 19: $K_{chl}$ should be $k_{chl}$, as in equation 33.

p.15, l.20: 'than' should be 'that'

p.20, l.15, and several later places: The name of the Hovmöller diagram is consequently mis-spelled as Hov-Muller diagram

**References**

Hauck, J; Völker, C (2015): Rising atmospheric $CO_2$ leads to large impact of biology on Southern Ocean $CO_2$ uptake via changes of the Revelle factor. Geophysical Research Letters, 42(5), 1459-1464, doi:10.1002/2015GL063070

Racault, M.F.; Le Quéreé, C.; Buitenhuis, E.; Sathyendranath, S.; Platt, T. (2012), Phytoplankton phenology in the global ocean. Ecol. Indic. 14, 152–163.

Schneider, B.; Bopp, L.; Gehlen, M.; Segschneider, J.; Frölicher, T. L.; Cadule, P.; . . . Joos, F. (2008). Climate-induced interannual variability of marine primary and export production in three global coupled climate carbon cycle models. Biogeosciences, 5(2), 597–614. doi:10.5194/bg-5-597-2008

Siegel, D.; Doney, S.; Yoder, J. (2002), The North Atlantic spring phytoplankton bloom and Sverdrup's critical depth hypothesis. Science 2002, 296, 730–733.

Tagliabue, A., O. Aumont, R. DeAth, J. P. Dunne, S. Dutkiewicz, E. Galbraith, K. Misumi, J. K. Moore, A. Ridgwell, E. Sherman, et al. (2016), How well do global ocean biogeochemistry models simulate dissolved iron distributions?, Global Biogeochem. Cycles, 30, 149–174, doi:10.1002/2015GB005289.

Wanninkhof, R. (1992), Relationship between wind speed and gas exchange over the ocean, J. Geophys. Res., 97(C5), 7373–7382, doi:10.1029/92JC00188.

---

## Author Comment (AC1) · 16 Apr 2019

**Description and evaluation of the Diat-HadOCC model v1.0: the ocean biogeochemical component of HadGEM2-ES**

**Ian Totterdell**

**Response to Reviewer #1**

I thank the Reviewer for their perceptive and detailed comments. I have endeavoured to make the changes to the manuscript that they suggested, which make it a much more complete description of the model and its performance, but it has not been possible to make changes to the model – the resources have just not been available due to development work for the new Earth System Model. It would be good to retune the parameters, and especially to upgrade the representation of iron, but there is no prospect of that happening, unfortunately.

I have therefore changed the introduction to make it clear that this is a complete documentation of the ocean biogeochemical model that was used as a component of HadGEM2-ES (which produced the UK Met Office's contribution to CMIP5), and an evaluation of its performance in the context for which it was designed. There is value in this, not just because the database of CMIP5 simulations have been and are still being used to examine climate responses and projections, but also because the Diat-HadOCC model's strong points, and its weaknesses, can be of interest and relevance to those looking to construct a model in the future.

In the revised manuscript I have included validations of the ocean interior for all the nutrients as well as carbon, alkalinity, oxygen and AOU. I have also included Taylor diagrams of many of the state variables, and noted the exact statistics. I have also significantly expanded the Conclusions section, discussing the model's performance in an objective way in the light of the paper's purpose (i.e., a complete description of a model that has been used for a particular experiment).

Referring to the reviewer's serious concerns:

(i) Status of Diat-HadOCC model: As recounted in the new Conclusions section, there are no plans to develop the Diat-HadOCC model further in the UKESM community; if HadGEM2-ES is used in the future, Diat-HadOCC will be a part of that, but all the community's resources are concentrated on UKESM1 (featuring the MEDUSA model as the ocean biogeochemical component). As Kwiatkowski et al. (2014, Biogeosciences v11 pp7291-7304) describe, Diat-HadOCC was, along with the earlier HadOCC model and MEDUSA, considered for inclusion in UKESM1, but since it was not chosen it will not be developed further.

(ii) Retuning and re-parameterisation: As explained above there is no prospect of the model being re-parameterised, due to resources being committed elsewhere. I certainly agree that the model could benefit from such an action, although at the time of the model's initial development there was significant sensitivity analysis and investigation of parameter space before the values shown here were settled on. The requirement, for operational reasons, to perform such analyses in the full 3-D model limited the scope, however. Some of the reviewer's criticisms of the lack of differentiation between the diatom and non-diatom components, and especially their behaviour in the simulations, stem from the bug in the silicate variable (which meant that that nutrient was never limiting) and also from the poor results of the crude iron model. The iron model itself was developed at a time (c2007; the model was fixed by early 2009) when much less was known about the marine iron cycle than today. That is particularly true in a quantitative sense, and hard numbers of rates and ligand concentrations are needed if the model is to be used predictively. But an iron model produced today would certainly look very different. Finally on this concern, I have added a brief description of the DMS parameterisation used.

(iii) Validation of ocean interior: now provided.

Referring to the more specific points:

(a) "Pages 2-3: the set of equations is nice but…": I have split the equations up into the appropriate parts of the manuscript, as suggested.

(b) "Page 3, line 21: …zooplankton mortality…": I have put the rationale for making the zooplankton mortality dependent on the iron limitation in the text. Although it is clear that no single zooplankton individual or species will be affected in that way, the model zooplankton Zp does not represent any one species (indeed, at different times and places it stands in for single-celled protozoans or multi-life-stage copepods) and this parameterisation attempts to account for the fact that in iron-limited areas diatoms will be more heavily-silicified and less palatable to zooplankton, and the only zooplankton that will be able to graze on them will be larger types with longer lifetimes and lower mortalities. This representation was championed at an early stage of the model's development by the late Prof Mike Fasham.

(c) "Page 4, eq. 15-16 …temperature effect…": the effect is not removed above 20C, and there is no threshold, but the curves are set so that the maximum is at 20C.

(d) "Page 5, lines 6-7: The optical scheme …": The text reports the version used for HadGEM2-ES in CMIP6; that used the re-parameterised scheme, as had earlier versions back to the first simulations using the original HadOCC model, including those described in [Cox et al 2000]. Since all those model versions had layer boundaries at 10m and 20m depth it made sense to use the re-parameterised form. However, in the other main implementation of the Diat-HadOCC model, the [IMARNET] comparison study reported in Kwiatkowski et al., a more flexible approach was utilised, keeping the 5m and 23m boundaries from the original (published) version of the Anderson light model even when they did not coincide with layer boundaries. In this approach, the model layers in which the light-boundaries occur are split in two at the appropriate point (for the purposes of light and primary production calculations only) a separate calculations made, the results subsequently being combined for the layer total. A note to this effect is now included in the relevant text.

(e) "Page 5, line 9: I think there is a typo there. It should be equation 17, shouldn't it ?": No, the text refers to Equation 17 of Anderson (1993), rather than (former) Eqn 16 in this paper.

(f) "Page 8, line 8: Zooplankton grazing…": A good point, and some discussion has been added to this text.

(g) "Page 8, Eqs 44-47: Zooplankton here is converted…": the biomass-equivalence terminology is a way of creating a stoichiometry-independent label for the phytoplankton and zooplankton state variables. This has special significance for the food types, as they have to be summed, but it is also the case that the grazing needs of a zooplankton will to first importance depend on its biomass rather than its N-content. Using the terminology for Zp means that the modelled grazing rate does not vary if the C:N ratio is changed. In these simulations the zooplankton C:N ratio is the same as it has been in other model versions, so the same grazing rates can be used as in those versions.

(h) "Page 9, Eq 48: The equation should be rewritten. There is a typo there.": Agreed, I missed that one. Typo corrected.

(i) "Page 10, lines 11-20: …detritus that reach the bottom…": This was a pragmatic solution to a problem first identified in earlier ESM versions. The ocean bathymetry featured a number of narrow canyons (some were two layers deep) which were too narrow to be advectively flushed; other than sinking in, the only connection with the bulk of the ocean was by diffusion. If the detritus was remineralised only into the bottom box it was found that, even after more than 5000m of sinking and pelagic remineralisation, the concentration of dissolved nitrate (and DIC) grew to unrealistic values; by spreading the remineralisation over the bottom three boxes this ceased to be a problem. It was reasoned that this would not make the several-hundred-year return time (to euphotic layers) much different, and it was worth degrading the concentrations in the bottom layers a bit to avoid the extreme canyon problem. A short comment has been added to the text about this.

(j) "Page 10, bottom paragraph: Iron is not tracked in the detritus.": During model development it was found that the surface layers were losing dissolved iron at an excessive rate; remineralising it immediately from the detritus was the method chosen. I recognise now that this is not supported by the observations, but the data was much less clear at the time (the Boyd paper was published 8 years after the model began running its first simulations, for example), and here I am describing the model that was used. I have however included text noting this (and other) issues with the iron model in both model description section and in the conclusions, and accept that the iron model is a very weak part of the Diat-HadOCC model.

(k) "Page 11, Eqs 77-80: The notation should be detailed": Agreed; now done.

(l) "Pages 11-14: I am not convinced that such a high level of details is required here": the precise equations for the equilibrium constants are indeed available elsewhere but since the carbon chemistry is a key part of the model it is good for the sake of completeness to have all details in one place. Bacastow's (1990) secant method of similar triangles is less easy to source online and the notation is not the clearest, so I think there is value in retaining the full details.

(m) "Page 16, lines 5-6: the author says that CO2 and DMS are exchanged…": A brief description of the DMS sub-model has been included.

(n) "Page 18, lines 5-6: The silicate and iron fields are said to drift quite significantly…": further details have been added. Basically, the drift is primarily in the surface layers, with a steady increase in the N Atlantic (and in other high-Fe areas).

(o) "Page 19, lines 3-7: As acknowledged by the author, chlorophyll concentrations are largely overestimated in the model…": As explained, it is not feasible to retune the model, and this paper aims to describe the model used for CMIP5. Further comments about the problems with the model results in these variables have been added to the conclusions, however.

(p) "Page 19, section 4.1.2: The model simulates an almost equal contribution…": A fuller discussion has been added to the results section, addressing these points. Many of the similarities are caused by the silicate problem meaning that diatoms are never si-limited.

(q) "Page 20, lines 13-14: Diatoms are simulated to have a better success in oligotrophic areas…": discussion of these points added.

(r) "Page 20, line 20: The author says that diatoms are more resistant to grazing…": comment has been amended.

(s) "Page 20, line 25: The author claims that the pco2 fields look very similar. I think this is quite optimistic!": Umm, yes. A more objective, less optimistic description has been provided.

(t) "Page 21, line 1-2: pCO2 levels are overestimated just south of 45∘ S…": The comment about over-estimated PP was not made as a suggested cause of the high pCO2, rather to suggest that there are (un-specified) factors at play in that region which might cause both errors. I have re-worded this comment to make it more explicit and less mis-leading.

(u) "Page 21, lines 7-10: On figure 11, one can see that the model simulates a pCO2 maximum in summer in the North Atlantic Ocean…": The model's primary production in the spring and summer is low compared to observations in the N Atlantic, so less CO2 is taken up. In the real ocean the biological drawdown out-weighs the increasing temperature so pCO2 is lower in summer than at the end of winter, but in the model the temperature increase wins. This has been added to the relevant place in the results section.

(v) "Page 22: Alkalinity is not shown…": agreed. Plots of surface alkalinity and meridional sections are now presented and discussed.

(w) "Page 23, lines 23-32: the silicon cycle is bugged in the model…": The problem with the silicon processes has been identified as a too-high value for the detrital-silicate remin/dissolution parameter, which means too much is remineralised in the upper water-column leaving too little for the lower water-column. It is indeed very annoying, but unfortunately it has not been (and will not be) possible to re-run the simulation, even in the limited manner suggested by the reviewer. I think comparing the seasonal cycle to observations can be of some value, although clearly in areas where there would be expected to be silicate-limitation at any part of the year the unlimited model will show a larger amplitude than it should; the text has been altered to acknowledge that.

(x) "Pages 23-24, the iron cycle part": I have included a plot of a section comparable to the Geotraces A2 section and discussed the differences. As noted by the reviewer the surface values are much too high, and it is clear that the iron sub-model does not work well; this is now objectively discussed in the Conclusions section. Certainly much more is

known now, both qualitatively and quantitatively, about the ocean iron cycle than was known when the model was developed, and if the model was being put together now a very different iron model would be included.

(y) "Page 25, line 22-25: This is the opposite to what has been found in previous studies such as Bopp et al. (2005) and Marinov et al (2010)": Again, this is due to the silicate problem meaning that there is not enough silicate-limitation (both in the simulated present-day and in the simulated future). Therefore the diatoms avoid much of the limitation that would cause them to lose out to the non-diatoms. This is now discussed in the Conclusions section.

---

## Author Comment (AC2) · 16 Apr 2019

**Description and evaluation of the Diat-HadOCC model v1.0: the ocean biogeochemical component of HadGEM2-ES**

**Ian Totterdell**

**Response to Reviewer #2**

I thank the Reviewer for their considered comments.

I have endeavoured to re-work the assessment of the model as suggested by the Reviewer. It is now more definitively written as a complete and detailed description of the model that was used, as part of the HadGEM2-ES Earth System Model, to run simulations for the CMIP5 experiment, and the validation of how the model simulates the present day ocean ecosystem and carbon cycle and how it responds to climate change forcing are focussed on evaluating that model, and not on presenting the results of those simulations.

1. Major Comments

1.1 Evaluation insufficient: I have significantly added to the model evaluation, including both statistical comparisons (Taylor diagrams and the corresponding correlations and standard deviations, for both the annual means and the seasonal cycles) and showing meridional sections in both the Atlantic and Pacific oceans for the nutrients (including dissolved iron, though Atlantic only) and for DIC, alkalinity and dissolved oxygen. I have updated the data products that I use for comparison, to WOA 2013 and GLODAPv2. I have also included plots of AOU, and discussed. I thank the Reviewer particularly for this advice, it makes the evaluation much more complete.

1.2 Iron cycle: The motivation for iron concentration affecting the feeding preferences lies in what the model's zooplankton state variable – it does not represent any one species, or even type, of zooplankton, but the dominant type in a place and time. When diatoms are iron-stressed they become more heavily silicified and are therefore less palatable to zooplankton; therefore the preference for them decreases. Likewise, the dominant type of zooplankton in areas where there are heavily-silicified diatoms will shift towards larger zooplankton with longer lifetimes and lower mortalities. The description in the text has been expanded to explain this clearly.
The return of iron to solution without going through detritus was a pragmatic choice made at a time when it was difficult to keep enough iron in the surface waters to enable adequate levels of production; subsequent independent changes to the land surface model, made late on in the development of HadGEM2-ES greatly increased the dust (and iron) input at the surface, but the choice was not revisited due to time constraints. A longer discussion of the iron model has been included.

1.3 Sensitivity tests: the focus of the model description has been sharpened to be more definitively on the version of the model that was part of HadGEM2-ES in the CMIP5 experiments. As such, the text now discusses the model choices and their impacts in a fuller, but only qualitative, way.
This model was tested with variations of the sinking speed and remin rate parameters, and wider tests were done with earlier model versions (including the HadOCC model). At the time accelerated methods of testing (e.g. Khatiwala's Transport Matrix method) were not available, so with long spinups being needed and the deep circulation uncertain only a few parameter combinations could be tested. However, the Martin et al. power-law was found to be an acceptable fit, and as a published figure it was decided to use that.

1.4 pCO2 cycle: I have added a discussion about this.

1.5 Eastern Eq Pacific nitrate discrepancy: the high primary productivity is the cause of low nitrate in the model; iron limitation is not strong enough. However it has been found that artificially reducing the PP in that area does not lead to a much higher NO3 concentration, but instead to a much wider area of nitrate, still well below the observed values. This seems to be due to the low resolution (1/3deg N-S at the equator, but coarsening out to 1deg by 30deg N/S) and the high isopycnal diffusion (required for dynamical and stability reasons).

1.6 Optimism-bias: the abstract and conclusions have been re-written with greater objectivity.

**2. Other Comments**

2.1 Equations: I have split the equations up to sit separately in the appropriate model description sub-sections, and added more comments about the terms in each. Thanks for the suggestion.

2.2 Iron half-sat: I have changed the terminology used from "half-saturation" to "scale-factor".

2.3 Redfield: I have made the terminology more clear, while maintaining the link to the parameters in the equations.

**3. Minor Comments**

3.1 Alk and Nitrate: The equation (#12 in the original manuscript, #4 in the revision) accurately describes the FORTRAN code; following Goldman & Brewer (1980), uptake of $NO_3^-$ ions is linked to release of $OH^-$ ions, so the factor is chosen to be 1.0

3.2 Temperature factor: that text refers to the option not to vary the growth-rates with temperature.

3.3 Re-analysis forcing: I would argue that the sentence is relevant, because although this paper describes the model particularly in the context of its implementation in HadGEM2-ES and use in CMIP5 simulations, it can also be used in other settings (e.g Kwiatkowski et al, 2014), and it is relevant to mention how the inputs have to be adapted to run the model in such cases.

3.4 Ph_min: description added

3.5 LgF: description added

3.6 Silicate bug: it would be better to re-run the simulation without the bug, but the (significant) resources needed to do so have not been available, nor will they be available in the future. I include Fig 19 (orig manuscript, Fig 26 in the revision) as an evaluation plot: an evaluation that the model does badly in.

---

## Author Comment (AC3) · 16 Apr 2019

**Description and evaluation of the Diat-HadOCC model v1.0: the ocean biogeochemical component of HadGEM2-ES**

**Ian Totterdell**

**Response to Reviewer #3**

I thank the Reviewer for their thoughtful comments.

1. General comments.

I agree with the Reviewer that it is valuable for a model that has been used in an important collaborative modelling experiment such as CMIP5 to be fully documented, "warts and all", even belatedly. In this revised manuscript I have also tried to discuss more objectively (less optimism-bias) the performance as shown by the results, and tried to evaluate the successes, or otherwise, of the different sub-models.

As suggested by this Reviewer (and the others) I have extended my evaluation by including additional plots: I have added evaluation plots of alkalinity, dissolved oxygen and AOU. These include meridional sections of nitrate, DIC, alkalinity and oxygen, and also dissolved iron (a comparison to the Geotraces A2 section); this last comparison allows the particular failings of the iron sub-model to be identified (values much too high in the surface). I have also added a number of Taylor diagrams to show the summary statistics of the model fields fit to data, for annual mean and often for seasonal cycle.

The iron sub-model is a weak point of the Diat-HadOCC model, but it is the one used, and was (necessarily) developed before much of what is now known about the marine iron-cycle had been discovered (especially in quantitative terms). But examination of its failings may help future models avoid them!

2. Questions and suggestions for improvement

(i) seasonal cycle evaluation: I have looked at how well a sine-curve fits the seasonal cycle of various biological variables: my criterion for it being valid was that the variance of the residual (after the best-fitting sine-curve had been subtracted from the raw seasonal cycle) should be less than half that of the original cycle. In all model variables the fit is valid at almost all ocean points; in the corresponding data fields the fit is valid at most ocean points except in the case of nitrate and silicate, where the fit is valid at most mid- and high-latitudes points but not at most tropical and sub-tropical points. I have adapted the plots to only show values where the criterion is satisfied.

(ii) zooplankton grazing prefs and mortality as a function of iron: the reviewer's guess is correct about the reasoning for the modelled dependency, and the model description has been changed to explain this more clearly. Any award for the cleverness of the parametrization should be given, sadly posthumously, to Prof Mike Fasham who developed the scheme (but never published it) around the time the DH model was first being developed; credit is given to him in the text.

(iii) The iron model: when the model was being developed there was a problem with retaining enough iron in the surface waters to allow any significant primary production, so the decision was made to forcibly remineralise it high in the water-column; implicit removal to the deep ocean by scavenging would be the main means of vertical transport. Though that scavenged iron was not released back into the water-column (but instead implicitly lost to the sediments) that was not deemed to be a major issue, because the major iron input to the surface waters was likely to be from dust, there was at the time little data to validate against and the surface values were what mattered (for production in the model). Subsequent late changes to the land surface scheme changed the amount of dust and iron received by the ocean but the ocean remineralisation choice was not revisited due to time constraints. The result of that was that many areas of the ocean, including the Southern Ocean, were not iron-limited as much as they should have been.

(iv) satellite chlorophyll: the data are the GlobColor fields as produced by that project, with no adaptions made. For the (newly-included) Taylor diagrams only ocean points where there is data are considered.

(v) pCO2 values: the alkalinity fields are now shown, compared to the GLODAPv2 data, and discussed.

(vi) air-sea flux: I agree that the only new ingredient is the piston velocity, but because the air-sea flux is such a key quantity in the ocean carbon cycle I think there is value in showing it separately.

(vii) silicate bug: this is not caused by an inherent problem with the model equations; rather, the value of the detrital silicate remineralisation/dissolution parameter was too high. This meant too much was remineralised and became dissolved silicate in the surface ocean, leaving too little for the deep ocean, and the surface concentrations increased throughout the simulations. Despite having a number of tools to monitor the ocean ecosystem while the runs were underway this was somehow not picked up until a very late stage.

(viii) pCO2 seasonal cycle in the future ocean: as surmised, the buffer factor does change to cause this effect. A comment on that has been added to the text.

3. Minor remarks

(a) page 7, line 19: change has been made
(b) p.15, l.20: change has been made
(c) Hovmöller name has been corrected; thank you for pointing this out.

---

## Author Comment (AC4) · 16 Apr 2019

Thank you for your helpful comment, encouraging me to use the latest and most comprehensive version of the GLODAP database and the gridded product, and also pointing out an out-of-date link.

In the revised manuscript I have used GLODAPv2 for all comparisons of DIC, and also for the newly-added comparisons of Total Alkalinity. I have also updated the references and the links in the text, as advised, and removed the sentence about the data product not including the Arctic.

---

## Author Response (AR1)

**Description and evaluation of the Diat-HadOCC model v1.0: the ocean biogeochemical component of HadGEM2-ES**

**Ian Totterdell**

**Response to Editor**

I thank the Reviewers for their detailed and perceptive comments. I have found them very useful in the substantial revision to the manuscript that I have undertaken. Their comments particularly focussed on the evaluation of the model, particularly on the lack of evaluation of the deep ocean and a lack of statistical indices. I have now included both these measures, and also, as requested included plots showing the evaluation of alkalinity, oxygen and AOU. I have also adapted the way I present the seasonal cycles of a number of quantities in the evaluation, making it clear where my method is not significant. Finally, I have updated the data products that I have used for evaluation to WOA 2013 and GLODAPv2 as appropriate.

In the text I have given a more objective assessment of how well the model performs (the Reviewers unanimously identified optimism-bias in the original manuscript), and substantially expanded the description and discussion of the results and the conclusion, now examining where the model can be considered a success and where it has weaknesses. The Reviewers' direction has definitely made this a better manuscript.

However, there is one area of criticism that I have not been able to respond to, namely the call by two of the Reviewers to retune and re-run the model. As I explain in my individual responses, it has just not proved possible to do that; the computing resources have just not been available, and nor are they going to be available in the foreseeable future. I have no control over that. Therefore I have emphasised in the introduction and the conclusions that this manuscript is presenting and describing a model that *was* used as part of an Earth System Model to run simulations for the CMIP5 experiment, and as such is an important record.

I hope these revisions enable the manuscript to be published.

Below is a marked-up version of the manuscript, showing the changes made in response to all the comments.

```
%% Copernicus Publications Manuscript Preparation Template for LaTeX
Submissions
%% ---------------------------------
%% This template should be used for copernicus.cls
%% The class file and some style files are bundled in the Copernicus
Latex Package which can be downloaded from the different journal
webpages.
%% For further assistance please contact the Copernicus Publications at:
publications@copernicus.org
%% http://publications.copernicus.org

%% Please use the following documentclass and Journal Abbreviations for
Discussion Papers and Final Revised Papers.

%% 2-Column Papers and Discussion Papers
\documentclass[gmd, manuscript]{copernicus}

%% Journal Abbreviations (Please use the same for Discussion Papers and
Final Revised Papers)

% Archives Animal Breeding (aab)
% Atmospheric Chemistry and Physics (acp)
% Advances in Geosciences (adgeo)
% Advances in Statistical Climatology, Meteorology and Oceanography
(ascmo)
% Annales Geophysicae (angeo)
% ASTRA Proceedings (ap)
% Atmospheric Measurement Techniques (amt)
% Advances in Radio Science (ars)
% Advances in Science and Research (asr)
% Biogeosciences (bg)
% Climate of the Past (cp)
% Drinking Water Engineering and Science (dwes)
% Earth System Dynamics (esd)
% Earth Surface Dynamics (esurf)
% Earth System Science Data (essd)
% Fossil Record (fr)
% Geographica Helvetica (gh)
% Geoscientific Instrumentation, Methods and Data Systems (gi)
% Geoscientific Model Development (gmd)
% Geothermal Energy Science (gtes)
% Hydrology and Earth System Sciences (hess)
% History of Geo- and Space Sciences (hgss)
% Journal of Sensors and Sensor Systems (jsss)
% Mechanical Sciences (ms)
% Natural Hazards and Earth System Sciences (nhess)
% Nonlinear Processes in Geophysics (npg)
% Ocean Science (os)
% Proceedings of the International Association of Hydrological Sciences
(piahs)
% Primate Biology (pb)
% Scientific Drilling (sd)
```

```
% SOIL (soil)
% Solid Earth (se)
% The Cryosphere (tc)
% Web Ecology (we)
% Wind Energy Science (wes)

%% \usepackage commands included in the copernicus.cls:
%\usepackage[german, english]{babel}
%\usepackage{tabularx}
%\usepackage{cancel}
%\usepackage{multirow}
%\usepackage{supertabular}
%\usepackage{algorithmic}
%\usepackage{algorithm}
%\usepackage{amsthm}
%\usepackage{float}
%\usepackage{subfig}
%\usepackage{rotating}

\begin{document}\sloppy
%\begin{document}

\title{Description and evaluation of the Diat-HadOCC model v1.0: the
ocean biogeochemical component of HadGEM2-ES}

% \Author[affil]{given_name}{surname}

\Author[1]{Ian}{Totterdell}
%\Author[]{}{}
%\Author[]{}{}

\affil[1]{Met Office, Fitzroy Road, Exeter, EX1 3PB, UK}
%\affil[]{ADDRESS}

%% The [] brackets identify the author with the corresponding
affiliation. 1, 2, 3, etc. should be inserted.

\runningtitle{The Diat-HadOCC model}

\runningauthor{I.~J.~Totterdell}

\correspondence{Dr Ian Totterdell (ian.totterdell@metoffice.gov.uk)}

\received{}
\pubdiscuss{} %% only important for two-stage journals
\revised{}
\accepted{}
\published{}

%% These dates will be inserted by Copernicus Publications during the
typesetting process.
```

\firstpage{1}

\maketitle

\begin{abstract}
%Diat-HadOCC is by far the greatest model the world has ever seen.

[revised manuscript text omitted]

%
\frac{d\,FeT}{d\,t} &=& (\;ph_{resp} \cdot R^{Ph}_{c2n}\;+\;dm_{resp}
\cdot R^{Dm}_{c2n}\;+\;ph_{mort} \cdot R^{Ph}_{c2n}\;+\;dm_{mort} \cdot
R^{Dm}_{c2n}\;+\;grz_{DIC}\;+\;grz_{DtC}\; \;dtc_{grz} \nonumber \\
%& &\;+\;dm_{mort} \cdot R^{Dm}_{c2n}\;+\;grz_{DIC}\;+\;grz_{DtC}\;
\;dtc_{grz} \nonumber \\
%& &\;+\;zp_{lin} \cdot R^{Zp}_{c2n}\;+\;zp_{mort} \cdot R^{Zp}_{c2n}\;
\;ph_{PP} \cdot R^{Ph}_{c2n} \nonumber \\
 & &\;+\;zp_{lin} \cdot R^{Zp}_{c2n}\;+\;zp_{mort} \cdot R^{Zp}_{c2n}\;
\;ph_{PP} \cdot R^{Ph}_{c2n}\; \;dm_{PP} \cdot R^{Dm}_{c2n}\;) \cdot
R^{eco}_{fe2c}\;+\;[\,fe_{dust}\,]\; \;fe_{adsorp} \\
%
\frac{d\,Ph}{d\,t} &=&\;ph_{PP}\;-\;ph_{resp}\;-\;ph_{mort}\;-\;ph_{grz}
%
\end{eqnarray}
\begin{eqnarray}
%
\frac{d\,Dm}{d\,t} &=&\;dm_{PP}\;-\;dm_{resp}\;-\;dm_{mort}\;-
\;dm_{grz}\;-\;[\,dm_{sink}\,] \\
%
\frac{d\,DmSi}{d\,t} &=&\;dm_{PP} \cdot R^{Dm}_{si2n}\; \;dmsi_{mort}\;-
\;dmsi_{grz}\; \;[\,dmsi_{sink}\,] \\
%
\frac{d\,Zp}{d\,t} &=&\;grz_{Zp}\;-\;zp_{lin}\;-\;zp_{mort} \\
%
\frac{d\,DtN}{d\,t} &=&\;ph_{mort} \cdot (1 f_{nmp})\;+\;dm_{mort} \cdot
(1 f_{nmp})\;+\;grz_{DtN}\;+\;zp_{mort} \cdot (1
f_{zmrt})\;+\;dm_{bedmrt} \nonumber \\
%& &\;+\;zp_{mort} \cdot (1 f_{zmrt})\;+\;dm_{bedmrt}\; \;dtn_{grz}
\nonumber \\
 & &\; \;dtn_{grz}\; \;dtn_{remin}\; \;[\,dtn_{sink}\,] \\
%
\frac{d\,DtSi}{d\,t} &=&\;dmsi_{mort}\;+\;grz_{dtsi}\;+\;dmsi_{bedmrt}\;-
\;dtsi_{remin}\;-\;[\,dtsi_{sink}\,] \\
%& &\; \;[\,dtsi_{sink}\,] \\
```

```
%
\frac{d\,DtC}{d\,t} &=&\;ph_{mort} \cdot (1-f_{nmp}) \cdot
R^{Ph}_{c2n}\;+\;dm_{mort} \cdot (1-f_{nmp}) \cdot
R^{Dm}_{c2n}\;+\;grz_{DtC}\;+\;zp_{mort} \cdot (1-f_{zmrt}) \cdot
R^{Zp}_{c2n} \nonumber \\
%& &\;+\;grz_{DtC}\;+\;zp_{mort} \cdot (1-f_{zmrt}) \cdot R^{Zp}_{c2n}
\nonumber \\
 & &\;+\;dm_{bedmrt} \cdot R^{Dm}_{c2n}\; \;dtc_{grz}\; \;dtc_{remin}\;
\;[\,dtc_{sink}\,] \\
%
%\end{eqnarray}
%\begin{eqnarray}\label{eqn-SiProc1} \\
%
\frac{d\,DIC}{d\,t} &=&\;ph_{resp} \cdot R^{Ph}_{c2n}\;+\;dm_{resp} \cdot
R^{Dm}_{c2n}\;+\;ph_{mort} \cdot f_{nmp} \cdot R^{Ph}_{c2n}\;+\;
dm_{mort} \cdot f_{nmp} \cdot R^{Dm}_{c2n}\;+\;grz_{DIC} \nonumber \\
%& &\;+\; dm_{mort} \cdot f_{nmp} \cdot
R^{Dm}_{c2n}\;+\;grz_{DIC}\;+\;zp_{lin} \cdot R^{Zp}_{c2n} \nonumber \\
%& &\;+\;zp_{mort} \cdot f_{zmrt} \cdot
R^{Zp}_{c2n}\;+\;dtc_{remin}\;+\;dtc_{bedrmn} \nonumber \\
 & &\;+\;zp_{lin} \cdot R^{Zp}_{c2n}\;+\;zp_{mort} \cdot f_{zmrt} \cdot
R^{Zp}_{c2n}\;+\;dtc_{remin}\;+\;dtc_{bedrmn}\;+\;crbnt \nonumber \\
 & &\;-\;ph_{PP} \cdot R^{Ph}_{c2n}\;-\;dm_{PP} \cdot
R^{Dm}_{c2n}\;+\;[\,CO2_{asf}\,] \label{eqn-DICProc1}
%
\end{eqnarray}
\begin{eqnarray}
%
\\
%
%\frac{d\,Zp}{d\,t} &=&\;grz_{Zp}\;-\;zp_{lin}\;-\;zp_{mort} \\
%
\frac{d\,TAlk}{d\,t} &=&\;2 \cdot crbnt\;-\;\frac{d\,DIN}{d\,t}
\label{eqn-TAlkProc1} \\
%
\frac{d\,Oxy}{d\,t} &=&\;[\,Oxy_{asf}\,]\;-
\;\left(\,\frac{d\,DIC}{d\,t}\;-\;crbnt\;-\;[\,CO2_{asf}\,]\,\right)
\cdot R^{eco}_{o2c}\;+\;resetO 2 \label{eqn-OxyProc1}
%
\end{eqnarray}

The terms in Equation~\ref{eqn-DINProc1} show that the concentration of
dissolved inorganic nitrogen is increased by, in order: a release of
nitrogen associated with respiration by misc-phytoplankton (to keep the
cell's molecular C:N ratio constant: Equation~\ref{eqn-phresp}); a
corresponding release associated with diatom respiration
(Equation~\ref{eqn-dmresp}); fractions of the nitrogen released by the
natural mortalities of misc-phytoplankton and of diatoms (the rest of the
nitrogen in each case passes to sinking detritus $DtN$, see
Equations~\ref{eqn-phmort} and \ref{eqn-dmmort}); a release of nitrogen
due to grazing by zooplankton on misc-phytoplankton, diatoms and detritus
(Equation~\ref{eqn-grzDIN}); losses from zooplankton (mainly associated
with respiration; Equation~\ref{eqn-zplin}); a fraction of the loss due
to zooplankton mortality (natural and due to unmodelled grazing by higher
trophic levels; Equation~\ref{eqn-zpmort}); and nitrogen returned to the
dissolved state by the remineralization of sinking detritus in the water-
column (Equation~\ref{eqn-dtnRmn}) and at the sea-floor
(Equation~\ref{eqn-dtXBdrmn}). Conversely, the final two terms show that
```

the concentration is decreased by uptake by misc-phytoplankton and diatoms to fuel photosynthesis and primary production (respectively Equations~\ref{eqn-PhPP1} and \ref{eqn-DmPP1}). The processes of nitrogen deposition from the atmosphere, inflow from rivers and estuaries, release from sediments, nitrogen fixation and denitrification are not included in the Diat-HadOCC model.

Equation~\ref{eqn-SiProc1} shows that the concentration of dissolved silicate is increased by the dissolution of detrital silicate in the water-column (Equation~\ref{eqn-dtsiRmn}) and at the sea-floor (Equation~\ref{eqn-dtXBdrmn}), while it is decreased by uptake by diatoms to produce opaline shells in association with growth (Equation~\ref{eqn-DmPP1}; the Si:N ratio $R^{Dm}_{si2n}$ is a function of the dissolved iron concentration following Equation~\ref{eqn-FeFctr1}). As with DIN, there are no inputs/losses of Si from/to the atmosphere, rivers, estuaries or sediments.

Each of the processes increasing or decreasing the dissolved inorganic nitrogen concentration has a counterpart that increases or decreases the dissolved inorganic carbon concentration; Equation~\ref{eqn-DICProc1} shows those processes and also the two processes that affect $DIC$, namely the formation and dissolution of solid calcium carbonate ($crbnt$, Equation~\ref{eqn-crbnt}) and the air-sea flux of CO$_2$ (Equation~\ref{eqn-Xasf}). Apart from the air-sea flux of CO$_2$ there are no other inputs/losses of inorganic carbon to the ocean.

In this model, biologically-mediated changes to the total alkalinity are associated with either the formation and dissolution of solid calcium carbonate or the uptake and release of dissolved inorganic nitrogen; Equation~\ref{eqn-TAlkProc1} shows how these processes are related to the alkalinity. Because the carbonate ion $CO_3^{2+}$ has two charges the change in the alkalinity due to $crbnt$ is double the change in $DIC$, and of opposite sign. Although uptake by phytoplankton of dissolved nitrate does not directly change the alkalinity it is usually associated with a balancing release of $OH^-$ ions which does change it \citep{goldman80}. In the model all the $DIN$ taken up is assumed to be nitrate, but in the real ocean some of the nutrient will be dissolved ammonia, $NH_4^+$, which is associated with a release of $H^+$ ions that change the alkalinity in the opposite sense to the $OH^-$ ions; the model's omission of ammonium ions is not a great problem as any that is taken up for growth will likely have been produced locally shortly before, given that ammonium has a short residence time in the upper water-column.

Dissolved oxygen is included in the model as a diagnostic tracer: its concentration is changed biological processes (as well as physical and chemical ones) but does not affect any other model state variable. It has particular value as a diagnostic of the respiration of organic matter at depth in the water-column, but also allows for the simulation of oxygen-minimum zones, and their evolution under climate change. It is assumed for the model that all respiration of organic matter is aerobic, so the same O:C ratio $R^{eco}_{o2c}$ can be used for all ecosystem processes, including both uptake and release of $O_2$; the second term in Equation~\ref{eqn-OxyProc1} (i.e. within the large brackets) connects such oxygen fluxes to those of organic carbon. The first term in that equation relates to the air-sea flux of oxygen. The third term, $resetO_2$, is included to prevent the dissolved oxygen concentration going negative: at the end of each time-step, if the combination of

physical fluxes and biological processes have taken the concentration in any grid-cell below zero, the concentration is re-set to zero and the amount that has been added to the model recorded. The column inventory of such re-set additions is calculated and subtracted from the surface layer; because that layer is in close contact with the atmosphere this adjustment should never reduce the surface concentration to zero (and in the CMIP5 simulations never came close to doing so anywhere). This approach was adopted in the model primarily to prevent negative concentrations of dissolved $O_2$ while conserving the global $O_2$ inventory, but it can be loosely related to real-world processes: in the model aerobic respiration continues at a rate independent of the oxygen concentration, but in low-oxygen zones in the real ocean anaerobic respiration that is slower and that produces methane rather than CO$_2$ would replace it. The methane produced will mix along isopycnals and vertically, and while some will escape to the atmosphere and some will be oxidised to CO$_2$ in deep but more oxygen-rich waters a major location for its oxidation is in the surface ocean, removing O$_2$ from the water there.

\subsection{Diatoms and misc-Phytoplankton}

\begin{eqnarray}
%
\frac{d\,Ph}{d\,t} &=&\;ph_{PP}\;-\;ph_{resp}\;-\;ph_{mort}\;-\;ph_{grz} \label{eqn-PhProc1} \\
%
\frac{d\,Dm}{d\,t} &=&\;dm_{PP}\;-\;dm_{resp}\;-\;dm_{mort}\;-\;dm_{grz}\;-\;[\,dm_{sink}\,] \label{eqn-DmProc1} \\
%
\frac{d\,DmSi}{d\,t} &=&\;dm_{PP} \cdot R^{Dm}_{si2n}\;-\;dmsi_{mort}\;-\;dmsi_{grz}\;-\;[\,dmsi_{sink}\,] +
\label{eqn-DmSiProc1}
%
\end{eqnarray}

\subsection{Growth of diatoms and misc-Phytoplankton}

[revised manuscript text omitted]

\subsection{Zooplankton and grazing}

\begin{equation}
%\frac{d\,Zp}{d\,t} &=&\;grz_{Zp}\;-\;zp_{lin}\;-\;zp_{mort} \\
\frac{d\,Zp}{d\,t} = \;grz_{Zp}\;-\;zp_{lin}\;-\;zp_{mort} \label{eqn-dZpdt}
\end{equation}

Zooplankton biomass (quantified by its nitrogen content) is increased by the grazing (of misc-phytoplankton, diatoms and detrital particles; see Equation~\ref{eqn-grzZp}) and decreased by losses such as respiration (Equation~\ref{eqn-zplin}) and by density-dependent predation by the un-modelled higher trophic levels (Equation~\ref{eqn-zpmort}).

The grazing function used in the Diat-HadOCC model differs from that used in the HadOCC model in that it uses a `switching' grazer similar to that used in Fasham et al. (1990; hereafter FDM90). It is noted that some authors \citep[e.g. ][]{gentleman03} recommend against using such a formulation because it can lead to reduced intake when food resources are increasing. The single zooplankton consumes diatoms, misc-Phytoplankton and (organic) detrital particles. As in FDM90 the realised preference $dprf_{X}$ for each food type depends on that type's abundance and on the base preferences $bprf_{X}$:
\begin{eqnarray}
%

```
dpr\!f_{denom} &=&\;bprf_{Dm} \cdot R^{Dm}_{b2n} \cdot Dm\;+\;bprf_{Ph}
\cdot R^{Ph}_{b2n} \cdot Ph\;+\;bprf_{Dt} \cdot (\,R^{DtN}_{b2n} \cdot
DtN\,+\,R^{DtC}_{b2c} \cdot DtC\,) \label{eqn-dprfdnm} \\
%& &\;+\;bprf_{Dt} \cdot (\,R^{DtN}_{b2n} \cdot DtN\,+\,R^{DtC}_{b2c}
\cdot DtC\,) \\
%
dprf_{Dm} &=&\;\frac{bprf_{Dm} \cdot R^{Dm}_{b2n} \cdot Dm}{dprf_{denom}}
\label{eqn-dprfDm} \\
%
dprf_{Ph} &=&\;\frac{bprf_{Ph} \cdot R^{Ph}_{b2n} \cdot Ph}{dprf_{denom}}
\label{eqn-dprfPh} \\
%
dprf_{Dt} &=&\;\frac{bprf_{Dt} \cdot (\,R^{DtN}_{b2n} \cdot
DtN\,+\,R^{DtC}_{b2c} \cdot DtC\,)}{dprf_{denom}}
 \label{eqn-dprfDt}
%
\end{eqnarray}
where, if $M_{N}$ and $M_{C}$ are the respective atomic weights of
nitrogen and carbon (14.01 and 12.01 g Mol$^{-1}$) and $R^{Rdfld}_{c2n}$
is the Redfield C:N ratio (106 Mol C : 16 Mol N), then the $R^{X}_{b2Y}$
terms convert from nitrogen or carbon units to biomass units that allow
the various potential food items to be compared:
\begin{eqnarray}
%
E &=& ( M_{N} + M_{C} \cdot R^{Rdfld}_{c2n} )^{-1} \nonumber \\
%
R^{Ph}_{b2n} &=& E \cdot ( M_{N} + M_{C} \cdot R^{Ph}_{c2n} ) \label{eqn-
rb2nPh} \\
%
R^{Dm}_{b2n} &=& E \cdot ( M_{N} + M_{C} \cdot R^{Dm}_{c2n} ) \label{eqn-
rb2nDm} \\
%
R^{Zp}_{b2n} &=& E \cdot ( M_{N} + M_{C} \cdot R^{Zp}_{c2n} ) \label{eqn-
rb2nZp} \\
%
R^{DtN}_{b2n} &=& E \cdot M_{N} \label{eqn-rb2nDtN} \\
%
R^{DtC}_{b2c} &=& E \cdot M_{C}
 \label{eqn-rb2nDtC}
%
\end{eqnarray}
Note that the base preference values supplied (or calculated as a
function of iron-limitation) $bprf_{X}$ are normalised so that they sum
up to 1. The available food is:
%\begin{eqnarray}
\begin{equation}
%
%food &=& dprf_{Dm} \cdot R^{Dm}_{b2n} \cdot Dm\;+\;dprf_{Ph} \cdot
R^{Ph}_{b2n} \cdot Ph \nonumber \\
%& &\;+\;dprf_{Dt} \cdot (\,R^{DtN}_{b2n} \cdot DtN\,+\,R^{DtC}_{b2c}
\cdot DtC\,)
food = dprf_{Dm} \cdot R^{Dm}_{b2n} \cdot Dm\;+\;dprf_{Ph} \cdot
R^{Ph}_{b2n} \cdot Ph\;+\;dprf_{Dt} \cdot (\,R^{DtN}_{b2n} \cdot
DtN\,+\,R^{DtC}_{b2c} \cdot DtC\,) \label{eqn-food}
%
\end{equation}
%\end{eqnarray}
and the grazing rates on the various model state variables are:
```

```
\begin{eqnarray}
%
dm_{grz} &=&\;\frac{dprf_{Dm} \cdot Dm \cdot g_{max} \cdot R^{Zp}_{b2n}
\cdot Zp}{g_{sat}\;+\;food} \label{eqn-dmGrz} \\
%
dmsi_{grz} &=&\;\frac{dprf_{Dm} \cdot DmSi \cdot g_{max} \cdot
R^{Zp}_{b2n} \cdot Zp}{g_{sat}\;+\;food} \label{eqn-dmsiGrz} \\
%
ph_{grz} &=&\;\frac{dprf_{Ph} \cdot Ph \cdot g_{max} \cdot R^{Zp}_{b2n}
\cdot Zp}{g_{sat}\;+\;food} \label{eqn-phGrz} \\
%
dtn_{grz} &=&\;\frac{dprf_{Dt} \cdot DtN \cdot g_{max} \cdot R^{Zp}_{b2n}
\cdot Zp}{g_{sat}\;+\;food}
\label{eqn-dtnGrz} \\
%
dtc_{grz} &=&\;\frac{dprf_{Dt} \cdot DtC \cdot g_{max} \cdot R^{Zp}_{b2n}
\cdot Zp}{g_{sat}\;+\;food} \label{eqn-dtcGrz}
%
\end{eqnarray}
\begin{equation}
%
dtc_{grz} = frac{dprf_{Dt} \cdot DtC \cdot g_{max} \cdot R^{Zp}_{b2n}
\cdot Zp}{g_{sat}\;+\;food}
%
\end{equation}

A fraction $(1-f_{ingst})$ of the grazed material is not ingested: of
this, a fraction $f_{messy}$ returns immediately to solution as $DIN$ and
$DIC$ while the rest becomes detritus. All of the grazed diatom silicate
$DmSi$ immediately becomes detrital silicate $DtSi$. Of the organic
material that is ingested, a source-dependent fraction ($\beta^{X}$) of
the nitrogen and of the carbon is assimilatable while the remainder is
egested from the zooplankton gut as detrital nitrogen $DtN$ or carbon
$DtC$. The amount of assimilatable material that is actually assimilated
by the zooplankton $grz_{Zp}$ is governed by its C:N ratio compared to
that of the zooplankton: as much as possible is assimilated, with the
remainder passed out immediately as $DIN$ or $DIC$.
\begin{eqnarray}
%
assim_{N} &=&\;f_{ingst} \cdot (\,\beta^{Dm} \cdot
dm_{grz}\;+\;\beta^{Ph} \cdot ph_{grz}\;+\;\beta^{Dt} \cdot dtn_{grz})
\label{eqn-asimN} \\
%
assim_{C} &=&\;f_{ingst} \cdot (\,\beta^{Dm} \cdot R^{Dm}_{c2n} \cdot
dm_{grz}\;+\;\beta^{Ph} \cdot R^{Ph}_{c2n} \cdot ph_{grz}\;+\;\beta^{Dt}
\cdot dtn_{grz}) \label{eqn-asimC} \\
%& &\;+\;\beta^{Dt} \cdot dtn_{grz}) \\
%
grz_{Zp} &=&\;MIN
\left(\,assim_{N},\;\frac{assim_{C}}{R^{Zp}_{c2n}}\,\right) \label{eqn-
grzZp} \\
%
grz_{DtN} &=&\;(1-f_{ingst}) \cdot (1-f_{messy}) \cdot
(\,dm_{grz}\;+\;ph_{grz}\;+\;dtn_{grz}\,) \nonumber \\
%& &\;+\;dtn_{grz}\,)\;+\;f_{ingst} \cdot (\,(1-\beta^{Dm}) \cdot
dm_{grz} \nonumber \\
```

```latex
 & &\;+\;f_{ingst} \cdot (\,(1-\beta^{Dm}) \cdot dm_{grz}\;+\;(1-
\beta^{Ph}) \cdot ph_{grz}\;+\;(1-\beta^{Dt}) \cdot dtn_{grz}\,)
\label{eqn-grzDtN} \\
%
grz_{DtC} &=&\;(1-f_{ingst}) \cdot (1-f_{messy}) \cdot (\,R^{Dm}_{c2n}
\cdot dm_{grz}\;+\;R^{Ph}_{c2n} \cdot ph_{grz}\;+\;dtc_{grz}\,) \nonumber
\\
%& &\;+\;dtc_{grz}\,)\;+\;f_{ingst} \cdot (\,(1-\beta^{Dm}) \cdot
R^{Dm}_{c2n} \cdot dm_{grz} \nonumber \\
 & &\;+\;f_{ingst} \cdot (\,(1-\beta^{Dm}) \cdot R^{Dm}_{c2n} \cdot
dm_{grz}\;+\;(1-\beta^{Ph}) \cdot R^{Ph}_{c2n} \cdot ph_{grz}\;+\;(1-
\beta^{Dt}) \cdot dtc_{grz}\,) \\
%& &\;+\;\beta^{Dt} \cdot dtn_{grz}) \\
%
grz_{Zp} &=&\;MIN
\left(\,assim_{N},\;\frac{assim_{C}}{R^{Zp}_{c2n}}\,\right) \\
%
grz_{DtN} &=&\;(1-f_{ingst}) \cdot (1-f_{messy}) \cdot
(\,dm_{grz}\;+\;ph_{grz}\;+\;dtn_{grz}\,) \nonumber \\
%& &\;+\;dtn_{grz}\,)\;+\;f_{ingst} \cdot (\,(1-\beta^{Dm}) \cdot
dm_{grz} \nonumber \\
 & &\;+\;f_{ingst} \cdot (\,(1-\beta^{Dm}) \cdot dm_{grz}\;+\;(1-
\beta^{Ph}) \cdot ph_{grz}\;+\;(1-\beta^{Dt}) \cdot dtn_{grz}\,) \\
%
grz_{DtC} &=&\;(1-f_{ingst}) \cdot (1-f_{messy}) \cdot (\,R^{Dm}_{c2n}
\cdot dm_{grz}\;+\;R^{Ph}_{c2n} \cdot ph_{grz}\;+\;dtc_{grz}\,) \nonumber
\\
%& &\;+\;dtc_{grz}\,)\;+\;f_{ingst} \cdot (\,(1-\beta^{Dm}) \cdot
R^{Dm}_{c2n} \cdot dm_{grz} \nonumber \\
 & &\;+\;f_{ingst} \cdot (\,(1-\beta^{Dm}) \cdot R^{Dm}_{c2n} \cdot
dm_{grz}\;+\;(1-\beta^{Ph}) \cdot R^{Ph}_{c2n} \cdot ph_{grz}\;+\;(1-
\beta^{Dt}) \cdot dtc_{grz}\,) \\
%
\label{eqn-grzDtC} \\
%
grz_{DtSi} &=&\;dmsi_{grz} \label{eqn-grzDtSi} \\
%
grz_{DIN} &=&\;(1-f_{ingst}) \cdot f_{messy} \cdot
(\,dm_{grz}\;+\;ph_{grz}\;+\;dtn_{grz}\,)\;+\;MAX
\left(\,0,\;assim_{N}\;-\;\frac{assim_{C}}{ R^{Zp}_{c2n}}\,\right)
\label{eqn-grzDIN} \\
%& &\;+\;dtn_{grz}\,)\;+\;MAX \left(\,0,\;assim_{N}\;-\;\frac{assim_{C}}{
R^{Zp}_{c2n}}\,\right) \\
%
grz_{DIC} &=&\;(1-f_{ingst}) \cdot f_{messy} \cdot (\,R^{Dm}_{c2n} \cdot
dm_{grz}\;+\;R^{Ph}_{c2n} \cdot ph_{grz}\;+\;dtc_{grz}\,) \nonumber \\
 & &\;+\;MAX(\,0,\;assim_{C}\;-\;assim_{N} \cdot R^{Zp}_{c2n})
\label{eqn-grzDIC}
%
\end{eqnarray}

\subsection{Other processes}

The other loss terms for diatoms, misc-Phytoplankton and zooplankton are:
\begin{eqnarray}
%
dm_{resp} &=&\;\Pi^{Dm}_{resp} \cdot Dm \\
```

```latex
%&      &\;+\;dtn_{grz}\,)\;+\;MAX \left(\,0,\;assim_{N}\; \;\frac{assim_{C}}{
R^{Zp}_{c2n}}\,\right) \\
%
grz_{DIC} &=&\;(1-f_{ingst}) \cdot f_{messy} \cdot (\,R^{Dm}_{c2n} \cdot
dm_{grz}\;+\;R^{Ph}_{c2n} \cdot ph_{grz}\;+\;dtc_{grz}\,) \nonumber \\
 & &\;+\;MAX(\,0,\;assim_{C}\; \;assim_{N} \cdot R^{Zp}_{c2n})
%
\end{eqnarray}

\subsection{Other processes}

The other loss terms for diatoms, misc Phytoplankton and zooplankton are:
\begin{eqnarray}
%
dm_{resp} &=&\;\Pi^{Dm}_{resp} \cdot Dm \label{eqn-dmresp} \\
%
ph_{resp} &=&\;\Pi^{Ph}_{resp} \cdot Ph \label{eqn-phresp} \\
%
dm_{mort} &=&\;\Pi^{Dm}_{mort} \cdot Dm^{2} \label{eqn-dmmort} \\
%
dmsi_{mort} &=&\;\Pi^{Dm}_{mort} \cdot Dm \cdot DmSi \label{eqn-dmsimort}
\\
%
ph_{mort} &=&\;\Pi^{Ph}_{mort} \cdot Ph^{2} \hspace{2.7cm} (Ph >
ph_{min}) \nonumber \\
  &=&\;0 \hspace{4.3cm} (Ph < ph_{min}) \\
%
ph_{mort} &=&\;\Pi^{Ph}_{mort} \cdot Ph^{2} \hspace{2.7cm} (Ph >
ph_{min}) \nonumber \\
  &=&\;0 \hspace{4.3cm} (Ph < ph_{min}) \label{eqn-phmort} \\
%
zp_{lin} &=&\;\Pi^{Zp}_{lin} \cdot Zp \label{eqn-zplin} \\
%
zp_{mort} &=&\;\Pi^{Zp}_{mort} \cdot Zp^{2} \label{eqn-zpmort}
%
\end{eqnarray}

%
\end{eqnarray}
```

In the above equations $ph_{min}$ is a set (low) concentration of $Ph$ below which the natural mortality of misc-Phytoplankton is set to zero; the inclusion of this term was a pragmatic and necessary choice in an early version of the model to prevent the misc-Phytoplankton dying out in certain parts of the seasonal cycle at high latitudes (it was not found to be necessary to include a similar term for diatoms). It can be rationalised as representing the ability of phytoplankton to enter a "cyst" state under certain stressful conditions.
Although respiration involves a release of carbon (as $CO_2$) the fixed C:N ratios used in the models for misc-Phytoplankton, Diatoms and Zooplankton require a balancing release of nitrogen from those model compartments. The "natural mortality" of both phytoplankton variables refers to cell-death, particularly including that caused by viral infections, which will be density-dependent. The $zp_{mort}$ refers primarily to zooplankton losses due to predation by un-modelled higher trophic levels, and is the closure term of the modelled ecosystem.

```latex
\subsubsection{Detrital sinking and remineralisation}

\begin{eqnarray}
%
\frac{d\,DtN}{d\,t} &=&\;ph_{mort} \cdot (1-f_{nmp})\;+\;dm_{mort} \cdot
(1-f_{nmp})\;+\;grz_{DtN}\;+\;zp_{mort} \cdot (1-
f_{zmrt})\;+\;dm_{bedmrt} \nonumber \\
%& &\;+\;zp_{mort} \cdot (1-f_{zmrt})\;+\;dm_{bedmrt}\;-\;dtn_{grz}
\nonumber \\
 & &\;-\;dtn_{grz}\;-\;dtn_{remin}\;-\;[\,dtn_{sink}\,] \label{eqn-
dDtNdt} \\
%
\frac{d\,DtSi}{d\,t} &=&\;dmsi_{mort}\;+\;grz_{DtSi}\;+\;dmsi_{bedmrt}\;-
\;dtsi_{remin}\;-\;[\,dtsi_{sink}\,] \label{eqn-dDtSidt} \\
%& &\;-\;[\,dtsi_{sink}\,] \\
%
\frac{d\,DtC}{d\,t} &=&\;ph_{mort} \cdot (1-f_{nmp}) \cdot
R^{Ph}_{c2n}\;+\;dm_{mort} \cdot (1-f_{nmp}) \cdot
R^{Dm}_{c2n}\;+\;grz_{DtC}\;+\;zp_{mort} \cdot (1-f_{zmrt}) \cdot
R^{Zp}_{c2n} \nonumber \\
%& &\;+\;grz_{DtC}\;+\;zp_{mort} \cdot (1-f_{zmrt}) \cdot R^{Zp}_{c2n}
\nonumber \\
 & &\;+\;dm_{bedmrt} \cdot R^{Dm}_{c2n}\;-\;dtc_{grz}\;-\;dtc_{remin}\;-
\;[\,dtc_{sink}\,] \label{eqn-dDtCdt}
%
\end{eqnarray}
```

All detrital material sinks at a constant speed $V_{Dt}$ at all depths.
Diatoms (and its associated silicate) sinks at a constant speed $V_{Dm}$
at all depths. Detrital remineralisation (of $DtN$ and $DtC$)is depth-
dependent, the specific rate varying as the reciprocal of depth but with
a maximum value. This functional form gives a depth variation of detritus
consistent with the \cite{martin87} power-law curve. Dissolution of opal
does not vary with depth.

```latex
\begin{eqnarray}
%
dtn_{remin} &=&\;DtN \cdot MIN
\left(\,\Pi^{DtN}_{rmnmx},\;\frac{\Pi^{DtN}_{rmndd}}{z}\,\right)
\label{eqn-dtnRmn} \\
%
dtc_{remin} &=&\;DtC \cdot MIN
\left(\,\Pi^{DtC}_{rmnmx},\;\frac{\Pi^{DtC}_{rmndd}}{z}\,\right)
\label{eqn-dtcRmn} \\
%
dtsi_{remin} &=&\;DtSi \cdot \Pi^{DtSi}_{rmn} \label{eqn-dtsiRmn} \\
%
dt(n,c,si)_{sink} &=&\;V_{Dt} \cdot \frac{d\,Dt(N,C,Si)}{d\,z}
\label{eqn-dtXSink} \\
%
d(m,msi)_{sink} &=&\;V_{Dm} \cdot \frac{d\,D(m,mSi)}{d\,z}
 \label{eqn-dXSink}
%
\end{eqnarray}
```
Since there are no sediments in the Diat-HadOCC model, all detritus that
sinks to the sea-floor is instantly remineralised to N, C or Si and
spread through the lowest three layers (above the sea-floor). Spreading
over the bottom three levels is a numerical artifice to prevent excessive
build-up of high concentrations (below regions of high primary

productivity and sinking detritus) in bathymetric canyons that are too narrow to support advection and so rely on weak vertical mixing to redistribute N, C or Si being introduced by the instant sea-floor remineralisation (such high concentrations would themselves be artifacts of the model). It is reasoned that where the ocean is (thousands of metres) deep the time required for dissolved inorganic nutrients and carbon to return to the euphotic zone will be dominated by the slow deep circulation and mixing, and shortening the path by at most a couple of levels will not significantly affect this time; while on the shallow shelves the instant transport upwards through two levels will actually partially mitigate the absence from the model of tidal mixing, which is very important in such environments in the real ocean. Diatoms (and associated silicate) that sink to the sea-floor instantly die and become $DtN$, $DtC$ and $DtSi$, as appropriate, in the lowest layer. Therefore, if $btmflx_{Y}$ is the value of [$Y_{sink}$] at the sea-floor:
\begin{eqnarray}
%
dt(n,c,si)_{bedrmn} &=&\;\frac{btmflx_{Dt(N,C,Si)}}{\Delta_{b3l}}
\hspace{1.15cm} (\,btm\;3\;lyrs\,) \nonumber \\
 &=&\;0 \hspace{3.6cm} (\,above\;btm\;3\;lyrs\,) \label{eqn-dtXBdrmn} \\
%
(dm,dmsi)_{bedmrt} &=&\;\frac{btmflx_{(dm,dmsi)}}{\Delta_{b1l}}
\hspace{1.35cm} (\,bottom\;lyr\,) \nonumber \\
 &=&\;0 \hspace{3.6cm} (\,other\;lyrs\,)
 \label{eqn-dXBdrmn}
%
\end{eqnarray}
where $btmflx_{X}$ is the sinking flux of $X$ to the sea-floor and $\Delta_{bMl}$ is the combined thickness of the bottom $M$ layers (of course, which layers those are will vary according to the location).

\subsubsection{The iron cycle}

\begin{eqnarray}
%\begin{equation}
%
\frac{d\,FeT}{d\,t} &=& (\;ph_{resp} \cdot R^{Ph}_{c2n}\;+\;dm_{resp}
\cdot R^{Dm}_{c2n}\;+\;ph_{mort} \cdot R^{Ph}_{c2n}\;+\;dm_{mort} \cdot
R^{Dm}_{c2n}\;+\;grz_{DIC}\;+\;grz_{DtC}\;-\;dtc_{grz} \nonumber \\
%& &\;+\;dm_{mort} \cdot R^{Dm}_{c2n}\;+\;grz_{DIC}\;+\;grz_{DtC}\;-
\;dtc_{grz} \nonumber \\
%& &\;+\;zp_{lin} \cdot R^{Zp}_{c2n}\;+\;zp_{mort} \cdot R^{Zp}_{c2n}\;-
\;ph_{PP} \cdot R^{Ph}_{c2n} \nonumber \\
 & &\;+\;zp_{lin} \cdot R^{Zp}_{c2n}\;+\;zp_{mort} \cdot R^{Zp}_{c2n}\;-
\;ph_{PP} \cdot R^{Ph}_{c2n}\;-\;dm_{PP} \cdot R^{Dm}_{c2n}\;) \cdot
R^{eco}_{fe2c}\;+\;[\,fe_{dust}\,]\;-\;fe_{adsorp} \label{eqn-dFeTdt}
%
%\end{equation}
\end{eqnarray}

[revised manuscript text omitted]

```latex
\begin{equation}
X_{asf} = Vp_X \cdot (X_{sat} - X_{surf}) \label{eqn-Xasf}
\end{equation}
```

The piston velocity (in m/s) is a function of the 10m wind-speed, $U$ (using the Wanninkhof 1992 formulation, normalised for a Schmidt number of 660), the gas-specific Schmidt number $Sch$ and the fraction of the grid-box area that is open water $A_{ow}$:

```latex
\begin{equation}
Vp_X = A_{ow} \cdot ( f_U \cdot U^2 \times 0.01/3600.0 ) \cdot
(Sch_X/660)^{-1/2} \label{eqn-Vpx}
\end{equation}
```

where $f_U$ is a coefficient taking the value 0.31 if wind-speed averaged over a day or less is used (e.g. in a coupled model) or 0.39 if monthly-mean wind-speed is used \citep{wanninkhof92}.

In the case of oxygen O$_{2,surf}$ is the model oxygen concentration, while the surface ocean is assumed to be fully saturated in equilibrium so O$_{2,sat}$ is equal to the solubility $C_{O_2}$ (calculated in units of ml $O_2$/l, and converted to model units before use). That is calculated using Equation 8 of \citep{garcia92}, but removing the spurious "$A_3 \cdot T_s^2$" term found at the end of the first line (as in the o2sato.f subroutine in the OCMIP-2 Biotic-HOWTO documentation, available at http://ocmip5.ipsl.jussieu.fr/OCMIP/phase2/simulations/Biotic/boundcond/o2sato.f). The solubility coefficients used in the OCMIP-2 subroutine, originally from \cite{benson84} and recommended by \cite{garcia92}, are used here. Note that in HadGEM2-ES the sea-level pressure is assumed to be always 1 atmosphere, everywhere. Therefore the equation is:

```latex
\begin{eqnarray}
C_{O_2} &=& exp( 2.00907 + 3.22014T_s + 4.05010T_s^2 + 4.94457 T_s^3 -
0.256847T_s^4 + 3.88767T_s^5 \nonumber \\
```

```
      & & - S \cdot ( 6.24523 + 7.37614T_s + 10.3410T_s^2 + 8.17083T_s^3
) \times 10^{-3} - 4.88682 \times 10^{-3} \cdot S^2 ) \label{eqn-CO}
\end{eqnarray}
```
where sea-surface temperature $T$ has units of $^{\circ}$C, salinity $S$ has units of permil and where $T_s = ln[(298.15 - T)(273.15 + T)^{-1}]$. $C$`_2}$ can be converted to units of mol $O_2$/m$^3$ by dividing by the molar volume, 22.3916 l/mol. The Schmidt number is calculated according to \cite{keeling98}:
```
\begin{equation}
Sch_{CO_2} = 1638.0 - 81.83T_l + 1.483T_l^2 - 0.008004T_l^3 \label{eqn-SchO2}
\end{equation}
```
where $T_l = max( -2.0, min( 40.0, T ) )$, protecting the calculation from crashing if the physical ocean model should produce unreasonably low or high sea-surface temperatures.

[revised manuscript text omitted]

\begin{eqnarray}
A_{C,u} &=& [HCO_3^-] + 2[CO_3^{2-}] \label{eqn-ACuDef} \\
 &=& A_{T,u} - Q_W \cdot \chi_{x,i} + Q_p / \chi_{x,i} - B_T/
\left(1+\frac{Q_B}{\chi_{x,i}} \right)
 \label{eqn-ACuNum}
\end{eqnarray}
where
\begin{eqnarray}
Q_p &=& \sqrt{K_1 \cdot K_2} \\
\label{eqn-Qp} \\
Q_r &=& \sqrt{\frac{K_1}{K_2}} \\
\label{eqn-Qr} \\
Q_B &=& \frac{Q_p}{K_B} \\
\label{eqn-QB} \\

```latex
Q_W &=& \frac{K_W}{ZQ_p} \label{eqn-QW} \\
\chi &=& \frac{ZQ_p}{[H^+]}
\end{eqnarray}

 \label{eqn-Chi}
\end{eqnarray}
```

Equations 86~\ref{eqn-DIC} and 95~\ref{eqn-ACuDef} can be re-arranged and combined with equations 87, 89, 97, 98Equations~\ref{eqn-K1def}, \ref{eqn-K2def}, \ref{eqn-Qp}, \ref{eqn-Qr} and 101\ref{eqn-Chi} to give: a quadratic in $\chi_{y,i}$:

```latex
\begin{equation}
(2 DIC_u - A_{C,u}) \cdot \chi_{y,i}^2 - ZQ_r \cdot ( A_{c,u} - DIC_u )
\cdot \chi_{y,i} - A_{C,u} = 0 \label{eqn-quadChiYI}
\end{equation}
```

which has the solution

```latex
\begin{equation}
\chi_{y,i} = 0.5 ( ZQ_r \cdot ( A_{c,u} - DIC_u ) + \sqrt{( ZQ_r^2 \cdot
( A_{c,u} - DIC_u )^2 + 4A_{C,u} \cdot (2 DIC_u - A_{C,u}) ) })/(2 DIC_u
- A_{C,u})
 \label{eqn-solnChiYI}
\end{equation}
```

When $\chi_{y,i}$ and $\chi_{x,i}$ are equal the value of $\chi$ that is consistent with both the $A_{C,u}$ and the $DIC_u$ values (for the current temperature and salinity) has been found, so [H$_2$CO$_3$] can be found from equations 86, 87 and 89.Equations~\ref{eqn-DIC}, \ref{eqn-K1def} and \ref{eqn-K2def}. While the two estimates of $\chi$ are not equal however, the secant method of similar triangles \citep{acton70} is used to find an updated estimate $\chi_{x,i+1}$ for input into the next iteration of equation 96Equation~\ref{eqn-ACuNum} by minimising $\chi_y - \chi_x$. The two similar triangles are right-angled and have sides of length $(\chi_{x,i+1}-\chi_{x,i}, \chi_{y,i}-\chi_{x,i})$ and $(\chi_{x,i+1}-\chi_{x,i-1}, \chi_{y,i-1}-\chi_{x,i-1})$ respectively; equating the ratios of these two triangles' sides and re-arranging gives

```latex
\begin{equation}
\chi_{x,i+1} = \frac{\chi_{x,i-1} \cdot \chi_{y,i} - \chi_{x,i} \cdot
\chi_{y,i-1}}{(\chi_{y,i}-\chi_{y,i-1}) - (\chi_{x,i}-\chi_{x,i-1}}
 \label{eqn-ChiXIp1}
\end{equation}
```

This calculation can be iterated until the fractional change in successive estimates is less than a certain amount (e.g. 10$^{-5}$). However, in the implementation used for HadGEM2-ES the calculation was iterated eight times; it had been found that the convergence criterion was always satisfied in 6 iterations, and given the computer architecture it was more computationally efficient to run that way than to repeatedly test for convergence.

Once the carbonic acid concentration has been determined (and converted back to model units) it can be used as $CO_{2,surf}$ in the air-sea flux calculation. Other diagnostic quantities can also be calculated: $pCO_2$ and $pH$ (the latter from the H$^+$ concentration).

```latex
\section{Description of experiments}
```

The Diat-HadOCC model formed the ocean biogeochemical component of the HadGEM2-ES Earth System model \citep{collins11}, which is part of the HadGEM2 family of coupled climate models \citep{hadgem2dt11}. Full details of the model set-up for the experiments described here can be found in those references, but a brief description is given here.

The atmospheric physical model has a horizontal resolution of 1.25$^{\circ}$ latitude by 1.875$^{\circ}$ longitude, and a vertical resoltion of 38 layers (to a height of 39 km).
A timestep of 30 minutes is used.
Eight species of aerosol are included in the atmosphere, as well as a representation of mineral dust (described in more detail below). The UK Chemistry and Aerosols (UKCA) model \citep{oconnor14} describes the atmospheric chemistry.
MOSES II \citep{essery03} is used for the land surface scheme, with additional processes and components as described in papers about the derived JULES scheme by \cite{best11} and \cite{clark11}. The hydrology includes a river-routing sub-model based on the TRIP scheme \citep{oki98}, which supplies freshwater (but not nutrients, carbon or alkalinity) to the ocean.
The TRIFFID dynamic vegetation model (Cox, 2001; Clark et al.\, 2011) and a four-pool implementation of the RothC soil carbon model (Coleman and Jenkinson 1996,1999) are used to represent the terrestrial carbon cycle.
TRIFFID calculates the growth and phenology of five plant functional types (broad-leaf trees, needle-leaf trees, C3 grasses, C4 grasses and shrubs) so that the (terrestrial) Gross Primary Production (GPP), and the Net Primary Production (NPP) can be determined, and thereby also the terrestrial sources and sinks of atmospheric carbon.

The ocean physical model is based on that described in \cite{johns06}, with developments as detailed in the paper by \cite{hadgem2dt11}.
It has a longitudinal resolution of 1$^{\circ}$, while the latitudinal resolution is also 1$^{\circ}$ poleward of 30$^{\circ}$ (
[revised manuscript text omitted]

%% The following commands are for the statements about the availability of data sets and/or software code corresponding to the manuscript.
%% It is strongly recommended to make use of these sections in case data sets and/or software code have been part of your research the article is based on.

\codeavailability{Due to intellectual property right restrictions, the author cannot provide either the source code or documentation papers for the Unified Model (UM). The Met Office Unified Model is available for use under licence. A number of research organizations and national meteorological services use the UM in collaboration with the Met Office to undertake basic atmospheric process research, produce forecasts, develop the UM code and build and evaluate Earth system models. For further information on how to apply for a licence, see http://www.metoffice.gov.uk/research/modelling-systems/unified-model.} %% use this section when having only software code available

%\dataavailability{TEXT} %% use this section when having only data sets available

%\codedataavailability{TEXT} %% use this section when having data sets and software code available

%\appendix
%\section{}     %% Appendix A

%\subsection{}                                    %% Appendix A1, A2, etc.

%\authorcontribution{TEXT}

\begin{acknowledgements}
The development of the HadGEM2 family represents the work of a large number of people, to whom the author is indebted. This work was supported

by the Joint DECC/Defra Met Office Hadley Centre Climate Programme
(GA01101).
 The author also wishes to thank three anonymous reviewers for their
perceptive and detailed comments which greatly helped to improve the
manuscript.
\end{acknowledgements}

%% REFERENCES

%% The reference list is compiled as follows:

\begin{thebibliography}{}

%\bibitem[AUTHOR(YEAR)]{LABEL}
%REFERENCE

\bibitem[Acton(1970)]{acton70}

[revised manuscript text omitted]

\end{thebibliography}

%% Since the Copernicus LaTeX package includes the BibTeX style file
copernicus.bst,
%% authors experienced with BibTeX only have to include the following two
lines:
%%
%% \bibliographystyle{copernicus}
%% \bibliography{example.bib}
%%
%% URLs and DOIs can be entered in your BibTeX file as:
%%
%% URL = {http://www.xyz.org/~jones/idx_g.htm}
%% DOI = {10.5194/xyz}

%% LITERATURE CITATIONS
%%
%% command                          & example result
%% \citet{jones90}|                 & Jones et al. (1990)

```
%% \citep{jones90}|            & (Jones et al., 1990)
%% \citep{jones90,jones93}|    & (Jones et al., 1990, 1993)
%% \citep[p.~32]{jones90}|     & (Jones et al., 1990, p.~32)
%% \citep[e.g.,][]{jones90}|   & (e.g., Jones et al., 1990)
%% \citep[e.g.,][p.~32]{jones90}| & (e.g., Jones et al., 1990, p.~32)
%% \citeauthor{jones90}|       & Jones et al.
%% \citeyear{jones90}|         & 1990

\clearpage
%
\pagebreak

%% FIGURES

%% ONE-COLUMN FIGURES

%%f
\begin{figure}[t]
%\includegraphics[width=15.6cm]{p05B21_DiatHadOCCFe_fldg.eps}
%\includegraphics[width=15.6cm]{fig01.eps}
\includegraphics[width=15.6cm]{fg01.eps}
\caption{Diagram of the Diat-HadOCC model components and flows of
nitrogen, carbon, silicon and iron}
\label{fig-DHdiag}
\end{figure}
%
\clearpage
%
\pagebreak
%
\begin{figure}[t]
%\includegraphics[width=12.3cm]{p15407b_4a_phyc_8c201_chl1_Am.eps}
%\includegraphics[width=12.3cm]{fig02.eps}
\includegraphics[width=12.3cm]{fg02.eps}
\caption{Comparison of surface chlorophyll: upper panel, mean over the
years 2010-9 inclusive from the model, Historical+ RCP8.5 scenario; lower
panel, mean over 1998-2007 from GlobColor, with further processing as
described in \citep{ford12}. Units are mg Chl m$^{-3}$$}
\label{fig-4chl1}
\end{figure}
%
\clearpage
\pagebreak
%
\begin{figure}[t]
%\includegraphics[width=12.4cm]{p15407c_4b_phyc_8c201_chl1_hmt.eps}
%\includegraphics[width=12.4cm]{fig03.eps}
\includegraphics[width=12.4cm]{fg03.eps}
\caption{Seasonal cycle of global zonal mean surface chlorophyll, in mg
Chl m$^{-3}$: top panel, average over the years 2010-9 inclusive from the
model, Historical+RCP8.5 scenario; middle, the same but scaled by factor
0.213/0.812 (=0.262) so that the model mean matches the observations;
bottom, satellite-derived data from GlobColor, averaged over 1998-2007
inclusive.}
\label{fig-4chl2}
\end{figure}
%
\clearpage
```

```latex
\pagebreak
%
\begin{figure}[t]
%\includegraphics[width=17.8cm]{p15407d_4c_phyc_chl1_scAm_scPh.eps}
\%\includegraphics[width=17.8cm]{fig04.eps}
%\includegraphics[width=17.8cm]{p18704a_fg04_Chl_seasCyc.eps}
\includegraphics[width=17.8cm]{fg04.eps}
\caption{The amplitude (left-hand panels; units are mg Chl m$^{-3}$) and
phase (right-hand panels; units are 'fraction of year') of the seasonal
cycle of surface chlorophyll in the model (upper panels; average over
years 2010-9, Historical+RCP8.5 scenario, amplitude scaled by factor of
0.213/0.812) and in the GlobColor data (lower panels; average over years
1998-2007). The amplitude has been determined by finding the best-fitting
sine-curve through the monthly-mean values of the average cycle at each
point, and the phase refers to the fraction of the year when the fitted
curve is at its maximum.}
. Points are left white if the variance of the residual (after the best-
fitting sine-curve has been removed) is more than half that of the
original seasonal cycle.}
\label{fig-4chl3}
\end{figure}
%
\clearpage
\pagebreak
%
\begin{figure}[t]
%\includegraphics[width=17.8cm]{p18B08a_fg05_srfcChl_TD_scldUnscl.eps}
\includegraphics[width=17.8cm]{fg05.eps}
\caption{Taylor diagrams of model surface chlorophyll compared to the
GlobColor product. Solid symbols represent correlations and standard
deviations from points in all parts of the ocean (except inland seas),
while open symbols have had a mask applied to remove the Arctic Ocean and
two grid-boxes around the coast, as explained in the text. Squares
represent the annual mean of all points, while circles, up-pointing
triangles and down-pointing triangles respectively represent the mid-
point, amplitude and phase of the sine-curve that best fits the seasonal
cycle (where the variance of the residual is less than half the variance
of the cycle). The diagram on the left uses the raw model results, while
that on the right uses the model chlorophyll scaled to give a comparable
global mean to the observations (again as explained in the text).}
\label{fig-2TDchl}
\end{figure}
%
\clearpage
\pagebreak
%
\begin{figure}[t]
%\includegraphics[width=12.4cm]{p15405a_phyd_phym_8c201_aMn.eps}
%\includegraphics[width=12.4cm]{p16A04_phyd_phym_8c201_aMn_ed.eps}
\%\includegraphics[width=12.4cm]{fig05.eps}
\includegraphics[width=12.4cm]{fg06.eps}
\caption{Phytoplankton surface biomass (in mMol N m$^{-3}$), averaged
over the model years 2010-2019 inclusive, for (upper panel) Diatoms, and
(lower panel) misc-Phytoplankton.}
\label{fig-4phy1}
\end{figure}
%
\clearpage
```

```latex
\pagebreak
%
\begin{figure}[t]
%\includegraphics[width=17.6cm]{p15405e_phyd_phym_8c201_scMn_Ph.eps}
%\includegraphics[width=17.6cm]{p17316c_phyd_phym_8c201_scMn_Ph.eps}
%\includegraphics[width=17.6cm]{fig06.eps}
%\includegraphics[width=17.6cm]{p18914a_fg07_DtmMPhy_seasCyc.eps}
\includegraphics[width=17.6cm]{fg07.eps}
\caption{Phytoplankton surface biomass mean seasonal cycle, averaged over
model years 2010 to 2019 inclusive, for (upper panels) Diatoms and (lower
panels) misc-Phytoplankton. Left-hand panels show the amplitude (in mMol
N m$^{-3}$) and the right-hand panels the phase (in fraction of calendar
year).
). Only points where the residual variance is less than half the original
are shown.}
\label{fig-4phy2}
\end{figure}
%
\clearpage
\pagebreak
%
\begin{figure}[t]
%\includegraphics[width=12.4cm]{p15405b_phyd_phym_8c201_hvt.eps}
%\includegraphics[width=12.4cm]{p16A04a_phyd_phym_8c201_hvt_hva.eps}
%\includegraphics[width=17.4cm]{p16A04a_phyd_phym_8c201_hvt_hva.eps}
%\includegraphics[width=17.4cm]{fig07.eps}
\includegraphics[width=17.4cm]{fg08.eps}
\caption{Phytoplankton surface biomass (in mMol N m$^{-3}$), zonal mean
(taken globally for left-hand panels, across Atlantic basin only for
right-hand panels), averaged for each month over the model years 2010-
2019 inclusive: upper panels, Diatoms; lower panels, misc-Phytoplankton}
\label{fig-4phy3}
\end{figure}
%
\clearpage
\pagebreak
%
\begin{figure}[t]
%\includegraphics[width=9.5cm]{p15407e_4f_ipp_ipd_ipm_mn.eps}
%\includegraphics[width=9.5cm]{p17316a_4f_ipp_ipd_ipm_mn.eps}
%\includegraphics[width=9.5cm]{fig08.eps}
\includegraphics[width=9.5cm]{fg09.eps}
\caption{Primary Production (g C m$^{-2}$ d$^{-1}$), depth-integrated,
averaged over the model years 2010-2019 inclusive: bottom panel, PP by
misc-Phytoplankton; middle panel, that by Diatoms; top panel, total by
both phytoplankton types}
%\caption{Primary Production (mg C m$^{-2}$ d$^{-1}$), depth-integrated,
averaged over the model years 2010-2019 inclusive: bottom panel, PP by
misc-Phytoplankton; middle panel, that by Diatoms; top panel, total by
both phytoplankton types}
\label{fig-4pp1}
\end{figure}
%
\clearpage
\pagebreak
%
\begin{figure}[t]
%\includegraphics[width=12.4cm]{p15407f_4g_ipp_ipd_ipm_hmt.eps}
```

```
%\includegraphics[width=17.4cm]{p16A04b_4g_ipp_ipd_ipm_hmt_hma.eps}
%\includegraphics[width=12.4cm]{p16A04b_4g_ipp_ipd_ipm_hmt_hma.eps}
%\includegraphics[width=17.4cm]{p17316b_4g_ipp_ipd_ipm_hmt_hma.eps}
%\includegraphics[width=17.4cm]{fig09.eps}
\includegraphics[width=17.4cm]{fg10.eps}
\caption{Primary Production (g C m$^{-2}$ d$^{-1}$), depth-integrated,
zonally-meaned, averaged for each month over the model years 2010-2019
inclusive: bottom panels, PP by misc-Phytoplankton; middle, that by
Diatoms; top, total by both phytoplankton types. The left-hand panels
show global zonal means, while the right-hand panels show zonal means
across the Atlantic basin only.}
%\caption{Primary Production (mg C m$^{-2}$ d$^{-1}$), depth-integrated,
zonally-meaned, averaged for each month over the model years 2010-2019
inclusive: bottom panels, PP by misc-Phytoplankton; middle, that by
Diatoms; top, total by both phytoplankton types. The left-hand panels
show global zonal means, while the right-hand panels show zonal means
across the Atlantic basin only.}
\label{fig-4pp2}
\end{figure}
%
\clearpage
\pagebreak
%
\begin{figure}[t]
%\includegraphics[width=12.1cm]{p15409a_dissic_2010s_glodap_mn.eps}
%\includegraphics[width=12.1cm]{p16A11a_dissic_2010s1990sGlodap_mn.eps}
%\includegraphics[width=9.5cm]{p16A11a_dissic_2010s1990sGlodap_mn.eps}
%\includegraphics[width=9.5cm]{fig15.eps}
%\includegraphics[width=9.5cm]{p18711a_fg11_DIC_srf199m201GDv2.eps}
\includegraphics[width=9.5cm]{fg11.eps}
\caption{Surface concentration of Dissolved Inorganic Carbon (mMol C
m$^{-3}$): top panel, model field averaged over model years 2010-2019
inclusive; middle, model field averaged over model years 1990-1999
inclusive; bottom, the gridded field from the GLODAPv2 database}
\label{fig-4dic1}
\end{figure}
%
\clearpage
\pagebreak
%
\begin{figure}[t]
%\includegraphics[width=12.1cm]{p18928j_fg12_TCO2_PrflAP_R201_Gd2m.eps}
\includegraphics[width=12.1cm]{fg12.eps}
\caption{Meridional sections of DIC: Upper panels show sections in the
Atlantic Ocean along 330$^{\circ}$, lower panels Pacific Ocean sections
along 190$^{\circ}$; left panels show model concentrations averaged over
2010-2019, right panels show concentrations from the GLODAPv2 gridded
product.}
\label{fig-4dicSec}
\end{figure}
%
\clearpage
\pagebreak
%
\begin{figure}[t]
%\includegraphics[width=12.1cm]{p15409c_dissic_2010s_scAmPh.eps}
%\includegraphics[width=12.1cm]{fig16.eps}
%\includegraphics[width=12.1cm]{p18711b_fg13_DIC_seasCyc.eps}
```

```latex
\includegraphics[width=12.1cm]{fg13.eps}
\caption{Surface DIC, model seasonal cycle, averaged over model years
2010-2019 inclusive: upper panel, amplitude of cycle (mMol C m$^{-3}$);
lower panel, phase of cycle (fraction of year). Only points where the
residual variance is less than half the original are shown.}
\label{fig-4dic2}
\end{figure}
%
\clearpage
\pagebreak
%
\begin{figure}[t]
%\includegraphics[width=12.1cm]{p18925a_fg14_TAlk_srf199m201GDv2.eps}
\includegraphics[width=12.1cm]{fg14.eps}
\caption{Surface concentration of Total Alkalinity (mEq m$^{-3}$): top
panel, model field averaged over model years 2010-2019 inclusive; middle,
model field averaged over model years 1990-1999 inclusive; bottom, the
gridded field from the GLODAPv2 database.}
\label{fig-3talk1}
\end{figure}
%
\clearpage
\pagebreak
%
\begin{figure}[t]
%\includegraphics[width=12.1cm]{p18927a_fg15_TAlk_PrflAP_R201_Gd2m.eps}
\includegraphics[width=12.1cm]{fg15.eps}
\caption{Meridional sections of Total Alkalinity: Upper panels show
sections in the Atlantic Ocean along 330$^{\circ}$, lower panels Pacific
Ocean sections along 190$^{\circ}$; left panels show model concentrations
averaged over 2010-2019, right panels show concentrations from the
GLODAPv2 gridded product.}
\label{fig-4talkSec}
\end{figure}
%
\clearpage
\pagebreak
%
\begin{figure}[t]
%\includegraphics[width=12.1cm]{p18920a_fg16_TAlk_seasCyc.eps}
\includegraphics[width=12.1cm]{fg16.eps}
\caption{Surface Total Alkalinity, model seasonal cycle, averaged over
model years 2010-2019 inclusive: upper panel, amplitude of cycle (mEq
m$^{-3}$); lower panel, phase of cycle (fraction of year). Only points
where the residual variance is less than half the original are shown.}
\label{fig-2talkSCy}
\end{figure}
%
\clearpage
\pagebreak
%
\begin{figure}[t]
%\includegraphics[width=12.1cm]{p15421a_spco2_h2k_t2k_a.eps}
\%\includegraphics[width=12.1cm]{fig10.eps}
\includegraphics[width=12.1cm]{fg17.eps}
\caption{Surface ocean pCO2 (in ppmv): upper panel, model field averaged
over the model years 1990-2009 inclusive; lower panel, Takahashi gridded
field from data, annual mean, referenced to the year 2000}
```

```latex
\label{fig-4spc1}
\end{figure}
%
\clearpage
\pagebreak
%
\begin{figure}[t]
%\includegraphics[width=17.4cm]{p15421b_spco2_h2k_t2k_sc_AmPh.eps}
%\includegraphics[width=17.4cm]{p17316d_spco2_h2k_t2k_sc_AmPh.eps}
%\includegraphics[width=17.4cm]{fig11.eps}
%\includegraphics[width=17.4cm]{p18919a_fg18_pCO2_seasCyc.eps}
\includegraphics[width=17.4cm]{fg18.eps}
\caption{Surface ocean pCO2, seasonal cycle: upper panels, model,
averaged over model years 1990-2009 inclusive; lower panels, Takahashi
gridded data, referenced to the year 2000; left-hand panels, amplitude of
the cycle (ppmv); right-hand panels, phase of the cycle (in fraction of
year
). Only points where the residual variance is less than half the original
are shown.}
\label{fig-4spc2}
\end{figure}
%
\clearpage
\pagebreak
%
\begin{figure}[t]
\includegraphics[width=17.4cm]{fgzz.eps}
\caption{Fraction of the seasonal cycle of pCO$ 2$ that is not driven by
the temperature (and salinity) seasonal variation. The details of the
calculation are given in the text. Where the ratio is less than 0.5, the
temperature variation dominates, and where the ratio is greater than 0.5
the biological uptake/respiration (and the air-sea uptake) dominate.
Ratios greater than 1.0 indicate that the biologically-driven and
temperature-driven cycles are opposed.}
\label{fig-spcFrac}
\end{figure}
%
\clearpage
\pagebreak
%
\begin{figure}[t]
%\includegraphics[width=8.7cm]{p18B06e_fg19_TD_PCO2DICTAlk_VMAP.eps}
%\includegraphics[width=17.4cm]{fg19.eps}
\includegraphics[width=8.4cm]{fg19.eps}
\caption{Taylor diagram for surface DIC (orange), surface Total
Alkalinity (purple) and surface pCO2 (blue). Model DIC and Total
Alkalinity from the RCP8.5 simulation (meaned over the years 2010 to 2019
inclusive) are compared to the gridded fields from GLODAPv2, while model
pCO2 (meaned over 1990 to 2009 inclusive) is compared to the Takahashi
gridded data. Filled squares refer to the raw surface fields, and filled
circles, upward-pointing triangles and downward triangles respectively
refer to the mid-point, amplitude and phase of the sine-curve that best
fits the seasonal cycle (in points where the variance of the residual is
less than half that of the original cycle).}
\label{fig-TDcarb}
\end{figure}
%
\clearpage
```

\pagebreak
%
\begin{figure}[t]
%\includegraphics[width=12.1cm]{p15412d_fgco2_8c201_pc201_mn.eps}
\%\includegraphics[width=12.1cm]{fig12.eps}
\includegraphics[width=12.1cm]{fg20.eps}
\caption{Total air-to-sea flux of CO$_2$ (ng C m$^{-2}$ s$^{-1}$;
positive values into the ocean), mean over model years 2010-2019
inclusive: upper panel, total flux (natural cycle and anthropogenic
perturbation); lower panel, anthropogenic perturbation}
\label{fig-4asf1}
\end{figure}
%
\clearpage
\pagebreak
%
\begin{figure}[t]
%\includegraphics[width=17.9cm]{p15412a_fgco2_8c201_hmtaip.eps}
\%\includegraphics[width=17.9cm]{fig13.eps}
\includegraphics[width=17.9cm]{fg21.eps}
\caption{Total air-to-sea flux of CO$_2$ (ng C m$^{-2}$ s$^{-1}$),
seasonal cycle averaged for each month over the model years 2010-2019
inclusive, zonally-meaned: upper left panel, global zonal mean; upper
right, zonal mean of the Atlantic Ocean basin; lower left, zonal mean of
the Indian Ocean basin; lower right, zonal mean of the Pacific Ocean
basin}
\label{fig-4asf2}
\end{figure}
%
\clearpage
\pagebreak
%
\begin{figure}[t]
%\includegraphics[width=17.6cm]{p15412b_fgco2_8pc201_hmtaip.eps}
\%\includegraphics[width=17.6cm]{fig14.eps}
\includegraphics[width=17.6cm]{fg22.eps}
\caption{As Figure~\ref{fig-4asf2}, but for the air-to-sea flux of
anthropogenic CO$_2$ only (ng C m$^{-2}$ s$^{-1}$)}
\label{fig-4asf3}
\end{figure}
%
\clearpage
\pagebreak
%
\begin{figure}[t]
%\includegraphics[width=12.1cm]{p15409a_dissic_2010s_glodap_mn.eps}
%\includegraphics[width=12.1cm]{p16A11a_dissic_2010s1990sGlodap_mn.eps}
%\includegraphics[width=9.5cm]{p16A11a_dissic_2010s1990sGlodap_mn.eps}
\includegraphics[width=9.5cm]{fig15.eps}
\caption{Surface concentration of Dissolved Inorganic Carbon (mMol C
m$^{-3}$): top panel, model field averaged over model years 2010-2019
inclusive; middle, model field averaged over model years 1990-1999
inclusive; bottom, the gridded field from the GLODAP database}
\label{fig-4dic1}
\end{figure}
%
\clearpage
\pagebreak

```latex
%
\begin{figure}[t]
%\includegraphics[width=12.1cm]{p15409c_dissic_2010s_scAmPh.eps}
\includegraphics[width=12.1cm]{fig16.eps}
\caption{Surface DIC, model seasonal cycle, averaged over model years
2010-2019 inclusive: upper panel, amplitude of cycle (mMol C m$^{-3}$);
lower panel, phase of cycle (fraction of year)}
\label{fig-4dic2}
\end{figure}
%
\clearpage
\pagebreak
%
\begin{figure}[t]
%\includegraphics[width=12.2cm]{p15410a_no3_8c201_woa_scMn.eps}
%\includegraphics[width=12.2cm]{fig17.eps}
%\includegraphics[width=12.2cm]{p18712a_fg23_NO3_srf201mWOA13.eps}
\includegraphics[width=12.2cm]{fg23.eps}
\caption{Surface dissolved nitrate (mMol N m$^{-3}$): upper panel, model
field averaged over model years 2010-2019 inclusive; lower panel, the
gridded field from the 20052013 World Ocean Atlas}
\label{fig-4nit1}
\end{figure}
%
\clearpage
\pagebreak
%
\begin{figure}[t]
%\includegraphics[width=17.4cm]{p18A18z_fg24_NO3_PrflAP_R201_WOA.eps}
%\includegraphics[width=17.4cm]{fg24.eps}
\includegraphics[width=15.0cm]{fg24.eps}
\caption{Meridional sections of disolved nitrate (mMol N m$^{-3}$): upper
panels show sections in the Atlantic Ocean along 330$^{\circ}$, lower
panels Pacific Ocean sections along 190$^{\circ}$; left panels show model
concentrations averaged over 2010-2019, right panels show concentrations
from the 2013 World Ocean Atlas gridded field.}
\label{fig-4nitSec}
\end{figure}
%
\clearpage
\pagebreak
%
\begin{figure}[t]
%\includegraphics[width=8.3cm]{p15410b_no3_8c201_woa_scAmPh.eps}
%\includegraphics[width=17.6cm]{p15410b_no3_8c201_woa_scAmPh.eps}
%\includegraphics[width=17.6cm]{fig18.eps}
%\includegraphics[width=17.6cm]{p18718a_fg25_NO3_seasCyc.eps}
\includegraphics[width=17.4cm]{fg25.eps}
\caption{Surface dissolved nitrate, seasonal cycle: upper panels, model
cycle, averaged over model years 2010-2019 inclusive; lower panels, the
cycle from the monthly gridded fields from the 20052013 World Ocean
Atlas; left-hand panels, the amplitude of the cycle (mMol N m$^{-3}$);
right-hand panels, the phase of the cycle (fraction of year)
). Only points where the residual variance is less than half the original
are shown.}
\label{fig-4nit2}
\end{figure}
%
```

```
\clearpage
\pagebreak
%
\begin{figure}[t]
%\includegraphics[width=12.1cm]{p15410d_si_8c201_woa_scMn.eps}
%\includegraphics[width=12.1cm]{fig19.eps}
%\includegraphics[width=12.1cm]{p18802a_fg26_SiO4_srf201mWOA13.eps}
\includegraphics[width=12.1cm]{fg26.eps}
\caption{Surface dissolved silicate (mMol Si m$^{-3}$): upper panel,
model field averaged over model years 2010-2019 inclusive; lower panel,
the gridded field from the 20052013 World Ocean Atlas}
\label{fig-4si1}
\end{figure}
%
\clearpage
\pagebreak
%
\begin{figure}[t]
%\includegraphics[width=17.4cm]{p18A19z_fg27_SiO4_PrflAP_R201_WOA.eps}
%\includegraphics[width=17.4cm]{fg27.eps}
\includegraphics[width=15.0cm]{fg27.eps}
\caption{Meridional sections of disolved silicate (mMol Si m$^{-3}$):
upper panels show sections in the Atlantic Ocean along 330$^{\circ}$,
lower panels Pacific Ocean sections along 190$^{\circ}$; left panels show
model concentrations averaged over 2010-2019, right panels show
concentrations from the 2013 World Ocean Atlas gridded field.}
\label{fig-4siSec}
\end{figure}
%
\clearpage
\pagebreak
%
\begin{figure}[t]
%\includegraphics[width=17.4cm]{p15410e_si_8c201_woa_scAmPh.eps}
%\includegraphics[width=17.4cm]{fig20.eps}
%\includegraphics[width=17.4cm]{p18802b_fg28_SiO4_seasCyc.eps}
\includegraphics[width=17.4cm]{fg28.eps}
\caption{Surface dissolved silicate, seasonal cycle: upper panels, model
cycle, averaged over model years 2010-2019 inclusive; lower panels, the
cycle from the monthly gridded fields from the 20052013 World Ocean
Atlas; left-hand panels, the amplitude of the cycle (mMol Si m$^{-3}$);
right-hand panels, the phase of the cycle (fraction of year)
). Only points where the residual variance is less than half the original
are shown.}
\label{fig-4si2}
\end{figure}
%
\clearpage
\pagebreak
%
\begin{figure}[t]
%\includegraphics[width=17.4cm]{p18A19x_fg29_O2_PrflAP_R201_WOA.eps}
%\includegraphics[width=17.4cm]{fg29.eps}
\includegraphics[width=15.0cm]{fg29.eps}
\caption{Meridional sections of disolved oxygen (mMol O$_2$ m$^{-3}$):
upper panels show sections in the Atlantic Ocean along 330$^{\circ}$,
lower panels Pacific Ocean sections along 190$^{\circ}$; left panels show
```

model concentrations averaged over 2010-2019, right panels show
concentrations from the 2013 World Ocean Atlas gridded field.}
\label{fig-4oxSec}
\end{figure}
%
\clearpage
\pagebreak
%
\begin{figure}[t]
%\includegraphics[width=17.4cm]{p18A19w_fg30_O2_thk_R201_WOA.eps}
\includegraphics[width=17.4cm]{fg30.eps}
\caption{Thickness (m) of the oxygen depletion zone in the water-column;
left panels from the model (averaged over the years 2010 to 2019
inclusive), right panels from the 2013 World Ocean Atlas. Upper and lower
panels show the extent of the water-column in which O$_2$ concentrations
are respectively below 50 and 100 mMol O$_2$ m$^{-3}$.}
\label{fig-4oxThk}
\end{figure}
%
\clearpage
\pagebreak
%
\begin{figure}[t]
%\includegraphics[width=17.4cm]{p18A18a_fg31_AOU_R201y_WOA13_1050m.eps}
\includegraphics[width=10.0cm]{fg31.eps}
\caption{Apparent Oxygen Utilisation (mMol O$_2$ m$^{-3}$): upper panel,
model field at 1045.6 m, averaged over years 2010 to 2019 inclusive;
lower panel, field at 1050 m from the 2013 World Ocean Atlas.}
\label{fig-2aouDep}
\end{figure}
%
\clearpage
\pagebreak
%
\begin{figure}[t]
%\includegraphics[width=17.4cm]{p18A22z_fg32_AOU_PrflAP_R201_WOA.eps}
\includegraphics[width=17.4cm]{fg32.eps}
\caption{Meridional sections of Apparent Oxygen Utilisation (mMol O$_2$
m$^{-3}$): upper panels show sections in the Atlantic Ocean along
330$^{\circ}$, lower panels Pacific Ocean sections along 190$^{\circ}$;
left panels show model concentrations averaged over 2010-2019, right
panels show concentrations from the 2013 World Ocean Atlas gridded
field.}
\label{fig-4aouSec}
\end{figure}
%
\clearpage
\pagebreak
%
\begin{figure}[t]
%\includegraphics[width=12.1cm]{p15411b_dfe_8c201_scMnAm.eps}
\%\includegraphics[width=12.1cm]{fig21.eps}
%\includegraphics[width=12.1cm]{p18814a_fg33_Fe_MnSeasCyc.eps}
\includegraphics[width=12.1cm]{fg33.eps}
\caption{Surface dissolved iron (uMol Fe m$^{-3}$): upper panel, model
field averaged over model years 2010-2019 inclusive; lower panel,
amplitude of the model seasonal cycle averaged over the same period

```latex
 (only points where the residual variance is less than half the original
are shown).}
\label{fig-4fe1}
\end{figure}
%
\clearpage
\pagebreak
%
\begin{figure}[t]
%\includegraphics[width=17.4cm]{p18A19y fg34 dFe PrflAP R201.eps}
\includegraphics[width=17.4cm]{fg34.eps}
\caption{Meridional sections of dissolved iron concentrations (uMol Fe
m$^{-3}$) from the model, averaged over years 2010 to 2019 inclusive:
left panels show sections in the Atlantic Ocean along 330$^{\circ}$,
right panels Pacific Ocean sections along 190$^{\circ}$.}
\label{fig-2FeSec}
\end{figure}
%
\clearpage
\pagebreak
%
\begin{figure}[t]
%\includegraphics[width=17.4cm]{p18B13a dFe sectGA02 Mdl eGeotrcs.eps}
%\includegraphics[width=17.4cm]{fg35.eps}
\includegraphics[width=17.0cm]{fg35.eps}
\caption{Sections of dissolved iron (uMol Fe m$^{-3}$) along the
eGEOTRACES GA02 transect (in the Atlantic Ocean): upper panel shows model
concentrations averaged over years 2010 to 2019 inclusive, lower panel is
reproduced from eGEOTRACES.}
\label{fig-2FeGeo}
\end{figure}
%
\clearpage
\pagebreak
%
\begin{figure}[t]
%\includegraphics[width=17.4cm]{p18B06d_fg35_TD_NO3SiO4O2AOU.eps}
\includegraphics[width=8.7cm]{fg36.eps}
\caption{Taylor diagram of nitrate (green), silicate (red), oxygen (blue)
and Apparent Oxygen Utilisation (purple); model values are averaged over
years 2010 to 2019 inclusive, observations are from the 2013 World Ocean
Atlas. Filled squares show the surface concentrations, open squares the
concentrations at the nearest depth level to 1050 m, and circles, upward-
pointing triangles and downward triangles respectively the mid-point,
amplitude and phase of the sine-curve that best fits the seasonal cycle
(only at points where the residual variance is less than half the
original). The filled and open stars show respectively the vertical
extent of the water-column where O$ 2$ concentration is below 50 and 100
mMol O$ 2$ m$^{-3}$.}
\label{fig-TD4nuts}
\end{figure}
%
\clearpage
\pagebreak
%
\begin{figure}[t]
%\includegraphics[width=17.5cm]{p15411a_no3_h8a_hmt.eps}
%\includegraphics[width=17.5cm]{fig22.eps}
```

```latex
\includegraphics[width=17.5cm]{fg37.eps}
\caption{Surface dissolved nitrate concentration (mMol N m$^{-3}$),
global zonal and annual means for model years 1860 to 2099, from the
CMIP5 Historical and RCP8.5 simulations, showing the response to changing
climatic forcing}
\label{fig-4ccNit1}
\end{figure}
%
\clearpage
\pagebreak
%
\begin{figure}[t]
%\includegraphics[width=12.6cm]{p15412f_chl_h8ap_hmap_zvap.eps}
+%\includegraphics[width=12.6cm]{fig23.eps}
\includegraphics[width=12.6cm]{fg38.eps}
\caption{Surface total chlorophyll concentration anomaly (mg Chl m$^{-
3}$), zonal and annual means for model years 1860 to 2099, from the CMIP5
Historical and RCP8.5 simulations: upper panel, zonal mean of the
Atlantic Ocean basin; lower panel, zonal mean of the Pacific Ocean basin.
The anomaly has been calculated by subtracting the surface chlorophyll
concentration field, meaned over the years 1860 to 2099 inclusive, as
produced by the piControl simulation from the annual means of the
Historical and RCP8.5 simulations}
\label{fig-4ccChl1}
\end{figure}
%
\clearpage
\pagebreak
%
\begin{figure}[t]
%\includegraphics[width=12.6cm]{p15412g_chl_8c209_8c201_hmap.eps}
+%\includegraphics[width=12.6cm]{fig24.eps}
\includegraphics[width=12.6cm]{fg39.eps}
\caption{Change in the seasonal cycle of surface chlorophyll
concentration in the CMIP5 RCP8.5 simulation: change is calculated
between the mean seasonal cycles of the model years 2090-2099 and 2010-
2019. Zonal means of the (upper panel) Atlantic Ocean basin and (lower
panel) Pacific Ocean basin}
\label{fig-4ccChl2}
\end{figure}
%
\clearpage
\pagebreak
%
\begin{figure}[t]
%\includegraphics[width=12.1cm]{p15412h_ipp_8c209_8c201_scMnAm.eps}
+%\includegraphics[width=12.1cm]{fig25.eps}
\includegraphics[width=12.1cm]{fg40.eps}
\caption{Change in the depth-integrated total Primary Production (mg C
m$^{-2}$ d$^{-1}$) in the RCP8.5 simulation: difference between the model
years 2090-2099 and 2010-2019. Upper panel: difference in decadal means;
lower panel: difference in amplitude of mean seasonal cycle}
\label{fig-4ccPP1}
\end{figure}
%
\clearpage
\pagebreak
%
```

```latex
\begin{figure}[t]
%\includegraphics[width=17.5cm]{p15412j_ipd_ipm_h8a_hmap.eps}
\%\includegraphics[width=17.5cm]{fig26.eps}
\includegraphics[width=17.5cm]{fg41.eps}
\caption{Change in annual mean depth-integrated Primary Production (mg C
m$^{-2}$ d$^{-1}$) during the model years 1860 to 2099 in the CMIP5
Historical and RCP8.5 simulations. Upper panels, PP by Diatoms; lower
panels, PP by misc-Phytoplankton; left-hand panels, Atlantic Ocean basin
zonal mean; right-hand panels, Pacific Ocean basin zonal mean}
\label{fig-4ccPP2}
\end{figure}
%
\clearpage
\pagebreak
%
\begin{figure}[t]
%\includegraphics[width=12.4cm]{p15421c_spco2_h8ahm_8c209_sc_MnAm.eps}
\%\includegraphics[width=12.4cm]{fig27.eps}
%\includegraphics[width=12.4cm]{fg42.eps}
\includegraphics[width=11.6cm]{fg42.eps}
\caption{Change in surface pCO$_2$ (ppmv) during the model years 1860 to
2099 in the CMIP5 Historical and RCP8.5 simulations. Top panel: the
anomaly over the period of the simulations, calculated by subtracting the
annual means of the piControl simulation from those of the Historical and
RCP8.5 simulations. Middle panel: the decadal mean anomaly during the
model years 2090-2099, calculated by subtracting the relevant years of
the piControl from those of the RCP8.5 simulation. Bottom panel: the
seasonal cycle amplitude anomaly averaged over the model years 2090-2099,
calculated as for the middle panel}
\label{fig-4ccSpc1}
\end{figure}
%
\clearpage
\pagebreak
%
\begin{figure}[t]
%\includegraphics[width=17.3cm]{p15413a_fgco2_h8a_t.eps}
\%\includegraphics[width=17.3cm]{fig28.eps}
\includegraphics[width=17.3cm]{fg43.eps}
%\caption{Time-evolution of the annual mean global total air-to-sea
CO$_2$ flux (ng C m$^{-2}$ s$^{-1}$) between model years 1860 and 2099 in
the CMIP5 Historical and RCP8.5 simulations}
\caption{Time-evolution of the annual mean global total air-to-sea CO$_2$
flux (Pg C yr$^{-1}$) between model years 1860 and 2099 in the CMIP5
Historical and RCP8.5 simulations}
\label{fig-4ccAsf0}
\end{figure}
%
\clearpage
\pagebreak
%
\begin{figure}[t]
%\includegraphics[width=12.4cm]{p15412e_fgco2_h8a_hm.eps}
\%\includegraphics[width=12.4cm]{fig29.eps}
\includegraphics[width=12.4cm]{fg44.eps}
\caption{Change in the annual mean total air-to-sea CO$_2$ flux (ng C
m$^{-2}$ s$^{-1}$) during model years 1860 to 2099 in the Historical and
```

```
RCP8.5 simulations. Top panel: global zonal mean; middle panel: Atlantic
Ocean basin zonal mean; bottom panel: Pacific Ocean basin zonal mean}
\label{fig-4ccAsf1}
\end{figure}
%
\clearpage
\pagebreak
%
\begin{figure}[t]
%\includegraphics[width=17.6cm]{p15412c_fgco2_8c209_hmtaip.eps}
\%\includegraphics[width=17.6cm]{fig30.eps}
\includegraphics[width=17.6cm]{fg45.eps}
\caption{The seasonal cycle (monthly means) of the total air-to-sea
CO$_2$ flux (ng C m$^{-2}$ s$^{-1}$) averaged over the model years 2090-
2099 inclusive. Zonal mean of: upper left panel, global ocean; upper
right, Atlantic Ocean basin; lower left, Indian Ocean basin; lower right,
Pacific Ocean basin}
\label{fig-4ccAsf2}
\end{figure}
%
\clearpage
\pagebreak
%
%
%\begin{figure}[t]
%\includegraphics[width=8.3cm]{FILE NAME}
%\caption{TEXT}
%\end{figure}
%
%%% TWO-COLUMN FIGURES
%
%%f
%\begin{figure*}[t]
%\includegraphics[width=12cm]{FILE NAME}
%\caption{TEXT}
%\end{figure*}
%
%
%%% TABLES
%%%
%%% The different columns must be seperated with a & command and should
%%% end with \\ to identify the column brake.
%
%%% ONE-COLUMN TABLE
%
\begin{table}
\caption{Diat-HadOCC model state variables}
\label{tbl-statevars}
\centering
\begin{tabular}{l l l}
\hline
 Symbol & Description & Units \\
\hline
$DIN$ & dissolved inorganic nitrogen & mmol-N / m$^{3}$ \\
$Si$ & silicic acid & mmol-Si / m$^{3}$ \\
$FeT$ & total dissolved iron & umol-Fe / m$^{3}$ \\
$Ph$ & miscellaneous (misc-) phytoplankton & mmol-N / m$^{3}$ \\
$Dm$ & diatom phytoplankton & mmol-N / m$^{3}$ \\
```

```latex
$DmSi$ & diatom silicate & mmol-Si / m$^{3}$ \\
$Zp$ & zooplankton & mmol-N / m$^{3}$ \\
$DtN$ & detrital nitrogen & mmol-N / m$^{3}$ \\
$DtSi$ & detrital silicate & mmol-Si / m$^{3}$ \\
$DtC$ & detrital carbon & mmol-C / m$^{3}$ \\
$DIC$ & dissolved inorganic carbon & mmol-C / m$^{3}$ \\
$TAlk$ & total alkalinity & meq / m$^{3}$ \\
$Oxy$ & dissolved oxygen & mmol-O2 / m$^{3}$ \\
\hline
\end{tabular}
\end{table}
%
\clearpage
\begin{table}
\caption{Polynomial coeffs relating $k$ to square root of pigment in
depth-range $L$}
\label{tbl-k_pigm}
\centering
\begin{tabular}{l l l l l l l}
\hline
 $L$ & $b_{0,L}$ & $b_{1,L}$ & $b_{2,L}$ & $b_{3,L}$ & $b_{4,L}$ &
$b_{5,L}$ \\
\hline
1 & 0.095934 & 0.039307 & 0.051891 & -0.020760 & 0.0043139 & -0.00035055
\\
2 & 0.026590 & 0.016301 & 0.073944 & -0.038958 & 0.0075507 & -0.00054532
\\
3 & 0.015464 & 0.14886 & -0.15711 & 0.15065 & -0.055830 & 0.0075811 \\
\hline
\end{tabular}
\end{table}
%
\clearpage
\begin{table}
\caption{Polynomial coeffs for $\frac{da^{\#}}{dz}$ as a function of
pigment and depth}
\label{tbl-ahash}
\centering
\begin{tabular}{l l}
\hline
$gcof_1$ = $g_1$ = 0.048014 & $gcof_6$ = $g_4$ = 0.0031095 \\
$gcof_2$ = $g_2$ = 0.00023779 & $gcof_7$ = $g_9$ = 0.0012398 \\
$gcof_3$ = $g_5$ = -0.0090545 & $gcof_8$ = $g_6$ = 0.0027974 \\
$gcof_4$ = $g_7$ = 0.00085217 & $gcof_9$ = $g_{10}$ = -0.00061991 \\
$gcof_5$ = $g_3$ = -0.023074 & $gcof_{10}$ = $g_8$ = -0.0000039804 \\
\hline
\end{tabular}
\end{table}
%
\clearpage
\begin{table}
\caption{Polynomial coeffs and rational function coeffs for psynth
calculation}
\label{tbl-omega}
\centering
\begin{tabular}{l l l l l l l}
\hline
Coeff & i=1 & 2 & 3 & 4 & 5 \\
```

```latex
\hline
$\Omega_i$ & 1.9004 & -0.28333 & 0.028050 & -0.0014729 & 0.000030841 \\
$\gamma_i$ & 1.62461 & 0.0045412 & 0.13140 & & \\
\hline
\end{tabular}
\end{table}
%
\clearpage
%\addtocounter{figure}{-1}\renewcommand{\thefigure}{\arabic{figure}a}
%\addtocounter{table}{-1}\renewcommand{\thetable}{\arabic{table}a}
\begin{table}
\caption{Parameter values used in CMIP5 simulations} \vspace{10pt}
\label{tbl-params}
\centering
%\begin{tabular}{c c p{0.3\textwidth} p{0.4\textwidth}}
%\begin{tabular}{c c p{0.35\textwidth} p{0.35\textwidth}}
\begin{tabular}{c c p{0.32\textwidth} p{0.38\textwidth}}
%
\hline
 Param  & Value & Units & Description   \\
\hline
 & & & \\
%
%  $P^{Ph}_{m,r}$  & 1.5 & d$^{-1}$ & Max rate of photosynthesis; misc-Phyto, Fe-replete  \vspace{1pt} \\
  $P^{Ph}_{m,r}$  & 1.5 & d$^{-1}$ & Max rate of psynth; misc-Phyto, Fe-replete  \vspace{1pt} \\
%
%  $P^{Ph}_{m,l}$  & 1.5 & d$^{-1}$ & Max rate of photosynthesis; misc-Phyto, Fe-limited  \vspace{1pt} \\
  $P^{Ph}_{m,l}$  & 1.5 & d$^{-1}$ & Max rate of psynth; misc-Phyto, Fe-limited  \vspace{1pt} \\
%
  $P^{Dm}_{m,r}$  & 1.85 & d$^{-1}$ & Max rate of photosynthesis; diatom, Fe-replete  \vspace{1pt} \\
%
  $P^{Dm}_{m,l}$  & 1.11 & d$^{-1}$ & Max rate of photosynthesis; diatom, Fe-limited  \vspace{1pt} \\
%
  $\alpha^{Ph}$   & 0.02 & mg C (mg Chl)$^{-1}$ h$^{-1}$ ($\mu$Einst m$^{-2}$ s$^{-1}$)$^{-1}$ & Initial slope of the psynth-light curve; misc-Phyto \vspace{1pt} \\
%
  $\alpha^{Dm}$   & 0.02 & mg C (mg Chl)$^{-1}$ h$^{-1}$ ($\mu$Einst m$^{-2}$ s$^{-1}$)$^{-1}$ & Initial slope of the psynth-light curve; diatom  \vspace{1pt} \\
%
  $k^{Ph}_{DIN}$  & 0.1  & mMol N m$^{-3}$ & Half-saturation const, N uptake; misc-Phyto  \vspace{1pt} \\
%
  $k^{Dm}_{DIN}$  & 0.2  & mMol N m$^{-3}$ & Half-saturation const, N uptake; diatom  \vspace{1pt} \\
%
  $k^{Dm}_{Si}$   & 1.0  & mMol Si m$^{-3}$ & Half-saturation const, Si uptake; diatom  \vspace{1pt} \\
%
  $R^{Ph}_{c2n}$ & 6.625 & mMol C (mMol N)$^{-1}$ & Molar C:N ratio, misc-Phyto \vspace{1pt} \\
```

```latex
%
  $R^{Dm}_{c2n}$ & 6.625 & mMol C (mMol N)$^{-1}$ & Molar C:N ratio,
diatom  \vspace{1pt} \\
%
  $R^{Zp}_{c2n}$ & 5.625 & mMol C (mMol N)$^{-1}$ & Molar C:N ratio,
zoopl  \vspace{1pt} \\
%
  $R^{Dm}_{si2n,r}$ & 0.606 & mMol Si (mMol N)$^{-1}$ & Molar Si:N ratio,
diatom, Fe-replete  \vspace{1pt} \\
%
  $R^{Dm}_{si2n,l}$ & 0.606 & mMol Si (mMol N)$^{-1}$ & Molar Si:N ratio,
diatom, Fe-limited  \vspace{1pt} \\
%
  $R^{Ph}_{c2chl,0}$ & 40.0 & mg C (mg Chl)$^{-1}$ & default
Carbon:Chlorophyll ratio, misc-Phyto  \vspace{1pt} \\
%
  $R^{Ph}_{c2chl,min}$ & 20.0 & mg C (mg Chl)$^{-1}$ & minimum
Carbon:Chlorophyll ratio, misc-Phyto  \vspace{1pt} \\
%
  $R^{Ph}_{c2chl,max}$ & 200.0 & mg C (mg Chl)$^{-1}$ & maximum
Carbon:Chlorophyll ratio, misc-Phyto  \vspace{1pt} \\
%
  $R^{Dm}_{c2chl,0}$ & 40.0 & mg C (mg Chl)$^{-1}$ & default
Carbon:Chlorophyll ratio, diatom  \vspace{1pt} \\
%
  $R^{Dm}_{c2chl,min}$ & 20.0 & mg C (mg Chl)$^{-1}$ & minimum
Carbon:Chlorophyll ratio, diatom  \vspace{1pt} \\
%
  $R^{Dm}_{c2chl,max}$ & 200.0 & mg C (mg Chl)$^{-1}$ & maximum
Carbon:Chlorophyll ratio, diatom  \vspace{1pt} \\
%
  $g_{max}$  & 0.8 & d$^{-1}$ & Max specific rate of zooplankton grazing
\vspace{1pt}  \\
%
  $g_{sat}$  & 0.5 & nMol N m$^{-3}$ & Half-saturation const for zoopl
grazing  \vspace{1pt} \\
%
  $bprf_{Ph}$ & 0.45 & (none) & Zoopl base feeding preference for misc-
Phyto  \vspace{1pt} \\
%
%  $bprf_{Dm,r}$ & 0.45 & (none) & Zoopl base feeding preference for
diatom, Fe-replete conditions  \vspace{1pt} \\
  $bprf_{Dm,r}$ & 0.45 & (none) & Zoopl base feeding pref: diatom, Fe-
replete  \vspace{1pt} \\
%
%  $bprf_{Dm,l}$ & 0.45 & (none) & Zoopl base feeding preference for
diatom, Fe-limited conditions  \vspace{1pt} \\
  $bprf_{Dm,l}$ & 0.45 & (none) & Zoopl base feeding pref: diatom, Fe-
limited  \vspace{1pt} \\
%
  $bprf_{Dt}$ & 0.10 & (none) & Zoopl base feeding preference for
detritus  \vspace{1pt} \\
%
  $F_{ingst}$  & 0.77 & (none) & Fraction of food that is ingested
\vspace{1pt} \\
%
%  $F_{messy}$  & 0.1 & (none) & Fraction of non-ingested food to
dissolved nutrient/carbon  \vspace{1pt} \\
```

```latex
  $F_{messy}$  & 0.1 & (none) & Frac of non-ingstd food to dslvd
nutrient/carbon  \vspace{1pt} \\
%
%  $\beta^{Ph}$  & 0.9 & (none) & Fraction of ingested misc-Phyto that
can be assimilated  \vspace{1pt} \\
%  $\beta^{Ph}$  & 0.9 & (none) & Frac of ingested misc-Phyto that can be
assimilated  \vspace{1pt} \\
  $\beta^{Ph}$  & 0.9 & (none) & Assimilate-able frac of ingested misc-
Phyto  \vspace{1pt} \\
%
%  $\beta^{Dm}$  & 0.9 & (none) & Fraction of ingested diatom that can be
assimilated  \vspace{1pt} \\
  $\beta^{Dm}$   & 0.9 & (none) & Frac of ingested diatom that can be
assimilated  \vspace{1pt} \\
%
%  $\beta^{Dt}$  & 0.7 & (none) & Fraction of ingested detritus that can
be assimilated  \vspace{1pt} \\
  $\beta^{Dt}$   & 0.7 & (none) & Frac of ingested detritus that can be
assimilated  \vspace{1pt} \\
%
\end{tabular}
\end{table}
%\pagebreak
\clearpage
\addtocounter{table}{-1}\renewcommand{\thetable}{\arabic{table}a}
\begin{table}
\caption{Parameter values used in CMIP5 simulations (cont)} \vspace{10pt}
%\label{tbl-params}
\centering
%\begin{tabular}{c c p{0.3\textwidth} p{0.4\textwidth}}
%\begin{tabular}{c c p{0.35\textwidth} p{0.35\textwidth}}
\begin{tabular}{c c p{0.32\textwidth} p{0.38\textwidth}}
%
\hline
 Param  & Value & Units & Description    \\
\hline
%
  $\Pi^{Ph}_{resp}$ & 0.05 & d$^{-1}$ & misc-Phyto respiration, specific
rate  \vspace{1pt} \\
%
  $\Pi^{Dm}_{resp}$ & 0.0 & d$^{-1}$ & Diatom respiration, specific rate
\vspace{1pt} \\
%
  $\Pi^{Ph}_{mort}$ & 0.05 & d$^{-1}$ (mMol N m$^{-3}$)$^{-1}$ & misc-
Phyto mortality, density-dep rate  \vspace{1pt} \\
%
%  $ph_{min}$ & 0.01 & mMol N m$^{-3}$ & misc-Phyto conc below which
misc-Phyto mortality is zero  \vspace{1pt} \\
  $ph_{min}$ & 0.01 & mMol N m$^{-3}$ & misc-Phyto conc below which
mortality is zero  \vspace{1pt} \\
%
  $\Pi^{Dm}_{mort}$ & 0.04 & d$^{-1}$ (mMol N m$^{-3}$)$^{-1}$ & Diatom
mortality, density-dep rate  \vspace{1pt} \\
%
  $\Pi^{Zp}_{lin}$  & 0.05 & d$^{-1}$ & Zooplankton losses, specific rate
\vspace{1pt} \\
%
```

```
%  $\Pi^{Zp}_{mort}$  & 0.3 & d$^{-1}$ (mMol N m$^{-3}$)$^{-1}$ & Zoopl.
mortality, density-dep  \vspace{1pt} \\
  $\Pi^{Zp}_{mort,r}$  & 0.3 & d$^{-1}$ (mMol N m$^{-3}$)$^{-1}$ & Zoopl.
mortality, density-dep, Fe-replete  \vspace{1pt} \\
%
  $\Pi^{Zp}_{mort,l}$  & 0.3 & d$^{-1}$ (mMol N m$^{-3}$)$^{-1}$ & Zoopl.
mortality, density-dep, Fe-deplete  \vspace{1pt} \\
%
  $F_{nmp}$ & 0.01 & (none) & Fraction of mortality to dissolved nutrient
\vspace{1pt} \\
%
  $F_{zmort}$ & 0.67 & (none) & Fraction of zoopl mortality to dissolved
nutrient  \vspace{1pt} \\
%
  $V_{Dt}$  & 10.0 & m d$^{-1}$ & Sinking speed, detritus  \vspace{1pt}
\\
%
  $\Pi^{DtC}_{rmndd}$  & 8.58 & m d$^{-1}$ & Detrital remineralisation
rate factor, carbon  \vspace{1pt} \\
%
  $\Pi^{DtC}_{rmnmx}$  & 0.125 & d$^{-1}$ & Max detrital remineralisation
rate, carbon  \vspace{1pt} \\
%
  $\Pi^{DtN}_{rmndd}$  & 8.58 & m d$^{-1}$ & Detrital remineralisation
rate factor, nitrogen  \vspace{1pt} \\
%
  $\Pi^{DtN}_{rmnmx}$  & 0.125 & d$^{-1}$ & Max detrital remineralisation
rate, nitrogen  \vspace{1pt} \\
%
  $\Pi^{DtSi}_{rmn}$  & 0.05 & d$^{-1}$ & Detrital silicate (opal)
remin/dissolution rate  \vspace{1pt} \\
%
  $V_{Dm}$  & 1.0 & m d$^{-1}$ & Diatom sinking speed  \vspace{1pt} \\
%
  $R^{eco}_{fe2c}$ & 0.025 & $\mu$Mol Fe (mMol C)$^{-1}$ & Molar Fe:C
ratio for ecosystem  \vspace{1pt} \\
%
%  $k_{FeT}$ & 0.2 & $\mu$Mol Fe m$^{-3}$ & Half-saturation factor for
Fe-limitation  \vspace{1pt} \\
  $k_{FeT}$ & 0.2 & $\mu$Mol Fe m$^{-3}$ & Scale factor for Fe-limitation
\vspace{1pt} \\
%
  $LgT$  & 1.0 & $\mu$Mol m$^{-3}$ & Total ligand concentration
\vspace{1pt} \\
%
  $K_{FeL}$  & 200.0 & ($\mu$Mol m$^{-3}$)$^{-1}$ & Fe-ligand partition
function  \vspace{1pt} \\
%
  $\Pi^{FeF}_{ads}$  & 5.0$\times$10$^{-5}$ & d$^{-1}$ & Adsorption rate
of iron onto particles  \vspace{1pt} \\
%
  $R^{eco}_{o2c}$  & 1.302 & mMol O$_{2}$ (mMol C)$^{-1}$ & Molar
O$_{2}$:C ratio for ecosystem  \vspace{1pt} \\
%
%  $R^{Ph}_{cc2pp}$  & 0.0195 & mMol CaCO$_{3}$ (mMol C)$^{-1}$ & Molar
ratio of carbonate formation to organic production, misc-Phyto
\vspace{1pt} \\
```

```
%   $R^{Ph}_{cc2pp}$  & 0.0195 & mMol CaCO$_{3}$ (mMol C)$^{-1}$ & Molar
ratio carbnt frmtn : organic prodn, misc-Phyto  \vspace{1pt} \\
  $R^{Ph}_{cc2pp}$  & 0.0195 & mMol CaCO$_{3}$ (mMol C)$^{-1}$ & Misc-
Phyto molar ratio, carbnt frmtn:organic prodn  \vspace{1pt} \\
%
  $Z_{lys}$ & 2113.0 & m & Depth of lysocline  \vspace{1pt} \\
%
\hline
\end{tabular}
\end{table}
%
%%t
%\begin{table}[t]
%\caption{TEXT}
%\begin{tabular}{column = lcr}
%\tophline
%
%\middlehline
%
%\bottomhline
%\end{tabular}
%\belowtable{} % Table Footnotes
%\end{table}
%
%%% TWO-COLUMN TABLE
%
%%t
%\begin{table*}[t]
%\caption{TEXT}
%\begin{tabular}{column = lcr}
%\tophline
%
%\middlehline
%
%\bottomhline
%\end{tabular}
%\belowtable{} % Table Footnotes
%\end{table*}
%
%
%%% NUMBERING OF FIGURES AND TABLES
%%%
%%% If figures and tables must be numbered 1a, 1b, etc. the following
command
%%% should be inserted before the begin{} command.
%
%\addtocounter{figure}{-1}\renewcommand{\thefigure}{\arabic{figure}a}
%
%
%%% MATHEMATICAL EXPRESSIONS
%
%%% All papers typeset by Copernicus Publications follow the math
typesetting regulations
%%% given by the IUPAC Green Book (IUPAC: Quantities, Units and Symbols
in Physical Chemistry,
%%% 2nd Edn., Blackwell Science, available at:
http://old.iupac.org/publications/books/gbook/green_book_2ed.pdf, 1993).
%%%
```

```
%%% Physical quantities/variables are typeset in italic font (t for time,
T for Temperature)
%%% Indices which are not defined are typeset in italic font (x, y, z, a,
b, c)
%%% Items/objects which are defined are typeset in roman font (Car A, Car
B)
%%% Descriptions/specifications which are defined by itself are typeset
in roman font (abs, rel, ref, tot, net, ice)
%%% Abbreviations from 2 letters are typeset in roman font (RH, LAI)
%%% Vectors are identified in bold italic font using \vec{x}
%%% Matrices are identified in bold roman font
%%% Multiplication signs are typeset using the LaTeX commands \times (for
vector products, grids, and exponential notations) or \cdot
%%% The character * should not be applied as mutliplication sign
%
%
%%% EQUATIONS
%
%%% Single-row equation
%
%\begin{equation}
%
%\end{equation}
%
%%% Multiline equation
%
%\begin{align}
%& 3 + 5 = 8\\
%& 3 + 5 = 8\\
%& 3 + 5 = 8
%\end{align}
%
%
%%% MATRICES
%
%\begin{matrix}
%x & y & z\\
%x & y & z\\
%x & y & z\\
%\end{matrix}
%
%
%%% ALGORITHM
%
%\begin{algorithm}
%\caption{…}
%\label{a1}
%\begin{algorithmic}
%…
%\end{algorithmic}
%\end{algorithm}
%
%
%%% CHEMICAL FORMULAS AND REACTIONS
%
%%% For formulas embedded in the text, please use \chem{}
%
```

```
%%% The reaction environment creates labels including the letter R, i.e.
(R1), (R2), etc.
%
%\begin{reaction}
%%% \rightarrow should be used for normal (one-way) chemical reactions
%%% \rightleftharpoons should be used for equilibria
%%% \leftrightarrow should be used for resonance structures
%\end{reaction}
%
%
%%% PHYSICAL UNITS
%%%
%%% Please use \unit{} and apply the exponential notation

\end{document}
```

---

## Referee Report (RR1)

**Comments on "*Description and evaluation of the Diat-HadOCC model v1.0: the ocean biogeochemical component of HadGEM2-ES*"**
by Ian Totterdell, submitted to Geosci. Model Dev. (gmd-2017-90).

This submission consists in a much improved version of the manuscript. I appreciate the efforts devoted by the author in meeting our concerns and questions.

The aim of the paper is now more clearly stated and a throughout analysis of the results is provided; also are the model description and the sequence of processes easier to follow.

However I have two main criticisms. First too much emphasis is given to the different behavior of diatoms and other phytoplankton species. Indeed, given that the iron and silicate cycles suffer from severe shortcomings discussing differences in productivity and seasonality of the two groups does not rely on robust results. Further, the manuscript is quite long. Focusing on robust and essential processes would allow reducing its length.

These points and some others are detailed below; they should be thoroughly addressed before the text be accepted for publication.

**Major comments**

1. Several problems with the iron ans silicate fields prevent a different behavior of misc-phytoplankton and diatoms. These problems are a significant drift in both fields (page 21, lines 20–21), an inadequate Fe recycling scheme, and a too high silicate dissolution rate (page 23, line 19). As a result, the silicate and iron fields are hardly limiting (Section 4.1.8 and Figs 27, 34). Therefore, diatoms do not behave much differently than misc-phytoplankton. The differences observed in primary production between the two groups is mostly due to the maximum growth rate of diatoms being larger than that of the other phytoplankton.

   In consequence, Figs 6,7, 8, 9, 10, and 42 should only illustrate the total primary production and biomass.

2. Iron and silicate

   - How large is the drift in silicate and iron (" However, there were still significant drifts in the silicate and dissolved iron fields."; Page 21, lines 20–21)?

   - Fig. 29 should be eliminated; indeed, analyzing the seasonal cycle in these circumstances is pointless. Related information should be removed from Fig. 37.

   - It is not obvious from the panels of Fig. 34 that there are significant areas where iron is limiting at certain times of the seasonal cycle (page 32, lines 22–24). At, perhaps, the exception of north and equatorial Pacific, the amplitude appears to be much less than the background value.

3. While I do agree that some methane is produced in the ocean (page 4, lines 24–26), this mostly occurs in sediments. In the open ocean the main process allowing the decay of organic matter in low-oxygen areas is denitrification (e.g., Gruber, 2008). Oxygen would be consumed during the first step of nitrification (Zehr and Kudela, 2011).

4. Some modeling choices are not accompanied with any evidence. Adequate references should be provided for:

- iron-dependency of growth rates, zooplankton feeding preference, and Si:N ratio in diatoms (page 5, lines 10–15).
- adaptation of phytoplankton growth rates to the average temperature (page 5, lines 29–30).

5. The declared aim of this manuscript is providing a detailed description of the biogeochemical component of the HadGEM2-ES model used for the CMIP5 simulations. Hence, model features which were not activated for those experiments should not be described since the present work does not offer any opportunity of evaluating their performance. Therefore several parts should be adapted:

- Lines 8 to 27 on page 5 could be much reduced by only discussing the active dependency of diatom growth rate on iron (with adequate reference); in parallel Table 5 could also be shortened.
- Sections 2.1.2 (C to Chl ratio) as well as the corresponding terms in Table 5 should be eliminated. That the Carbon to Chlorophyll ratio is constant and identical for misc-Phyto and diatoms should be mentioned adequately on lines 4–5 of page 8.

6. The description of the DMS model (page 19, lines 24–28) is very reduced. Is there any impact of the ocean DMS flux on atmospheric processes? Is this model fully described in another publication?

7. Alkalinity

- A rain ratio of 0.0195 is very low, usually it is of the order of 0.1 or more (Tsunogai and Noriki, 1991). Why is such a low value selected?
- Considering the respective roles of soft tissue production and $CaCO_3$ in driving alkalinity such a small rain ratio gives an utmost importance to the usually small contribution from nitrate uptake. Indeed, assuming that 1 mol of nitrate is used for soft tissue production, and that diatoms and misc-phytoplankton are equally contributing, then the change in alkalinity would be

$$\Delta TA = -0.5 \times 2 \times 6.625 \times 0.0195 + 1 = +0,871$$

where the role of soft tissue dominates – contrary to what is stated on page 26 (end of line 9).
- The lack of vertical contrast in Alkalinity predicted by the model (page 26, lines 21–28, and Fig. 15) is not surprising considering that the rain ratio is low and that the lysocline is fixed and at identical depth everywhere. The conclusion on lines 25–28 should be revised.

8. With respect to ocean carbon cycle an important process is the export production. Is this quantity measurable in the present experiments? How does it compare to other estimates (models or field studies)?

**Other comments**

1. Structure

   - Most of the material on pages 7 and 8 (photosynthesis sub-mode) and section 2.3.4 should be moved to annexes.
   - Section 4.1.8 (nutrients) should precede/accompany sections 4.1.1 to 4.1.3
   - The description of results for $O_2$ and AOU should come after that of nitrate, silicate, and iron. Should constitute a specific section.

2. Figures

   - Fig. 4: phase is not discussed in the text; the right panels should be eliminated
   - Fig. 5 is discussed nowhere in the text
   - Fig. 16 is not needed; there is no discussion of this illustration
   - Fig. 20 does not come in order in the text
   - Fig. 34: eliminate lower panel
   - Fig. 35 and related discussion on page 32 could be eliminated without any information loss
   - Fig. 37 does not come in order in the text
   - Fig. 42 and associated discussion should be eliminated

3. Miscellaneous

   - page 4, line 10: its concentration is changed **by** biological processes
   - page 5, line 31: phytopl**a**nkton
   - page 6, lines 4–5: what is meant by "(and the temperature factor is actively used)"?
   - page 6, line 23: ... and instantaneous production calcul**a**ted ...
   - page 32, lines 32–33: units for eGEOTRACES data are nMolFe/kg, not nMolFe/m$^3$
   - Page 32, last line is unfinished

**References**

Gruber, N. (2008). Nitrogen in the Marine Environment. Elsevier. pp. 135. ISBN 978-0-12-372522-6

Tsunogai, S., and S. Noriki (1991). Particulate fluxes of carbonate and organic carbon in the ocean. Is the marine biological activity working as a sink of the atmospheric carbon?, Tellus, 43, 256–266.

Zehr, J. P., and R.M. Kudela (2011). Nitrogen cycle of the open ocean: From genes to ecosystems. Annual Review of Marine Science. 3: 197225. doi:10.1146/annurev-marine-120709-142819

---

## Author Response (AR3)

**Description and evaluation of the Diat-HadOCC model v1.0: the ocean biogeochemical component of HadGEM2-ES**

**Ian Totterdell**

**2nd response to Editor**

I thank the Reviewers for their helpful comments (and for spotting a number of proof-reading errors that I missed).

The greatest changes were in response to the comments of Reviewer #2. In the results section where the model is evaluated I have reduced the emphasis on the separate results of Diatoms and misc-Phytoplankton (because parameter choices made them very similar) and instead concentrated on the total phytoplankton biomass and production. I also show less results for dissolved silicate and dissolved iron, since the results were compromised by poor parameter choices and model structure, respectively. As a result of these changes the number of figures has decreased by 4. At the behest of this reviewer I have also moved some of the more detailed parts of the model description into annexes (I tried to make them appendices but doing so changed the numbering of the figures for some reason).

I should also point out that it appears that the only versions of latex available on our computer system do not put line numbers on the manuscript (the abstract still has them because it is unchanged).

I hope these revisions enable the manuscript to be published.

**Description and evaluation of the Diat-HadOCC model v1.0: the ocean biogeochemical component of HadGEM2-ES**

**Ian Totterdell**

**2ⁿᵈ response to Reviewer #2**

I thank the Reviewer for their thoughtful comments and advice; and I am pleased that they considered the revised manuscript to be an improvement.

In particular I thank the reviewer for the detailed analysis and discussion of the alkalinity (Major comment 7), which corrects a hasty analysis error that I had put into the manuscript on the matter.

In this second revision I have reduced the emphasis given, in the evaluation section, to the growth of the diatoms and misc-Phytoplankton separately. I have also reduced the space given to the silicate and iron results, as suggested, because they suffer from poor choices of parameters. This reduces the length of the manuscript a bit. I have taken much of the advice on structure into account, and where I have not I have explained my choice.

**Major comments:**

1. "Several problems with the iron and silicate fields…" As stated above, I have replaced most plots that separately showed the biomass and productivity of the Diatoms and misc-Phytoplankton, for decadal means, seasonal cycles and future projections. The plots have been replaced by a smaller number of plots showing such information regarding the total phytoplankton biomass and production. The relevant text has been altered to reflect this. However Fig 6 (number is unchanged from previous manuscript) retains the separate Diatom and Misc-Phyto biomass plots, and has an extra panel added showing the total phyto biomass, because I thought it was useful to demonstrate the similarity once. The new Fig 7 shows the seasonal cycle of the total phyto, replacing former Figs 7 and 8. New Fig 8 shows the total primary productivity (mean and seasonal cycle), replacing former Figs 9 and 10. In the future projections section, new Fig 38 shows the change in total PP in the Atlantic and Pacific basins, replacing the 4 panels (split by basin and phyto type) in the former Fig 42.

2. Iron and Silicate:
(a) values for the drift in the silicate field and the iron field are now given (page 15, lines 7 to 10).
(b) Former Fig 29 (seasonal cycle of surface Si) has been removed. However, I have left the points in the corresponding Taylor diagram (new Fig 29, former Fig 34) for reasons given in the text.
(c) areas where iron is seasonally limiting: I checked that such areas exist, the plot below shows the number of months per year when the surface iron is less than the Fe scale factor (for limitation). (The plot has not been added to the manuscript, just supplied for this response).

[Figure]

3. Methane description: I have removed the invalid justification for how Diat-HadOCC avoids producing negative oxygen concentrations, but I have retained the description of how it was pragmatically done.

4. Modelling choices:
(a) iron-dependency of growth-rates: I have expanded (somewhat) the explanation how the parameterisation of iron-dependency was arrived at – being basically a modelling "trick" to account for changing ecosystem structure in a very simple model – and what it does, and does not, imply. I have referenced the earlier modelling work by Dr Mike Fasham, aimed at fitting a simple model to the results of the SOIREE iron fertilisation experiment, which inspired the formulation that Diat-HadOCC uses.
(b) In the CMIP5 simulations, we chose to switch the temperature dependence of phytoplankton growth off, though the option to use the Eppley (1972) formulation had been coded, because the effects of running with it on were extreme – it clearly does not describe differences between populations living in different temperature waters. I have removed any reference to temperature adaption from the text, just describing the choice that was made.

5. Inactive features
(a) iron-dependency: I have chosen to leave the full description in the text; much of it is needed in any case to describe how the diatom growth rate varied, and while the other dependencies were not used in CMIP5 they were examined at an earlier stage in model development and may be useful (as a clever option of taking into account implicit changes to ecosystem structure caused by iron) for any other modeller wanting to use Diat-HadOCC.
(b) Since the carbon:chlorophyll ratio was held constant in the CMIP5 runs, I have removed the description of the parameterisation from the main text. However, since the same parameterisation has been used in other experiments, not described here, I have moved it to an annex.

6. DMS sub-model: I have provided a reference (Halloran et al., 2010) where the DMS sub-model used in HadGEM2-ES is fully described.

7. Alkalinity:
(a) The ratio with a value of 0.0195 is not the rain-ratio, it is the ratio of calcium carbonate production to misc-Phyto organic production. I have made this much clearer in the relevant text. Given an export ratio of around 16% (see later) of which roughly half is due to misc-Phyto a rain-ratio of about 0.053 is obtained.
(b) I agree with your analysis and have changed the text accordingly.
(c) I agree with your conclusion and have changed the text accordingly.

8. Export flux: I have added a short sub-section giving the total sinking flux, the export ratio, and the effective rain-ratio. I compare the export flux (5.58 PgC/yr) to estimates from models and satellite-based estimates. Our results are within the range of values given, but at the low end (given our low primary production, this is not surprising).

**Other comments:**

1. Structure
(a) I have moved the suggested sections of model description, which are in fact very detailed, to Annexes (and also the description of the C:Chl formulation, as mentioned earlier).
(b) Nutrients: I have left the nutrients section where it was (after DIC, TAlk, pCO2 and air-sea flux) because I wanted to order the sections on importance for the CMIP5 experiment: the phytoplankton and the carbon fluxes are the most important, the nutrients are to some extent a means to that end.
(c) Oxygen/AOU: I have however split the description of the Oxygen and AOU results into their own section (and also moved the section on dissolved iron, which previously was placed after them, to follow immediately after the Silicate sub-section).

2. Figures:
(a) Fig 4, Phase: The omission of any discussion about phase in Fig 4 was an error, so rather than remove the panel I have included the text that I had intended.
(b) Fig 5: likewise the discussion of the Taylor diagram in Fig 5 was also omitted in error, and has now been included as originally intended.
(c) former Fig 16: Again, discussion of this figure (now Fig 14 due to some earlier figures being removed) was omitted in error, and has now been included.

(d) former Fig 20 out of order: (now Fig 18, but not moved); this Taylor diagram shows results for several quantities associated with the carbon cycle, and I feel there is value in showing the results on one diagram (it also reduces the number of figures). That means it has to go out-of-order for at least some of the quantities, and I chose to put it *after* all the quantities concerned.

(e) former Fig 34 (now Fig 27): lower panel removed as suggested.

(f) former Fig 35 and associated text: removed as suggested.

(g) former Fig 37 (now Fig 29): as in point (d), this Taylor diagram shows results for many quantities, and I chose to place it after the nutrients and before the oxygen/AOU plots.

(h) former Fig 42: I have replaced the four panels (split between both Atlantic and Pacific basins and the two phytoplankton types) with just two panels, still split between the Atlantic and Pacific basins but now for total primary production only. This is now Fig 38. I have also altered the discussion in the text.

3. Miscellaneous:

(a) typo corrected

(b) typo corrected

(c) text has been removed as a result of other changes.

(d) typo corrected

(e) units corrected

(f) the last line of (former) page 32 has been restored to its full length by correcting the latex error that removed it.

**Description and evaluation of the Diat-HadOCC model v1.0: the ocean biogeochemical component of HadGEM2-ES**

**Ian Totterdell**

**2nd Response to Reviewer #3**

I thank the Reviewer for their helpful comments, and I am pleased that they were happy with the first set of revisions. The further set of revisions, in response to comments by another reviewer, have not reduced the detail in the model description, but have made the evaluation of the results more focussed.

I have corrected the typos, and replaced "outwith" with the more standard "outside of".

Page 26, lines 25ff: Regarding the low TAlk in the deep Pacific, the ratio of calcite formation to organic production by misc-Phyto was set rather low, and produces a rain-ratio of only 0.053 (cf 0.1). Therefore the amount of alkalinity being introduced to the water below the lysocline is significantly lower than it should have been. I have changed some of the related text to explain this.

Page 32, end: the missing text (due to a latex error) has been restored.

Page 36, line 6: the max Photosynth rate is low (and the grazer parameters set to compensate). Unfortunately, during many rounds of sensitivity analyses I never found a higher rate that gave a better overall fit globally.

[revised manuscript text omitted]

$$\nu_n \equiv 1 + Z_n$$

$$DLCO0_n \equiv \nu_n - \nu_{n-1} \tag{71}$$

$$DLCO1_n \equiv (\nu_n \cdot \log(\nu_n) - \nu_n) - (\nu_{n-1} \cdot \log(\nu_{n-1}) - \nu_{n-1}) \tag{72}$$

$$DLCO2_n \equiv (\nu_n \cdot (\log(\nu_n))^2 - 2\nu_n \cdot \log(\nu_n) + 2\nu_n) - (\nu_{n-1} \cdot (\log(\nu_{n-1}))^2 - 2\nu_{n-1} \cdot \log(\nu_{n-1}) + 2\nu_{n-1}) \tag{73}$$

$$DLCO3_n \equiv (\nu_n \cdot (\log(\nu_n))^3 - 3\nu_n \cdot (\log(\nu_n))^2 + 6\nu_n \cdot \log(\nu_n) - 6\nu_n) - (\nu_{n-1} \cdot (\log(\nu_{n-1}))^3$$
$$- 3\nu_{n-1} \cdot (\log(\nu_{n-1}))^2 + 6\nu_{n-1} \cdot \log(\nu_{n-1}) - 6\nu_{n-1}) \tag{74}$$

In the above equations $R_{c2chl}^{Ph}$ is the carbon to chlorophyll ratio (units: mgC mgChl$^{-1}$), which is either calculated according to Equation 81 or fixed, $w_C$ is the molecular weight of carbon, 12.01 mg Mol$^{-1}$, and $Z_n$ is the depth (in metres) of the base of layer n, with $Z_0 = 0.0$m. Note that the $gcof$ coefficients relate to the 'g' coefficients in TRA93's Equations 18 and 21, but are numbered in a different order, as shown in Table 3; in TRA93 they were ordered by the total exponent of $c$ and $\nu$ combined, but the Diat-HadOCC model (like the HadOCC model) orders them by the exponent of $\nu$.

Based on TRA93's Equation 29 (itself derived from work described in Platt et al., 1990) the primary production for each phytoplankton type ($Dm$ or $Ph$) in layer n during a whole day can then be calculated using a fitted 5th-order polynomial. In that equation, a quantity shown as $(\alpha_{max}^B \cdot a_n^{\#} \cdot I_{n,\Phi,1}/P_m^B)$ is calculated; Platt et al.'s polynomial is fitted for values of that quantity between 0.0 and 15.8 and the fitted function oscillates wildly outside that range, but in the model the value of the corresponding quantity can be larger than 15.8. Therefore a rational function with non-oscilliatory behaviour was calculated (Geoff Evans, pers. comm) which matches the 5th-order polynomial at an input of 15.8 in both value and first derivative, and

this is used for higher input values. For phytoplankton type $X$ and layer n (of thickness $\Delta_n$):

$$solbio_n \ \cong \ solbio_{n-1} \cdot exp(-k_n \cdot \Delta_n) \tag{75}$$

$$psmaxs_n^X \ \cong \ P_n^X \cdot R_{c2chl}^X / 24 \tag{76}$$

$$V_a \ \cong \ \alpha_{mx}^X \cdot astar_n / psmaxs_n^X \tag{77}$$

$$V_b \ \cong \ V_a \cdot solbio_{n-1}$$

$$V_c \ \cong \ V_a \cdot solbio_n$$

$$V_d \ \cong \ MIN(15.8, V_b)$$

$$V_e \ \cong \ MIN(15.8, V_c)$$

$$V_f \ \cong \ MAX(15.8, V_b)$$

$$V_g \ \cong \ MAX(15.8, V_c)$$

[revised manuscript text omitted]